

SciPost Phys. Lect. Notes 68 (2023)

# Introduction to the theory of open quantum systems

**Piotr Szańkowski**⋆

Institute of Physics, Polish Academy of Sciences,
al. Lotników 32/46, PL 02-668 Warsaw, Poland

⋆ piotr.szankowski@ifpan.edu.pl

## Abstract

This manuscript is an edited and refined version of the lecture script for a one-semester graduate course given originally at the PhD school in the Institute of Physics of Polish Academy of Sciences in the Spring/Summer semester of 2022. The course expects from the student only a basic knowledge on graduate-level quantum mechanics. The script itself is largely self-contained and could be used as a textbook on the topic of open quantum systems. The program of this course is based on a novel approach to the description of the open system dynamics: It is showed how the environmental degrees of freedom coupled to the system can be represented by a multi-component quasi-stochastic process. Using this representation one constructs the super-quasi-cumulant (or super-qumulant) expansion for the system's dynamical map—a parametrization that naturally lends itself for the development of a robust and practical perturbation theory. Thus, even an experienced researcher might find this manuscript of some interest.

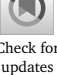

# 1  Introduction

Any quantum system in contact with any other system (its environment) counts as an *open system*; therefore, the theory of open systems is, essentially, a theory about everything (although, not the theory of everything). This poses a conundrum: how to teach the practical methods useful in solving concrete problems while properly conveying the extent of the theory's scope? On the one hand, if one places too much emphasis on the general formalism abstracted from concrete systems, then one risks being locked in on the level of generality that, indeed, encompasses all possible open system, but is unspecific to the point of being useless in practice. On the other hand, if one focuses too much on concrete systems (usually, those that can be solved completely), then the students might fall under the illusion that these few examples are, in a sense, universal models of all open systems. To find a sweet spot between these two extremes, I believe, one has to present the theory with a properly designed *parametrization* of the open system dynamics. This is, I think, the most important lesson thought to us by the pioneers of this field. Indeed, the GKLS master equation (or the Lindblad master equation)—the discovery of which was the catalyst for the open systems to become a distinct field of physics—is, essentially, a parametrization of a class of dynamics [1]. Basically, the GKLS form is a way to translate an abstract mathematical conditions for physical validity of quantum state transformations into a form that is both more amenable to physical interpretation and relatively easy to solve (thus, also relatively easy to teach). And precisely because of that the GKLS form had such an impact, even though it applies only to a narrow class of dynamics. This line of thinking convinces me that the framework that focuses on the development of the parametrizations for open system dynamics has a lot of potential and is worth further pursuit. Here, my goal was to develop the course program that is built around a novel form of parametrization that is designed with practical problems in mind and is applicable to *any* open system dynamics.

The course (and the novel parametrization) is based on the concepts I have been developing over last 2-3 years and a hefty portion of the material is yet unpublished at the time of writing. As a result, the curriculum presented in these lecture notes departs significantly from the program of traditional textbooks and monographs on the subject of open system theory (e.g., [2,3]). However, the traditional topics are not left out—they have been integrated into the body of the course, but, perhaps, in an unorthodox order.

The script includes assignments for the students to think about after the lecture. Each assignment is accompanied with a detailed solution and some commentary (the solutions can found in Appendix A). The point of these assignments is two-fold: On the one hand, they are meant to give students an opportunity to gain same hands-on experience with the presented topics. On the other hand, the assignments (and provided solutions) can be though of as a supplementary material that is still important but had to be cut to maintain the natural flow of the lecture.

My hope is that these notes can serve not only as a textbook for an initiate to the subject, but also as an introduction to the novel approach to open quantum systems that could be of some interest even to experienced researcher.

# 2  Dynamics of closed quantum systems

## 2.1  Unitary evolution, Hamiltonian, and Schrödinger equation

The fundamental axiom of quantum mechanics declares that the evolving state of a quantum system is represented in the mathematical formalism of the theory by the state-ket $|\Psi(t)\rangle$—a unit vector in a Hilbert space. When the system is insulated, so that its energy is conserved,

the time evolution is governed by the transformation

$$|\Psi(t + \Delta t)\rangle = \hat{U}(\Delta t)|\Psi(t)\rangle\,, \tag{1}$$

where the *evolution operator* $\hat{U}$ is *generated* by the Hamiltonian $\hat{H}$—a hermitian operator (i.e. $\hat{H}^\dagger = \hat{H}$) representing the energy of the system—according to the rule

$$\hat{U}(t) = e^{-it\hat{H}} = \hat{1} + \sum_{k=1}^{\infty} \frac{(-it)^k}{k!}\hat{H}^k\,. \tag{2}$$

A number of important properties follow from the definition (2):

1. The state-ket satisfies the *Schrödinger equation*,

$$\frac{d}{dt}|\Psi(t)\rangle = \frac{d\hat{U}(t-s)}{dt}|\Psi(s)\rangle = -i\hat{H}\hat{U}(t-s)|\Psi(s)\rangle = -i\hat{H}|\Psi(t)\rangle\,. \tag{3}$$

In fact, it is equally valid to treat (3) as a fundamental principle, and then the form of the evolution operator $\hat{U}(t-s)$ follows as a unique solution to the differential equation with the initial condition $|\Psi(t)\rangle|_{t=s} = \hat{U}(0)|\Psi(s)\rangle = |\Psi(s)\rangle$.

Moreover, since the Schrödinger equation is true for any initial state-ket $|\Psi(s)\rangle$, we can simply drop it and rewrite (3) as a *dynamical equation* for the evolution operator itself:

$$\frac{d}{dt}\hat{U}(t) = -i\hat{H}\hat{U}(t), \tag{4}$$

with the initial condition $\hat{U}(0) = \hat{1}$.

2. The evolution operator satisfies the *composition rule*,

$$\hat{U}(t)\hat{U}(s) = \hat{U}(t+s). \tag{5}$$

This can be shown algebraically, or we can prove it using the Schrödinger equation, instead. Indeed, if we set $|\Psi(s)\rangle = \hat{U}(s)|\Psi(0)\rangle$ as an initial condition in Eq. (3), then we know that the solution at time $t + s$ is $|\Psi(t + s)\rangle = \hat{U}(t)|\Psi(s)\rangle$. But this must equal $|\Psi(t + s)\rangle = \hat{U}(t + s)|\Psi(0)\rangle$—the solution at time $t + s$ when the initial condition is set to $|\Psi(0)\rangle$; therefore, $\hat{U}(t + s) = \hat{U}(t)\hat{U}(s)$. In other words, the composition rule is equivalent to the Schrödinger equation.

3. Evolution operator is unitary,

$$\hat{U}^\dagger(t) = \hat{U}(-t) = \hat{U}^{-1}(t). \tag{6}$$

The first equality follows directly from the hermicity of the generator,

$$\hat{U}^\dagger(t) = \left(\hat{1} + \sum_{k=1}^{\infty} \frac{(-it)^k}{k!}\hat{H}^k\right)^\dagger = \hat{1} + \sum_{k=1}^{\infty} \frac{(it)^k}{k!}\hat{H}^k = \hat{U}(-t). \tag{7}$$

The second equality is obtained by using the composition rule

$$\hat{U}(t)\hat{U}(-t) = \hat{U}(-t)\hat{U}(t) = \hat{U}(0) = \hat{1}. \tag{8}$$

**Assignment 1 (Solution)** *Use the definition* (2) *to calculate the evolution operator for the Hamiltonian*

$$\hat{H} = \frac{\omega}{2}\hat{\sigma}_z\,, \tag{9}$$

*where $\hat{\sigma}_z$ is one of the Pauli operators,*

$$\hat{\sigma}_z = \begin{bmatrix} 1 & 0 \\ 0 & -1 \end{bmatrix}; \quad \hat{\sigma}_x = \begin{bmatrix} 0 & 1 \\ 1 & 0 \end{bmatrix}; \quad \hat{\sigma}_y = \begin{bmatrix} 0 & -i \\ i & 0 \end{bmatrix}.$$

## 2.2 Time-dependent Hamiltonians and time-ordering operation

### 2.2.1 The time-dependent Schrödinger equation and its solution

When the system is not insulated and the energy is not conserved, then its energy operator—the Hamiltonian—is no longer a constant, $\partial \hat{H}(t)/\partial t \neq 0$. For such a system, the dynamics of state-ket are governed by the time-dependent version of the Schrödinger equation

$$\frac{d}{dt}|\Psi(t)\rangle = -i\hat{H}(t)|\Psi(t)\rangle. \tag{10}$$

As it turns out, the solution cannot be obtained with the formula (2) because, in general, $[\hat{H}(t), \hat{H}(s)] \neq 0$ for $t \neq s$.

**Assignment 2 (Solution)** *Show that* $\exp[-it\hat{H}(t)]|\Psi(0)\rangle$ *or* $\exp[-i\int_0^t \hat{H}(s)ds]|\Psi(0)\rangle$ *are not the solution to time-dependent Schrödinger equation.*

Given the initial condition $|\Psi(s)\rangle$, the proper solution to time-dependent Schrödinger equation is given by

$$|\Psi(t)\rangle = |\Psi(s)\rangle + \sum_{k=1}^{\infty}(-i)^k \int_s^t du_1 \int_s^{u_1} du_2 \cdots \int_s^{u_{k-1}} du_k \, \hat{H}(u_1)\cdots\hat{H}(u_k)|\Psi(s)\rangle$$

$$= \left(\hat{1} + \sum_{k=1}^{\infty}(-i)^k \int_s^t du_1 \cdots \int_s^{u_{k-1}} du_k \, \hat{H}(u_1)\cdots\hat{H}(u_k)\right)|\Psi(s)\rangle. \tag{11}$$

And thus, the evolution operator for system with time-dependent Hamiltonian reads,

$$\hat{U}(t,s) = \hat{1} + \sum_{k=1}^{\infty}(-i)^k \int_s^t du_1 \cdots \int_s^{u_{k-1}} du_k \, \hat{H}(u_1)\cdots\hat{H}(u_k). \tag{12}$$

**Assignment 3 (Solution)** *Verify by direct calculation that Eq. (11) is a solution to the time-dependent Schrödinger equation and show it satisfies the time-dependent dynamical equation*

$$\frac{d}{dt}\hat{U}(t,s) = -i\hat{H}(t)\hat{U}(t,s), \tag{13}$$

*with the initial condition* $\hat{U}(s,s) = \hat{1}$.

Conventionally, the evolution operator is written in the terms of so-called *time-ordering operation* or *time-ordered exp*,

$$\hat{U}(t,s) = Te^{-i\int_s^t \hat{H}(u)du}. \tag{14}$$

Let us try to motivate the time-ordered notation, and in particular, the appearance of exp given that we have started from Eq. (12).

Consider a c-number exp with a normal function as its argument instead of an operator,

$$e^{-i\int_s^t f(u)du} = 1 + \sum_{k=1}^{\infty}\frac{(-i)^k}{k!}\left(\int_s^t f(u)du\right)^k = 1 + \sum_{k=1}^{\infty}\frac{(-i)^k}{k!}\int_s^t du_1 \cdots \int_s^t du_k \prod_{i=1}^k f(u_i). \tag{15}$$

The time integrals are *unordered*: variables $u_i$ are independent, and each one goes from $s$ to $t$. Now, for each multiple integral over set $\{u_1, \ldots, u_k\}$ there are as many ways to sort the variables into descending order as there are permutations of the set of indexes $\{1, \ldots, k\}$, i.e. $k!$; e.g., for $k = 2$ we have

$$\int_s^t du_1 du_2 = \int_s^t du_1 du_2 \left[\theta(u_1 - u_2) + \theta(u_2 - u_1)\right] = \int_s^t du_1 \int_s^{u_1} du_2 + \int_s^t du_2 \int_s^{u_2} du_1, \qquad (16)$$

so that there are $2! = 2$ orderings (here, $\theta(x) = 1$ if $x > 0$ and 0 otherwise).

**Assignment 4 (Solution)** *Find all $3! = 6$ orderings for $\int_s^t du_1 du_2 du_3$.*

If $\Sigma(1, \ldots, k)$ is the set of all permutations of the set of indices $\{1, \ldots, k\}$, then we can write

$$\int_s^t du_1 \cdots \int_s^t du_k \prod_{i=1}^k f(u_i) = \sum_{\sigma \in \Sigma(1, \ldots, k)} \int_s^t du_{\sigma(1)} \cdots \int_s^{u_{\sigma(k-1)}} du_{\sigma(k)} \prod_{i=1}^k f(u_i)$$

$$= \int_s^t du_1 \cdots \int_s^{u_{k-1}} du_k \sum_{\sigma \in \Sigma(1, \ldots, k)} \prod_{i=1}^k f(u_{\sigma^{-1}(i)}), \qquad (17)$$

where we have simply changed the names of variables in each integral $u_{\sigma(i)} \to u_i$ [and thus, $f(u_i) \to f(u_{\sigma^{-1}(i)})$].

To clarify the notation: permutations $\sigma \in \Sigma(1, \ldots, k)$ are understood as a mapping of indices, e.g., for permutation $\sigma$ defined as $\{1, 2, \ldots, k\} \to \{k, \ldots, 2, 1\}$, we would write $\sigma(1) = k$, $\sigma(2) = k-1$, etc., and the inverse map would go as $\sigma^{-1}(k) = 1$, $\sigma^{-1}(k-1) = 2$, etc.. The inverse mapping always exist because permutation is a one-to-one map.

Continuing, since $f$'s are numbers that commute (obviously), each product $f(u_{\sigma^{-1}(1)}) \cdots f(u_{\sigma^{-1}(k)})$ in Eq.(17) can be rearranged to be equal to $f(u_1) \cdots f(u_k)$ (remember, each permutation is a one-to-one map of indexes), so that we get

$$\int_s^t du_1 \cdots \int_s^{u_{k-1}} du_k \sum_{\sigma \in \Sigma(1, \ldots, k)} \prod_{i=1}^k f(u_{\sigma^{-1}(i)}) = \int_s^t du_1 \cdots \int_s^{u_{k-1}} du_k \prod_{i=1}^k f(u_i) \sum_{\sigma \in \Sigma(1, \ldots, k)}$$

$$= k! \int_s^t du_1 \cdots \int_s^{u_{k-1}} du_k \prod_{i=1}^k f(u_i). \qquad (18)$$

As we can see, the factorial cancels with the denominator, and we get

$$e^{-i \int_s^t f(s) du} = 1 + \sum_{k=1}^\infty (-i)^k \int_s^t du_1 \cdots \int_s^{u_{k-1}} du_k f(u_1) \cdots f(u_k), \qquad (19)$$

that is, we can write the exp (15) as an time-ordered exp where the integration variables are sorted in the descending order; note that we could do this because the integrand $f$ is a number for which $f(t)f(s) = f(s)f(t)$.

When we go back to operators $\hat{H}(t)$, the Hamiltonians at different times, in general, do not commute $\hat{H}(t)\hat{H}(s) \neq \hat{H}(s)\hat{H}(t)$, so we cannot freely switch between ordered and unordered versions of operator exp,

$$\hat{1} + \sum_{k=1}^\infty (-i)^k \int_s^t du_1 \cdots \int_s^{u_{k-1}} du_k \hat{H}(u_1) \cdots \hat{H}(u_k) \neq \hat{1} + \sum_{k=1}^\infty \frac{(-i)^k}{k!} \left(\int_s^t \hat{H}(u) du\right)^k. \qquad (20)$$

**Assignment 5 (Solution)** *Compute the evolution operator for $\hat{H}(t) = \hat{H}_0$ (a constant Hamiltonian) and $\hat{H}(t) = f(t)\hat{H}_0$ and show that in these cases the time-order exp reduces to unordered exp.*

**Assignment 6 (Solution)** *Show that $\hat{U}^\dagger(t,s) = \hat{U}(s,t)$.*

### 2.2.2 Time-ordering as an operation

Formally, it is sufficient to treat $T\exp[-i\int_s^t \hat{H}(u)du]$ as a one big complicated symbol and write

$$Te^{-i\int_s^t \hat{H}(u)du} \equiv \hat{1} + \sum_{k=1}^\infty (-i)^k \int_s^t du_1 \cdots \int_s^{u_{k-1}} du_k\, \hat{H}(u_1)\cdots\hat{H}(u_k). \tag{21}$$

However, it is also permissibly to think of $T$ as a linear operation that acts on products of parameter dependent operators $\hat{H}(u)$—in fact, this is the typical approach to time-ordering. Within this interpretation, the action of $T$ in the time-ordered exp is resolved by first expanding exp into power series, and then acting with $T$ term-by-term using its linearity,

$$\begin{aligned}
Te^{-i\int_s^t \hat{H}(u)du} &= T\left\{\hat{1} + \sum_{k=1}^\infty \frac{(-i)^k}{k!}\left(\int_s^t \hat{H}(u)du\right)^k\right\} \\
&= \hat{1} + \sum_{k=1}^\infty \frac{(-i)^k}{k!} \int_s^t du_1\cdots du_k\, T\left\{\hat{H}(u_1)\cdots\hat{H}(u_k)\right\}.
\end{aligned} \tag{22}$$

The action of $T\{\hat{H}(u_1)\cdots\hat{H}(u_k)\}$ is then defined as follows: return the product but rearrange operators $\hat{H}(u_i)$ from left to right in the descending order of the value of their arguments, e.g., if $t > s$, then $T\{\hat{H}(t)\hat{H}(s)\} = T\{\hat{H}(s)\hat{H}(t)\} = \hat{H}(t)\hat{H}(s)$.

   Hence, to resolve the action of time-ordering operation in Eq. (22), first, one has to establish the order of the integration variables. This can be done by approaching the problem in the same way as we did when we transformed $\exp[-i\int_s^t f(u)du]$ into time-ordered exp, i.e. by ordering the initially unordered time integrals,

$$\begin{aligned}
\int_s^t du_1\cdots du_k\, T\left\{\hat{H}(u_1)\cdots\hat{H}(u_k)\right\} &= \int_s^t du_1\cdots\int_s^{u_{k-1}} du_k \sum_{\sigma\in\Sigma(1,\ldots,k)} T\left\{\hat{H}(u_{\sigma(1)})\cdots\hat{H}(u_{\sigma(k)})\right\} \\
&= k! \int_s^t du_1\cdots\int_s^{u_{k-1}} du_k\, \hat{H}(u_1)\cdots\hat{H}(u_k).
\end{aligned} \tag{23}$$

Since the integrals are ordered (the upper limit of $u_i$ is capped by the value of $u_{i-1}$), we now know that $u_1 > u_2 > \cdots > u_k$ is always true. Then, the time-ordering operation rearranges each product $\hat{H}(u_{\sigma(1)})\cdots\hat{H}(u_{\sigma(k)})$ into $\hat{H}(u_1)\cdots\hat{H}(u_k)$—we have done essentially the same thing as we did previously with $f(u_{\sigma(1)})\cdots f(u_{\sigma(k)})$, except, then, we could do it without $T$.

   Using $T$ as a standalone operation can simplify certain proofs; here, we will utilize it to show that the evolution operator satisfies the composition rule. First, consider an operator $\exp[\hat{A}+\hat{B}]$, where $\hat{A},\hat{B}$ are some non-commuting operators. It is well known that, unlike c-number function $\exp[a+b]$, the operator exp does not factorize easily,

$$e^{\hat{A}+\hat{B}} \neq e^{\hat{A}}e^{\hat{B}} \quad \text{when} \quad [\hat{A},\hat{B}] \neq 0. \tag{24}$$

Even though the powers and factorials in the series expansion of $\exp[\hat{A}+\hat{B}]$ align just right to give us factorization, the problem is that on the RHS all the $\hat{A}$'s are on the left of all the $\hat{B}$'s, which is definitely not the case on the LHS. Of course, if $\hat{A}$ and $\hat{B}$ commuted (or they where numbers), then this would not be any problem; in other words, $\exp[\hat{A}+\hat{B}]$ does not factorize because of the ordering of products of $\hat{A}$ and $\hat{B}$. Bearing this in mind, let us now go back to the evolution operator; to show the composition rule, first, we split the time integral into two segments,

$$\hat{U}(t,0) = Te^{-i\int_0^t \hat{H}(u)du} = Te^{-i\left(\int_s^t + \int_0^s\right)\hat{H}(u)du} = Te^{-i\int_s^t \hat{H}(u)du - i\int_0^s \hat{H}(v)dv}, \qquad (25)$$

where $t > s > 0$. Now, we would like to factorize the exp, but we know from the previous discussion, that factorization does not work because of the ordering. However, as long as the exp is under $T$, we do not have to worry about this issue because the time-ordering operation will fix everything at the end. Indeed, the rule of thumb is that operators inside $T$ can be treated as commuting c-cumbers because the proper ordering is automatically taken care of. Hence, we can write

$$Te^{-i\int_s^t \hat{H}(u)du - i\int_0^s \hat{H}(v)dv} = T\left\{e^{-i\int_s^t \hat{H}(u)du}e^{-i\int_0^s \hat{H}(v)dv}\right\}. \qquad (26)$$

Since variable $u$ of the first integral is always grater than $v$ of the second, we know that $T$ will never change the order of exps, and thus, the time-ordering acts on each exp *independently*. Therefore, we get

$$T\left\{e^{-i\int_s^t \hat{H}(u)du}e^{-i\int_0^s \hat{H}(v)dv}\right\} = \left(Te^{-i\int_s^t \hat{H}(u)du}\right)\left(Te^{-i\int_0^s \hat{H}(v)dv}\right) = \hat{U}(t,s)\hat{U}(s,0), \qquad (27)$$

i.e. the composition rule.

A word of caution. When you construct your proofs using the $T$-as-operation approach, effectively, you are trying to replace algebra with words. It is true that the algebraic transformations involved in some of the proofs can be very laborsome; skipping the grind with clever arguments based on one's understanding of $T$-as-operation is, for sure, a tempting proposition. However, there are risks involved with replacing a procedural algebra with words: it is very easy to fall into a trap of making false arguments that sound correct but are extremely difficult to verify without going back to algebra. In best case scenario, incorrect arguments lead to incorrect results, which hopefully look suspicious enough for one to investigate further and catch the error. A more dangerous case is when incorrect arguments lead to correct result. When this happens, then, since there is no reason to doubt the correctness of the used arguments, one might start reusing them to "solve" more problems—a disaster in the making! Fortunately, at no point one is forced to take the $T$-as-operation route—everything can be shown by doing the algebra on the series representation of time-ordered exps (21). Our recommendation is to stick with algebra and avoid $T$-as-operation approach; it is much safer that way and, with some practice, not so laborsome as it seems.

**Assignment 7 (Solution)** *Using only algebraic methods prove the composition rule,*

$$\hat{U}(t,w)\hat{U}(w,s) = \hat{U}(t,s). \qquad (28)$$

Finally, note that the composition rule plus $\hat{U}^\dagger(t,s) = \hat{U}(s,t)$ proves that $\hat{U}(t,s)$ is unitary:

$$\hat{U}^\dagger(t,s)\hat{U}(t,s) = \hat{U}(s,t)\hat{U}(t,s) = \hat{U}(s,s) = \hat{1}, \qquad (29)$$

$$\hat{U}(t,s)\hat{U}^\dagger(t,s) = \hat{U}(t,s)\hat{U}(s,t) = \hat{U}(t,t) = \hat{1}. \qquad (30)$$

## 2.3 Disentanglement theorem

Consider the system with the Hamiltonian

$$\hat{H} = \hat{H}_0 + \hat{V}, \tag{31}$$

such that the dynamics generated by $\hat{H}_0$ alone is already solved, i.e. the evolution operator

$$\hat{U}_0(t) = e^{-it\hat{H}_0}, \tag{32}$$

is a known quantity. The idea is to "factor out" or to *disentangle* the known part $\hat{U}_0$ from the full evolution operator,

$$\hat{U}(t) = e^{-it(\hat{H}_0+\hat{V})} = e^{-it\hat{H}_0}\hat{U}_I(t,0), \tag{33}$$

the remainder $\hat{U}_I(t,0)$ is the evolution operator in so-called *interaction picture*: for time-independent $\hat{H}$, $\hat{U}_I(t,0)$ is given in the terms of $\hat{V}_I(t)$—the interaction picture of $\hat{V}$,

$$\hat{V}_I(t) \equiv \hat{U}_0^\dagger(t)\hat{V}\hat{U}_0(t) = \hat{U}_0(-t)\hat{V}\hat{U}_0(t), \tag{34}$$

$$\hat{U}_I(t,0) = Te^{-i\int_0^t \hat{V}_I(s)ds} = \hat{1} + \sum_{k=1}^\infty (-i)^k \int_0^t ds_1 \cdots \int_0^{s_{k-1}} ds_k \hat{V}_I(s_1)\cdots\hat{V}_I(s_k). \tag{35}$$

Note that, here, $T$ is understood to affect the ordering of $\hat{V}_I(t)$ operators; soon it will become more apparent why we are pointing out this seemingly obvious fact.

**Assignment 8 (Solution)** *Prove the disentanglement theorem by transforming the dynamical equation (4) to the interaction picture.*

Now, let us consider the interaction picture for time-dependent Hamiltonian,

$$\hat{H}(t) = \hat{H}_0(t) + \hat{V}(t). \tag{36}$$

In this case, the known evolution operator is given by the standard time-ordered exponential,

$$\hat{U}_0(t,0) = \hat{1} + \sum_{k=1}^\infty (-i)^k \int_0^t ds_1 \cdots \int_0^{s_{k-1}} ds_k \hat{H}_0(s_1)\cdots\hat{H}_0(s_k) \equiv T_{H_0} e^{-i\int_0^t \hat{H}_0(s)ds}, \tag{37}$$

where we have introduced $T_{H_0}$ that explicitly specifies which operator products can be rearranged by the time-ordering operation. With this new notation we can now give the result for the interaction picture of the evolution operator,

$$\hat{V}_I(s) = \hat{U}_0^\dagger(s,0)\hat{V}(s)\hat{U}_0(s,0) = \hat{U}_0(0,s)\hat{V}(s)\hat{U}_0(s,0); \tag{38}$$

$$\hat{U}_I(t,0) = T_{V_I} e^{-i\int_0^t \hat{V}_I(s)ds} = \hat{1} + \sum_{k=1}^\infty (-i)^k \int_0^t ds_1 \cdots \int_0^{s_{k-1}} ds_k \hat{V}_I(s_1)\cdots\hat{V}_I(s_k). \tag{39}$$

Let us now explain why we need $T_{V_I}$. Imagine that we stick with the "universal" $T$ that time-orders products of any operators, then we write

$$Te^{-i\int_0^t \hat{V}_I(s)ds} = T\left\{\hat{1} - i\int_0^t \hat{V}_I(s)ds + \ldots\right\} = \hat{1} - i\int_0^t ds\, T\left\{\hat{U}_0^\dagger(s,0)\hat{V}(s)\hat{U}_0(s,0)\right\} + \ldots \tag{40}$$

Here, we have expanded the exp into power series up to the first-order terms, and then, we have substituted the full form of $\hat{V}_I$. If we treat $T$ as a "universal" time-ordering operation (i.e. in the sense that it acts on all time-dependent operators), then it makes sense to think that $T$ also affects the order within composite operators such as $\hat{V}_I(t)$ and $\hat{U}_0(t,0)$. But, since all $\hat{H}_0(t)$ inside $\hat{U}_0^\dagger(s,0)$ are earlier than $s$, then the "universal" $T$ would give us

$$
\begin{aligned}
T e^{-i\int_0^t \hat{U}_0^\dagger(s,0)\hat{V}(s)\hat{U}_0(s,0)ds} &= \hat{1} - i\int_0^t ds\, T\left\{\hat{U}_0^\dagger(s,0)\hat{V}(s)\hat{U}_0(s,0)\right\} + \dots \\
&= \hat{1} - i\int_0^t ds\,\hat{V}(s)\hat{U}_0^\dagger(s,0)\hat{U}_0(s,0) + \dots \\
&= \hat{1} - i\int_0^t \hat{V}(s)ds - \int_0^t ds_1\int_0^{s_1} ds_2\,\hat{V}(s_1)\hat{V}(s_2) + \dots \\
&= T e^{-i\int_0^t \hat{V}(s)ds},
\end{aligned}
\tag{41}
$$

i.e. we have just "shown" that the operator exp factorizes like c-number exp,

$$
T e^{-i\int_0^t\left[\hat{H}_0(s)+\hat{V}(s)\right]ds} = \left(T e^{-i\int_0^t \hat{H}_0(s)ds}\right)\left(T e^{-i\int_0^t \hat{V}(s)ds}\right),
\tag{42}
$$

which is, of course, wrong!

The "specialized" time-ordering $T_{V_I}$ is allowed to rearrange only $\hat{V}_I(t)$'s that are treated as atomic expressions (i.e. the internal ordering of the component operators is not affected by $T_{V_I}$), and thus, it gives us the correct result,

$$
\begin{aligned}
T_{V_I} e^{-i\int_0^t \hat{V}_I(s)ds} &= \hat{1} + \sum_{k=1}^{\infty}(-i)^k \int_0^t ds_1\cdots\int_0^{s_{k-1}} ds_k\,\hat{V}_I(s_1)\cdots\hat{V}_I(s_k) \\
&= \hat{1} - i\int_0^t \hat{U}_0(0,s)\hat{V}(s)\hat{U}_0(s,0)ds \\
&\quad - \int_0^t ds_1\int_0^{s_1} ds_2\,\hat{U}(0,s_1)\hat{V}(s_1)\hat{U}_0(s_1,s_2)\hat{V}(s_2)\hat{U}_0(s_2,0) + \dots
\end{aligned}
\tag{43}
$$

**Assignment 9 (Solution)** *Prove the disentanglement theorem for time-dependent Hamiltonian using only algebraic methods.*

**Assignment 10 (Solution)** *Show that*

$$
\hat{U}(t,s) = \hat{U}_0(t,s)T_{V_I}e^{-i\int_s^t \hat{V}_I(u,s)du} = \hat{U}_0(t,0)\left(T_{V_I}e^{-i\int_s^t \hat{V}_I(u)du}\right)\hat{U}_0^\dagger(s,0),
$$

*where*

$$
\hat{V}_I(u,s) \equiv \hat{U}_0^\dagger(u,s)\hat{V}(u)\hat{U}_0(u,s) \neq \hat{V}_I(u), \quad \text{when} \quad s \neq 0.
$$

## 2.4 Computability of unitary evolution operator

Formal equations and definitions are fine and all, but what if one wishes to apply the theory and actually compute an evolution operator? For time-independent Hamiltonians the problem can be reduced to the diagonalization of $\hat{H}$: if $D = \{|n\rangle\}_n$ is the basis in which the Hamiltonian

is a diagonal matrix, $\hat{H} = \sum_n E_n |n\rangle\langle n|$, then

$$
\begin{aligned}
e^{-it\hat{H}} &= \sum_{k=0}^{\infty} \frac{(-it)^k}{k!} \hat{H}^k = \sum_{k=0}^{\infty} \frac{(-it)^k}{k!} \sum_{n_1,\ldots n_k} E_{n_1} \cdots E_{n_k} |n_1\rangle\langle n_1|n_2\rangle\langle n_2| \cdots |n_k\rangle\langle n_k| \\
&= \sum_{k=0}^{\infty} \frac{(-it)^k}{k!} \sum_{n_1,\ldots n_k} E_{n_1} \cdots E_{n_k} |n_1\rangle \delta_{n_1,n_2} \cdots \delta_{n_{k-1},n_k} \langle n_k| \\
&= \sum_n |n\rangle \left( \sum_{k=0}^{\infty} \frac{(-it)^k}{k!} E_n^k \right) \langle n| = \sum_n e^{-itE_n} |n\rangle\langle n|,
\end{aligned}
\tag{44}
$$

or in the matrix notation

$$
e^{-it\hat{H}} = \exp\left( -it \begin{bmatrix} E_1 & & \\ & E_2 & \\ & & \ddots \end{bmatrix}_D \right) = \begin{bmatrix} e^{-itE_1} & & \\ & e^{-itE_2} & \\ & & \ddots \end{bmatrix}_D .
\tag{45}
$$

Bear in mind that the diagonalization is not always feasible because, e.g., the dimension of ket-space is just too large. The issue is worst when the Hamiltonian is time-dependent, because, in that case, it is impossible to find a basis in which $\hat{H}(t)$ is diagonal for all $t$ (indeed, if it was possible then $[\hat{H}(t), \hat{H}(s)] = 0$).

When the diagonalization fails, the best one can count on is to compute the operator numerically using some kind of iterative method. A naive approach would be to take the definition (2) or (12), and simply cut the infinite series at some finite order $k_0$,

$$
\hat{U}(t,0) \approx \hat{1} + \sum_{k=1}^{k_0} (-i)^k \int_0^t ds_1 \cdots \int_0^{s_{k-1}} ds_k \, \hat{H}(s_1) \cdots \hat{H}(s_k).
\tag{46}
$$

If $k_0$ is not too large this is something that could be computed numerically with a moderate difficulty. However, such an approximation is very crude and it works reasonably well only in a short time limit $t \to 0$, then cutting on $k_0 = 1$ or $k_0 = 2$ could be a good idea. In all other cases it should be avoided.

If not the truncated series, then what is the alternative? A typical approach used in many fields of physics is to integrate numerically a differential equation that defines the problem; of course, in our case that equation would be the dynamical equation (13).

**Assignment 11 (Solution)** *Write a code that integrates numerically the dynamical equation*

$$
\frac{d}{dt} \hat{U}(t,0) = -i \frac{\mu}{2} \left[ \cos(\omega t) \hat{\sigma}_x - \sin(\omega t) \hat{\sigma}_y \right] \hat{U}(t,0),
$$

*with the initial condition $\hat{U}(0,0) = \hat{1}$ and constant $\mu$, $\omega$.*

*Compare the results obtained with three integration methods: (i) the Euler method, (ii) the Runge-Kutta method, and (iii) the Crank-Nicolson method.*

## 2.5 Density matrices

As it turns out, not all states of quantum systems can be describe with state-kets $|\Psi\rangle$; in general, the system state is encoded in the *density matrix* $\hat{\rho}$—a linear operator acting on the system's state-ket Hilbert space that has the following properties

1. $\hat{\rho}$ is non-negative, $\forall_{|\phi\rangle} \langle\phi|\hat{\rho}|\phi\rangle \geqslant 0$;

2. $\hat{\rho}$ is hermitian, $\hat{\rho}^\dagger = \hat{\rho}$.

   Note that hermiticity is already implied by the non-negativity because all non-negative operators can be written as $\hat{\rho} = \hat{O}^\dagger \hat{O}$, and in this form, $\hat{\rho}$ is obviously hermitian;

3. $\hat{\rho}$ has unit trace, $\mathrm{tr}(\hat{\rho}) = 1$.

The simplest example of density matrix are so-called *pure* states,

$$\text{pure state:} \quad \hat{\rho} = |\Psi\rangle\langle\Psi|, \tag{47}$$

where $\hat{\rho}$ is a projection operator onto state-ket $|\Psi\rangle$. The conditions 1-3 are indeed met by this form: (1) a projector is a non-negative operator, $\langle\phi||\Psi\rangle\langle\Psi||\phi\rangle = \langle\phi|\Psi\rangle\langle\phi|\Psi\rangle^* = |\langle\phi|\Psi\rangle|^2 \geqslant 0$ for any $|\phi\rangle$; (2) Hermicity is already guaranteed by the non-negativity, but it is equally simply to check it explicitly, $(|\Psi\rangle\langle\Psi|)^\dagger = (\langle\Psi|^\dagger)(|\Psi\rangle)^\dagger = |\Psi\rangle\langle\Psi|$; (3) the unit trace follows from the normalization of state-kets, $\mathrm{tr}(|\Psi\rangle\langle\Psi|) = \langle\Psi||\Psi\rangle\langle\Psi||\Psi\rangle = |\langle\Psi|\Psi\rangle|^2 = 1$.

In general, density matrix represents a *mixed states* and $\hat{\rho}$ has a form of a 'mixture' of pure states,

$$\text{mixed state:} \quad \hat{\rho} = \sum_i p_i |\Psi_i\rangle\langle\Psi_i|; \quad \forall_i p_i > 0 \quad \text{and} \quad \sum_i p_i = 1, \tag{48}$$

and the set $\{|\Psi_i\rangle\}_i$ is some collection of state-kets. A mixed state satisfies conditions 1-3 because it is a convex combination of non-negative unit trace operators:

$$(1) \langle\phi| \sum_i p_i |\Psi_i\rangle\langle\Psi_i||\phi\rangle = \sum_i p_i |\langle\phi|\Psi_i\rangle|^2 \geqslant 0,$$

since $p_i > 0$;

$$(2) (\sum_i p_i |\Psi_i\rangle\langle\Psi_i|)^\dagger = \sum_i p_i (|\Psi_i\rangle\langle\Psi_i|)^\dagger = \sum_i p_i |\Psi_i\rangle\langle\Psi_i|;$$

$$(3) \mathrm{tr}(\sum_i p_i |\Psi_i\rangle\langle\Psi_i|) = \sum_i p_i = 1.$$

Importantly, the decomposition of mixed state $\hat{\rho}$ into convex mixture turns out not to be unique. That is, according to Schrödinger-Hughston-Jozsa-Wooters theorem (also known as the *purification* theorem), there is an infinite number of ways to decompose a non-pure density matrix $\hat{\rho}$ into a combination of form (48),

$$\hat{\rho} = \sum_i p_i |\Psi_i\rangle\langle\Psi_i| = \sum_k p_k' |\Psi_k'\rangle\langle\Psi_k'| = \sum_m p_m'' |\Psi_m''\rangle\langle\Psi_m''| = \dots, \tag{49}$$

where $\forall_i p_i > 0$, $\sum_i p_i = 1$, and $\forall_k p_k' > 0$, $\sum_k p_k' = 1$, etc. This unintuitive feature of mixed states can be illustrated with a simple example. Consider a general mixed state $\hat{\rho}$ in a form (48) such that the state-kets $\{|\Psi_i\rangle\}_i$ it has been decomposed into are not necessarily mutually orthogonal, i.e. $\langle\Psi_i|\Psi_j\rangle \neq \delta_{ij}$. On the other hand, $\hat{\rho}$ is a Hermitian operator, and thus, it can always be diagonalized,

$$\hat{\rho} = \sum_n \rho_n |n\rangle\langle n|, \tag{50}$$

and $\forall_n \rho_n \geqslant 0$ (because $\hat{\rho}$ is non-negative), and $\sum_n \rho_n = 1$ (because of unit trace). Note that the spectral decomposition (50) also counts as a convex mixture, albeit a special one because the eigenkets $\{|n\rangle\}_n$ are all mutually orthogonal, $\langle n|n'\rangle = \delta_{n,n'}$. Therefore, the state-kets $\{|\Psi_i\rangle\}$

cannot be the same as the eigenkets $\{|n\rangle\}_n$, but nevertheless, both the spectral decomposition and the original mixture represent the same density matrix,

$$\hat{\rho} = \sum_i p_i |\Psi_i\rangle\langle\Psi_i| = \sum_n \rho_n |n\rangle\langle n|. \tag{51}$$

Due to positivity and normalization of coefficients $p_i$, the form (48) of mixed state suggests a probabilistic interpretation: with probability $p_i$, the system is found in the corresponding pure state-ket $|\Psi_i\rangle$. This could describe an experimental situation where the system is being initialized by some classical apparatus. The apparatus has the ability to prepare the system is any one of the pure states $|\Psi_i\rangle$, but it also exhibits random behavior where it picks its setting $i$ with probability $p_i$. Because of this randomness, we cannot know which particular state was picked, so, we are forced to account for this uncertainty by describing the prepared state with a mixed density matrix.

The problem is, of course, that the decomposition of mixed state is not unique. The interpretation we sketched above is correct if our system is actually prepared by real apparatus. However, it is incorrect to say that every mixed state is mixed because it was prepared by some random classical apparatus. Later we will show different origins of "mixedness" of quantum states. An interesting observation is that it is impossible to determine the origin of mixedness just by examining the density matrix.

When the state is described with density matrix $\hat{\rho}$ (mixed or pure), the expectation value of observable $\hat{A}$ is calculated according to

$$\langle A \rangle = \mathrm{tr}\left(\hat{A}\hat{\rho}\right). \tag{52}$$

For pure states, this rule reduces to familiar formula

$$\hat{\rho} = |\Psi\rangle\langle\Psi| \ : \ \langle A \rangle = \mathrm{tr}\left(\hat{A}|\Psi\rangle\langle\Psi|\right) = \langle\Psi|\hat{A}|\Psi\rangle, \tag{53}$$

which shows the origin of state-ket-based description of quantum systems.

The time-evolution of the state described with density matrix is defined by the evolution operator $\hat{U}(t,s) = T_H \exp[-i \int_s^t \hat{H}(u)du]$ we discussed previously,

$$\hat{\rho}(t) = \hat{U}(t,s)\hat{\rho}(s)\hat{U}^\dagger(t,s). \tag{54}$$

In the case of pure states $\hat{\rho}(s) = |\Psi(s)\rangle\langle\Psi(s)|$, the evolution reduces to the evolution of state-kets we discussed previously,

$$\hat{U}(t,s)|\Psi(s)\rangle\langle\Psi(s)|\hat{U}^\dagger(t,s) = \left(\hat{U}(t,s)|\Psi(s)\rangle\right)\left(\hat{U}(t,s)|\Psi(s)\rangle\right)^\dagger = |\Psi(t)\rangle\langle\Psi(t)|. \tag{55}$$

This is another piece of explanation for the state-ket-based description of states.

The time-evolution rule (54) leads to the *von Neumann equation*—the Schrödinger equation analogue for density matrices,

$$\begin{aligned}
\frac{d}{dt}\hat{\rho}(t) &= \frac{d\hat{U}(t,s)}{dt}\hat{\rho}(s)\hat{U}^\dagger(t,s) + \hat{U}(t,s)\hat{\rho}(s)\left(\frac{d\hat{U}(t,s)}{dt}\right)^\dagger \\
&= -i\hat{H}(t)\hat{U}(t,s)\hat{\rho}(s)\hat{U}^\dagger(t,s) + \hat{U}(t,s)\hat{\rho}(s)\left(-i\hat{H}(t)\hat{U}(t,s)\right)^\dagger \\
&= -i\hat{H}(t)\hat{\rho}(t) + i\hat{\rho}(t)\hat{H}(t) \\
&= -i[\hat{H}(t),\hat{\rho}(t)],
\end{aligned} \tag{56}$$

where we have used the dynamical equation (13) to calculate the derivatives of evolution operators.

Of course, all that we have learned about the evolution operators $\hat{U}$ can be applied to time-evolving density matrix, e.g., passing to the interaction picture is achieved by switching the pictures in both evolution operators,

$$
\begin{aligned}
\hat{\rho}_I(t) = \hat{U}_0^\dagger(t,0)\hat{\rho}(t)\hat{U}_0(t,0) &= \hat{U}_0^\dagger(t,0)\left(\hat{U}_0(t,0)\hat{U}_I(t,0)\hat{\rho}\,\hat{U}_I^\dagger(t,0)\hat{U}_0^\dagger(t,0)\right)\hat{U}_0(t,0) \\
&= \hat{U}_I(t,0)\hat{\rho}\,\hat{U}_I^\dagger(t,0).
\end{aligned}
\tag{57}
$$

However, this in not always the optimal way; now, we will show the description that is better suited to dealing with density matrices.

## 2.6 Unitary maps

### 2.6.1 Super-operators

Mathematically, the Schrödinger equation is a linear differential equation where on the RHS a linear operator $\hat{H}(t)$ acts on a vector $|\Psi(t)\rangle$ that is an element of the system's Hilbert space. The operators themselves, including the density matrix $\hat{\rho}(t)$, also form a vectors space; indeed, if $\hat{A}$ and $\hat{B}$ are operators acting on state-kets, so is their linear combination $\hat{C} = \alpha\hat{A} + \beta\hat{B}$ (here, $\alpha, \beta$ are complex numbers). Therefore, operators, and the density matrix $\hat{\rho}(t)$ in particular, can formally be considered as vectors. Moreover, this abstract vector space of operators can be easily upgraded to Hilbert space when we equip it with the dot product; a typical choice is the so-called Hilbert-Schmidt inner product defined using the trace operation,

$$
(\hat{A}|\hat{B}) = \text{tr}\left(\hat{A}^\dagger\hat{B}\right).
\tag{58}
$$

Since we now know that $\hat{\rho}(t)$ can be viewed as a vector, then it is perfectly viable to think of the commutator on RHS of the von Neumann equation (56) as a linear operator that acts on vectors in the space of operators—a *super-operator*, as we will call them,

$$
\frac{d}{dt}\hat{\rho}(t) = -i[\hat{H}(t), \hat{\rho}(t)] = -i[\hat{H}(t), \bullet]\hat{\rho}(t) \equiv -i\mathcal{H}(t)\hat{\rho}(t).
\tag{59}
$$

Here, we have adopted the notation where a "bullet" symbol $\bullet$ is used as a placeholder for the argument of super-operator when it acts on operators on its right, e.g.,

$$
\left(\hat{A}\bullet + \bullet\hat{A}\right)\hat{B} = \hat{A}\hat{B} + \hat{B}\hat{A}.
\tag{60}
$$

In addition, as the "hat" symbol $\hat{A}$ is used to denote operators, here, the calligraphic font will be used to denote super-operators, $\mathcal{A}$.

Using the language of super-operators we recognize that the von Neumann equation is, essentially, of the same form as the the Schrödinger equation (i.e. both are linear differential equations). Since, "the same equation has the same solution", the time-evolution of density matrix can be written in the terms of evolution super-operator acting on the initial state. The "shape" of this super-operator, $\mathcal{U}(t,s)$, is the same as the evolution operator $\hat{U}(t,s)$: a time-ordered exp generated by the super-operator on the RHS of the differential equation, $\mathcal{H}(t) = [\hat{H}(t), \bullet]$ in this case. Therefore, the state represented by a density matrix evolves as

$$
\hat{\rho}(t) = T_\mathcal{H}e^{-i\int_s^t \mathcal{H}(u)du}\hat{\rho}(s) = \mathcal{U}(t,s)\hat{\rho}(s).
\tag{61}
$$

We will call $\mathcal{U}$ the *unitary map* (soon it will become clear why "unitary"); its explicit form reads

$$\mathcal{U}(t,s) = T_{\mathcal{H}} e^{-i\int_s^t \mathcal{H}(u)du}$$

$$= \bullet + \sum_{k=1}^{\infty} (-i)^k \int_s^t du_1 \cdots \int_s^{u_{k-1}} du_k \, \mathcal{H}(u_1) \cdots \mathcal{H}(u_k)$$

$$= \bullet + \sum_{k=1}^{\infty} (-i)^k \int_s^t du_1 \cdots \int_s^{u_{k-1}} du_k \, [\hat{H}(u_1), \bullet] \cdots [\hat{H}(u_k), \bullet]$$

$$= \bullet + \sum_{k=1}^{\infty} (-i)^k \int_s^t du_1 \cdots \int_s^{u_{k-1}} du_k \, [\hat{H}(u_1), [\hat{H}(u_2), \ldots [\hat{H}(u_k), \bullet] \ldots ]], \qquad (62)$$

where the standalone $\bullet$ is simply the super-operator identity, $\bullet\hat{A} = \hat{A}$. Of course, we can also compare Eq. (54) and Eq. (61), and immediately arrive at an interesting identity

$$\mathcal{U}(t,s) = \hat{U}(t,s) \bullet \hat{U}^{\dagger}(t,s) = \hat{U}(t,s) \bullet \hat{U}(s,t). \qquad (63)$$

Some properties of $\mathcal{U}$ are the same as those of $\hat{U}$:

1. Since the generator $\mathcal{H}(t)$ is Hermitian,

$$(\hat{A}|\mathcal{H}(t)\hat{B}) = \text{tr}\left(\hat{A}^{\dagger}[\hat{H}(t), \hat{B}]\right) = \text{tr}\left([\hat{A}^{\dagger}, \hat{H}(t)]\hat{B}\right) = \text{tr}\left([\hat{H}(t), \hat{A}]^{\dagger}\hat{B}\right) = (\mathcal{H}(t)\hat{A}|\hat{B}),$$

   $\mathcal{U}$ is unitary: $\mathcal{U}^{\dagger}(t,s) = \mathcal{U}^{-1}(t,s) = \mathcal{U}(s,t)$ (this is why it is in its name!);

2. As a time-ordered exp, $\mathcal{U}$ satisfies the composition rule,

$$\mathcal{U}(t,u)\mathcal{U}(u,s) = \mathcal{U}(t,s); \qquad (64)$$

3. As a time-ordered exp, $\mathcal{U}$ also satisfies the dynamical equation,

$$\frac{d}{dt}\mathcal{U}(t,s) = -i\mathcal{H}(t)\mathcal{U}(t,s), \qquad (65)$$

   with the initial condition $\mathcal{U}(s,s) = \bullet$.

Some properties are new (they would not make sense for operators on ket-space),

(i) $\mathcal{U}$ preserves the trace of density matrix,

$$\text{tr}(\mathcal{U}(t,s)\hat{\rho}(s)) = \text{tr}(\hat{U}(t,s)\hat{\rho}(s)\hat{U}^{\dagger}(t,s)) = \text{tr}(\hat{U}^{\dagger}(t,s)\hat{U}(t,s)\hat{\rho}(s)) = \text{tr}(\hat{\rho}(s)); \qquad (66)$$

(ii) $\mathcal{U}$ preserves positivity of the density matrix,

$$\mathcal{U}(t,s)\hat{\rho}(s) = \hat{U}(t,s)\hat{O}_s^{\dagger}\hat{O}_s\hat{U}^{\dagger}(t,s) = \left[\hat{O}_s\hat{U}^{\dagger}(t,s)\right]^{\dagger}\left[\hat{O}_s\hat{U}^{\dagger}(t,s)\right] = \hat{O}_t^{\dagger}\hat{O}_t. \qquad (67)$$

(iii) $\mathcal{U}$ preserves the purity of the density matrix,

$$\mathcal{U}(t,s)\hat{\rho}(s) = \mathcal{U}(t,s)\sum_n \rho_n |n\rangle\langle n| = \sum_n \rho_n \hat{U}(t,s)|n\rangle \left(\hat{U}(t,s)|n\rangle\right)^{\dagger} = \sum_n \rho_n |n(t)\rangle\langle n(t)|. \qquad (68)$$

That is, the eigenvalues of $\mathcal{U}(t,s)\hat{\rho}(s)$ are the same as the eigenvalues of the initial $\hat{\rho}(s)$; eigenkets have been changed, $|n\rangle \to |n(t)\rangle = \hat{U}(t,s)|n\rangle$, but in such a way that they are still mutually orthonormal, $\langle n(t)|n'(t)\rangle = \langle n|\hat{U}^{\dagger}(t,s)\hat{U}(t,s)|n'\rangle = \langle n|n'\rangle = \delta_{n,n'}$. Therefore, a pure state always remain pure, and mixed state remains mixed.

**Assignment 12 (Solution)** *Find the interaction picture with respect to $\hat{H}_0(t)$ of the unitary map for the time-dependent Hamiltonian $\hat{H}(t) = \hat{H}_0(t) + \hat{V}(t)$.*

### 2.6.2  Matrix representation of super-operators

As we said previously, the space of operators is a Hilbert space, and thus, similarly to the space of state-kets, one can find an orthonormal basis, $\mathcal{B} = \{\hat{E}_n\}_n$, such that

$$(\hat{E}_n | \hat{E}_m) = \text{tr}\left(\hat{E}_n^\dagger \hat{E}_m\right) = \delta_{n,m}, \tag{69}$$

and use it to decompose any operator,

$$\hat{A} = \sum_n \hat{E}_n(\hat{E}_n | \hat{A}) = \sum_n a_n \hat{E}_n. \tag{70}$$

The following is true for any orthonormal basis,

$$\bullet = \sum_n \hat{E}_n(\hat{E}_n | \bullet) = \sum_n \hat{E}_n \, \text{tr}\left(\hat{E}_n^\dagger \bullet\right). \tag{71}$$

This is, of course, the decomposition of identity: a property guaranteed for any vector space with inner product; e.g., in the state-ket space it takes on the form,

$$\hat{1} = \sum_n |n\rangle\langle n|, \tag{72}$$

and $\{|n\rangle\}$ is some basis. By comparing the two manifestations of the identity, we see that $(\hat{E}_n | \bullet) = \text{tr}(\hat{E}_n^\dagger | \bullet)$ is the operator-space analogue of ket-space bra $\langle n|$—the dual vector of state-ket $|n\rangle$.

Now, we can use the decomposition of identity to construct a matrix representation of super-operator,

$$\mathcal{A} = \bullet \mathcal{A} \bullet = \sum_n \hat{E}_n \, \text{tr}(\hat{E}_n^\dagger \bullet) \mathcal{A} \sum_m \hat{E}_m \, \text{tr}(\hat{E}_m^\dagger \bullet) = \sum_{n,m} \hat{E}_n \, (\mathcal{A}_{nm}) \, \text{tr}\left(\hat{E}_m^\dagger \bullet\right), \tag{73}$$

where the matrix element are given by

$$\mathcal{A}_{nm} = (\hat{E}_n | \mathcal{A}\hat{E}_m) = \text{tr}\left(\hat{E}_n^\dagger \mathcal{A}\hat{E}_m\right). \tag{74}$$

The formula (73) is the super-operator analogue of matrix representation of operators,

$$\hat{A} = \sum_{n,m} |n\rangle \, \langle n|\hat{A}|m\rangle \, \langle m|. \tag{75}$$

**Assignment 13 (Solution)** *Show that* $(\mathcal{C}^\dagger)_{nm} = \mathcal{C}_{mn}^*$.

**Assignment 14 (Solution)** *Find the super-operator matrix representation of quantum expectation value,*

$$\langle A(t)\rangle = \text{tr}\left[\hat{A}\hat{\rho}(t)\right] = \text{tr}\left[\hat{A}\mathcal{U}(t,0)\hat{\rho}\right],$$

*where $\hat{A}$ is a hermitian operator ($\hat{A}^\dagger = \hat{A}$).*

### 2.6.3  Computablility of unitary maps

There are no fundamental issues with the computability of unitary maps. Indeed, the dynamical equation (65) allows for numerical integration, and the Hermicity of time-independent generators $\mathcal{H} = [\hat{H}, \bullet]$ enables the diagonalization approach when the generator is time-independent.

The one problem with computability is purely technical. If the Hilbert space of the system is $d$-dimensional, then the dimension of the operator space is exponentially larger, $d^2$, and thus, while the evolution operator $\hat{U}$ is a $d \times d$ matrix, the unitary map $\mathcal{U}$ is $d^2 \times d^2$ dimensional—a significantly larger matrix. Numerical integration or diagonalization involving such monstrously sized matrices might start to pose real problems even for quite powerful hardware. Fortunately, this difficulty can be entirely circumvented because the unitary map is made out of smaller unitary operators, $\mathcal{U}(t,0) = \hat{U}(t,0) \bullet \hat{U}^\dagger(t,0)$. Therefore, in practice, it is much more efficient to compute $\hat{U}$, take its hermitian conjugate $\hat{U}^\dagger$, and then use them to assemble $\mathcal{U}$ instead of integrating the dynamical equation directly.

# 3 Stochastic dynamics of quantum systems

## 3.1 Basics of stochastic theory and practice

### 3.1.1 Stochastic variables

Formally, the stochastic variable $X$ (also known as random variable) is a function from a set of possible outcomes of a random event to real numbers. For example, let us choose a coin toss as the random event under consideration; the set of outcomes of the event has two elements, $\Omega_{\text{coin}} = \{\text{Heads}, \text{Tails}\}$. The stochastic variable $X_{\text{coin}}$ that uses $\Omega_{\text{coin}}$ as its set of outcomes is defined by the mappings

$$X_{\text{coin}} : \begin{cases} \text{Heads} & \rightarrow & x_h; \\ \text{Tails} & \rightarrow & x_t. \end{cases}$$

The purpose of stochastic variables is to "inject" the information about the random event into standard mathematical expressions. The stochastic variable $X$ itself is *not* a number, but it is supposed to be used as a place-holder that becomes a number when the random event produces a specific outcome. Take for example a function $f$ from real numbers to real numbers. The standard mode of handling $f$ is, of course, to supply it with a real number input $x$ to obtain a real number $f(x)$ as an output. When you write $f(X)$ ($X$ is a stochastic variable), then you create a new stochastic variable that further processes the information passed on by $X$, e.g.,

$$f(X_{\text{coin}}) : \begin{cases} \text{Heads} & \rightarrow & f(x_h); \\ \text{Tails} & \rightarrow & f(x_t). \end{cases}$$

However, writing down functions of stochastic variables and waiting for a random event to happen is not the typical way the stochastic variables are used in practice (although, they sometimes are). When you need to quantify something about the random event—which could be some complex physical process, like a coin toss—you want to abstract it to something that captures the essence of the problem (randomness, whole set of outcomes, etc.) and is also compatible with your typical mathematical formalism. An accepted approach is to work with *probabilities* of outcomes, which, roughly speaking, condense all the details about the course of the event into relative frequencies of its outcomes. The stochastic variable brings about an additional level of abstraction: since we already have a variable $X$ that correlates outcomes with numbers, we can simply skip straight to probabilities of $X$ taking on the specific value $x$; thus, we arrive at the concept of the *probability distribution* of $X$, $P_X(x)$.

The probability distributions $P_X(x)$ works in the following way. The function accepts as its inputs the values taken on by the stochastic variable; in more technical terms, the domain of the distribution coincides with the image of the variable, $\text{Dom}(P_X) = \text{Img}(X)$. For example, in the case of $X_{\text{coin}}$, the domain of its probability distribution $P_{X_{\text{coin}}}(x)$ has two elements, $\{x_h, x_t\}$.

It is customary, however, to redefine $P_X(x)$ so that its domain is not restricted, and instead it is set to zero for all the inputs outside of $\text{Img}(X)$; we will follow this convention from this point. The value the distribution assigns to a given input $x$ is, intuitively speaking, found by counting the number of the occurrences of outcome for which $X$ becomes $x$ in $N$ instances of the event, dividing this number by $N$, and going to the limit $N \to \infty$; e.g., for $X_{\text{coin}}$, assuming the coin is fair, we would get $P_{X_{\text{coin}}}(x_h) = P_{X_{\text{coin}}}(x_t) = 1/2$ as there would be an equal number of heads and tails in a long enough trial of coin tosses.

Given the description of probability distribution we sketched above, $P_X(x)$, as a function, has to satisfy the following conditions:

1. $P_X(x)$ is non-negative,

$$\forall_x P_X(x) \geqslant 0 \,, \tag{76}$$

   because the values of $P_X(x)$ are obtained by counting the occurrences of outcomes;

2. $P_X(x)$ is normalized to unity,

$$\sum_{x \in \text{Img}(X)} P_X(x) = 1, \text{ when } \text{Img}(X) \text{ is a discrete set, or}$$

$$\int_{-\infty}^{\infty} P_X(x) dx = 1, \text{ when } \text{Img}(X) \text{ is not discrete,} \tag{77}$$

   because the total number of occurrences of *any* outcome in $N$ instances of the event always equals $N$.

Conversely, any function that satisfies the above can serve as a probability distribution of some stochastic variable.

The probability distribution $P_X(x)$ allows one to answer all the important questions regarding the statistics of the stochastic variable (and thus, of the random event it represents); e.g., the probability that, when the event happens, $X$ will be converted into a value greater than certain $x_0$ is given by

$$\text{Prob}(x > x_0) = \int_{x_0}^{\infty} P_X(x) dx \; \left( \sum_{x > x_0} P_X(x), \text{ when } X \text{ is discrete} \right). \tag{78}$$

An important and very useful class of quantities derived from probability distributions are the *expectation values*. The simplest one is the *average value* of stochastic variable $X$,

$$\overline{X} = \int P_X(x) x \, dx \,. \tag{79}$$

The average value is an estimate for the arithmetic average of a large number of samples of $X$; in other words, $\overline{X}$ is a number that the values of $X$ will scatter around when the stochastic variable is repeatedly converter into numbers by the instances of the random event. The extent of this scattering is quantified by another expectation value, the *dispersion*,

$$
\begin{aligned}
\overline{X^2 - (\overline{X})^2} &= \int dx P_X(x) \left[ x - \overline{X} \right]^2 \\
&= \int dx P_X(x) \left[ x^2 - 2x\overline{X} + \overline{X}^2 \right] \\
&= \int dx P_X(x) x^2 - 2\overline{X} \int dx P_X(x) x + \overline{X}^2 \int dx P_X(x) \\
&= \overline{X^2} - \overline{X}^2 \,.
\end{aligned}
\tag{80}
$$

In general, one can compute an average $\overline{(\ldots)}$ of any function of stochastic variable,

$$\overline{f(X)} = \int dx P_X(x) f(x),\tag{81}$$

and its interpretation, obviously, depends on the form of function $f$.

### 3.1.2 Stochastic processes

A *stochastic process* is an extension of the concept of stochastic variable: a process $F(t)$ is a map from the set of outcomes to the real-valued *functions* of $t$ (usually interpreted as time). Hence, when the random event happens, depending on the outcome, the stochastic process $F(t)$ converts into a corresponding function $f(t)$–the *trajectory* of the process.

Since the image of stochastic process constitutes of functions, the probability distribution of process's trajectories is a *functional* (a function that takes functions as an input), $P_F[f]$. Of course, the distribution is non-negative, $\forall_f P_F[f] \geqslant 0$, and it is normalized,

$$\int P_F[f][Df] = 1,\tag{82}$$

where $\int \ldots [Df]$ is a functional integration.

The probability distribution $P_F[f]$ (and functional integration) can be considered as a purely formal tool that is almost never used in practical calculations. Instead, in virtually all real applications, it is sufficient to employ *joint probability distributions*

$$P_F^{(k)}(f_1, t_1; f_2, t_2; \ldots; f_k, t_k),\tag{83}$$

an infinite family $\{P_F^{(k)}\}_{k=1}^\infty$ of standard non-negative functions of $k$ pairs of inputs (the real value $f_i$ and the time $t_i$). Each joint distribution is defined only for the consecutive times $t_1 > t_2 > \cdots > t_k$ and it describes the probability that trajectories of process $F(t)$ reach value $f_k$ at time $t_k$, followed by value $f_{k-1}$ at time $t_{k-1}$, etc.. A helpful way to visualize how this works is to imagine a shutter that closes at times $t_i$ for an instant leaving open only a tiny slit at coordinate $f_i$. Such an obstacle course will block any trajectory that does not happen to be passing through the slit each time the shutter closes and $P_F^{(k)}(f_1, t_1; \ldots; f_k, t_k)$ is the probability that a trajectory drawn at random from $P_F[f]$ will pass through all the slits.

The members of the family $\{P_F^{(k)}\}_{k=1}^\infty$ are all related to each other via *Chapman-Kolmogorov equation*,

$$\int df_i P_F^{(k)}(f_1, t_1; \ldots; f_k, t_k) = P_F^{(k-1)}(\ldots; f_{i-1}, t_{i-1}; f_{i+1}, t_{i+1}; \ldots),\tag{84}$$

for $k > 1$ and $\int df_1 P_F^{(1)}(f_1, t_1) = 1$ when $k = 1$. Equation (84), also know as the *consistency relation*, can be explained in the terms of the closing shutter interpretation of joint probabilities. If $P_F^{(k)}(\ldots; f_i, t_i; \ldots)$ is the probability that trajectory passes through slit placed at the coordinate $f_i$ at time $t_i$, then $P_F^{(k)}(\ldots; f_i, t_i; \ldots) + P_F^{(k)}(\ldots; f_i', t_i; \ldots)$ is the probability that the trajectory passes through slit at $f_i$ or $f_i'$; adding more terms simply opens more options for the trajectory to slip through. Therefore, the integral in Eq. (84), where $f_i$ goes over all coordinate values, means that the LHS is the probability that, at time $t_i$, the trajectory passes through any point—in other words, LHS describes the same situation as if the shutter never closed at time $t_i$. But if the shutter is wide open at $t_i$, then the probability of passing through should be described by the joint distribution with one less pair of inputs between $f_{i-1}, t_{i-1}$ and $f_{i+1}, t_{i+1}$, i.e. the RHS.

To illustrate how to work with joint distributions let us calculate the average value of the process at certain time $t_0$. Formally, the average is defined analogously to the average of stochastic variable: it is a weighted average of values $f(t_0)$ where the weight of each value is given by the probability of drawing the corresponding trajectory $P_F[f]$,

$$\overline{F(t_0)} = \int P_F[f]f(t_0)[Df]. \tag{85}$$

One the other hand, we obtain the same result when, instead, we calculate the average by weighting values $f_0$ with the probability that the trajectories pass through the slit set at time $t_0$ at the coordinate $f_0$, i.e. the $k = 1$ joint probability distribution $P_F^{(1)}(f_0, t_0)$,

$$\overline{F(t_0)} = \int P_F^{(1)}(f_0, t_0)f_0 \, df_0. \tag{86}$$

Hence, one can calculate the average using standard integration! Another example is the *auto-correlation function* of process $F(t)$,

$$\begin{aligned}
C_F(t_1, t_2) &= \overline{\left(F(t_1) - \overline{F(t_1)}\right)\left(F(t_2) - \overline{F(t_2)}\right)} = \overline{F(t_1)F(t_2)} - \overline{F(t_1)}\,\overline{F(t_2)} \\
&= \int P_F[f]f(t_1)f(t_2)[Df] - \overline{F(t_1)}\,\overline{F(t_2)} \\
&= \theta(t_1 - t_2)\iint P_F^{(2)}(f_1, t_1; f_2, t_2)f_1 f_2 \, df_1 df_2 \\
&\quad + \theta(t_2 - t_1)\iint P_F^{(2)}(f_1, t_2; f_2, t_1)f_1 f_2 \, df_1 df_2 - \overline{F(t_1)}\,\overline{F(t_2)}. \tag{87}
\end{aligned}$$

We used here the step function, $\theta(x) = 1$ if $x > 0$ and 0 otherwise, to cover the two options that $t_1 > t_2$ or $t_2 > t_1$, because joint distributions are defined only for ordered time arguments.

In general, any *moment* of stochastic process can be calculated using joint probability distributions instead of the functional $P_F[f]$,

$$\overline{F(t_1)\cdots F(t_k)} = \int P_F[f]f(t_1)\cdots f(t_k)[Df] = \int P_F^{(k)}(f_1, t_1; \ldots; f_k, t_k)f_1 \cdots f_k df_1 \cdots df_k, \tag{88}$$

assuming $t_1 > t_2 > \cdots > t_k$. Note that even though the joint probabilities are defined only for chronologically ordered time arguments, the moments of stochastic process are completely symmetric with respect to times; indeed, take for example the second moment and assume that $t_1 > t_2$,

$$\begin{aligned}
\overline{F(t_2)F(t_1)} &= \int P_F[f]f(t_2)f(t_1)[Df] \\
&= \int P_F[f]f(t_1)f(t_2)[Df] = \int P_F^{(2)}(f_1, t_1; f_2, t_2)f_1 f_2 df_1 df_2 \\
&= \overline{F(t_1)F(t_2)}. \tag{89}
\end{aligned}$$

That is, because the trajectory values are numbers, they can always be rearranged under the functional integral into the chronological order.

### 3.1.3 Example: random telegraph noise

An example of stochastic process for which the joint probability distributions can be written in terms of analytical formulas is the *random telegraph noise* (RTN), $R(t)$, also known as the *dichotomic process*. The trajectories of RTN have only two values, $r(t) \in \{1, -1\}$, and, over time, the process switches randomly between those two values at the rate $w$. The $k$th joint probability distribution for $R$ can be written as

$$P_R^{(k)}(r_1, t_1; \ldots; r_k, t_k) = P_R^{(1)}(r_k, t_k) \prod_{j=1}^{k-1} \left( \frac{1 + r_j r_{j+1} e^{-2w(t_j - t_{j+1})}}{2} \right), \quad \text{for} \quad r_j = \pm 1, \quad (90)$$

and the first probability distribution reads

$$P_R^{(1)}(r, t) = \frac{1 + p\, r e^{-2wt}}{2}, \quad (91)$$

with $p \in [-1, 1]$.

The simplest extension of the standard RTN is a dichotomic process $R'(t)$ that switches between some other pair of values $r_\pm$; such process is obtained from $R(t)$ by rescaling and adding a constant,

$$R'(t) = \frac{r_+ - r_-}{2} R(t) + \frac{r_+ + r_-}{2}. \quad (92)$$

Another generalization of $R(t)$ is a dichotomic process where the rate of switching from $-1$ to $1$ is *not* the same as the rate for switching from $1$ to $-1$. The joint probabilities for such process preserve the factorized structure of the standard RTN but are much more complicated to write down, and so, we will not list them here. Finding their explicit form is deferred to the assignment.

**Assignment 15 (Solution)** *Calculate the average value and the auto-correlation function for random telegraph noise.*

**Assignment 16 (Solution)** *Note the following property of joint distributions of RTN,*

$$\sum_{r_1, r_2 = \pm 1} r_1 r_2 P_R^{(k+2)}(r_1, t_1; \ldots; r_k, t_k) = e^{-2w(t_1 - t_2)} P_R^{(k)}(r_3, t_3; \ldots; r_k, t_k).$$

*Prove it and use it to show that the moments of the process are given by*

$$\overline{R(t_1) \cdots R(t_{2k})} = \prod_{j=1}^{k} e^{-2w(t_{2j-1} - t_{2j})}, \quad (93)$$

$$\overline{R(t_1) \cdots R(t_{2k+1})} = p e^{-2wt_{2k+1}} \prod_{j=1}^{k} e^{-2w(t_{2j-1} - t_{2j})}, \quad (94)$$

*where $t_i > t_{i+1}$ for all $i$.*

**Assignment 17 (Solution)** *Check that RTN is a Markovian process, i.e. using the Bayes law, show that the probability for the trajectory to reach value $r_1$ at time $t_1$ depends only on the value $r_2$ attained in the latest moment in past $t_2 < t_1$, but not on all the earlier values $r_3$ at $t_3$, ..., $r_k$ at $t_k$ ($t_1 > t_2 > t_3 > \cdots > t_k$).*

**Assignment 18 (Solution)** *Calculate the joint probability distributions for the dichotomic process $T(t)$ that switches between 1 and $-1$ but the rates of switching are unequal.*

*Process $T(t)$ is Markovian and its first conditional probability, $P_T^{(1|1)}(r,t|r',t') = u_{r,r'}(t-t')$, is defined by the rate equation,*

$$\dot{u}_{r_3,r_1}(t) = \sum_{r_2=\pm 1} \left[ w_{r_3,r_2} u_{r_2,r_1}(t) - w_{r_2,r_3} u_{r_3,r_1}(t) \right],$$

*where $w_{r,r'}$ is the rate of switching from $r'$ to $r$.*

**Assignment 19 (Solution)** *Write a code to generate trajectories of RTN.*

### 3.2 Stochastic dynamics

The quantum system undergoes *stochastic dynamics* when its time evolution reads

$$\hat{\rho}(t) = \overline{T_{\mathcal{H}} e^{-i\int_0^t \mathcal{H}[F](s)ds}} \hat{\rho} = \overline{\mathcal{U}[F](t,0)}\hat{\rho} = \int P_F[f]\mathcal{U}[f](t,0)[Df]\hat{\rho}, \tag{95}$$

where $\hat{\rho} = \hat{\rho}(0)$ is the initial state and the generator of the time-ordered exp,

$$\mathcal{H}[F](t) = [\hat{H}[F](t), \bullet], \tag{96}$$

is a stochastic super-operator due to the functional dependence of the Hamiltonian $\hat{H}$ on a stochastic process $F(t)$. We will refer to this evolution super-operator as a *stochastic map*, and we will use for it a shorthand notation

$$\overline{\mathcal{U}}(t,0) \equiv \overline{T_{\mathcal{H}} e^{-i\int_0^t \mathcal{H}[F](s)ds}} = \overline{\mathcal{U}[F](t,0)}, \tag{97}$$

when its is clear from context what is the form of the stochastic generator $\mathcal{H}[F](s)$.

The stochastic maps are sometimes called the *random unitary* maps; it is an apt name because $\overline{\mathcal{U}}$ has a form of a weighted average of unitary maps. Indeed, in Eq. (95), under the functional integral, we have the probability distribution weighting the trajectory-wise time-ordered exps [trajectory-wise means here that $F(t)$ was converted into $f(t)$] that have the form of a standard unitary maps generated by the corresponding time-dependent Hamiltonian $\hat{H}[f](t)$. However, even though each trajectory-wise map is unitary, their average—i.e. the stochastic map—generally loses this property. We will come back to this point later down the line.

There are a few reasons why one would want to, or need to, describe the evolution of a quantum system in terms of stochastic map. One physical setting where the use of stochastic dynamics is warranted is when the system in question is coupled to the external field but we, as an observer, do not possess the full knowledge about the field. In such a case, we take our ignorance into account by switching to the probabilistic description and represent the field with a stochastic process $F(t)$—hence, the stochastic Hamiltonian $\hat{H}[F](t)$. Within this model, the result of any single-shot measurement we perform on the system can be explained assuming unitary evolution,

$$\hat{\rho}(t|f) = T_{\mathcal{H}} e^{-i\int_0^t \mathcal{H}[f](s)ds} \hat{\rho}, \tag{98}$$

where $\hat{\rho}(t|f)$ is the state of the system at time $t$ under the condition that the external field was $f(s)$. However, we do not know which trajectory $f(s)$ of process $F(s)$ was realized in a given measurement, we only know the probability $P_F[f]$ that its was $f(s)$. Of course, the statistics of the measurements we have carried out have to incorporate this additional degree

of uncertainty; in particular, if the expectation value of a measured observable $\hat{A}$ for a given trajectory $f(s)$ reads $\langle A(t|f)\rangle = \text{tr}[\hat{A}\hat{\rho}(t|f)]$, then the expectation value for our measurements is

$$\langle A(t)\rangle = \int P_F[f]\langle A(t|f)\rangle[Df] = \text{tr}\left\{\hat{A}\int P_F[f]\hat{\rho}(t|f)[Df]\right\} = \text{tr}\left[\hat{A}\overline{\mathcal{U}}(t,0)\hat{\rho}\right]. \quad (99)$$

That is, we are describing the state of the system using stochastic dynamics! As we have already indicated, this was only one of many possible interpretations of stochastic dynamics; we will come back to this topic later down the line in Chapter 7.

Finally, note that unitary map is a special, although trivial, case of stochastic map; if the stochastic process $F(t)$ is, actually, not stochastic at all and $P_F[f] = \delta[f - f_0]$, then

$$\overline{\mathcal{U}}(t,0) = \int \delta[f - f_0]T_{\mathcal{H}}e^{-i\int_0^t \mathcal{H}[f](s)ds}[Df] = T_{\mathcal{H}}e^{-i\int_0^t \mathcal{H}[f_0](s)ds} = \mathcal{U}(t,0). \quad (100)$$

**Assignment 20 (Solution)** *Verify that $\overline{\mathcal{U}}$ is a proper map from density matrices to density matrices, i.e. check if the operator $\hat{\rho}(t) = \overline{\mathcal{U}}(t,0)\hat{\rho}$ is positive and have unit trace.*

### 3.3 Stochastic maps and joint probability distributions

We will focus here on stochastic Hamiltonians of the form

$$\hat{H}[F](t) = \hat{H}_0 + \lambda F(t)\hat{V}, \quad (101)$$

where $\hat{H}_0$ and $\hat{V}$ are constant (non-stochastic) hermitian operators. We will not discuss any more complex dependencies, like, e.g., a multi-component stochastic processes,

$$\hat{H}[F_x, F_y, F_z](t) = \sum_{i=x,y,z} \lambda F_i(t)\hat{V}_i,$$

or time-non-local couplings,

$$\hat{H}[F](t) = \int_{-\infty}^t \lambda F(\tau)\hat{V}(t - \tau)d\tau,$$

etc. All the key points about stochastic dynamics can already be made even with the simplest form of the Hamiltonian (101).

The stochastic map $\overline{\mathcal{U}}$ can always be expressed in terms of the family of joint probability distributions $\{P_F^{(k)}\}_{k=1}^\infty$. In particular, for the stochastic Hamiltonian of form (101) the language of joint probability distributions fits very naturally. To show this, first, we switch to the interaction picture with respect to $\hat{H}_0$ (see Assignment 12 in Sec. 2.6.1),

$$\overline{\mathcal{U}}(t,0) = \overline{T_{\mathcal{H}}e^{-i\int_0^t \mathcal{H}[F](s)ds}} = e^{-it[\hat{H}_0,\bullet]}\overline{T_{\mathcal{V}_I}e^{-i\lambda\int_0^t F(s)\mathcal{V}_I(s)ds}}$$

$$\equiv e^{-it[\hat{H}_0,\bullet]}\overline{\mathcal{U}}_I(t,0), \quad (102)$$

Since $\hat{H}_0$ is non-stochastic we could bring $\exp(-it[\hat{H}_0,\bullet])$ outside the average. In what remains we have the coupling super-operator,

$$\mathcal{V}_I(s) = [e^{is\hat{H}_0}\hat{V}e^{-is\hat{H}_0}, \bullet], \quad (103)$$

being simply multiplied by the stochastic process $F(t)$. Next, we expand the time-ordered exp and we carry out the average term-by-term,

$$\overline{\mathcal{U}_I(t,0)} = \bullet + \overline{\sum_{k=1}^{\infty}(-i\lambda)^k \int_0^t ds_1 \cdots \int_0^{s_{k-1}} ds_k \, F(s_1)\mathcal{V}_I(s_1)\cdots F(s_k)\mathcal{V}_I(s_k)}$$

$$= \bullet + \sum_{k=1}^{\infty}(-i\lambda)^k \int_0^t ds_1 \cdots \int_0^{s_{k-1}} ds_k \, \overline{F(s_1)\cdots F(s_k)} \, \mathcal{V}_I(s_1)\cdots \mathcal{V}_I(s_k). \qquad (104)$$

As a result, the average exp is now transformed into a series involving the consecutive moments of the process—the *moment series*. We have shown previously that a $k$th moment $\overline{F(s_1)\cdots F(s_k)}$ can be calculated using $k$th joint distribution $P_F^{(k)}(f_1,s_1;\ldots;f_k,s_k)$ [see Eq. (88)].

### 3.4 Example of stochastic dynamics

#### 3.4.1 System setup

We will consider a two-level system—a qubit—driven by an external field represented by the random telegraph noise $R(t)$,

$$\hat{H}_Q[R](t) = \frac{\lambda}{2}\hat{\sigma}_z R(t). \qquad (105)$$

Here, the Pauli operator is given by

$$\hat{\sigma}_z = \begin{bmatrix} 1 & 0 \\ 0 & -1 \end{bmatrix}_B = |\uparrow\rangle\langle\uparrow| - |\downarrow\rangle\langle\downarrow|, \qquad (106)$$

and $B = \{|\downarrow\rangle, |\uparrow\rangle\}$ is basis in the state-ket space that is composed of eigenkets of $\hat{\sigma}_z$.

The important thing to note here is that the stochastic Hamiltonian (105) commutes with itself at different times, $[\hat{H}_Q[R](t_1), \hat{H}_Q[R](t_2)] = 0$, which, of course, implies that the super-generator

$$\mathcal{H}_Q[R](t) = [\hat{H}_Q[R](t), \bullet] = \lambda R(t)[\tfrac{1}{2}\hat{\sigma}_z, \bullet] \equiv \lambda R(t)\mathcal{S}_z, \qquad (107)$$

also commutes,

$$\mathcal{H}_Q[R](t_1)\mathcal{H}_Q[R](t_2) = \lambda^2 R(t_1)R(t_2)\mathcal{S}_z\mathcal{S}_z = \mathcal{H}_Q[R](t_2)\mathcal{H}_Q[R](t_1). \qquad (108)$$

Therefore, the time-ordering operation does nothing for the exp,

$$\overline{\mathcal{U}(t,0)} = \overline{T_{\mathcal{H}_Q} e^{-i\int_0^t \mathcal{H}_Q[R](s)ds}} = \overline{e^{-i\lambda\left(\int_0^t R(s)ds\right)\mathcal{S}_z}} = \bullet + \sum_{k=1}^{\infty}\frac{(-i)^k}{k!}\overline{\left(\lambda\int_0^t R(s)ds\right)^k} \mathcal{S}_z^k. \qquad (109)$$

This simplifies things tremendously because it is now possible to calculate the matrix of the map exactly, provided that we can diagonalize the generator $\mathcal{S}_z$.

#### 3.4.2 Matrix representation

To proceed further, we have to pick a basis $\mathcal{B}$ in the space of operators; a good first choice is the set of projectors $|e\rangle\langle e|$ and coherences $|e\rangle\langle e'|$ made out of the kets and bras of the state-ket basis $B$,

$$\mathcal{B} = \{|\uparrow\rangle\langle\uparrow|, |\downarrow\rangle\langle\downarrow|, |\uparrow\rangle\langle\downarrow|, |\downarrow\rangle\langle\uparrow|\}. \qquad (110)$$

We can now decompose any operator in this basis.

Later on we will need the initial state $\hat{\rho}$ to act on with the stochastic map, so we should decompose it first,

$$
\begin{aligned}
\hat{\rho} &= \sum_{|e\rangle,|e'\rangle\in B} |e\rangle\langle e|\hat{\rho}|e'\rangle\langle e'| = \left[\begin{array}{cc} \langle\uparrow|\hat{\rho}|\uparrow\rangle & \langle\uparrow|\hat{\rho}|\downarrow\rangle \\ \langle\downarrow|\hat{\rho}|\uparrow\rangle & \langle\downarrow|\hat{\rho}|\downarrow\rangle \end{array}\right]_{B} \\
&= \sum_{\hat{E}\in\mathcal{B}} \operatorname{tr}\left(\hat{E}^{\dagger}\hat{\rho}\right)\hat{E} = \langle\uparrow|\hat{\rho}|\uparrow\rangle|\uparrow\rangle\langle\uparrow| + \langle\downarrow|\hat{\rho}|\downarrow\rangle|\downarrow\rangle\langle\downarrow| + \langle\uparrow|\hat{\rho}|\downarrow\rangle|\uparrow\rangle\langle\downarrow| + \langle\downarrow|\hat{\rho}|\uparrow\rangle|\downarrow\rangle\langle\uparrow| \\
&= \left[\begin{array}{c} \langle\uparrow|\hat{\rho}|\uparrow\rangle \\ \langle\downarrow|\hat{\rho}|\downarrow\rangle \\ \langle\uparrow|\hat{\rho}|\downarrow\rangle \\ \langle\downarrow|\hat{\rho}|\uparrow\rangle \end{array}\right]_{\mathcal{B}} .
\end{aligned} \tag{111}
$$

The column matrix notation used here underlines that $\hat{\rho}$ is a vector in the space of operators.

We proceed to construct the matrix representation for the super-generator $\mathcal{S}_z$ using the relation (73),

$$
\begin{aligned}
\mathcal{S}_z &= \sum_{\hat{E},\hat{E}'\in\mathcal{B}} \hat{E}\operatorname{tr}\left(\hat{E}^{\dagger}\mathcal{S}_z\hat{E}'\right)\operatorname{tr}\left((\hat{E}')^{\dagger}\bullet\right) \\
&= \left[\begin{array}{cccc} \operatorname{tr}(|\uparrow\rangle\langle\uparrow|\mathcal{S}_z|\uparrow\rangle\langle\uparrow|) & \operatorname{tr}(|\uparrow\rangle\langle\uparrow|\mathcal{S}_z|\downarrow\rangle\langle\downarrow|) & \operatorname{tr}(|\uparrow\rangle\langle\uparrow|\mathcal{S}_z|\uparrow\rangle\langle\downarrow|) & \operatorname{tr}(|\uparrow\rangle\langle\uparrow|\mathcal{S}_z|\downarrow\rangle\langle\uparrow|) \\ \operatorname{tr}(|\downarrow\rangle\langle\downarrow|\mathcal{S}_z|\uparrow\rangle\langle\uparrow|) & \operatorname{tr}(|\downarrow\rangle\langle\downarrow|\mathcal{S}_z|\downarrow\rangle\langle\downarrow|) & \operatorname{tr}(|\downarrow\rangle\langle\downarrow|\mathcal{S}_z|\uparrow\rangle\langle\downarrow|) & \operatorname{tr}(|\downarrow\rangle\langle\downarrow|\mathcal{S}_z|\downarrow\rangle\langle\uparrow|) \\ \operatorname{tr}(|\downarrow\rangle\langle\uparrow|\mathcal{S}_z|\uparrow\rangle\langle\uparrow|) & \operatorname{tr}(|\downarrow\rangle\langle\uparrow|\mathcal{S}_z|\downarrow\rangle\langle\downarrow|) & \operatorname{tr}(|\downarrow\rangle\langle\uparrow|\mathcal{S}_z|\uparrow\rangle\langle\downarrow|) & \operatorname{tr}(|\downarrow\rangle\langle\uparrow|\mathcal{S}_z|\downarrow\rangle\langle\uparrow|) \\ \operatorname{tr}(|\uparrow\rangle\langle\downarrow|\mathcal{S}_z|\uparrow\rangle\langle\uparrow|) & \operatorname{tr}(|\uparrow\rangle\langle\downarrow|\mathcal{S}_z|\downarrow\rangle\langle\downarrow|) & \operatorname{tr}(|\uparrow\rangle\langle\downarrow|\mathcal{S}_z|\uparrow\rangle\langle\downarrow|) & \operatorname{tr}(|\uparrow\rangle\langle\downarrow|\mathcal{S}_z|\downarrow\rangle\langle\uparrow|) \end{array}\right]_{\mathcal{B}} \\
&= \left[\begin{array}{cccc} 0 & 0 & 0 & 0 \\ 0 & 0 & 0 & 0 \\ 0 & 0 & 1 & 0 \\ 0 & 0 & 0 & -1 \end{array}\right]_{\mathcal{B}} .
\end{aligned} \tag{112}
$$

As it turns out, $\mathcal{S}_z$ is already diagonal in basis $\mathcal{B}$—no need to diagonalize it manually (note that super-operators in this space are $4 \times 4$ matrices, so not all of them can be diagonalized analytically). The matrix representation of the map follows immediately,

$$
\begin{aligned}
\overline{\mathcal{U}}(t,0) &= \overline{e^{-i\left(\lambda\int_0^t R(s)ds\right)\mathcal{S}_z}} \\
&= \left[\begin{array}{cccc} 1 & 0 & 0 & 0 \\ 0 & 1 & 0 & 0 \\ 0 & 0 & \overline{e^{-i\lambda\int_0^t R(s)ds}} & 0 \\ 0 & 0 & 0 & \overline{e^{+i\lambda\int_0^t R(s)ds}} \end{array}\right]_{\mathcal{B}} \equiv \left[\begin{array}{cccc} 1 & 0 & 0 & 0 \\ 0 & 1 & 0 & 0 \\ 0 & 0 & W(t)^* & 0 \\ 0 & 0 & 0 & W(t) \end{array}\right]_{\mathcal{B}} .
\end{aligned} \tag{113}
$$

Let us see how this map changes the state of the qubit,

$$
\begin{aligned}
\hat{\rho}(t) &= \overline{\mathcal{U}}(t,0)\hat{\rho} \\
&= \left[\begin{array}{cccc} 1 & 0 & 0 & 0 \\ 0 & 1 & 0 & 0 \\ 0 & 0 & W(t)^* & 0 \\ 0 & 0 & 0 & W(t) \end{array}\right]_{\mathcal{B}} \left[\begin{array}{c} \langle\uparrow|\hat{\rho}|\uparrow\rangle \\ \langle\downarrow|\hat{\rho}|\downarrow\rangle \\ \langle\uparrow|\hat{\rho}|\downarrow\rangle \\ \langle\downarrow|\hat{\rho}|\uparrow\rangle \end{array}\right]_{\mathcal{B}} = \left[\begin{array}{c} \langle\uparrow|\hat{\rho}|\uparrow\rangle \\ \langle\downarrow|\hat{\rho}|\downarrow\rangle \\ \langle\uparrow|\hat{\rho}|\downarrow\rangle W(t)^* \\ \langle\downarrow|\hat{\rho}|\uparrow\rangle W(t) \end{array}\right]_{\mathcal{B}} \\
&= \left[\begin{array}{cc} \langle\uparrow|\hat{\rho}|\uparrow\rangle & \langle\uparrow|\hat{\rho}|\downarrow\rangle W(t)^* \\ \langle\downarrow|\hat{\rho}|\uparrow\rangle W(t) & \langle\downarrow|\hat{\rho}|\downarrow\rangle \end{array}\right]_{B} .
\end{aligned} \tag{114}
$$

As we can see, the stochastic map modifies only the off-diagonal terms of the density matrix.

### 3.4.3 Coherence function

A very simple form of the Hamiltonian have reduced our problem to a straightforward multiplication of the off-diagonal elements of $\hat{\rho}$ by the *coherence function $W(t)$*. However, even though the "quantum part" was trivial, we are still facing a difficult task of calculating the stochastic average of the phase factor,

$$
\begin{aligned}
W(t) = \overline{e^{+i\lambda \int_0^t R(s)ds}} &= 1 + \sum_{k=1}^{\infty} \frac{(i\lambda)^k}{k!} \int_0^t ds_1 \cdots ds_k \overline{R(s_1)\cdots R(s_k)} \\
&= 1 + \sum_{k=1}^{\infty} (i\lambda)^k \int_0^t ds_1 \cdots \int_0^{s_{k-1}} ds_k \sum_{r_1,\ldots,r_k=\pm 1} r_1 \cdots r_k P_R^{(k)}(r_1,s_1;\ldots;s_k,t_k),
\end{aligned}
\tag{115}
$$

where we have switched to ordered time integrals because joint probability distributions are defined only for a consecutive sequence of timings, $s_1 > s_2 > \cdots > s_k$.

There are only a precious few types of stochastic processes for which $W(t)$ can be computed analytically. Fortunately, the random telegraph noise is one of those lucky processes,

$$
W(t) = e^{-wt} \left[ \cosh\left(wt\sqrt{1-\tfrac{\lambda^2}{w^2}}\right) + \left(\frac{w+ip\lambda}{\sqrt{w^2-\lambda^2}}\right) \sinh\left(wt\sqrt{1-\tfrac{\lambda^2}{w^2}}\right) \right].
\tag{116}
$$

As you can see, the course of the evolution of $W$ changes its character depending on the relation between the amplitude $\lambda$ and the rate of switching $w$; in particular, when $\lambda > w$ the hyperbolic cosh and sinh turn into the corresponding harmonic functions, thus overlaying an oscillatory behavior with the exponential decay. Whatever the ratio $\lambda/w$, the long-time behavior of $W$ is a rapid decay,

$$
W(t) \xrightarrow{t\to\infty}
\begin{cases}
e^{-wt}, & \lambda > w; \\
e^{-w\left(1-\sqrt{1-\frac{\lambda^2}{w^2}}\right)t}, & \lambda < w.
\end{cases}
\tag{117}
$$

As a result, the stochastic map, at long times, erases the off-diagonal elements of the density matrix,

$$
\hat{\rho}(t) \xrightarrow{t\to\infty}
\begin{bmatrix}
\langle \uparrow|\hat{\rho}|\uparrow\rangle & 0 \\
0 & \langle \downarrow|\hat{\rho}|\downarrow\rangle
\end{bmatrix}_B.
\tag{118}
$$

**Assignment 21 (Solution)** *Calculate the coherence function for random telegraph noise.*

### 3.4.4 Pure dephasing dynamics

The above example can be easily generalized. Consider a stochastic Hamiltonian of a form

$$
\hat{H}[F](t) = \lambda F(t)\hat{V} = \lambda F(t) \sum_n v_n |n\rangle\langle n|,
\tag{119}
$$

where $B = \{|n\rangle\}_n$ is the basis of eigenkets of $\hat{V}$, $\hat{V}|n\rangle = v_n|n\rangle$ and $F(t)$ is some stochastic process. Since $\hat{H}[F](t)$ commutes at different times, again, the time-ordering is unnecessary,

$$
\overline{\mathcal{U}}(t,0) = \overline{e^{-i\lambda \int_0^t F(s)ds\,\mathcal{V}}},
\tag{120}
$$

where the time-independent super-generator reads,

$$
\mathcal{V} = [\hat{V}, \bullet].
\tag{121}
$$

Analogously to our qubit example, the matrix of $\mathcal{V}$ in the basis made out kets and bras of $B$, $\mathcal{B} = \{|n\rangle\langle m|\}_{n,m}$, is diagonal; or, equivalently, $|n\rangle\langle m|$ are eigenoperators of super-operator $\mathcal{V}$

$$\mathcal{V}|n\rangle\langle m| = \hat{V}|n\rangle\langle m| - |n\rangle\langle m|\hat{V} = (v_n - v_m)|n\rangle\langle m|. \tag{122}$$

Note that only the coherences $|n\rangle\langle m|$ ($n \neq m$) correspond to non-zero eigenvalues.

The resultant action of the stochastic map onto the initial state is summarized as follows

$$\langle n|\overline{\mathcal{U}}(t,0)\hat{\rho}|m\rangle = \begin{cases} \langle n|\hat{\rho}|n\rangle, & m = n; \\ \langle n|\hat{\rho}|m\rangle W_{nm}(t), & m \neq n, \end{cases} \tag{123}$$

where

$$W_{nm}(t) = \overline{e^{-i\lambda(v_n - v_m)\int_0^t F(s)ds}}. \tag{124}$$

This type of dynamics is often called the *pure dephasing* because the evolution consists of average phase factors,

$$\overline{e^{-ig\int_0^t F(s)ds}} = \overline{e^{-i\Phi}}, \tag{125}$$

where $\Phi = g\int_0^t F(s)ds$ is a stochastic phase. The intuition is that such an average, which can be written as a sum over samples of stochastic phase,

$$\overline{e^{-i\Phi}} = \lim_{N \to \infty} \frac{1}{N} \sum_{j=1}^{N} e^{-i\phi_j}, \tag{126}$$

tends to zero when the phases are large enough. This tendency is explained by the observation that, for large phases, and when the number of summands tend to infinity, for every phase factor pointing in a certain direction (recall that $\exp(-i\phi_j)$ can be viewed as a unit vector in the complex plane) there will be another one that points in a roughly opposite direction canceling with the first. Phenomenon like this is called the *dephasing*.

## 3.5 Dynamical mixing with stochastic maps

One of the key properties of unitary dynamics is that it preserves the purity of the state [see Eq. (68)]: if we start with a pure state $\hat{\rho} = |\Psi\rangle\langle\Psi|$, it will always remain pure, for as long as the dynamics of the system are only unitary. So, where the mixed states come from? Back in Sec. 2.5, where we have introduced the concept, we had to resort to the example of an uncertain state preparation procedure, but we could not specify any *dynamical process* that would lead to "mixing" of the density matrix because, at that point, we had only unitary maps at our disposal.

The example we have analyzed in Sec. 3.4 shows that the stochastic dynamics could model such a dynamical "mixing"; indeed, the qubit stochastic map generated by Hamiltonian (105), with its ability to erode the off-diagonal elements of the initial density matrix, can transform over time an initially pure state into a mixture, e.g., if we set

$$\hat{\rho} = \frac{1}{2}\begin{bmatrix} 1 & 1 \\ 1 & 1 \end{bmatrix}_B = \left(\frac{|\uparrow\rangle + |\downarrow\rangle}{\sqrt{2}}\right)\left(\frac{\langle\uparrow| + \langle\downarrow|}{\sqrt{2}}\right), \tag{127}$$

then, in a long time limit, it evolves into a mixture,

$$\hat{\rho}(t) \xrightarrow{t \to \infty} \frac{1}{2}\begin{bmatrix} 1 & 0 \\ 0 & 1 \end{bmatrix}_B = \frac{1}{2}|\uparrow\rangle\langle\uparrow| + \frac{1}{2}|\downarrow\rangle\langle\downarrow| = \frac{1}{2}\hat{1}. \tag{128}$$

In fact, it is a general property of the stochastic maps that they do *not* preserve the purity of the state; more than that, the stochastic dynamics cannot purify the state, they can only make it more mixed. To show this, first, we have to somehow quantify the purity (or mixedness) of states. A typical measure of purity is the *entropy* of the state defined as

$$S(\hat{\rho}) = -\operatorname{tr}[\hat{\rho}\ln(\hat{\rho})] = -\sum_n \rho_n \ln(\rho_n), \tag{129}$$

where $\rho_n$ are the eigenvalues of $\hat{\rho}$. For pure states, the entropy is zero—the minimal value possible,

$$S(|\Psi\rangle\langle\Psi|) = -\ln(1) = 0. \tag{130}$$

For finite dimensional systems, the entropy reaches its maximal value for the completely mixed state $\hat{\rho} = (1/d)\hat{1}$,

$$S\left(\frac{1}{d}\hat{1}\right) = -\frac{1}{d}\ln\left(\frac{1}{d}\right)\sum_{n=1}^{d} = \ln(d), \tag{131}$$

where $d$ is the dimension of the ket-space (e.g., $d = 2$ for qubits). All other states sit somewhere in between those extremes, $0 \leqslant S(\hat{\rho}) \leqslant \ln(d)$. With entropy as a measure, we can now grade the purity of a given state: the larger the entropy, the less pure the state (or, the more mixed the state, if one prefers).

As a sanity check, we will show now that the unitary dynamics preserves the purity of states. Since the unitary maps does not change the eigenvalues of the density matrix [see Eq. (68)], they also do not change the entropy,

$$S(\mathcal{U}(t,0)\hat{\rho}) = -\sum_n \rho_n \ln(\rho_n) = S(\hat{\rho}), \tag{132}$$

therefore, the purity of $\hat{\rho}$ remains unchanged.

An important property of the entropy is that it is a *concave* function,

$$S\left(\sum_i c_i \hat{A}_i\right) \geqslant \sum_i c_i S(\hat{A}_i),$$

for any linear combination where $c_i \geqslant 0$, $\sum_i c_i = 1$ and $\hat{A}_i$ are hermitian. Since an integral (even a functional one) is ultimately a sum, and the probability distribution functionals $P_F[f]$ are positive and normalized, we have

$$S\left(\overline{\mathcal{U}}(t,0)\hat{\rho}\right) = S\left(\int P_F[f]\mathcal{U}[f](t,0)\hat{\rho}[Df]\right)$$
$$\geqslant \int P_F[f]S(\mathcal{U}[f](t,0)\hat{\rho})[Df] = \left(\int P_F[f][Df]\right)S(\hat{\rho}) = S(\hat{\rho}). \tag{133}$$

That is, the entropy of the state *at the conclusion* of the stochastic dynamics never decreases below its initial level; in other words, stochastic maps cannot make the initial state more pure, only more mixed (or leave the purity as it is, like, e.g., for completely mixed state $\hat{\rho} \propto \hat{1}$).

## 3.6 Computability of stochastic maps

### 3.6.1 No dynamical equation for free

There are two main difficulties that prevent one from easily computing a stochastic map. First, we have the standard problem of time-ordered super-operators; second, the time-ordered exp

also has to be averaged. Previously, we have circumvented the issues with time-ordering by integrating the dynamical equation; let us check if we can defeat the average in a similar manner. To see if there is a valid dynamical equation for the stochastic map we calculate the derivative and check if the map reconstitutes itself on the RHS,

$$
\begin{aligned}
\frac{d}{dt}\overline{\mathcal{U}}_I(t,0) = \frac{d}{dt}\overline{T_{\mathcal{V}_I}e^{-i\lambda\int_0^t F(s)\mathcal{V}_I(s)ds}} &= \overline{\frac{d}{dt}T_{\mathcal{V}_I}e^{-i\lambda\int_0^t F(s)\mathcal{V}_I(s)ds}} \\
&= -i\lambda\mathcal{V}_I(t)\overline{F(t)T_{\mathcal{V}_I}e^{-i\lambda\int_0^t F(s)\mathcal{V}_I(s)ds}}.
\end{aligned}
\tag{134}
$$

Unfortunately, we did not get a dynamical equation because $F(t)$ that has "spilled" out of the exp was caught by the average, and so, the stochastic map could not be reassembled.

To clarify what exactly is the nature of our problem, let us focus for the time being on the pure dephasing map (see, Sec 3.4.4). Since with pure dephasing we do not have to bother with time-ordering and non-commuting super-operators we can now isolate the issues with the stochastic average. The only time-dependent part of the map are the coherence functions of a form $W(t) = \overline{\exp[i\lambda\int_0^t F(s)ds]}$ that multiply the off-diagonal matrix elements of the state [see Eq. (123)], therefore, they are the things that we should differentiate,

$$
\begin{aligned}
\frac{d}{dt}W(t) &= i\lambda\overline{F(t)e^{-i\lambda\int_0^t F(s)ds}} \\
&= i\lambda\overline{F(t)} + \sum_{k=1}^{\infty}\frac{(i\lambda)^{k+1}}{k!}\int_0^t ds_1\cdots ds_k\overline{F(t)F(s_1)\cdots F(s_k)}.
\end{aligned}
\tag{135}
$$

Predictably, we are encountering the same kind of problem as with the general stochastic map: $W$ does not reconstituting itself on the RHS because an additional $F(t)$ is caught by the average. This could be fixed if the moments followed some factorization pattern so that we could extract a common factor and reconstitute from what is left the series expansion of the averaged exp. What we are talking here about is something similar to the RTN case, where the moments factorized in a certain way [see Eqs. (93) and (94)] and, as a result, we did obtain a kind of pseudo-dynamical equation (A.5). There, the problem was that the obtained differential equation was non-local in time, and thus, much more difficult to compute; time-local dynamical equations is what we really need.

### 3.6.2 Cumulant expansion

The "factorization patterns" in moments can be formally analyzed using the concept of *cumulant expansion*. Consider the following implicit definition,

$$
\overline{e^{i\lambda\int_0^t F(s)ds}} \equiv \exp\left[\sum_{k=1}^{\infty}\frac{(i\lambda)^k}{k!}\int_0^t C_F^{(k)}(s_1,\ldots,s_k)ds_1\cdots ds_k\right],
\tag{136}
$$

where the argument of the exp on RHS is called the *cumulant series*,

$$
\chi_F(t) = \sum_{k=1}^{\infty}\frac{(i\lambda)^k}{k!}\int_0^t C_F^{(k)}(s_1,\ldots,s_k)ds_1\cdots ds_k,
\tag{137}
$$

and its constituents, $C_F^{(k)}$, are the *cumulants* of process $F(t)$.

The explicit form of cumulants is found be expanding the LHS and RHS of Eq. (136) into power series in $\lambda$ and comparing the terms of corresponding orders. For example, the first

three cumulants found in this way are as follows,

$$C_F^{(1)}(s) = \overline{F(s)};\tag{138a}$$

$$C_F^{(2)}(s_1, s_2) = \overline{F(s_1)F(s_2)} - \overline{F(s_1)}\,\overline{F(s_2)} = C_F(s_1, s_2);\tag{138b}$$

$$C_F^{(3)}(s_1, s_2, s_3) = \overline{F(s_1)F(s_2)F(s_3)} + \overline{F(s_1)}\,\overline{F(s_2)}\,\overline{F(s_3)}$$
$$- \overline{F(s_1)}\,\overline{F(s_2)F(s_3)} - \overline{F(s_2)}\,\overline{F(s_1)F(s_3)} - \overline{F(s_3)}\,\overline{F(s_1)F(s_2)}.\tag{138c}$$

Note that the second cumulant $C_F^{(2)}$ equals the auto-correlation function, see Eq. (87). In general, a cumulant of order $k$, $C_F^{(k)}$, is given by a combination of products of moments of orders no greater than $k$, arranged in such a way that the total order of each product adds up to $k$.

Because of their relation to moments, cumulants can be considered as *statistically irreducible*; let us explain what we mean by that with an example. Consider a process composed of two independent contributions, $F(t) = A(t) + B(t)$. The statistical independence between $A(t)$ and $B(t)$ means that their moments factorize,

$$\overline{A(s_1)\cdots A(s_a)B(u_1)\cdots B(u_b)} = \overline{A(s_1)\cdots A(s_a)}\,\overline{B(u_1)\cdots B(u_b)}.\tag{139}$$

When we calculate the cumulant series for this example, we find that

$$\overline{e^{i\lambda\int_0^t [A(s)+B(s)]ds}} = \overline{e^{i\lambda\int_0^t A(s)ds}e^{i\lambda\int_0^t B(s)ds}}$$

$$= \sum_{n,m}\frac{(i\lambda)^{n+m}}{n!m!}\int_0^t ds_1\cdots ds_n du_1\cdots du_m \overline{A(s_1)\cdots A(s_n)}\,\overline{B(u_1)\cdots B(u_m)}$$

$$= \overline{e^{i\lambda\int_0^t A(s)ds}}\,\overline{e^{-i\lambda\int_0^t B(u)du}}$$

$$= e^{\sum_{k=1}^\infty \frac{(i\lambda)^k}{k!}\int_0^t C_A^{(k)}(s_1,\ldots,s_k)ds_1\cdots ds_k}e^{\sum_{k=1}^\infty \frac{(i\lambda)^k}{k!}\int_0^t C_B^{(k)}(s_1,\ldots,s_k)ds_1\cdots ds_k}$$

$$= e^{\chi_A(t)}e^{\chi_B(t)},\tag{140}$$

that is, there are no mixed cumulants $C_{A+B}^{(k)}$ where the moments of $A$ and the moments of $B$ would contribute together to one cumulant series. How does this happen? Upon closer inspection, one notices that $k$th cumulant can be viewed as a $k$th moment minus all possible factorizations of this moment into smaller moments; so, if we try to compute, for example, the 2nd-order cumulant for $F(t) = A(t) + B(t)$ we get,

$$C_{A+B}^{(2)}(s_1, s_2) = \overline{(A(s_1)+B(s_1))(A(s_2)+B(s_2))} - \left(\overline{A(s_1)+B(s_1)}\right)\left(\overline{A(s_2)+B(s_2)}\right)$$

$$= \overline{A(s_1)A(s_2)} - \overline{A(s_1)}\,\overline{A(s_2)} + \overline{B(s_1)B(s_2)} - \overline{B(s_1)}\,\overline{B(s_2)}$$

$$+ \overline{A(s_1)B(s_2)} - \overline{A(s_1)}\,\overline{B(s_2)} + \overline{B(s_1)A(s_2)} - \overline{B(s_1)}\,\overline{A(s_2)}$$

$$= C_A^{(2)}(s_1, s_2) + C_B^{(2)}(s_1, s_2),\tag{141}$$

because $\overline{A(t)B(s)} = \overline{A(t)}\,\overline{B(s)}$ for independent $A$, $B$, and the mixed moments cancel with the factorized corrections. Essentially, a cumulant is a moment that is "reduced" by all the redundant information that can be found in other moments. Therefore, the statistical information remaining in the cumulant cannot be found anywhere else—in other words, cumulants are statistically irreducible.

Having introduced the cumulants, we can now use them to obtain the dynamical equation

for $W$. Formally, we can write the derivative as

$$
\begin{aligned}
\frac{d}{dt}W(t) &= \frac{d}{dt}e^{\chi_F(t)} = \dot{\chi}_F(t)W(t) \\
&= \left( i\lambda\overline{F(t)} + \sum_{k=2}^{\infty}\frac{(i\lambda)^k}{(k-1)!}\int_0^t C_F^{(k)}(t,s_2,\ldots,s_k)ds_2\cdots ds_k \right)W(t) \\
&= \left( i\lambda\overline{F(t)} + \sum_{k=2}^{\infty}(i\lambda)^k\int_0^t ds_2\cdots\int_0^{s_{k-1}}ds_k\, C_F^{(k)}(t,s_2,\ldots,s_k) \right)W(t).
\end{aligned}
\tag{142}
$$

By the very definition of cumulant series, the averaged exponent must reconstitute itself on the RHS—this is the whole point of cumulant expansion. However, unless we can actually compute the cumulant series as a whole, this is not a dynamical equation we can readily work with. Clearly, cumulant expansion does not change anything unless we start employing some approximation schemes. While it is true that cutting the moment series is a poor approximation, an analogous truncation for the cumulant series is another story. For example, when we decide to cut the series at order $k_0 = 2$, then

$$
\frac{d}{dt}W(t) \approx \left( i\lambda\overline{F(t)} - \lambda^2\int_0^t C_F^{(2)}(t,s)ds \right)W(t),
\tag{143}
$$

and, of course, the dynamical equation can be solved with ease,

$$
\overline{e^{i\lambda\int_0^t F(s)ds}} \approx e^{i\lambda\int_0^t \overline{F(s)}ds}e^{-\frac{1}{2}\lambda^2\int_0^t C_F^{(2)}(s_1,s_2)ds_1 ds_2}.
\tag{144}
$$

(Actually, we could, instead, simply calculate the cumulant series—it now has only two terms—and take the exp of it, but the point is made.)

Cutting the cumulant series indeed leads to "factorization patterns" in moments. If we take, for example, the cut at $k_0 = 2$, and we assume for simplicity that $\overline{F(t)} = 0$, then the 4th moment reads

$$
\overline{F(s_1)\cdots F(s_4)} = \overline{F(s_1)F(s_2)}\;\overline{F(s_3)F(s_4)} + \overline{F(s_1)F(s_3)}\;\overline{F(s_2)F(s_4)} + \overline{F(s_1)F(s_4)}\;\overline{F(s_2)F(s_3)}.
\tag{145}
$$

All higher moments look similar, they are all made out of products of the auto-correlation function (which equals the second moment when the average is zero); it makes sense, because the auto-correlation function (or the second cumulant) is the only quantity that is left when we cut the cumulant series on the second order. In general, when one cuts the series at some finite order, then the higher-order moments will factorize in such a way that in Eq. (135) it becomes possible to extract the common factor and reconstitute the averaged exp from the remainder.

There are some constraints to cutting the cumulant series that should be kept in mind. The theorem states that a valid cumulant series—i.e. the series that represents an actual stochastic process—has either infinitely many non-zero cumulants or all cumulants beyond the second are zero. Is it, then, even permissible to cut at some order higher than two? The answer is yes. Even though, formally, a process where only $2 < k_0 < \infty$ cumulants are non-zero does not exist, for as long as the truncated series is a good approximation of the complete series, the coherence function $W$ will not exhibit any aberrant behavior, like, e.g., diverging to values larger than 1. However, when the approximation ceases to be accurate and the contribution from the cut cumulants becomes significant, then a non-physical divergences that would be compensated for by all the neglected cumulants might start to creep in.

The processes with, at most, two non-zero cumulants are called *Gaussian*, and are quite popular in applications. (The RTN is, of course, non-Gaussian, and that is why we could not

get for it a proper dynamical equation.) One reason why Gaussian processes are so widely used is because they are simple: all one needs to define them is an auto-correlation function of almost any shape. Another reason is that Gaussian processes are quite accurate models in many physical and non-physical settings due to phenomenon explained by the *central limit theorem*. A basic version of the theorem establishes that stochastic process $G(t)$ composed of a combination of $N$ statistically independent processes

$$G(t) = \sum_{i=1}^{N} \frac{1}{\sqrt{N}} F_i(t), \tag{146}$$

becomes Gaussian in the limit $N \to \infty$, even when the constituents $F_i(t)$ are non-Gaussian. The least complicated way to arrive at this result is to, in addition to the independence, also assume that $F_i(t)$ have zero average and are all identical, in the sense that their individual cumulants of the respective orders are equal,

$$\forall_{k, i \neq j} C_{F_i}^{(k)} = C_{F_j}^{(k)} \equiv C_{F_0}^{(k)}, \quad C_{F_0}^{(1)} = 0. \tag{147}$$

Then, when we calculate a cumulant of $G(t)$ we get

$$C_G^{(k)} = \frac{1}{(\sqrt{N})^k} C_{F_1 + \dots + F_N}^{(k)} = \frac{1}{(\sqrt{N})^k} \sum_{i=1}^{N} C_{F_i}^{(k)} = \frac{1}{N^{\frac{k}{2}-1}} C_{F_0}^{(k)}, \tag{148}$$

where, due to statistical independence, the cumulants of a sum of $F_i(t)$ reduced to the sum of individual cumulants. Now, we pass to the limit and, due to factor $1/N^{\frac{k}{2}-1}$, only $k = 2$ cumulant survives,

$$\lim_{N \to \infty} C_G^{(k)} = \begin{cases} C_{F_0}^{(2)}, & \text{for} \quad k = 2; \\ 0, & \text{for} \quad k \neq 2, \end{cases} \tag{149}$$

In other words, $G(t)$ is a Gaussian process with zero average. The general point is that the central limit theorem-style of argument can be applied more broadly: when the process is a combination of large number of approximately independent and *weak* constituents (the $1/\sqrt{N}$ scaling made them weak in the above example), then the second-order cumulant of that process tends to dominate over the higher-order cumulants.

Now we can come back to general stochastic maps. As it turns out, the cumulant expansion of stochastic processes can be incorporated into the description of a map even when the coupling super-operators need time-ordering,

$$\overline{\mathcal{U}}_I(t, 0) = \overline{T_{\mathcal{V}_I} e^{-i\lambda \int_0^t F(s) \mathcal{V}_I(s) ds}}$$
$$= T_{\mathcal{V}_I} \exp\left[ \sum_{k=1}^{\infty} (-i\lambda)^k \int_0^t ds_1 \cdots \int_0^{s_{k-1}} ds_k \, C_F^{(k)}(s_1, \dots, s_k) \mathcal{V}_I(s_1) \cdots \mathcal{V}_I(s_k) \right]. \tag{150}$$

**Assignment 22 (Solution)** *Verify* (150).

### 3.6.3 Super-cumulant expansion

Assume the process $F(t)$ in stochastic Hamiltonian $\hat{H}[F](t)$ is Gaussian with zero average,

$$\overline{\mathcal{U}}_I(t, 0) = \overline{T_{\mathcal{V}_I} e^{-i\lambda \int_0^t F(s) \mathcal{V}_I(s) ds}} = T_{\mathcal{V}_I} e^{-\lambda^2 \int_0^t ds_1 \int_0^{s_1} ds_2 \, C_F^{(2)}(s_1, s_2) \mathcal{V}_I(s_1) \mathcal{V}_I(s_2)}. \tag{151}$$

Previously, we have shown that, for a Gaussian processes (or any other cut of the cumulant series), one gets the dynamical equation for $W$; the question is, will this also solve the problem for stochastic map? Let us check,

$$
\begin{aligned}
\frac{d}{dt}\overline{\mathcal{U}}_I(t,0) &= \frac{d}{dt}T_{\mathcal{V}_I}e^{-\lambda^2\int_0^t ds_1\int_0^{s_1}ds_2\,C_F^{(2)}(s_1,s_2)\mathcal{V}_I(s_1)\mathcal{V}_I(s_2)} \\
&= T_{\mathcal{V}_I}\left\{\frac{d}{dt}e^{-\lambda^2\int_0^t ds_1\int_0^{s_1}ds_2\,C_F^{(2)}(s_1,s_2)\mathcal{V}_I(s_1)\mathcal{V}_I(s_2)}\right\} \\
&= T_{\mathcal{V}_I}\left\{-\lambda^2\int_0^t ds\,C_F^{(2)}(t,s)\mathcal{V}_I(t)\mathcal{V}_I(s)\,e^{-\lambda^2\int_0^t ds_1\int_0^{s_1}ds_2\,C_F^{(2)}(s_1,s_2)\mathcal{V}_I(s_1)\mathcal{V}_I(s_2)}\right\} \\
&= -\lambda^2\int_0^t ds\,C_F^{(2)}(t,s)\mathcal{V}_I(t)\,T_{\mathcal{V}_I}\left\{\mathcal{V}_I(s)e^{-\lambda^2\int_0^t ds_1\int_0^{s_1}ds_2\,C_F^{(2)}(s_1,s_2)\mathcal{V}_I(s_1)\mathcal{V}_I(s_2)}\right\}. \quad (152)
\end{aligned}
$$

We have extracted $\mathcal{V}_I(t)$ form under the time-ordering operation because $t$ is the largest time (it is the uppermost limit of all integrals). However, we could not do the same for $\mathcal{V}_I(s)$, and thus, the map did not reconstitute itself on RHS. The answer is then "no": assuming that $F(t)$ is Gaussian is not enough to give us the dynamical equation. Essentially, we could not get the equation because the time-ordering we are using is too "fine", we would not have any issues if $T$ treated the whole argument of the exp as an atomic expression; let us see what can be done about it.

Consider the following implicit definition of *super-cumulants* $\{\mathcal{C}^{(k)}\}_{k=1}^\infty$,

$$
\begin{aligned}
\overline{\mathcal{U}}_I(t,0) &= \overline{T_{\mathcal{V}_I}e^{-i\lambda\int_0^t F(s)\mathcal{V}_I(s)ds}} \\
&= T_{\mathcal{V}_I}e^{\sum_{k=1}^\infty(-i\lambda)^k\int_0^t ds_1\cdots\int_0^{s_{k-1}}ds_k\,C_F^{(k)}(s_1,\ldots,s_k)\mathcal{V}_I(s_1)\cdots\mathcal{V}_I(s_k)} \\
&\equiv T_{\mathcal{C}^{(k)}}e^{\sum_{k=1}^\infty(-i\lambda)^k\int_0^t \mathcal{C}^{(k)}(s)ds} \\
&= \bullet + \sum_{n=1}^\infty\int_0^t ds_1\cdots\int_0^{s_{n-1}}ds_n\left(\sum_{k=1}^\infty(-i\lambda)^k\mathcal{C}^{(k)}(s_1)\right)\cdots\left(\sum_{k=1}^\infty(-i\lambda)^k\mathcal{C}^{(k)}(s_n)\right), \quad (153)
\end{aligned}
$$

with the time-ordering operation $T_{\mathcal{C}^{(k)}}$ that does the chronological sorting for all $\mathcal{C}^{(k)}(s)$, e.g., if $t > s$, then $T_{\mathcal{C}^{(k)}}\left\{\mathcal{C}^{(k_1)}(s)\mathcal{C}^{(k_2)}(t)\right\} = \mathcal{C}^{(k_2)}(t)\mathcal{C}^{(k_1)}(s)$ for any $k_1, k_2$;

The explicit form of super-cumulant is found by expanding both sides of Eq. (153) into power series, applying the corresponding time-orderings, and comparing the terms of equal orders in $\lambda$. Below we list a first few super-cumulants (for simplicity, we will assume that $F(t)$ is Gaussian and has zero average),

$$\mathcal{C}^{(1)}(s) = 0; \quad (154a)$$

$$\mathcal{C}^{(2)}(s) = \mathcal{V}_I(s)\int_0^s C_F^{(2)}(s,s_1)\mathcal{V}_I(s_1)ds_1; \quad (154b)$$

$$\mathcal{C}^{(3)}(s) = 0; \quad (154c)$$

$$
\begin{aligned}
\mathcal{C}^{(4)}(s) = \mathcal{V}_I(s)\int_0^s ds_1\int_0^{s_1}ds_2\int_0^{s_2}ds_3 \\
\times\Big[&C_F^{(2)}(s,s_2)C_F^{(2)}(s_1,s_3)\big(\mathcal{V}_I(s_1)\mathcal{V}_I(s_2)-\mathcal{V}_I(s_2)\mathcal{V}_I(s_1)\big)\mathcal{V}_I(s_3) \\
+&C_F^{(2)}(s_1,s_4)C_F^{(2)}(s_2,s_3)\big(\mathcal{V}_I(s_1)\mathcal{V}_I(s_2)\mathcal{V}_I(s_3)-\mathcal{V}_I(s_3)\mathcal{V}_I(s_1)\mathcal{V}_I(s_2)\big)\Big]. \quad (154d)
\end{aligned}
$$

Like the cumulants, the super-cumulant expansion is a tool we can utilize to construct computable maps that can serve as an approximation to the given stochastic map $\overline{\mathcal{U}}_I$. Since the

time-ordering $T_{\mathcal{C}^{(k)}}$ in Eq. (153) treats all $\mathcal{C}^{(k)}$ on an equal footing, we can apply to the ordered exp the disentanglement theorem, e.g.,

$$
\begin{aligned}
\overline{\mathcal{U}}_I(t,0) &= T_{\mathcal{C}^{(k)}} e^{\sum_{k=1}^{\infty}(-i\lambda)^k \int_0^t \mathcal{C}^{(k)}(s)ds} \\
&= T_{\mathcal{C}^{(k)}} e^{\sum_{k=1}^{n}(-i\lambda)^k \int_0^t \mathcal{C}^{(k)}(s)ds + \sum_{k'=n+1}^{\infty}(-i\lambda)^{k'} \int_0^t \mathcal{C}^{(k')}(s)ds} \\
&= \overline{\mathcal{U}}^{(n)}(t,0) T_{\mathcal{C}^{(k)}_{I(n)}} e^{\sum_{k=n+1}^{\infty}(-i\lambda)^k \int_0^t \mathcal{C}^{(k)}_{I(n)}(s)ds} \\
&= \overline{\mathcal{U}}^{(n)}(t,0) + (-i\lambda)^{n+1} \int_0^t \overline{\mathcal{U}}^{(n)}(t,s)\mathcal{C}^{(n+1)}(s)\overline{\mathcal{U}}^{(n)}(s,0)ds + \dots,
\end{aligned}
\tag{155}
$$

where $\mathcal{C}^{(k)}_{I(n)}(s) = \overline{\mathcal{U}}^{(n)}(0,s)\mathcal{C}^{(k)}(s)\overline{\mathcal{U}}^{(n)}(s,0)$, and

$$
\overline{\mathcal{U}}^{(n)}(t,s) = T_{\mathcal{C}^{(k)}} e^{\sum_{k=1}^{n}(-i\lambda)^k \int_s^t \mathcal{C}^{(k)}(u)du}.
\tag{156}
$$

The map generated by the disentangled super-cumulants $\overline{\mathcal{U}}^{(n)}$ satisfies a dynamical equation,

$$
\frac{d}{dt}\overline{\mathcal{U}}^{(n)}(t,s) = \left(\sum_{k=1}^{n}(-i\lambda)^k \mathcal{C}^{(k)}(t)\right)\overline{\mathcal{U}}^{(n)}(t,s); \quad \overline{\mathcal{U}}^{(n)}(s,s) = \bullet,
\tag{157}
$$

and it also satisfies the composition rule,

$$
\overline{\mathcal{U}}^{(n)}(t,u)\overline{\mathcal{U}}^{(n)}(u,s) = \overline{\mathcal{U}}^{(n)}(t,s).
\tag{158}
$$

All this, of course, makes $\overline{\mathcal{U}}^{(n)}$ computable. Therefore, if one could argue that the contribution from higher-order super-cumulants is small and it could be neglected, then $\overline{\mathcal{U}}^{(n)}$ would give us a computable approximation to $\overline{\mathcal{U}}_I$.

Let us pause for a minute to appreciate the usefulness of the specialized time-orderings $T_{\mathcal{A}}$. Recall, that the stochastic map for Gaussian, zero-average $F(t)$ is given by

$$
\overline{\mathcal{U}}_I(t,0) = T_{\mathcal{V}_I} e^{-\lambda^2 \int_0^t ds_1 \int_0^{s_1} ds_2 \, C_F^{(2)}(s_1,s_2)\mathcal{V}_I(s_1)\mathcal{V}_I(s_2)} = T_{\mathcal{V}_I} e^{-\lambda^2 \int_0^t \mathcal{C}^{(2)}(s_1)ds_1},
\tag{159}
$$

where, in the argument of exp, we have simply recognized the formula for the second-order super-cumulant, Eq. (154b). However, even though the exponent in the map above is formally identical to the exponent in

$$
\overline{\mathcal{U}}^{(2)}(t,0) = T_{\mathcal{C}^{(k)}} e^{-\lambda^2 \int_0^t \mathcal{C}^{(2)}(s)ds},
\tag{160}
$$

the two maps are, obviously, not the same,

$$
T_{\mathcal{C}^{(k)}} e^{-\lambda^2 \int_0^t \mathcal{C}^{(2)}(s)ds} \neq T_{\mathcal{V}_I} e^{-\lambda^2 \int_0^t \mathcal{C}^{(2)}(s)ds}.
\tag{161}
$$

In fact, $\overline{\mathcal{U}}^{(2)}$ is only a second-order approximation to $\overline{\mathcal{U}}_I$. Of course, the difference between them is the "fineness" of the time-ordering—$T_{\mathcal{C}^{(k)}}$ is more coarse-grained than $T_{\mathcal{V}_I}$, for the former the whole exponent is an atomic expression, while the latter delves into the innards of $\mathcal{C}^{(2)}$.

Even though the specialized time-ordering $T_{\mathcal{A}}$ is, essentially, only a form of notation convention, without it, the very concept of super-cumulant expansion would not be possible. Ask yourself, how would you even approach this construction if we had only "universal" $T$ in Eqs. (159) and (160)? Actually, you do not have to imagine, you can see for yourself how, back in a day, people tried [4].

**Assignment 23 (Solution)** *Find the 4th super-cumulant $\mathcal{C}^{(4)}$ for non-Gaussian $F(t)$; assume that $\overline{F(t)} = 0$.*

### 3.6.4 Sample average

The stochastic average can be approximated by an arithmetic average over finite number of samples,

$$\overline{\mathcal{U}}(t,0) = \int P_F[f]\mathcal{U}[f](t,0)[Df] = \lim_{N\to\infty} \frac{1}{N}\sum_{j=1}^{N}\mathcal{U}[f_j](t,0) \approx \frac{1}{N_0}\sum_{j=1}^{N_0}\mathcal{U}[f_j](t,0), \quad (162)$$

where $\{f_j(t)\}_{j=1}^{N_0}$ is a set of $N_0$ sample trajectories of process $F(t)$. Therefore, the problem of computability of the stochastic map is reduced to trivial (but laborious, when the number of samples is large) computation of the set of trajectory-wise unitary maps, $\{\mathcal{U}[f_j](t,0)\}_{j=1}^{N_0}$. The issue is the access to samples: the sample average can be computed only when one can get their hands on the sufficiently large set of sampled trajectories.

When the process $F(t)$ is known, one can make an attempt at generating the trajectories numerically. However, as we have seen in Assignment 19 of Sec. 3.1.3, an algorithm for efficient numerical trajectory generation is defined only for Markovian processes. If $F(t)$ is non-Markovian, there are no general purpose methods that are guaranteed to work for all cases.

When the numerical generation is unfeasible, there still might be an alternative way to assemble the set of samples. In certain physical situations that facilitate the stochastic map description of the dynamics, it is possible to directly measure the sample trajectories; this is not guaranteed for all stochastic dynamics, however. We will elaborate on this topic more later down the line.

**Assignment 24 (Solution)** *Consider the stochastic Hamiltonian,*

$$\hat{H}[F](t) = \frac{\Omega}{2}\hat{\sigma}_z + \frac{\lambda}{2}F(t)\hat{\sigma}_x.$$

*Compute the stochastic map generated by this Hamiltonian using the sample average and the 2nd-order super-cumulant expansion methods for $F(t) = R_0(t)$—the random telegraph noise with zero average ($p = 0$ in $P_R^{(k)}$'s). In addition, do the same for*

$$F(t) = G(t) = \sum_{i=1}^{N}\frac{1}{\sqrt{N}}R_i(t),$$

*where $R_i(t)$ are independent, zero-average RTNs and $N \gg 1$, so that $G(t)$ approaches Gaussian process.*

## 4 Dynamics of open quantum systems

### 4.1 Open quantum systems

So far, we have only considered *closed* quantum systems, i.e. systems the dynamics of which are completely independent of all other quantum systems constituting the rest of the Universe. We are underlining here the independence from other *quantum* systems, because, previously, we have implicitly allowed for our system to be influenced by *external fields*, which could be interpreted as a coupling with classical system. When there is a need to factor in the contribution from other quantum degrees of freedom, then the rule is to take the *outer product* ⊗ of Hilbert spaces associated with each system. In practice, this means that the basis for the composite system (or the total system) is constructed by taking the outer product of basis

kets in the constituent systems, e.g., if $B_S = \{|n\rangle\}_n$ is a basis in the ket-space of our system of interest $S$, and $B_E = \{|j\rangle\}_j$ is a basis of the system $E$ encompassing all other systems that we want to take into account, then the set

$$B_{SE} = \{|n\rangle \otimes |j\rangle\}_{n,j}, \tag{163}$$

is a basis in the ket-space of the total system $SE$.

Given the basis $B_{SE}$, we can decompose any state-ket of a total system,

$$|\Psi_{SE}\rangle = \sum_{n,j} \psi_{nj}|n\rangle \otimes |j\rangle, \quad |\Phi_{SE}\rangle = \sum_{n,j} \phi_{nj}|n\rangle \otimes |j\rangle, \tag{164}$$

we can compute the inner product,

$$\langle\Psi_{SE}|\Phi_{SE}\rangle = \sum_{n,j}\sum_{n',j'} \psi^*_{nj}\phi_{n'j'}\big((\langle n| \otimes \langle j|)(|n'\rangle \otimes |j'\rangle)\big) = \sum_{n,j}\sum_{n',j'} \psi^*_{nj}\phi_{n'j'}\langle n|n'\rangle\langle j|j'\rangle$$

$$= \sum_{n,j}\sum_{n'j'} \psi^*_{nj}\phi_{n'j'}\delta_{n,n'}\delta_{j,j'} = \sum_{n,j} \psi^*_{nj}\phi_{nj}, \tag{165}$$

and we can calculate how operators in the total space,

$$\hat{A}_{SE} = \sum_{n,j}\sum_{n'j'} |n\rangle \otimes |j\rangle A_{nj,n'j'}\langle n'| \otimes \langle j'| = \sum_{n,j}\sum_{n',j'} A_{nj,n'j'}|n\rangle\langle n'| \otimes |j\rangle\langle j'|, \tag{166}$$

act on state-kets,

$$\hat{A}_{SE}|\Psi_{SE}\rangle = \sum_{n,j}\sum_{n',j'} A_{nj,n'j'}|n\rangle\langle n'| \otimes |j\rangle\langle j'| \sum_{n'',j''} \psi_{n''j''}|n''\rangle \otimes |j''\rangle$$

$$= \sum_{n,j}\sum_{n',j'} A_{nj,n'j'} \sum_{n'',j''} \psi_{n''j''}\big((\langle n'|n''\rangle|n\rangle) \otimes (\langle j'|j''\rangle|j\rangle)\big)$$

$$= \sum_{n,j}\left(\sum_{n',j'} A_{nj,n'j'}\psi_{n'j'}\right)|n\rangle \otimes |j\rangle. \tag{167}$$

Mixed states of the total system are described with density matrices in $SE$-operator space,

$$\hat{\rho}_{SE} = \sum_{n,j}\sum_{n'j'} \rho_{nj,n'j'}|n\rangle\langle n'| \otimes |j\rangle\langle j'|, \tag{168}$$

and the expectation value of obervables $\hat{A}_{SE}$ are computed according to standard rules,

$$\langle A_{SE}\rangle = \text{tr}\big(\hat{A}_{SE}\hat{\rho}_{SE}\big) = \sum_{n,j}\big((\langle n| \otimes \langle j|)\hat{A}_{SE}\hat{\rho}_{SE}(|n\rangle \otimes |j\rangle)\big). \tag{169}$$

An important case are obervables that depend only on one part of the composite system; such observables are represented by operators of the form,

$$( \, S\text{-only observable} \, ) = \hat{A}_S \otimes \hat{1}. \tag{170}$$

Since $A_S$ is independent of $E$, its expectation value is determined by only a fraction of information encoded in the total state of $SE$,

$$
\begin{aligned}
\langle A_S \rangle = \mathrm{tr}\left(\hat{A}_S \otimes \hat{1}\,\hat{\rho}_{SE}\right) &= \sum_{n,j}(\langle n|\otimes\langle j|)(\hat{A}_S \otimes \hat{1})\hat{\rho}_{SE}(|n\rangle\otimes|j\rangle) \\
&= \sum_{n,j}(\langle n|\otimes\langle j|)(\hat{A}_S \otimes \hat{1})\left(\sum_{n',j'}\sum_{n'',j''}\rho_{n'j',n''j''}|n'\rangle\langle n''|\otimes|j'\rangle\langle j''|\right)(|n\rangle\otimes|j\rangle) \\
&= \sum_{n}\langle n|\hat{A}_S\left(\sum_{j}\sum_{n',j'}\sum_{n'',j''}\rho_{n'j',n''j''}\langle j|j'\rangle\langle j''|j\rangle|n'\rangle\langle n''|\right)|n\rangle \\
&= \sum_{n}\langle n|\hat{A}_S\left(\sum_{n',n''}\left(\sum_{j}\rho_{n'j,n''j}\right)|n'\rangle\langle n''|\right)|n\rangle \\
&\equiv \sum_{n}\langle n|\hat{A}_S\,\mathrm{tr}_E\left(\hat{\rho}_{SE}\right)|n\rangle = \mathrm{tr}_S\left(\hat{A}_S\,\mathrm{tr}_E(\hat{\rho}_{SE})\right) = \mathrm{tr}_S\left(\hat{A}_S\hat{\rho}_S\right).
\end{aligned}
\tag{171}
$$

Here, the operator in $S$ subspace obtained as a partial trace over $E$ of $\hat{\rho}_{SE}$,

$$
\hat{\rho}_S = \mathrm{tr}_E\left(\hat{\rho}_{SE}\right) = \left(\sum_{j}\rho_{n'j,n''j}\right)|n'\rangle\langle n''|,
\tag{172}
$$

is positive with unit trace, and thus, it serves as a density matrix for the subsystem $S$. It is called the *reduced density matrix*, or simply, the state of $S$. In a special case when the state of $SE$ has a product form

$$
\hat{\rho}_{SE} = \hat{\rho}_S \otimes \hat{\rho}_E,
\tag{173}
$$

we say that $S$ and $E$ are uncorrelated, and the expectation values of $S$-only observables behave as if $E$ did not exist,

$$
\mathrm{tr}_S(\hat{A}_S\hat{\rho}_S) = \mathrm{tr}\left[(\hat{A}_S \otimes \hat{1})(\hat{\rho}_S \otimes \hat{\rho}_E)\right].
\tag{174}
$$

When one focuses exclusively on $S$-only observables, but $S$ is not independent of the other system $E$, then it is said that $S$ is an *open system*. In such a case, it is customary to refer to $E$ as the *environment* of $S$.

## 4.2 Dynamical maps

If we are interested in open systems, then we need to figure out how we should go about describing their dynamics. The most straightforward approach would be to simply take the Hamiltonian of the total $SE$ system,

$$
\hat{H}_{SE} = \hat{H}_S \otimes \hat{1} + \hat{1} \otimes \hat{H}_E + \hat{V}_{SE},
\tag{175}
$$

(any $\hat{H}_{SE}$ can be written in this form: $\hat{H}_{S/E}$ would generate evolution for $S$ and $E$ if they were independent, $\hat{V}_{SE}$ is the coupling and the reason why $S$ and $E$ are not independent), then to compute the unitary map for the whole system,

$$
\mathcal{U}_{SE}(t,0) = e^{-it[\hat{H}_{SE},\bullet]},
\tag{176}
$$

then use the map to evolve the initial state, $\hat{\rho}_{SE}(t) = \mathcal{U}_{SE}(t,0)\hat{\rho}_{SE}$, and, at the end, calculate the partial trace,

$$
\hat{\rho}_S(t) = \mathrm{tr}_E\left(\mathcal{U}_{SE}(t,0)\hat{\rho}_{SE}\right).
\tag{177}
$$

**Assignment 25 (Solution)** *Show that*

$$e^{-it[\hat{H}_S \otimes \hat{1} + \hat{1} \otimes \hat{H}_E, \bullet]} = e^{-it[\hat{H}_S, \bullet]} \otimes e^{-it[\hat{H}_E, \bullet]}.$$  (178)

**Assignment 26 (Solution)** *Find the state $\hat{\rho}_S(t)$ of a qubit S open to the environment E composed of a single qubit. The total Hamiltonian is given by*

$$\hat{H}_{SE} = \frac{\lambda}{2} \hat{\sigma}_z \otimes \hat{\sigma}_z,$$

*and the initial state is*

$$\hat{\rho}_{SE} = |X\rangle\langle X| \otimes |X\rangle\langle X|,$$

*where $|X\rangle = (|1\rangle + |-1\rangle)/\sqrt{2}$ and $\hat{\sigma}_z |\pm 1\rangle = \pm |\pm 1\rangle$.*

Of course, the brute force approach described above can only work if $S$ and $E$ are sufficiently simple systems; typically, this is not the case. In fact, making the distinction between the open system $S$ and its environment $E$ makes practical sense only when the environment is so complex that it is no longer feasible to compute the total map or even fully characterize $E$. Then, the question becomes: what is the minimal amount of information about the environment $E$ necessary to describe the dynamics of open system $S$ with a satisfactory level of accuracy. Hopefully, this amount is much less than the computed unitary map for the total system.

The most natural (and familiar) way of tackling this question starts with rephrasing the problem in the language of *dynamical maps*—transformations of the reduced density matrices of $S$ that sends the initial state to $\hat{\rho}_S(t)$ while taking into account all the relevant influence from $E$,

$$\hat{\rho}_S(0) = \mathrm{tr}_E(\hat{\rho}_{SE}) \xrightarrow{\text{dynamical map}} \hat{\rho}_S(t) = \mathrm{tr}_E(\mathcal{U}_{SE}(t,0)\hat{\rho}_{SE}),$$  (179)

In other words, we want to find the form of a super-operator $\mathcal{M}(t,0)$ on the space of density matrices of $S$ such that

$$\hat{\rho}_S(t) = \mathcal{M}(t,0)\hat{\rho}_S(0).$$  (180)

When we decide to go with such a formulation, we make an implicit assumption that $\mathcal{M}(t,0)$ would play an analogous role to unitary or stochastic maps: on the one hand, $\mathcal{M}$ should encapsulate the dynamical laws governing the evolution of states of $S$, and on the other hand, these laws are supposed to be abstracted from the states themselves. That is, we expect $\mathcal{M}$ to be independent of $\hat{\rho}_S(0)$ so that its action changes all initial states according to the same set of rules. Such an approach allows to categorize the open system by the dynamical laws rather than the particular initial state; this is how we tend to think about closed systems as well: the closed system is identified with Hamiltonian and the initial states are just different hats the system can wear while maintaining its identity.

The issue is that, for general initial state of the total system $\hat{\rho}_{SE}$, it is impossible to obtain $\mathcal{M}$ that is independent of $\hat{\rho}_S(0)$. To see this, let us decompose an arbitrary $\hat{\rho}_{SE}$ into an uncorrelated product and a remainder that quantifies the correlations between $S$ and $E$:

$$\hat{\rho}_{SE} = \hat{\rho}_S \otimes \hat{\rho}_E + (\hat{\rho}_{SE} - \hat{\rho}_S \otimes \hat{\rho}_E) \equiv \hat{\rho}_S \otimes \hat{\rho}_E + \hat{\rho}_{\text{corr}},$$  (181)

such that $\hat{\rho}_S, \hat{\rho}_E \geqslant 0$ and $\mathrm{tr}_{S/E}(\hat{\rho}_{S/E}) = 1$, so that $\hat{\rho}_S(0) = \mathrm{tr}_E(\hat{\rho}_{SE}) = \hat{\rho}_S$ (such decomposition is always possible). Then, according to its definition, the reduced state of $S$ is given by

$$\begin{aligned}
\hat{\rho}_S(t) &= \mathcal{M}(t,0)\hat{\rho}_S(0) = \mathrm{tr}_E(\mathcal{U}_{SE}(t,0)\hat{\rho}_{SE}) \\
&= \mathrm{tr}_E(\mathcal{U}_{SE}(t,0) \bullet \otimes \hat{\rho}_E)\hat{\rho}_S(0) + \mathrm{tr}_E(\mathcal{U}_{SE}(t,0)\hat{\rho}_{\text{corr}}).
\end{aligned}$$  (182)

The first term has the desired form of a dynamical map: the super-operator acting on $\hat{\rho}_S(0)$ is independent of the state of $S$. The second term is the problematic one; $\hat{\rho}_{\mathrm{corr}}$ cannot be independent of $\hat{\rho}_S(0)$ (e.g., $\hat{\rho}_{\mathrm{corr}}$ must be chosen in such a way that it plus $\hat{\rho}_S(0) \otimes \hat{\rho}_E$ is positive) and the action of the total unitary map, in general, changes $\hat{\rho}_{\mathrm{corr}}$ so that it no longer disappears under the partial trace. Therefore, the correlation part might remain as a $\hat{\rho}_S(0)$-dependent contribution to map $\mathcal{M}$. The only way to solve this problem in all cases is to demand that the initial total state is fully uncorrelated, $\hat{\rho}_{SE} = \hat{\rho}_S \otimes \hat{\rho}_E$, then

$$\hat{\rho}_S(t) = \mathrm{tr}_E\left(\mathcal{U}_{SE}(t,0)\hat{\rho}_S \otimes \hat{\rho}_E\right) \equiv \langle\mathcal{U}\rangle(t,0)\hat{\rho}_S\,, \tag{183}$$

and the $S$-only super-operator

$$\langle\mathcal{U}\rangle(t,0) = \mathrm{tr}_E\left(\mathcal{U}_{SE}(t,0) \bullet \otimes \hat{\rho}_E\right)\,, \tag{184}$$

is independent of the initial state of $S$ and can be considered as a proper dynamical map that describes the dynamics of $S$ induced by coupling to $E$.

From this point on we will always assume that the initial state of the total system has the product form, as this is the only case that is compatible with the dynamical map formulation of open system dynamics. This assumption might seem restrictive, but, in actuality, it accurately describes typical physical settings where one would employ the open system theory. In practical applications one assumes that the subsystem $S$ in under full control of the experimenter (as opposed to the environment that is largely beyond one's capability to wrangle it). This is why we talk about obervables of form $\hat{A}_S \otimes \hat{1}$: to do measurements on a system, first one has to be able to exert certain degree of control over it; so measurement on $S$ are doable, but not so much on $E$. The state preparation procedures are a crucial part of any control package and any procedure that sets a subsystem in a definite state by necessity also severs all correlations with other subsystems. Therefore, a product initial state of $SE$ is an inevitable consequence of a experimental setup for measurements of $S$-only observables.

**Assignment 27 (Solution)** *Calculate the entropy of state of the open qubit system from Assignment 26.*

### 4.3 Joint quasi-probability distributions

There are deeply rooted structural parallels between dynamical and stochastic maps we have been analyzing so far. These analogies are not readily apparent, however—we have to do some work to bring them up to the light. Once we do that, we will be able to easily transfer to dynamical maps virtually all the tricks and methods we have already developed for stochastic dynamics; one of those transferable methods is the super-cumulant expansion, which is vital from the point of view of map computability.

Consider the Hamiltonian of a composite system,

$$\hat{H}_{SE} = \hat{H}_S \otimes \hat{1} + \hat{1} \otimes \hat{H}_E + \lambda \hat{V} \otimes \hat{F}\,. \tag{185}$$

We will investigate the dynamical map derived from $\hat{H}_{SE}$ for system $S$ side-by-side with a stochastic dynamics generated by the corresponding stochastic Hamiltonian

$$\hat{H}_S[F](t) = \hat{H}_S + \lambda F(t)\hat{V}\,.$$

Recall that the stochastic map can be written as an averaged trajectory-wise unitary evolution,

$$\overline{\mathcal{U}}_I(t,0) = \int P_F[f]\mathcal{U}_I[f](t,0)[Df] = \int P_F[f]\hat{U}_I[f](t,0) \bullet \hat{U}_I[f](0,t)[Df]\,, \tag{186}$$

where the unitary map $\mathcal{U}_I[f](t,0) = \hat{U}_I[f](t,0) \bullet \hat{U}_I^\dagger[f](t,0) = \hat{U}_I[f](t,0) \bullet \hat{U}_I[f](0,t)$ and the trajectory-wise evolution operator is given by the standard time-ordered exp,

$$\hat{U}_I[f](t,0) = T_{V_I} e^{-i\lambda \int_0^t f(s)\hat{V}_I(s)ds}. \tag{187}$$

As it turns out, the dynamical map in the interaction picture,

$$\langle\mathcal{U}\rangle(t,0) = e^{-it[\hat{H}_S,\bullet]} \text{tr}_E\left(T_{\mathcal{V}_{SE}} e^{-i\lambda\int_0^t \mathcal{V}_{SE}(s)ds} \bullet \otimes\hat{\rho}_E\right) \equiv e^{-it[\hat{H}_S,\bullet]}\langle\mathcal{U}\rangle_I(t,0), \tag{188}$$

$$\mathcal{V}_{SE}(s) = \left[\left(e^{is\hat{H}_S} \otimes e^{is\hat{H}_E}\right)\hat{V}\otimes\hat{F}\left(e^{-is\hat{H}_S}\otimes e^{-is\hat{H}_E}\right),\bullet\right] = [\hat{V}_I(s)\otimes\hat{F}_I(s),\bullet], \tag{189}$$

can also be written as a functional integral,

$$\langle\mathcal{U}\rangle_I(t,0) = \iint Q_F[f,\bar{f}]\hat{U}_I[f](t,0)\bullet\hat{U}_I[\bar{f}](0,t)[Df][D\bar{f}]. \tag{190}$$

Here, the functional $Q_F[f,\bar{f}]$ is complex-valued, so it cannot be interpreted as a probability distribution, but rather as a *quasi-probability* distribution for a pair of real-valued trajectories, $(f(t),\bar{f}(t))$. One of the trajectories, $f(t)$, plays the role of a driving external field for the evolution operator that goes forwards in time $\hat{U}_I[f](t,0)$, while the other trajectory, $\bar{f}(t)$, drives the evolution going backwards in time, $\hat{U}_I^\dagger[\bar{f}](t,0) = \hat{U}_I[\bar{f}](0,t)$. Note that when the quasi-probability starts to behave as a proper probability distribution, $Q_F[f,\bar{f}] \to \delta[f-\bar{f}]P_F[f]$, we see that dynamical map reduces to stochastic map. We will come back to this link between open system and stochastic dynamics later on in Chapter 7.

Predictably, the functional integral formulation (190) is not really useful in practical applications. If we want to do a proper analysis, we should switch from quasi-probability functional to *joint quasi-probability distributions* $\{Q_F^{(k)}\}_{k=1}^\infty$—an analogue of joint probability distributions $\{P_F^{(k)}\}_{k=1}^\infty$. The construction of joint quasi-probabilities begins with the $E$-side coupling operator and its spectral decomposition,

$$\hat{F} = \sum_n f(n)|n\rangle\langle n| = \sum_{f\in\Omega(F)} f \sum_n \delta_{f,f(n)}|n\rangle\langle n| = \sum_{f\in\Omega(F)} f\,\hat{P}(f); \tag{191a}$$

$$\hat{F}_I(s) = e^{is\hat{H}_E}\hat{F}e^{-is\hat{H}_E} = \sum_{f\in\Omega(F)} f\,e^{is\hat{H}_E}\hat{P}(f)e^{-is\hat{H}_E} = \sum_{f\in\Omega(F)} f\,\hat{P}_I(f,s). \tag{191b}$$

Here, $\Omega(F)$ is the set of all unique eigenvalues, and $\hat{P}(f)$ are the projector operators [i.e. $\hat{P}(f)\hat{P}(f') = \delta_{f,f'}\hat{P}(f)$] onto the degenerate subspaces of $\hat{F}$, i.e. the subspaces spanned by the eigenkets of $\hat{F}$ that correspond to the same eigenvalue $f$. Of course, partitioning the basis of eigenkets into degenerate subspaces does not interfere with the decomposition of identity,

$$\sum_{f\in\Omega(F)} \hat{P}(f) = \sum_n\left(\sum_{f\in\Omega(F)}\delta_{f,f(n)}\right)|n\rangle\langle n| = \sum_n |n\rangle\langle n| = \hat{1}. \tag{192}$$

The dynamical map can then be written as a *quasi-moment* series,

$$\begin{aligned}
\langle\mathcal{U}\rangle_I(t,0) = &\bullet + \sum_{k=1}^\infty (-i\lambda)^k \int_0^t ds_1 \cdots \int_0^{s_{k-1}} ds_k \\
&\sum_{f_1,\bar{f}_1\in\Omega(F)} \cdots \sum_{f_k,\bar{f}_k\in\Omega(F)} Q_F^{(k)}(f_1,\bar{f}_1,s_1;\ldots;f_k,\bar{f}_k,s_k)\mathcal{W}(f_1,\bar{f}_1,s_1)\cdots\mathcal{W}(f_k,\bar{f}_k,s_k) \\
\equiv &\sum_{k=0}^\infty (-i\lambda)^k \left\langle \int_0^t ds_1 \mathcal{W}(F(s_1),\bar{F}(s_1),s_1)\cdots\int_0^{s_{k-1}} ds_k \mathcal{W}(F(s_k),\bar{F}(s_k),s_k)\right\rangle \\
= &\left\langle T_{\mathcal{W}} e^{-i\lambda\int_0^t \mathcal{W}(F(s),\bar{F}(s),s)ds}\right\rangle,
\end{aligned} \tag{193}$$

where the super-operator in $S$-operator space is defined as

$$\mathcal{W}(f,\bar{f},s) = f\hat{V}_I(s)\bullet - \bullet\hat{V}_I(s)\bar{f}, \tag{194}$$

and the joint quasi-probability distribution of order $k$ is given by

$$Q_F^{(k)}(f_1,\bar{f}_1,s_1;\dots;f_k,\bar{f}_k,s_k) = \text{tr}_E\left[\hat{P}_I(f_1,s_1)\cdots\hat{P}_I(f_k,s_k)\,\hat{\rho}_E\,\hat{P}_I(\bar{f}_k,s_k)\cdots\hat{P}_I(\bar{f}_1,s_1)\right]. \tag{195}$$

The subscript $F$ indicates that the quasi-probability is associated with the coupling operator $\hat{F}_I(s)$.

**Assignment 28 (Solution)** *Prove the joint quasi-probability representation of the dynamical map* (193).

To hammer home the analogy with stochastic dynamics, compare the quasi-probability representation with the moment series for $\overline{\mathcal{U}}_I$,

$$\begin{aligned}
\overline{\mathcal{U}}_I(t,0) &= \bullet + \sum_{k=1}^{\infty}(-i\lambda)^k\int_0^t ds_1\cdots\int_0^{s_{k-1}} ds_k\sum_{f_1}\cdots\sum_{f_k} \\
&\quad\times P_F^{(k)}(f_1,s_1;\dots;f_k,s_k)[f_1\hat{V}_I(s_1),\bullet]\cdots[f_k\hat{V}_I(s_k),\bullet] \\
&= \sum_{k=0}^{\infty}(-i\lambda)^k\int_0^t ds_1[F(s_1)\hat{V}_I(s_1),\bullet]\cdots\int_0^{s_{k-1}} ds_k[F(s_k)\hat{V}_I(s_k),\bullet] \\
&= \sum_{k=0}^{\infty}(-i\lambda)^k\int_0^t ds_1\mathcal{W}(F(s_1),F(s_1),s_1)\cdots\int_0^{s_{k-1}} ds_k\mathcal{W}(F(s_k),F(s_k),s_k) \\
&= \overline{T_{\mathcal{W}}e^{-i\lambda\int_0^t\mathcal{W}(F(s),F(s),s)ds}}, \tag{196}
\end{aligned}$$

where we assumed that process $F(t)$ in the stochastic Hamiltonian is discrete (sums instead of integrals over $f_i$) and we have used the fact that

$$[f\hat{V}_I(s),\bullet] = \mathcal{W}(f,f,s).$$

The structure of the moment and quasi-moment series are essentially identical; formally, they differ by the method of averaging the exp: we have the *quasi-average* $\langle\dots\rangle$ for dynamical map and the stochastic average $\overline{(\dots)}$ for stochastic map. In fact, if we were to substitute an ordinary probability distributions for joint quasi-probability distributions,

$$Q_F^{(k)}(f_1,\bar{f}_1,s_1;\dots) \to \delta_{f_1,\bar{f}_1}\cdots\delta_{f_k,\bar{f}_k}P_F^{(k)}(f_1,s_1;\dots),$$

then the dynamical map would simply turn into the stochastic map.

The members of the family of joint quasi-probability distributions $\{Q_F^{(k)}\}_{k=1}^{\infty}$, analogously to joint probability distributions, are all related to each other via their version of Chapman-Kolmogorov equation (the consistency relation),

$$\sum_{f_i,\bar{f}_i}Q_F^{(k)}(f_1,\bar{f}_1,s_1;\dots;f_k,\bar{f}_k,s_k) = Q_F^{(k-1)}(\dots;f_{i-1},\bar{f}_{i-1},s_{i-1};f_{i+1},\bar{f}_{i+1},s_{i+1};\dots), \tag{197}$$

for $k > 1$ and $\sum_{f,\bar{f}}Q_F^{(1)}(f,\bar{f},s) = 1$ for $k = 1$.

**Assignment 29 (Solution)** *Prove consistency relation* (197).

**Assignment 30 (Solution)** *Find the explicit form of quasi-moments (assume $t > s$):*

$$\langle F(t) \rangle = \sum_{f_1, \bar{f}_1} Q_F^{(1)}(f_1, \bar{f}_1, t) f_1 \, ;$$

$$\langle \bar{F}(t) \rangle = \sum_{f_1, \bar{f}_1} Q_F^{(1)}(f_1, \bar{f}_1, t) \bar{f}_1 \, ;$$

$$\langle F(t)F(s) \rangle = \sum_{f_1, \bar{f}_1} \sum_{f_2, \bar{f}_2} Q_F^{(2)}(f_1, \bar{f}_1, t; f_2, \bar{f}_2, s) f_1 f_2 \, ;$$

$$\langle F(t)\bar{F}(s) \rangle = \sum_{f_1, \bar{f}_1} \sum_{f_2, \bar{f}_2} Q_F^{(2)}(f_1, \bar{f}_1, t; f_2, \bar{f}_2, s) f_1 \bar{f}_2 \, ;$$

$$\langle \bar{F}(t)F(s) \rangle = \sum_{f_1, \bar{f}_1} \sum_{f_2, \bar{f}_2} Q_F^{(2)}(f_1, \bar{f}_1, t; f_2, \bar{f}_2, s) \bar{f}_1 f_2 \, ;$$

$$\langle \bar{F}(t)\bar{F}(s) \rangle = \sum_{f_1, \bar{f}_1} \sum_{f_2, \bar{f}_2} Q_F^{(2)}(f_1, \bar{f}_1, t; f_2, \bar{f}_2, s) \bar{f}_1 \bar{f}_2 \, .$$

**Assignment 31 (Solution)** *Find the explicit form of super-quasi-moments,*

$$\langle \mathcal{W}(F(t), \bar{F}(t), t) \rangle \, , \quad \langle \mathcal{W}(F(t), \bar{F}(t), t)\mathcal{W}(F(s), \bar{F}(s), s) \rangle \, , \ (t > s) \, .$$

## 4.4 Super-cumulant expansion for dynamical maps

Previously, we have shown that stochastic maps and dynamical maps are structurally analogous; we have reached this conclusion by comparing the moment and quasi-moment series representation of the respective maps. Essentially, the two maps differ by the method of averaging: the stochastic average $\overline{(\ldots)}$ over joint probability distributions $P_F^{(k)}$ that produces moments, versus the quasi-average $\langle \ldots \rangle$ over joint quasi-probability distributions $Q_F^{(k)}$ that produces quasi-moments. At this point it should not be surprising that there is also a "quasi" analogue of cumulants—*quasi-cumulants*, or *qumulants*, if you will.

First, note that the stochastic cumulants can be written as a kind of average, e.g.,

$$\begin{aligned}
C_F^{(2)}(s_1, s_2) &= \overline{F(s_1)F(s_2)} - \overline{F(s_1)} \ \overline{F(s_2)} \\
&= \sum_{f_1, f_2} \left[ P_F^{(2)}(f_1, s_1; f_2, s_2) - P_F^{(1)}(f_1, s_1) P_F^{(1)}(f_2, s_2) \right] f_1 f_2 \\
&\equiv \sum_{f_1, f_2} \tilde{P}_F^{(2)}(f_1, s_1; f_2, s_2) f_1 f_2 \, .
\end{aligned} \tag{198}$$

In general, we have

$$C_F^{(k)}(s_1, \ldots, s_k) = \sum_{f_1, \ldots, f_k} \tilde{P}_F^{(k)}(f_1, s_1; \ldots; f_k, s_k) f_1 \cdots f_k \equiv \overline{\overline{F(s_1) \cdots F(s_k)}}, \tag{199}$$

where the symbol $\overline{\overline{(\ldots)}}$ stands for the *cumulant average*, and the *cumulant distributions* $\tilde{P}_F^{(k)}$ the cumulant average is carried out with, can be found by inspecting the the corresponding cumulants $C_F^{(k)}$ [see Eq. (138)]; e.g., the first three distributions are defined as,

$$\tilde{P}_F^{(1)}(f, s) = P_F^{(1)}(f, s), \tag{200a}$$

$$\tilde{P}_F^{(2)}(f_1, s_1; f_2, s_2) = P_F^{(2)}(f_1, s_1; f_2, s_2) - P_F^{(1)}(f_1, s_1) P_F^{(1)}(f_2, s_2), \tag{200b}$$

$$\begin{aligned}
\tilde{P}_F^{(3)}(f_1, s_1; f_2, s_2; f_3, s_3) &= P_F^{(3)}(f_1, s_1; f_2, s_2; f_3, s_3) + P_F^{(1)}(f_1, s_1) P_F^{(1)}(f_2, s_2) P_F^{(1)}(f_3, s_3) \\
&\quad - P_F^{(1)}(f_1, s_1) P_F^{(2)}(f_2, s_2; f_3, s_3) - P_F^{(1)}(f_2, s_2) P_F^{(2)}(f_1, s_1; f_3, s_3) \\
&\quad - P_F^{(1)}(f_3, s_3) P_F^{(2)}(f_1, s_1; f_2, s_2) \, .
\end{aligned} \tag{200c}$$

Using the concept of cumulant average we can rewrite the cumulant expansion of the stochastic map,

$$
\begin{aligned}
\overline{\mathcal{U}}_I(t,0) &= T_{\mathcal{V}_I} e^{\sum_{k=1}^{\infty}(-i\lambda)^k \int_0^t ds_1 \cdots \int_0^{s_{k-1}} ds_k \, C_F^{(k)}(s_1,\ldots,s_k)\mathcal{V}_I(s_1)\cdots\mathcal{V}_I(s_k)} \\
&= T_{\mathcal{V}_I} e^{\sum_{k=1}^{\infty}(-i\lambda)^k \int_0^t ds_1 \cdots \int_0^{s_{k-1}} ds_k \, \overline{F(s_1)\cdots F(s_k)}\mathcal{V}_I(s_1)\cdots\mathcal{V}_I(s_k)} \\
&= T_{\mathcal{W}} e^{\sum_{k=1}^{\infty}(-i\lambda)^k \overline{\int_0^t ds_1 \mathcal{W}(F(s_1),F(s_1),s_1)\cdots \int_0^{s_{k-1}} ds_k \mathcal{W}(F(s_k),F(s_k),s_k)}}.
\end{aligned}
\tag{201}
$$

For clarity, this is how one should interpret the cumulant average of super-operators,

$$
\begin{aligned}
&\overline{\mathcal{W}(F(s_1),F(s_1),s_1)\cdots\mathcal{W}(F(s_k),F(s_k),s_k)} \\
&\quad = \sum_{f_1,\ldots,f_k} \tilde{P}_F^{(k)}(f_1,s_1;\ldots;f_k,s_k)\mathcal{W}(f_1,f_1,s_1)\cdots\mathcal{W}(f_k,f_k,s_k) \\
&\quad = \sum_{f_1,\ldots,f_k} \tilde{P}_F^{(k)}(f_1,s_1;\ldots;f_k,s_k)f_1\cdots f_k \mathcal{V}_I(s_1)\cdots\mathcal{V}_I(s_k).
\end{aligned}
\tag{202}
$$

If we now take the quasi-moment representation of a dynamical map (193), and we replicate on it all the steps that previously lead from the moment series to the cumulant expansion (201) (see assignment 22 of Sec. 3.6.2) while replacing all stochastic averages with quasi-averages, we will inevitable end up with the *qumulant* expansion,

$$
\langle\mathcal{U}\rangle_I(t,0) = T_{\mathcal{W}} e^{\sum_{k=1}^{\infty}(-i\lambda)^k \langle\!\langle \int_0^t ds_1 \mathcal{W}(F(s_1),\bar{F}(s_1),s_1)\cdots \int_0^{s_{k-1}} ds_k \mathcal{W}(F(s_k),\bar{F}(s_k),s_k)\rangle\!\rangle},
\tag{203}
$$

where the *quasi-cumulant average* (or *qumulant average*) $\langle\!\langle\ldots\rangle\!\rangle$ operates as follows

$$
\begin{aligned}
&\langle\!\langle \mathcal{W}(F(s_1),\bar{F}(s_1),s_1)\cdots\mathcal{W}(F(s_k),\bar{F}(s_k),s_k)\rangle\!\rangle \\
&\quad = \sum_{f_1,\bar{f}_1}\cdots\sum_{f_k,\bar{f}_k} \tilde{Q}_F^{(2)}(f_1,\bar{f}_1,s_1;\ldots;f_k,\bar{f}_k,s_k)\mathcal{W}(f_1,\bar{f}_1,s_1)\cdots\mathcal{W}(f_k,\bar{f}_k,s_k),
\end{aligned}
\tag{204}
$$

and the *qumulant distributions* $\tilde{Q}_F^{(k)}$ are constructed according the same pattern as cumulant distributions $\tilde{P}_F^{(k)}$, e.g.,

$$
\tilde{Q}_F^{(1)}(f,\bar{f},s) = Q_F^{(1)}(f,\bar{f},s),
\tag{205a}
$$

$$
\tilde{Q}_F^{(2)}(f_1,\bar{f}_1,s_1;f_2,\bar{f}_2,s_2) = Q_F^{(2)}(f_1,\bar{f}_1,s_1;f_2,\bar{f}_2,s_2) - Q_F^{(1)}(f_1,\bar{f}_1,s_1)Q_F^{(1)}(f_2,\bar{f}_2,s_2),
\tag{205b}
$$

etc.. Essentially, the qumulant expansion is obtained form the cumulant expansion by replacing the cumulant average $\overline{(\ldots)}$ with the qumulant average $\langle\!\langle\ldots\rangle\!\rangle$; this is just another consequence of the *structural parallelism* between stochastic and dynamical maps.

**Assignment 32 (Solution)** *Find the first two qumulant distributions $\tilde{Q}_F^{(1)}$ and $\tilde{Q}_F^{(2)}$ without invoking the structural parallelism between stochastic and dynamical maps.*

The main purpose behind introducing the concept of qumulants is to use them as a jump-off point for the *super-qumulant* expansion—the analogue of super-cumulant expansion for stochastic map. Consider now an implicit definition of *super-qumulants* $\{\mathcal{L}^{(k)}\}_{k=1}^{\infty}$,

$$
\langle\mathcal{U}\rangle_I(t,0) = \left\langle T_{\mathcal{W}} e^{-i\lambda \int_0^t \mathcal{W}(F(s),\bar{F}(s),s)ds}\right\rangle \equiv T_{\mathcal{L}^{(k)}} e^{\sum_{k=1}^{\infty}(-i\lambda)^k \int_0^t \mathcal{L}^{(k)}(s)ds}.
$$

Of course, the explicit form of $\mathcal{L}^{(k)}$'s can be found in a standard way: expand the RHS into powers of $\lambda$ and compare them with the terms of corresponding order in the quasi-moment

series on the LHS. However, we do not have to do all that labor! Due to structural parallelism, we can obtain $\mathcal{L}^{(k)}$'s by simply taking the super-cumulants $\mathcal{C}^{(k)}$ and replacing the cumulant averages with the qumulant versions. In assignment 23 of Sec. 3.6.3 you have already found first four super-cumlants,

$$\mathcal{C}^{(1)}(s) = C_F^{(1)}(s)\mathcal{V}_I(s) = \overline{\overline{\mathcal{W}(F(s),F(s),s)}}, \tag{206a}$$

$$\mathcal{C}^{(2)}(s) = \mathcal{V}_I(s)\int_0^s C_F^{(2)}(s,u)\mathcal{V}_I(u)du = \int_0^s \overline{\overline{\mathcal{W}(F(s),F(s),s)\mathcal{W}(F(u),F(u),u)}}\,du, \tag{206b}$$

$$\mathcal{C}^{(3)}(s) = \int_0^s du_1\int_0^{u_1} du_2\,\overline{\overline{\mathcal{W}(F(s),F(s),s)\mathcal{W}(F(u_1),F(u_1),u_1)\mathcal{W}(F(u_2),F(u_2),u_2)}}$$

$$+ \int_0^s du_1\int_0^{u_1} du_2\,\overline{\overline{\mathcal{W}(F(s),F(s),s)\left[\overline{\mathcal{W}(F(u_1),F(u_1),u_1)},\mathcal{W}(F(u_2),F(u_2),u_2)\right]}}. \tag{206c}$$

$\vdots$

This gives us immediately the corresponding super-qumulants,

$$\mathcal{L}^{(1)}(s) = \left\langle\!\left\langle \mathcal{W}(F(s),\bar{F}(s),s)\right\rangle\!\right\rangle, \tag{207a}$$

$$\mathcal{L}^{(2)}(s) = \int_0^s \left\langle\!\left\langle \mathcal{W}(F(s),\bar{F}(s),s)\mathcal{W}(F(u),\bar{F}(u),u)\right\rangle\!\right\rangle du, \tag{207b}$$

$$\mathcal{L}^{(3)}(s) = \int_0^s du_1\int_0^{u_1} du_2\,\left\langle\!\left\langle \mathcal{W}(F(s),\bar{F}(s),s)\mathcal{W}(F(u_1),\bar{F}(u_1),u_1)\mathcal{W}(F(u_2),\bar{F}(u_2),u_2)\right\rangle\!\right\rangle$$

$$+ \int_0^s du_1\int_0^{u_1} du_2\,\left\langle\!\left\langle \mathcal{W}(F(s),\bar{F}(s),s)\left[\langle\mathcal{W}(F(u_1),\bar{F}(u_1),u_1)\rangle,\mathcal{W}(F(u_2),\bar{F}(u_2),u_2)\right]\right\rangle\!\right\rangle, \tag{207c}$$

$\vdots$

**Assignment 33 (Solution)** *Calculate the explicit forms of super-qumulants $\mathcal{L}^{(1)}(s)$ and $\mathcal{L}^{(2)}(s)$.*

**Assignment 34 (Solution)** *Find the quasi-probability representation of the dynamical map for general SE coupling*

$$\hat{V}_{SE} = \lambda\sum_\alpha \hat{V}_\alpha\otimes\hat{F}_\alpha,$$

*where $\hat{V}_\alpha$ and $\hat{F}_\alpha$ are hermitian operators.*
   *Note that this also includes couplings of form*

$$\hat{V}_{SE} = \lambda(\hat{S}\otimes\hat{E} + \hat{S}^\dagger\otimes\hat{E}^\dagger),$$

*where $\hat{S}$ and $\hat{E}$ are non-hermitian. Operator like this can always be transformed into a combination of hermitian products:*

$$\hat{S}\otimes\hat{E} + \hat{S}^\dagger\otimes\hat{E}^\dagger = \frac{\hat{S}+\hat{S}^\dagger}{\sqrt{2}}\otimes\frac{\hat{E}+\hat{E}^\dagger}{\sqrt{2}} + \frac{\hat{S}-\hat{S}^\dagger}{i\sqrt{2}}\otimes\frac{\hat{E}^\dagger-\hat{E}}{i\sqrt{2}}\,.$$

**Assignment 35 (Solution)** *Calculate the explicit forms of super-qumulants $\mathcal{L}^{(1)}(s)$ and $\mathcal{L}^{(2)}(s)$ for the general SE coupling,*

$$\hat{V}_{SE} = \lambda \sum_\alpha \hat{V}_\alpha \otimes \hat{F}_\alpha,$$

*where $\hat{V}_\alpha$ and $\hat{F}_\alpha$ are hermitian operators.*

## 4.5 An interpretation of joint quasi-probability distributions

So far we have been single-mindedly exploiting the structural parallelism between dynamical and stochastic maps to arrive as efficiently as possible at the super qumulant expansion. On our way there, we have identified the joint quasi-probability distributions $\{Q_F^{(k)}\}_{k=1}^\infty$; although, surely, they are a new and interesting object, up to this point we have looked at them only from purely utilitarian perspective. Now is the moment to pause for a minute and consider possible physical interpretations.

The choice of the prefix "quasi" in the name have been very much deliberate: joint quasi-probability distributions are similar to joint probability distributions in certain ways (both function as distributions to integrate with, and both satisfy analogous consistency relations) but—and this is vital—there are also key differences that completed change their respective natures. Let us illustrate what we have in mind here with an example. First and foremost, joint probability distributions $P_F^{(k)}(f_1, s_1; \ldots; f_k, s_k)$ are probability distributions for chronological sequences of form $(f_k, \ldots, f_1)$. The most natural physical interpretation is that such a sequence is a record of trajectory of process $F(t)$ captured by repeated observations at the consecutive moments in time $0 < s_k < s_{k-1} < \cdots < s_1$; hence, the sequence can be though of as a sample trajectory of $F(t)$. This observation has important practical implications. On the one hand, if one has access to functions $P_F^{(k)}$, then one can generate trajectories of $F(t)$ by drawing sample sequences from the probability distributions (see assignment 19 of Sec. 3.1.3). On the other hand, when one has access to sufficiently large ensemble of sample trajectories, then one can approximate the stochastic average over distribution $P_F^{(k)}$ with the sample average (see Sec. 3.6.4); the equivalence of sample average and average over distribution follows from the very definition of probability. Since functions $Q_F^{(k)}$ are complex-valued, they are *quasi*-probability distributions—they appear in formulas as if they were probability distributions, but cannot be actual probabilities because they are not non-negative. Consequently, it is impossible to use $Q_F^{(k)}$'s to generate something analogous to trajectories, nor to approximate the quasi-average with some sort of sample average. This is the key difference between the two types of distributions and the reason why, despite all the formal similarities, they are of a completely different natures.

What *is* the nature of joint quasi-probability distributions, then? Obviously, it is a tricky question to answer, but we may try to find an inspiration by looking for other contexts where quasi-probabilities appear naturally. One example of such context is the classic double-slit-type of experiments that have been often used to demonstrate, among other thing, the super-position principle and the effects of quantum interference. The most famous type of double-slit experiment is of the Young's experiment; it typically involves an emitter of particle beam shot towards a screen that has the ability to detect the position of particles hitting it. However, the direct path to the detector is blocked by an impassable barrier with a couple of slits punctured through. As a result, there are only a number of alternative paths—each passing through different slit—a particle can take to reach the screen. For quantum particles, those different paths are in super-position and they can interfere with each other. To avoid technical complications, inevitable when attempting to describe such a setup, we will consider instead a much simpler variant of double-slit experiment—the Stern-Gerlach (SG) experiment.

The SG experiment involves particles with spin degree of freedom, described with operator

$$\hat{S} = \sum_{m=-s}^{s} m|m\rangle\langle m|, \tag{208}$$

representing, say, the $z$-component of the spin (e.g., for $s = 1/2$ we would have $\hat{S} = \hat{\sigma}_z/2$). The beam of those particles is emitted towards the *SG apparatus* that employs the magnetic field gradient to deflect moving particles in the directions dependent on the value of $m$—the higher the value the steeper the deflection angle. Consequently, when the initial beam passes through the apparatus, it leaves split into $2s+1$ distinct beams, each characterized by a definite value of $m$. The detector screen is placed right behind the output ports of the SG apparatus; since the beam has been split, one is now able to correlate the position of the detected hit with the value of $m$, thus realizing the measurement of the spin. According to the rules of quantum mechanics, the probability of measuring a particular value $m$ is given by the modulus square of the *probability amplitude* for the corresponding process leading to the detection event,

$$\text{Prob}(m) = |\text{Amp}(m)|^2. \tag{209}$$

Assuming that the travel time from the emitter to the screen is $t$, and the initial state of the spin is given by $|\Psi\rangle = \sum_{m'} c_{m'}|m'\rangle$, and the Hamiltonian describing the evolution of the beam is $\hat{H}_{\text{beam}}$ (it includes the beam-splitting SG apparatus), we can write the amplitude as

$$\text{Amp}(m) = \langle m|e^{-it\hat{H}_{\text{beam}}}|\Psi\rangle = \sum_{m'=-s}^{s} c_{m'}\langle m|e^{-it\hat{H}_{\text{beam}}}|m'\rangle \equiv \sum_{m'=-s}^{s} c_{m'}\phi_t(m|m'). \tag{210}$$

We are not indicating here the spatial degrees of freedom because they do not have any impact on our discussion.

To turn this basic setup into proper double-slit experiment, we insert an additional SG apparatus somewhere in between the emitter and the detection screen (say, it takes time $u < t$ for particles to reach it). Right after the apparatus we place an impassible barrier with two slits punctured through in such a way that only $m = \pm s$ beams are not being blocked. Then, we install one more SG apparatus with the gradient orientation in the opposite direction so that the split beams recombine into a single beam. In this setup, the amplitude for the spin to be found in state $m$, provided that it has passed trough one of the slits, is calculated according to the product rule,

$$\text{Amp}(m|\pm s) = \langle m|e^{-i(t-u)\hat{H}_{\text{beam}}}|\pm s\rangle\langle \pm s|e^{-iu\hat{H}_{\text{beam}}}|\Psi\rangle = \sum_{m'=-s}^{s} c_{m'}\phi_{t-u}(m|\pm s)\phi_u(\pm s|m'). \tag{211}$$

The total amplitude is a sum of amplitudes for passing through each slit,

$$\text{Amp}(m) = \text{Amp}(m|+s) + \text{Amp}(m|-s) = \sum_{m'=-s}^{s} c_{m'} \sum_{m''=\pm s} \phi_{t-u}(m|m'')\phi_u(m''|m'), \tag{212}$$

and thus, the probability of measuring $m$ reads

$$\begin{aligned}
\text{Prob}(m) = |\text{Amp}(m)|^2 &= \Big|\sum_{m'=\pm s}\text{Amp}(m|m')\Big|^2 \\
&= \sum_{m'=\pm s} |\text{Amp}(m|m')|^2 + \text{Amp}(m|+s)\,\text{Amp}(m|-s)^* + \text{Amp}(m|-s)\,\text{Amp}(m|+s)^* \\
&\equiv \sum_{m'=\pm s} \text{Prob}(m|m') + \Phi(m|s,-s). \tag{213}
\end{aligned}$$

SciPost Phys. Lect. Notes 68 (2023)

We got here the standard result for double-slit experiment: $\text{Prob}(m|+s) + \text{Prob}(m|-s)$ is the classical probability for reaching the screen through two alternative paths, and the *interference* term $\Phi(m|s,-s)$ is responsible for all the quantum effects visible in the measured results.

Now we establish the connection with quasi-probabilities. Given the spectral decomposition of the spin operator $\hat{S}$, the Hamiltonian of the spin particle beam $\hat{H}_{\text{beam}}$, and the initial pure state, $\hat{\rho} = |\Psi\rangle\langle\Psi|$, we can define a joint quasi-probability distribution associated with $\hat{S}$,

$$
\begin{aligned}
&Q_S^{(k)}(m_1, \bar{m}_1, s_1; \ldots; m_k, \bar{m}_k, s_k) \\
&\quad = \text{tr}\left(\hat{P}_I(m_1, s_1) \cdots \hat{P}_I(m_k, s_k) \hat{\rho} \hat{P}_I(\bar{m}_k, s_k) \cdots \hat{P}_I(\bar{m}_1, s_1)\right) \\
&\quad = \text{tr}\left(e^{is_1 \hat{H}_{\text{beam}}} |m_1\rangle\langle m_1| e^{-is_1 \hat{H}_{\text{beam}}} \cdots e^{is_k \hat{H}_{\text{beam}}} |m_k\rangle\langle m_k| e^{-is_k \hat{H}_{\text{beam}}} |\Psi\rangle\right. \\
&\qquad \left. \times \langle\Psi| e^{is_k \hat{H}_{\text{beam}}} |\bar{m}_k\rangle\langle\bar{m}_k| e^{-is_k \hat{H}_{\text{beam}}} \cdots e^{is_1 \hat{H}_{\text{beam}}} |\bar{m}_1\rangle\langle\bar{m}_1| e^{-is_k \hat{H}_{\text{beam}}}\right) \\
&\quad = \delta_{m_1, \bar{m}_1} \langle m_1| e^{-i(s_1-s_2)\hat{H}_{\text{beam}}} |m_2\rangle \cdots \langle m_k| e^{-is_k \hat{H}_{\text{beam}}} |\Psi\rangle \\
&\qquad \times \left(\langle m_1| e^{-i(s_1-s_2)\hat{H}_{\text{beam}}} |\bar{m}_2\rangle \cdots \langle\bar{m}_k| e^{-is_k \hat{H}_{\text{beam}}} |\Psi\rangle\right)^* \\
&\quad = \delta_{m_1, \bar{m}_1} \sum_{m_{k+1}} \sum_{\bar{m}_{k+1}} c_{m_{k+1}} c_{\bar{m}_{k+1}}^* \prod_{i=1}^k \phi_{s_i - s_{i+1}}(m_i|m_{i+1}) \phi_{s_i - s_{i+1}}(\bar{m}_i|\bar{m}_{i+1})^*,
\end{aligned}
\tag{214}
$$

and we have set here $s_{k+1} = 0$. Observe that $Q_S^{(k)}$ has a form of a product of the elementary amplitudes $\phi_{s_i - s_{i+1}}(m_i|m_{i+1})$. To clarify this point, let us focus on the second-order quasi-probability,

$$
\begin{aligned}
Q_S^{(2)}(m_1, m_1, t; m_2, \bar{m}_2, u) &= \sum_{m_0} c_{m_0} \phi_{t-u}(m_1|m_2) \phi_u(m_2|m_0) \sum_{\bar{m}_0} c_{\bar{m}_0}^* \phi_{t-u}(m_1|\bar{m}_2)^* \phi_u(\bar{m}_2|\bar{m}_0)^* \\
&= \text{Amp}(m_1|m_2) \text{Amp}(m_1|\bar{m}_2)^*.
\end{aligned}
\tag{215}
$$

We see here how exactly a quasi-probability relates to probability amplitudes Amp we have been discussing so far. Consequently, both parts of the measurement probability $\text{Prob}(m)$ in our double-slit experiment can be rewritten in terms of quasi-probability distribution,

$$
\text{Prob}(m|\pm s) = Q_S^{(2)}(m, m, t; \pm s, \pm s, u);
\tag{216a}
$$

$$
\Phi(m|s, -s) = Q_S^{(2)}(m, m, t; +s, -s, u) + Q_S^{(2)}(m, m, t; -s, +s, u).
\tag{216b}
$$

Of course, this result can be generalized: Each additional SG apparatus and barrier-with-slits combo increases the order $k$ of used quasi-probabilities $Q_S^{(k)}$; each additional slit punctured through barrier adds more possible values of argument pair $m_i, \bar{m}_i$. In any case, the overall pattern always remains unchanged. First, the quasi-probability with matching sets of arguments, $\bar{m}_i = m_i$ for all $i$, equals the classical conditional probability for particles to get to the detector through alternative paths defined by the sequence of traversed slits $(m_1, m_2, \ldots, m_k)$,

$$
Q_S^{(k)}(m_1, m_1, t; m_2, m_2, u_2; \ldots; m_k, m_k, u_k) = \text{Prob}(m_1|m_2, \ldots, m_k).
\tag{217}
$$

Second, all the quasi-probabilities with one path $(m_1, m_2, \ldots, m_k)$ not matching the other path $(m_1, \bar{m}_2, \ldots, \bar{m}_k)$ (even if the only difference is one pair of arguments), contribute collectively to the interference term. One could even say that each individual mismatched quasi-probability, $Q_S^{(k)}(m_1, m_1, t; m_2, \bar{m}_2, u_2; \ldots)$, quantifies the quantum interference between the two alternative paths.

Finally, note how in our example the joint quasi-probability distributions appeared even though the SG experiment is a *closed* quantum system. This underlines an important feature of $\{Q_F^{(k)}\}_{k=1}^\infty$: joint quasi-probabilities are defined exclusively by the dynamical properties of the

system the observable $\hat{F}$ belongs to. Therefore, in the context of dynamical map, the quasi-probabilities that holistically describe the influence $E$ has on the dynamics of open system $S$, are completely independent of $S$ itself.

## 4.6 Central super-qumulants

While studying qumulants and super-qumulants (or cumulants and super-cumulants), it is not difficult to notice that the vast majority of formulas simplify tremendously when the average value $\langle F(t)\rangle = \mathrm{tr}_E(\hat{F}_I(t)\hat{\rho}_E)$ (or $\overline{F(t)}$) is set to zero. Here, we will show how to eliminate the contribution from the average even when $\langle F(t)\rangle \neq 0$ by switching the dynamical map to *central picture*.

We start by rewriting the total Hamiltonian of the $SE$ system,

$$
\begin{aligned}
\hat{H}_{SE} &= \hat{H}_S \otimes \hat{1} + \hat{1} \otimes \hat{H}_E + \lambda \hat{V} \otimes \hat{F} + \left(\lambda\langle F(t)\rangle\hat{V}\otimes\hat{1} - \lambda\langle F(t)\rangle\hat{V}\otimes\hat{1}\right) \\
&= \left(\hat{H}_S + \lambda\langle F(t)\rangle\hat{V}\right)\otimes\hat{1} + \hat{1}\otimes\hat{H}_E + \lambda\hat{V}\otimes(\hat{F} - \langle F(t)\rangle\hat{1}) \\
&= \hat{H}'_S(t)\otimes\hat{1} + \hat{1}\otimes\hat{H}_E + \lambda\hat{V}\otimes\hat{X}(t),
\end{aligned}
\tag{218}
$$

where

$$
\hat{H}'_S(t) = \hat{H}_S + \lambda\langle F(t)\rangle\hat{V}, \tag{219}
$$

$$
\hat{X}(t) = \hat{F} - \langle F(t)\rangle\hat{1} = \sum_f (f - \langle F(t)\rangle)\hat{P}(f). \tag{220}
$$

The operator $\hat{H}'_S$ can be though of as a modified Hamiltonian of the open system $S$, and the "centered" coupling $\hat{X}$ has zero average,

$$
\langle X(t)\rangle = \mathrm{tr}_E\left(e^{it\hat{H}_E}\hat{X}(t)e^{-it\hat{H}_E}\hat{\rho}_E\right) = \mathrm{tr}_E\left(\hat{F}_I(t)\hat{\rho}_E\right) - \langle F(t)\rangle\,\mathrm{tr}_E(\hat{\rho}_E) = 0. \tag{221}
$$

The next step is to disentangle from the dynamical map the $S$-only part of the total Hamiltonian, $\mathcal{H}'_S(t) = [\hat{H}'_S(t), \bullet]$,

$$
\langle\mathcal{U}\rangle(t,0) = \mathcal{U}'_S(t,0)\langle\mathcal{U}\rangle_c(t,0), \tag{222}
$$

where $\mathcal{U}'_S(t,0) = T_{\mathcal{H}'_S}\exp[-i\int_0^t \mathcal{H}'_S(s)ds]$ and $\langle\mathcal{U}\rangle_c$ is the map in central picture, that we got instead of the interaction picture which we would have obtained if we had disentangle only $[\hat{H}_S, \bullet]$. Since the eigenkets of $\hat{X}$ and $\hat{F}$ are identical, we can express the central picture map as a quasi-moment series with only slight modifications (compare with the assignment 28 of Sec. 4.3),

$$
\begin{aligned}
\langle\mathcal{U}\rangle_c(t,0) &= \sum_{k=0}^{\infty}(-i\lambda)^k \int_0^t ds_1 \cdots \int_0^{s_{k-1}} ds_k \sum_{f_1,\bar{f}_1}\cdots\sum_{f_k,\bar{f}_k} Q_F^{(k)}(f_1,\bar{f}_1,s_1;\ldots;f_k,\bar{f}_k,s_k) \\
&\quad \times \mathcal{W}_c(f_1 - \langle F(s_1)\rangle, \bar{f}_1 - \langle F(s_1)\rangle, s_1)\cdots\mathcal{W}_c(f_k - \langle F(s_k)\rangle, \bar{f}_k - \langle F(s_k)\rangle, s_k) \\
&= \sum_{k=0}^{\infty}(-i\lambda)^k \left\langle \int_0^t ds_1 \mathcal{W}_c(F(s_1) - \langle F(s_1)\rangle, \bar{F}(s_1) - \langle F(s_1)\rangle, s_1)\cdots\right. \\
&\quad \left.\cdots\int_0^{s_{k-1}} ds_k \mathcal{W}_c(F(s_k) - \langle F(s_k)\rangle, \bar{F}(s_k) - \langle F(s_k)\rangle, s_k)\right\rangle \\
&= \sum_{k=0}^{\infty}(-i\lambda)^k \left\langle \int_0^t ds_1 \mathcal{W}_c(X(s_1), \bar{X}(s_1), s_1)\cdots\int_0^{s_{k-1}} ds_k \mathcal{W}_c(X(s_k), \bar{X}(s_k), s_k)\right\rangle \\
&= \left\langle T_{\mathcal{W}} e^{-i\lambda\int_0^t \mathcal{W}_c(X(s),\bar{X}(s),s)ds}\right\rangle,
\end{aligned}
\tag{223}
$$

where the central picture of super-operators $\mathcal{W}$ are given by

$$\hat{V}_c(s) = \left(T_{H'_S}e^{-i\int_0^s \hat{H}'_S(u)du}\right)^\dagger \hat{V}\, T_{H'_S}e^{-i\int_0^s \hat{H}'_S(u)du} = \hat{U}'_S(0,s)\hat{V}\hat{U}'_S(s,0)\,, \tag{224a}$$

$$\mathcal{W}_c(x,\bar{x},s) = x\hat{V}_c(s)\bullet - \bullet \hat{V}_c(s)\bar{x}\,. \tag{224b}$$

Note that the central and interaction pictures are related via the following transformation,

$$\hat{V}_c(s) = \hat{U}'_S(0,s)e^{-is\hat{H}_S}\hat{V}_I(s)\,e^{is\hat{H}_S}\hat{U}'_S(s,0)\,, \tag{225a}$$

$$\mathcal{W}_c(x,\bar{x},s) = \mathcal{U}'_S(0,s)e^{-is[\hat{H}_S,\bullet]}\mathcal{W}(x,\bar{x},s)e^{is[\hat{H}_S,\bullet]}\mathcal{U}'_S(s,0)\,. \tag{225b}$$

The final step is to introduce the *central* super-qumulants $\mathcal{L}_c^{(k)}(s)$ in the usual fashion,

$$\left\langle T_{\mathcal{W}_c}e^{-i\lambda\int_0^t \mathcal{W}_c(X(s),\bar{X}(s),s)ds}\right\rangle \equiv T_{\mathcal{L}_c^{(k)}}e^{\sum_{k=1}^\infty (-i\lambda)^k \int_0^t \mathcal{L}_c^{(k)}(s)ds}\,. \tag{226}$$

We, then, find the consecutive $\mathcal{L}_c^{(k)}$ by comparing the terms of the corresponding orders in $\lambda$ on LHS and RHS.

As advertised, the first-order central super-qumulant vanishes,

$$\begin{aligned}
\mathcal{L}_c^{(1)}(s) &= \langle \mathcal{W}_c(X(s),\bar{X}(s),s)\rangle = \langle F(s) - \langle F(s)\rangle\rangle \hat{V}_c(s)\bullet - \bullet\hat{V}_c(s)\langle\bar{F}(s)-\langle F(s)\rangle\rangle\\
&= (\langle F(s)\rangle - \langle F(s)\rangle)\hat{V}_c(s)\bullet - \bullet\hat{V}_c(s)\left(\langle\bar{F}(s)\rangle - \langle F(s)\rangle\right) = 0\,.
\end{aligned} \tag{227}$$

This, of course, has ripple effects on all subsequent orders. The second central super-qumulant reduces to

$$\begin{aligned}
\mathcal{L}_c^{(2)}(s) &= \int_0^s \langle \mathcal{W}_c(X(s),\bar{X}(s),s)\mathcal{W}_c(X(u),\bar{X}(u),u)\rangle du - \int_0^s \mathcal{L}_c^{(1)}(s)\mathcal{L}_c^{(1)}(u)du\\
&= \int_0^s \left\langle \left(F(s)-\langle F(s)\rangle\right)\left(F(u)-\langle F(u)\rangle\right)\right\rangle\left(\hat{V}_c(s)\hat{V}_c(u)\bullet - \hat{V}_c(u)\bullet\hat{V}_c(s)\right)du\\
&\quad + \int_0^s \left\langle\left(\bar{F}(s)-\langle F(s)\rangle\right)\left(\bar{F}(u)-\langle F(u)\rangle\right)\right\rangle\left(\bullet\hat{V}_c(u)\hat{V}_c(s) - \hat{V}_c(s)\bullet\hat{V}_c(u)\right)du\\
&= \int_0^s \langle\langle F(s)F(u)\rangle\rangle\left(\hat{V}_c(s)\hat{V}_c(u)\bullet - \hat{V}_c(u)\bullet\hat{V}_c(s)\right)du\\
&\quad + \int_0^s \langle\langle\bar{F}(s)\bar{F}(u)\rangle\rangle\left(\bullet\hat{V}_c(u)\hat{V}_c(s) - \hat{V}_c(s)\bullet\hat{V}_c(u)\right)du\\
&= \int_0^s \langle\langle\mathcal{W}_c(F(s),\bar{F}(s),s)\mathcal{W}_c(F(u),\bar{F}(u),u)\rangle\rangle du\,, \tag{228}
\end{aligned}$$

or, using the shorthand notation,

$$\tilde{Q}_{i_1\cdots i_k}\mathcal{W}_{i_1}\cdots\mathcal{W}_{i_k} = \sum_{f_{i_1},\bar{f}_{i_1}}\cdots\sum_{f_{i_k},\bar{f}_{i_k}}\tilde{Q}_F^{(k)}(f_{i_1},\bar{f}_{i_1},s_{i_1};\ldots;f_{i_k},\bar{f}_{i_k},s_k)\mathcal{W}_c(f_{i_1},\bar{f}_{i_1},s_{i_1})\cdots\mathcal{W}_c(f_{i_k},\bar{f}_{i_k},s_{i_k})\,, \tag{229}$$

we can write it as

$$\mathcal{L}_c^{(2)}(s_1) = \int_0^{s_1} ds_2\,\tilde{Q}_{12}\mathcal{W}_1\mathcal{W}_2\,. \tag{230}$$

Then, normally very complex higher order super-qumulants, simplify significantly in the central picture,

$$\mathcal{L}_c^{(3)}(s_1) = \int_0^{s_1} ds_2 \int_0^{s_2} ds_3 \, \tilde{Q}_{123} \mathcal{W}_1 \mathcal{W}_2 \mathcal{W}_3 \,, \tag{231}$$

$$\mathcal{L}_c^{(4)}(s_1) = \int_0^{s_1} ds_2 \cdots \int_0^{s_4} ds_4 \, \tilde{Q}_{1234} \mathcal{W}_1 \mathcal{W}_2 \mathcal{W}_3 \mathcal{W}_4$$
$$+ \int_0^{s_1} ds_2 \cdots \int_0^{s_3} ds_4 \Big( \tilde{Q}_{13} \tilde{Q}_{24} \mathcal{W}_1 [\mathcal{W}_2, \mathcal{W}_3] \mathcal{W}_4 + \tilde{Q}_{14} \tilde{Q}_{23} \mathcal{W}_1 [\mathcal{W}_2 \mathcal{W}_3, \mathcal{W}_4] \Big). \tag{232}$$

The cost of switching to the central picture is rather minuscule: computing the transformation operator $\hat{U}_S'$ is a trivial problem (it satisfies a standard dynamical equation). The readily apparent benefits we have just showcased above are definitely worth this additional bit of work.

## 4.7 Computability of dynamical maps

Due to complexities of the environment $E$, the computation of dynamical maps is a notoriously difficult problem (even more so than it is for stochastic maps). Since, in general, dynamical maps do not satisfy any form of dynamical equation—the structural parallelism with stochastic maps can be invoked here to explain why it is the case—a systematic way forward is to *approximate* the central super-qumulant (cSQ) series with a closed-form super-operator, and by doing so, force a dynamical equation upon the map. Once we have the equation, we can integrate it numerically, and thus, compute a map approximating the real dynamical map. This is not the only way to achieve computability, mind you; there are myriad specialized methods to compute an approximated map, but, typically, they work only in a narrowly defined contexts. Here, we want to discuss a general purpose approach, hence the focus on the cSQ series.

A good starting point for most approximation schemes is to assume that $E$ is *Gaussian*, i.e. to neglect the contribution from qumulant distributions $\tilde{Q}_F^{(k)}$ with $k > 2$—an approximation analogous to cutting the cumulant series of stochastic process at the second order. The Gaussian environment means that the totality of the information about $E$ is contained in the trivial expectation value of the coupling $\langle F(t) \rangle$ and the second-order qumulants,

$$\langle\langle F(t)F(s) \rangle\rangle = \langle\langle \bar{F}(t)F(s) \rangle\rangle = \langle F(t)F(s) \rangle - \langle F(t) \rangle \langle F(s) \rangle = C_F(t,s) - iK_F(t,s) \,, \tag{233a}$$
$$\langle\langle \bar{F}(t)\bar{F}(s) \rangle\rangle = \langle\langle F(t)\bar{F}(s) \rangle\rangle = \langle \bar{F}(t)\bar{F}(s) \rangle - \langle F(t) \rangle \langle F(s) \rangle = C_F(t,s) + iK_F(t,s) \,; \tag{233b}$$

that can be expressed through two physically significant quantities,

$$C_F(s,u) = \frac{1}{2} \big( \langle F(t)F(s) \rangle + \langle \bar{F}(t)\bar{F}(s) \rangle \big) - \langle F(t) \rangle \langle F(s) \rangle$$
$$= \frac{1}{2} \mathrm{tr}_E \Big( \big\{ \hat{F}_I(t) - \langle F(t) \rangle \,, \, \hat{F}_I(s) - \langle F(s) \rangle \big\} \hat{\rho}_E \Big) \,; \tag{234}$$

$$K_F(s,u) = \frac{1}{2} i \big( \langle F(t)F(s) \rangle - \langle \bar{F}(t)\bar{F}(s) \rangle \big)$$
$$= \frac{1}{2} i \, \mathrm{tr}_E \Big( \big[ \hat{F}_I(t) - \langle F(t) \rangle \,, \, \hat{F}_I(s) - \langle F(s) \rangle \big] \hat{\rho}_E \Big). \tag{235}$$

These two objects, the *susceptibility* $K_F$ and the *correlation function* $C_F$ (and the *spectral density* $S(\omega)$, the Fourier transform of $C_F$), can often be measured—with the *linear response* technique to obtain the former, and with *noise spectroscopy* to get the latter. Moreover, when $E$ is in thermal equilibrium (i.e. $\hat{\rho}_E \propto e^{-\beta \hat{H}_E}$), then the spectral density and the susceptibility are related through the *Fluctuation-Dissipation theorem*, so that one can be obtained from the other, given the temperature of the environment.

**Assignment 36 (Solution)** *Prove the Fluctuation-Dissipation theorem,*

$$S(\omega) = i\left(\frac{1 + e^{-\beta\omega}}{1 - e^{-\beta\omega}}\right)\kappa(\omega),$$

*where $\beta = 1/k_{\mathrm{B}}T$, and*

$$S(\omega) = \int_{-\infty}^{\infty} C_F(\tau, 0) e^{-i\omega\tau} d\tau; \quad \kappa(\omega) = \int_{-\infty}^{\infty} K_F(\tau, 0) e^{-i\omega\tau} d\tau.$$

The Gaussianity is a natural approximation—or even an exact property, in some cases—for environments composed of indistinguishable bosons; some prominent examples of such systems include phononic excitations in solids and electromagnetic quantum fields. Gaussianity can also be expected in $E$ where the central limit theorem-style of argument is applicable, i.e. systems that are composed of large number of roughly independent and similar subsystems,

$$\hat{F} \approx \frac{1}{\sqrt{N}}\sum_{i=1}^{N} \hat{1}^{\otimes(i-1)} \otimes \hat{f} \otimes \hat{1}^{\otimes(N-i)}; \quad \hat{H}_E \approx \sum_{i=1}^{N} \hat{1}^{\otimes(i-1)} \otimes \hat{h} \otimes \hat{1}^{\otimes(N-i)}; \quad \hat{\rho}_E \approx \hat{\rho}^{\otimes N}. \tag{236}$$

An example of such $E$ would be a bath of nuclear spins in crystal lattices. Finally, note that the Gaussianity, or any other approximation to quasi-moments and qumulants, can be argued for independently of any properties of the open system $S$ because $Q_F^{(k)}$'s are defined exclusively by the coupling operator $\hat{F}$ and the dynamical properties of $E$.

Assuming that $E$ is Gaussian, the simplest approximation to cSQ series on the $S$ side is to truncate it at some finite order $n$ (note that for Gaussian $E$ odd-order cSQs are zero), so that

$$\langle \mathcal{U} \rangle_c(t, 0) \approx T_{\mathcal{L}_c^{(k)}} e^{\sum_{k=1}^{n}(-i\lambda)^{2k}\int_0^t \mathcal{L}_c^{(2k)}(s)ds} \equiv \langle \mathcal{U} \rangle^{(2n)}(t, 0), \tag{237}$$

where, of course, the approximating map $\langle \mathcal{U} \rangle^{(2n)}$ is computable because it satisfies the dynamical equation,

$$\frac{d}{dt}\langle \mathcal{U} \rangle^{(2n)}(t, s) = \left(\sum_{k=1}^{n}(-i\lambda)^{2k}\mathcal{L}_c^{(2k)}(t)\right)\langle \mathcal{U} \rangle^{(2n)}(t, s); \quad \langle \mathcal{U} \rangle^{(2n)}(s, s) = \bullet. \tag{238}$$

The justification for this strategy is based on the estimate of the order of magnitude for cSQs,

$$\lambda^{2k}\int_0^t \mathcal{L}_c^{(2k)}(s)ds \sim \lambda^{2k}t\tau_c^{2k-1}\left(\frac{\tau_c}{\tau_S}\right)^{k-1}; \tag{239}$$

let us now explain where this estimation comes from. Here, it is assumed the coupling operators has been defined in such a way that $||\mathcal{W}_c|| \sim 1$ so that the overall magnitude of the coupling is quantified by $\lambda$. Next, the *correlation time* $\tau_c$ is defined as a range of the correlation function,

$$C_F(t, s) \xrightarrow{|t-s| \gg \tau_c} 0. \tag{240}$$

More generally, $\tau_c$ is a time-scale on which the quasi-process $(F(t), \bar{F}(t))$ decorrelates,

$$\left.\begin{array}{r}\langle\!\langle \cdots F(t)\cdots F(s)\cdots \rangle\!\rangle \\ \langle\!\langle \cdots \bar{F}(t)\cdots F(s)\cdots \rangle\!\rangle \\ \langle\!\langle \cdots F(t)\cdots \bar{F}(s)\cdots \rangle\!\rangle \\ \langle\!\langle \cdots \bar{F}(t)\cdots \bar{F}(s)\cdots \rangle\!\rangle\end{array}\right\} \xrightarrow{|t-s| \gg \tau_c} 0. \tag{241}$$

This follows from the analogy with the statistical irreducibility of cumulants (see Sec. 3.6.2), that can also be applied to the the process evaluated at different points in time. Typically, the correlation between the values of stochastic process at different time points, say $F(t)$ and $F(s)$, weaken as the distance $|t-s|$ increases; when the correlations drop to zero, it means that $F(t)$ and $F(s)$ become statistically independent, and thus, the cumulant $\overline{\cdots F(t) \cdots F(s) \cdots}$ must vanish because it is statistically irreducible. Since the irreducibility is a consequence of the cumulant structure (see, e.g., Sec. 3.6.2), due to parallelism, it also applies to quasi-cumulants. This explains the factor $\lambda^{2k} t \tau_c^{2k-1}$ which comes for the $2k$-fold time integrations that are being restricted by the range of correlations withing the central qumulants constituting the cSQ. Note that this estimation would not work for non-central SQs because the contributions form the average $\langle F(s) \rangle$ interwoven into (non-central) qumulants would not restrict the time-integrals.

The second factor in Eq (239), $(\tau_c/\tau_S)^{k-1}$, estimates the contribution from commutators between $\mathcal{W}_c$'s at different times that appear in $\mathcal{L}_c^{(2k)}$'s [see, e.g., Eq. (232)]; these commutators have the following general form,

$$\sum_{f_2, \bar{f}_2} \sum_{f_3, \bar{f}_3} \tilde{Q}_F^{(2)}(f_1, \bar{f}_1, s_1; f_3, \bar{f}_3, s_3) \tilde{Q}_F^{(2)}(f_2, \bar{f}_2, s_2; f_4, \bar{f}_4, s_4) [\mathcal{W}_c(f_2, \bar{f}_2, s_2), \mathcal{W}_c(f_3, \bar{f}_3, s_3)]. \quad (242)$$

Of course, the distance between time arguments of $\mathcal{W}_c$ in question ($|s_2 - s_3|$ in the above example) is capped by $\tau_c$ due to the range of correlations in $E$ that we have discussed previously. Naturally, if the central picture does not change appreciably on the time-scale $\tau_c$, i.e.

$$\mathcal{W}_c(f, \bar{f}, s + \tau_c) = \mathcal{U}_S'(s, s + \tau_c) \mathcal{W}_c(f, \bar{f}, s) \mathcal{U}_S'(s + \tau_c, s) \approx \mathcal{W}_c(f, \bar{f}, s), \quad (243)$$

then the commutator will tend to zero because $\mathcal{W}_c$ at $s_2$ and $s_3$ will, basically, be the same super-operator. Here, we want to estimate the magnitude of the commutator when the change is small but still finite. To this end, we assume $E$ is *stationary*, i.e. $[\hat{\rho}_E, \hat{H}_E] = 0$, so that $\langle F(t) \rangle$ is time-independent, and thus, so is $\hat{H}_S'$; then, the generator of the central picture has spectral decomposition

$$\mathcal{H}_S' = \sum_{a,b} (\epsilon_a - \epsilon_b) |a\rangle\langle b| \operatorname{tr}_S (|b\rangle\langle a| \bullet), \quad (244)$$

defined by the modified energy levels of $S$, $\hat{H}_S'|a\rangle = \epsilon_a |a\rangle$. Now, let $\Omega$ be the maximal spacing between the levels, i.e.

$$\Omega = \max_{a,b} |\epsilon_a - \epsilon_b| \equiv \frac{2\pi}{\tau_S}, \quad (245)$$

which makes $\tau_S$ the fastest time scale associated with the central picture. The assumption that the central picture changes very little can be quantified as $\Omega \tau_c \ll 1$ or $\tau_c \ll \tau_S$, and it implies that the evolution of super-operators $\mathcal{W}_c$ on time-scale $\tau_c$ can be approximated with the lowest-order correction,

$$[\mathcal{W}_c(s + \tau_c), \mathcal{W}_c(s)] \approx \left[\mathcal{W}_c(s) + \tau_c \frac{d\mathcal{W}_c(s)}{ds}, \mathcal{W}_c(s)\right] = [-i\tau_c[\mathcal{H}_S', \mathcal{W}_c(s)], \mathcal{W}_c(s)]$$
$$= -i\tau_c \{\mathcal{H}_S', \mathcal{W}_c(s)^2\}, \quad (246)$$

where we used the fact that $\mathcal{U}_S'$ is a time-ordered exp generated by $\mathcal{H}_S'$. The matrix elements, $\mathcal{A}_{ab,a'b'} = \operatorname{tr}(|b\rangle\langle a| \mathcal{A} |a'\rangle\langle b'|)$, of the commutator then read

$$|([\mathcal{W}_c(s + \tau_c), \mathcal{W}_c(s)])_{ab,a'b'}| \approx \tau_c |(\epsilon_a - \epsilon_b + \epsilon_{a'} - \epsilon_{b'})(\mathcal{W}_c(s)^2)_{ab,a'b'}| \sim \Omega \tau_c \sim \frac{\tau_c}{\tau_S}, \quad (247)$$

which gives us our estimate.

A more advanced approximations to the cSQ series can be obtained with *diagramatic* techniques. The idea is to represent the series, and cSQs themselves, as a sum of irreducible diagrams. These are highly sophisticated techniques because inventing the set of rules for translating mathematical formulas into diagrams is more of an art form than an engineering problem; how one deals with it depends on the particular context and one's skill, intuition and overall technical aptitude. Historical examples show that a well formulated diagram rules can potentially expose some normally hidden patterns that could be then exploited to, e.g., analytically sum up a whole sub-series of diagrams. Such finds are the Holy Grail of diagramatic methods, as they allow to incorporate corrections of all orders of magnitudes.

# 5 Master equation

## 5.1 Completely positive maps

As we have defined it here, the objective of the open system theory is to find an efficient and accurate description of the dynamics of system $S$ induced by its coupling with the environment $E$. When the theory is viewed from this perspective, the central problem to solve is to derive from "first principles" the dynamical map that encapsulates all the relevant influence $E$ exerts onto $S$ and the effects the dynamics has on the state are simply the consequence of the map's action. Then, the main practical concern is to achieve the computability of the map while maintaining a high degree of accuracy and, simultaneously, minimizing the necessary input of information about $E$.

A different way to interpret the goals of open system theory is to focus on the effects dynamical maps have on the state of $S$ rather than the physical origins of the map. In such a framework, instead of deriving the dynamical map from first principles, one's objective is to determine what kind of map, treated as an abstract transformation, produces a desired effect on a state of system $S$. There are two main practical concerns regarding this approach. The first is to define the constraints on the form of an abstract map so that it could, at least in principle, correspond to some real dynamical map induced by interactions with actual physical environment. The second is to find a convenient parametrization of the valid abstract maps that would allow one to clearly define the expected effects it has on the states.

The industry standard answer to the first point is the so-called *Kraus representation*; consider the following relation,

$$\langle \mathcal{U} \rangle(t,0) = \mathrm{tr}_E\left(e^{-it[\hat{H}_{SE}, \bullet]} \bullet \otimes \hat{\rho}_E\right) = \mathrm{tr}_E\left(e^{-it\hat{H}_{SE}} \bullet \otimes \left(\sum_i \rho_j |j\rangle\langle j|\right) e^{it\hat{H}_{SE}}\right)$$

$$= \sum_{j,j'} \sqrt{\rho_j} \langle j'|e^{-it\hat{H}_{SE}}|j\rangle \bullet \langle j|e^{it\hat{H}_{SE}}|j'\rangle \sqrt{\rho_j} \equiv \sum_{j,j'} \hat{K}_{jj'} \bullet \hat{K}_{jj'}^\dagger, \quad (248)$$

where $B_E = \{|j\rangle\}_j$ is a basis in $E$ ket-space made out of eigenkets of $\hat{\rho}_E$ and the *Kraus operators* are defined as

$$\hat{K}_{jj'} = \sqrt{\rho_j} \langle j'|e^{-it\hat{H}_{SE}}|j\rangle = \sum_{n,n'} \sqrt{\rho_j} \left(\langle n'| \otimes \langle j'|e^{-it\hat{H}_{SE}}|n\rangle \otimes |j\rangle\right)|n'\rangle\langle n|$$

$$= \sum_{n,n'} |n'\rangle \sqrt{\rho_j} \left(e^{-it\hat{H}_{SE}}\right)_{n'j',nj} \langle n|, \quad (249)$$

($\{|n\rangle\}_n$ is a basis in $S$). The key property of the set $\{\hat{K}_{jj'}\}_{j,j'}$ is the relation,

$$
\begin{aligned}
\sum_{j,j'} \hat{K}^\dagger_{jj'} \hat{K}_{jj'} &= \sum_{j,j'} \rho_j \sum_{n,n'} |n'\rangle \left(e^{-it\hat{H}_{SE}}\right)^*_{nj',n'j} \langle n| \sum_{n'',n'''} |n'''\rangle \left(e^{-it\hat{H}_{SE}}\right)_{n'''j',n''j} \langle n''| \\
&= \sum_{j,j'} \rho_j \sum_{n,n',n''} |n'\rangle \left(e^{it\hat{H}_{SE}}\right)_{n'j,nj'} \left(e^{-it\hat{H}_{SE}}\right)_{nj',n''j} \langle n''| \\
&= \sum_j \rho_j \sum_{n',n''} |n'\rangle \left(e^{it\hat{H}_{SE}} e^{-it\hat{H}_{SE}}\right)_{n'j,n''j} \langle n''| = \sum_j \rho_j \sum_{n',n''} |n'\rangle \delta_{n',n''} \delta_{j,j} \langle n''| \\
&= \left(\sum_j \rho_j\right) \sum_{n'} |n'\rangle\langle n'| = \hat{1} \,. 
\end{aligned}
\tag{250}
$$

The representation of the dynamical map with the sum over Kraus operators is not unique, however, e.g., we could have picked a different basis $B'_E = \{|\mu\rangle\}_\mu$ to represent the partial trace,

$$
\mathrm{tr}_E(\hat{A}_{SE}) = \sum_\mu \langle \mu | \hat{A}_{SE} | \mu \rangle \,,
$$

or a different decomposition of the density matrix,

$$
\hat{\rho}_E = \sum_\alpha p_\alpha |\Psi_\alpha\rangle\langle\Psi_\alpha| \,,
$$

($p_\alpha \geqslant 0$ and $\sum_\alpha p_\alpha = 1$, but the kets $\{|\Psi_\alpha\rangle\}_\alpha$ are not necessarily mutually orthogonal, see Sec. 2.5), to obtain a different set of Kraus operators,

$$
\hat{K}_{\alpha\mu} = \sqrt{p_\alpha} \langle \mu | e^{-it\hat{H}_{SE}} | \Psi_\alpha \rangle \,, \quad \sum_{\alpha,\mu} \hat{K}^\dagger_{\alpha\mu} \hat{K}_{\alpha\mu} = \hat{1} \,,
\tag{251}
$$

and a different representation of the same dynamical map. In summary, any dynamical map *evaluated at a specific time $t$* has Kraus representation(s),

$$
\langle \mathcal{U} \rangle(t,0) = \sum_{j,j'} \hat{K}_{jj'} \bullet \hat{K}^\dagger_{jj'} = \sum_{\alpha,\mu} \hat{K}_{\alpha\mu} \bullet \hat{K}^\dagger_{\alpha\mu} = \dots
\tag{252}
$$

Conversely, according to the *Naimark's dilation theorem*, given a set of operators $\{\hat{K}_m\}_m$ in $S$ that satisfy $\sum_m \hat{K}^\dagger_m \hat{K}_m = \hat{1}$, it is always possible to find an unitary operator $\hat{U}_{SS'}$ that acts in a composite space $SS'$, and a ket in $S'$ space $|\Psi_{S'}\rangle$, such that

$$
\sum_m \hat{K}_m \bullet \hat{K}^\dagger_m = \mathrm{tr}_{S'}\left(\hat{U}_{SS'} \bullet \otimes |\Psi_{S'}\rangle\langle\Psi_{S'}| \hat{U}^\dagger_{SS'}\right) \,.
\tag{253}
$$

Of course, in such a setting, $S'$ can be formally treated as the environment to $S$ and $\hat{U}_{SS'}$ as an evolution operator generated by a corresponding $SS'$ Hamiltonian over some evolution time. Even though the system $S'$ and the evolution operator $\hat{U}_{SS'}$ are only postulated here, they could exist in principle, and that is enough. Indeed, for as long as one only observes system $S$ after the map $\sum_m \hat{K}_m \bullet \hat{K}^\dagger_m$ produces its effects, it is impossible to verify if $S'$ is real or not.

Kraus representation is also an answer to the question of parametrization for the effects-oriented view of dynamical maps. In particular, for qubit system $S$, any operator—including Kraus operators—can be decomposed into linear combination of Pauli matrices. It is easy to categorize what kind of effects can be produced by Kraus representations made out of simple combinations, e.g., $\{\hat{K}_1 = \sqrt{(1+p)/2}\,\hat{1}, \hat{K}_2 = \sqrt{(1-p)/2}\,\hat{\sigma}_z\}$ results in

$$
\sum_{m=1}^2 \hat{K}_m \hat{\rho}_S \hat{K}^\dagger_m = \left(\frac{1+p}{2}\right)\hat{\rho}_S + \left(\frac{1-p}{2}\right)\hat{\sigma}_z \hat{\rho}_S \hat{\sigma}_z = \begin{bmatrix} \langle\uparrow|\hat{\rho}_S|\uparrow\rangle & p\langle\uparrow|\hat{\rho}_S|\downarrow\rangle \\ p\langle\downarrow|\hat{\rho}_S|\uparrow\rangle & \langle\downarrow|\hat{\rho}_S|\downarrow\rangle \end{bmatrix} \,.
\tag{254}
$$

Kraus representation made out of more complex combinations would, in a sense, combine those simple effects. Thus, there are only relatively few options for Kraus representations that can produce distinct and unique effects. Unfortunately, for non-qubit $S$, the categorization of possible Kraus representations by their effects is not so clear cut. Historically, this was not a big problem because, originally, the development of the open system theory as a distinct field was, almost wholly, motivated by the context of quantum computers and quantum information processing, and thus, non-qubit systems were not often considered.

Traditionally, valid abstract maps, that is, maps with Kraus representations, are referred to as *completely positive* (CP) maps; therefore, "CP map" can be used as a shorthand for "dynamical map in $S$ obtained as a partial trace of the total unitary map in $SE$, acting on an uncorrelated initial state with fixed $\hat{\rho}_E$ and variable $\hat{\rho}_S$." However, as one might have guessed, there is some history behind this very specific nomenclature. The concept of *complete positivity* arose as quintessentially effects-oriented way to solve the problem of constraints for abstract maps. Here, we will review the basics of reasoning that leads naturally to the definition of CP.

Consider a physical system $S$ that has its dynamics defined by the map $\mathcal{M}_t$, i.e. if $\hat{\rho}_S$ is an arbitrary initial state of $S$, then $\mathcal{M}_t \hat{\rho}_S$ is the state at later time $t > 0$, $\hat{\rho}_S(t)$. In what follows we fully embrace the effects-oriented reading of the theory, and thus, we are willfully ignoring the first-principles origin of the map as a reduced unitary evolution of the system coupled to its environment. In that case, we have to somehow deduce which properties we should demand of $\mathcal{M}_t$ so that it can function as a valid dynamical map. The most obvious first step is to make sure that the output operator $\mathcal{M}_t \hat{\rho}_S$ is a proper density matrix. Therefore, at the very least, the map $\mathcal{M}_t$ should preserve the trace and positivity of the input state,

1. $\mathcal{M}_t$ is *trace-preserving*, i.e. $\forall_{t>0} \ \mathrm{tr}_S(\mathcal{M}_t \hat{\rho}_S) = \mathrm{tr}_S(\hat{\rho}_S)$;

2. $\mathcal{M}_t$ is *positive*, i.e. if $\hat{\rho}_S > 0$, then $\forall_{t>0} \ \mathcal{M}_t \hat{\rho}_S > 0$.

However, positivity and trace-preservation (PTP) alone turns out to be insufficient. To see what is missing, we have to take into consideration other physical systems governed by their respective dynamical laws. In particular, it is feasible to have two systems—our initial $S$ and the other $S'$—such that, starting from $t = 0$, their dynamical laws are independent of each other, i.e. if $\mathcal{M}_t$ is the PTP map of $S$ and $\mathcal{M}'_t$ is the PTP map of $S'$, then the total dynamics for $t > 0$ is described by $\mathcal{M}_t \otimes \mathcal{M}'_t$. We can leave unspecified what happened in the past, t<0; for now, all we need to know is the initial (as in, $t = 0$) state of the total $SS'$ system, $\hat{\rho}_{SS'}$. Even though we are assuming the dynamical laws are independent, the total map should, at least, be PTP; after all, the composition of physical systems is a physical system, and thus, $\mathcal{M}_t \otimes \mathcal{M}'_t \hat{\rho}_{SS'}$ should result in a proper density matrix of the composite system $SS'$. Nothing seems amiss if we assume that $S$ and $S'$ also start as uncorrelated systems, $\hat{\rho}_{SS'} = \hat{\rho}_S \otimes \hat{\rho}_{S'}$; indeed, the trace is preserved,

$$\mathrm{tr}_{SS'}\left(\mathcal{M}_t \otimes \mathcal{M}'_t \hat{\rho}_S \otimes \hat{\rho}_{S'}\right) = \mathrm{tr}_S\left(\mathcal{M}_t \hat{\rho}_S\right)\mathrm{tr}_{S'}\left(\mathcal{M}'_t \hat{\rho}_{S'}\right) = \mathrm{tr}_S(\hat{\rho}_S)\mathrm{tr}_{S'}(\hat{\rho}_{S'}) = \mathrm{tr}_{SS'}\left(\hat{\rho}_S \otimes \hat{\rho}_{S'}\right);$$

and so is positivity,

$$\mathcal{M}_t \otimes \mathcal{M}'_t \hat{\rho}_S \otimes \hat{\rho}_{S'} = (\mathcal{M}_t \hat{\rho}_S) \otimes \left(\mathcal{M}'_t \hat{\rho}_{S'}\right) > 0,$$

because the outer product $\otimes$ of positive operators is positive. We can even allow for classical correlations between $S$ and $S'$, $\hat{\rho}_{SS'} = \sum_i p_i \hat{\rho}_S^{(i)} \otimes \hat{\rho}_{S'}^{(i)}$ ($p_i \geqslant 0$, $\sum_i p_i = 1$), and still everything looks fine. The inadequacy of PTP is exposed only once we introduce non-classical correlations—the quantum entanglement—between $S$ and $S'$, i.e. we set $\hat{\rho}_{SS'}$ that is not of the form $\sum_i p_i \hat{\rho}_S^{(i)} \otimes \hat{\rho}_{S'}^{(i)}$. We can demonstrate the breakdown of PTP as the definite constraint with

the famous *Peres-Horodecki criterion* for non-entangled states. The test is defined as follows: First, one constructs the following transformation in $SS'$,

$$\mathcal{M}_{SS'} = \bullet \otimes (\bullet^T), \tag{255}$$

where $(\bullet^T)\hat{A} = \hat{A}^T$ is the matrix transposition—note here that the identity $\bullet$ and the transposition $\bullet^T$ are both PTP transformations. Second, one applies $\mathcal{M}_{SS'}$ to the subject of the test, the $SS'$ state $\hat{\rho}_{SS'}$. In the last step, one verifies if the resultant output is a positive operator. As we have discussed above, if the tested state $\hat{\rho}_{SS'}$ was only classically correlated, than the product of PTP transformations would produce a positive output. Therefore, a non-positive output $\bullet \otimes (\bullet^T)\hat{\rho}_{SS'}$ implies that $\hat{\rho}_{SS'} \neq \sum_i p_i \hat{\rho}_S^{(i)} \otimes \hat{\rho}_{S'}^{(i)}$ (although, the converse is true only for qubit $S$, $S'$ or for qubit $S$ and qutrit $S'$). So, the PTP requirement alone is so transparently inadequate to define the constraints on abstract maps that its failure is used to witness non-classical correlations! Clearly, there is more than PTP that one should require of physical dynamical maps $\mathcal{M}_t$; one way to "fix" the failure point with entangled composite systems is to introduce the third condition:

3. $\mathcal{M}_t$ is such that $\mathcal{M}_{SS_d} = \mathcal{M}_t \otimes \bullet$, a transformation constructed to act in composite system $SS_d$ with $d$-dimensional $S_d$, is positive for any $d$, $t > 0$.

The PTP transformation $\mathcal{M}_t$ that also satisfies the above is called a CP map—this is the standard mathematical definition of complete positivity. Of course, a total map $\mathcal{M}_t \otimes \mathcal{M}_t'$ made out of CP maps will produce a proper output density matrix even if the initial state is entangled. Also, a product of CP maps is CP itself, so the condition of complete positivity is consistent in this sense (unlike maps that are only PTP).

The definition of complete positivity presented here (i.e. PTP plus the condition 3) is mathematically equivalent to a plethora of many different sets of conditions; a good portion of those alternative definitions make an explicit reference to entangled states (see [5] for detailed discussion on this subject). However, from the point of view of our current discussion, there is one alternative definition that is of particular interest: the Choi's theorem asserts that all CP maps have Kraus representation and all maps with Kraus representation are CP. This is the origin of the nomenclature; since CP is mathematically equivalent to Kraus representation, it is okay to call maps with Kraus representation "CP maps".

## 5.2 Time-dependent Kraus representations

The set of Kraus operators constructed according to the "first principles" blueprint (249) represents a "snapshot" of the dynamical map taken at certain point in time $t$. When one consistently follows the blueprint for all points in time,

$$\forall_{t>0} \ \hat{K}_{jj'}(t) \equiv \sqrt{\rho_j} \langle j' | e^{-it\hat{H}_{SE}} | j \rangle, \tag{256}$$

then these $t$-dependent Kraus representations can be logically conjoined together under one family parametrized by the passage of time, $\{\hat{K}_{jj'}(t)\}_{j,j';t \geqslant 0}$, such that $\forall_{j,j'} \hat{K}_{jj'}(s) \to \hat{K}_{jj'}(t)$ when $s \to t$. As a result, one naturally upgrades the collection of disjointed temporal snapshots of the dynamical map into the Kraus representation of the whole dynamics of the open system.

But what about the effects-oriented point of view? How to introduce time-dependence when Kraus representations are abstracted from any underlying physical $SE$ dynamics? Of course, without the underpinning of unitary evolution of the total system, there is no reason for any form of temporal relation between two abstract representations; such a relation can be established only by introducing additional constraints on the form of Kraus operators. One such constraint, that is known to lead to useful parametrization, is to postulate that the map

represented by the $t$-dependent family of Kraus operators $\{\hat{K}_m(t)\}_{m,t\geqslant 0}$ obeys the simplest form of the composition rule,

$$\left(\sum_m \hat{K}_m(t)\bullet\hat{K}_m^\dagger(t)\right)\left(\sum_m \hat{K}_m(s)\bullet\hat{K}_m^\dagger(s)\right) = \sum_m \hat{K}_m(t+s)\bullet\hat{K}_m^\dagger(t+s).\qquad(257)$$

As we have learned previously (e.g., Sec. 2.1), this composition rule implies time-independent dynamical equation (in fact, the two are equivalent),

$$\frac{d}{dt}\sum_m \hat{K}_m(t)\bullet\hat{K}_m^\dagger(t) = \mathcal{L}_{\text{GKLS}}\sum_m \hat{K}_m(t)\bullet\hat{K}_m^\dagger(t);\quad \sum_m \hat{K}_m(0)\bullet\hat{K}_m^\dagger(0) = \bullet,\qquad(258)$$

where the super-operator $\mathcal{L}_{\text{GKLS}}$ is constant and it acts as a generator of the map, i.e.

$$\sum_m \hat{K}_m(t)\bullet\hat{K}_m^\dagger(t) = e^{t\mathcal{L}_{\text{GKLS}}}.\qquad(259)$$

The enforced composition rule by itself does not tell us anything else about $\mathcal{L}_{\text{GKLS}}$. However, $\hat{K}_m(t)$'s also satisfy the standard Kraus condition, $\forall_{t\geqslant 0}\sum_m \hat{K}_m^\dagger(t)\hat{K}_m(t) = \hat{1}$, which, of course, constraints the allowed forms of the generator; in particular, it was shown by the trio of V. Gorini, A. Kossakowski, and G. Sudarshan [6] and independently by G. Lindblad [7] (GKLS in alphabetic order) that $\mathcal{L}_{\text{GKLS}}$ must be of the *GKLS form*:

$$\mathcal{L}_{\text{GKLS}} = -i[\hat{H},\bullet] + \sum_{\alpha,\beta}\gamma_{\alpha,\beta}\left(\hat{L}_\alpha\bullet\hat{L}_\beta^\dagger - \frac{1}{2}\{\hat{L}_\beta^\dagger\hat{L}_\alpha,\bullet\}\right),\qquad(260)$$

where $\hat{H}$ is an arbitrary hermitian operator, $\{\hat{L}_\alpha\}_\alpha$ is an arbitrary set of operators, and—this is key—$\gamma_{\alpha,\beta}$ is a positive matrix. The resultant dynamical equation for the density matrix of $S$,

$$\frac{d}{dt}\hat{\rho}_S(t) = -i[\hat{H},\hat{\rho}_S(t)] + \sum_{\alpha,\beta}\gamma_{\alpha,\beta}\left(\hat{L}_\alpha\hat{\rho}_S(t)\hat{L}_\beta^\dagger - \frac{1}{2}\{\hat{L}_\beta^\dagger\hat{L}_\alpha,\hat{\rho}_S(t)\}\right),\qquad(261)$$

is called the *GKLS equation* or simply the *master equation* (see, e.g., [8] for a simplified proof).

The theorem that super-operator in the GKLS form generates CP map can be considered as the foundational result for the theory of open systems. It would not be an exaggeration to say that the discovery of GKLS form was the moment when the open system theory was born as a distinct field in physics. Indeed, see [9, 10] as an example of works on open systems where GKLS maps are treated as the central concept of the theory.

## 5.3 Master equation as a physical model

At the time, the result obtained by GKLS captured imagination of many; a mathematical characterization of valid dynamical maps (at least, of a class of maps with constant generators) is a supremely strong result indeed. Of course, people started to wonder how this result would apply to physics: can it be argued that dynamical maps in GKLS form are sufficiently accurate models of real open systems? And if they are, then what is the physical meaning behind operators $\hat{L}_\alpha$ and the matrix $\gamma_{\alpha,\beta}$? Essentially, this is a question of link between the effect-oriented and first principles approaches.

The first person to answer these questions was E.B. Davies, who demonstrated that the dynamics induced by the coupling with a generic environment tends to the GKLS form when one can justify the application of a sequence of intuitive approximations; here, we recreate their reasoning using the modern tools we have developed previously. To begin with, we assume that $E$ is initialized in the state of thermal equilibrium, $\hat{\rho}_E = e^{-\beta\hat{H}_E}/Z$—the immediate consequence is that $\hat{H}_S' = \hat{H}_S + \lambda\langle F\rangle\hat{V}$ is constant. Then, we apply the following approximations,

1. **Born approximation**: relaying on the estimate (239), we assume the *weak coupling* $\lambda \tau_c \ll 1$ and the separation of time scales $\tau_c \ll \tau_S$, so that we can truncate the cSQ series at the lowest order,

$$\langle \mathcal{U} \rangle_c(t, 0) \approx \langle \mathcal{U} \rangle^{(2)}(t, 0) = T_{\mathcal{L}_c^{(k)}} e^{-\lambda^2 \int_0^t \mathcal{L}_c^{(2)}(s) ds}. \tag{262}$$

2. **Markov approximation**: First, we exploit the stationarity of $E$, brought about by the initial thermal state, to shift the integration variable in the remaining cSQ,

$$\begin{aligned}
\int_0^t \mathcal{L}_c^{(2)}(s) ds &= \int_0^t ds [\hat{V}_c(s), \bullet] \int_0^s du \left( \langle\langle F(s) F(u) \rangle\rangle \hat{V}_c(u) \bullet - \bullet \hat{V}_c(u) \langle\langle \bar{F}(s) \bar{F}(u) \rangle\rangle \right) \\
&= \int_0^t ds \, \mathcal{U}_S'(-s)[\hat{V}, \bullet] \\
&\quad \times \int_0^s du \left( \langle\langle F(s-u) F \rangle\rangle \hat{V}_c(u-s) \bullet - \bullet \hat{V}_c(u-s) \langle\langle F(s-u) F \rangle\rangle^* \right) \mathcal{U}_S'(s) \\
&= \int_0^t ds \, \mathcal{U}_S'(-s)[\hat{V}, \bullet] \int_0^s d\tau \left( \langle\langle F(\tau) F \rangle\rangle \hat{V}_c(-\tau) \bullet - \bullet \hat{V}_c(-\tau) \langle\langle F(\tau) F \rangle\rangle^* \right) \mathcal{U}_S'(s),
\end{aligned} \tag{263}$$

where we used the relations

$$\begin{aligned}
\langle\langle \bar{F}(s) \bar{F}(u) \rangle\rangle &= \langle\langle F(s) F(u) \rangle\rangle^*, \\
\mathcal{U}_S'(s) &= e^{-is\mathcal{H}_S'} = e^{-is[\hat{H}_S', \bullet]} = \hat{U}_S'(s) \bullet \hat{U}_S'(-s), \\
\hat{U}_S'(s) &= e^{-is\hat{H}_S'} = e^{-is(\hat{H}_S + \lambda \langle F \rangle \hat{V})}, \\
[\hat{V}_c(s), \bullet] &= \mathcal{U}_S'(-s)[\hat{V}, \bullet] \mathcal{U}_S'(s), \\
\hat{V}_c(s-u) \bullet &= \mathcal{U}_S'(s)(\hat{V}_c(u) \bullet) \mathcal{U}_S'(-s), \\
\bullet \hat{V}_c(s-u) &= \mathcal{U}_S'(s)(\bullet \hat{V}_c(u)) \mathcal{U}_S'(-s), \\
\bullet &= \mathcal{U}_S'(-s) \mathcal{U}_S'(s).
\end{aligned}$$

Then, we apply the approximation where we extend the upper limit of the integral over $\tau$ to infinity,

$$\begin{aligned}
\int_0^t \mathcal{L}_c^{(2)}(s) ds &\approx \int_0^t ds \, \mathcal{U}_S'(-s)[\hat{V}, \bullet] \\
&\quad \times \left( \left( \int_0^\infty \langle\langle F(\tau) F \rangle\rangle \hat{V}_c(-\tau) d\tau \right) \bullet - \bullet \left( \int_0^\infty \langle\langle F(\tau) F \rangle\rangle \hat{V}_c(-\tau) d\tau \right)^\dagger \right) \mathcal{U}_S'(s) \\
&\equiv \int_0^t ds \, \mathcal{U}_S'(-s)[\hat{V}, \hat{\Lambda} \bullet - \bullet \hat{\Lambda}^\dagger] \mathcal{U}_S'(s) \equiv \int_0^t \mathcal{L}_R(s) ds, \tag{264}
\end{aligned}$$

where we have defined, now time-independent, operator

$$\hat{\Lambda} = \int_0^\infty \langle\langle F(\tau) F \rangle\rangle \hat{V}_c(-\tau) d\tau. \tag{265}$$

Given that we are already assuming the weak coupling, $\lambda \tau_c \ll 1$, the approximation is justified under the condition that $t$ is much larger than the correlation time $\tau_c$. When

$t \gg \tau_c$, the effective upper limit of the integral over $\tau$ is not $t$ but rather $\tau_c$—the range of qumulants. Then, we can use a crude estimate $|\langle\!\langle F(\tau)F\rangle\!\rangle| \sim \theta(\tau_c - \tau)$, and we write

$$\lambda^2 \left| \left\| \int_0^t \mathcal{L}_c^{(2)}(s)ds \right\| - \left\| \int_0^t \mathcal{L}_R(s)ds \right\| \right|$$

$$\sim \lambda^2 \left| \int_0^t ds \left( \int_0^s \theta(\tau_c - \tau)d\tau - \int_0^\infty \theta(\tau_c - \tau)d\tau \right) \right|$$

$$= \lambda^2 \left| \left( t\tau_c - \frac{1}{2}\tau_c^2 \right) - t\tau_c \right| = \frac{1}{2}\lambda^2\tau_c^2 \ll 1.$$

Hence, when $t \gg \tau_c$ and $\lambda\tau_c \ll 1$, the difference between the original cSQ and the approximating super-operator is negligible.

If we were to stop here, we would obtain the so-called *Redfield equation* for density matrix of $S$,

$$\frac{d}{dt}\hat{\rho}_S(t) = \frac{d}{dt}\langle\mathcal{U}\rangle(t,0)\hat{\rho}_S = \frac{d}{dt}\mathcal{U}_S'(t)\langle\mathcal{U}\rangle_c(t,0)\hat{\rho}_S$$

$$= -i\mathcal{H}_S'\mathcal{U}_S'(t)\langle\mathcal{U}\rangle_c(t,0)\hat{\rho}_S + \mathcal{U}_S'(t)\frac{d}{dt}\langle\mathcal{U}\rangle_c(t,0)\hat{\rho}_S$$

$$\approx -i\mathcal{H}_S'\langle\mathcal{U}\rangle(t,0)\hat{\rho}_S - \lambda^2\mathcal{U}_S'(t)\mathcal{U}_S'(-t)[\hat{V}, \hat{\Lambda}\bullet - \bullet\hat{\Lambda}^\dagger]\mathcal{U}_S'(t)\langle\mathcal{U}_c\rangle(t,0)\hat{\rho}_S$$

$$= -i[\hat{H}_S + \lambda\langle F\rangle\hat{V}, \hat{\rho}_S(t)] - \lambda^2[\hat{V}, \hat{\Lambda}\hat{\rho}_S(t) - \hat{\rho}_S(t)\hat{\Lambda}^\dagger], \quad (266)$$

and its formal solution reads

$$\hat{\rho}_S(t) = \exp\left(-it[\hat{H}_S + \lambda\langle F\rangle\hat{V}, \bullet] - \lambda^2 t[\hat{V}, \hat{\Lambda}\bullet - \bullet\hat{\Lambda}^\dagger]\right)\hat{\rho}_S. \quad (267)$$

3. **Secular approximation** (also known as **rotating wave approximation**): we introduce the frequency decomposition of the coupling operator obtained using the eigenkets of $\hat{H}_S' = \sum_a \epsilon_a |a\rangle\langle a|$,

$$\hat{V}_c(s) = \hat{U}_S'(-s)\hat{V}\hat{U}_S'(s) = \sum_{a,b} \hat{U}_S'(-s)|a\rangle\langle a|\hat{V}|b\rangle\langle b|\hat{U}_S'(s)$$

$$= \sum_{a,b} e^{is(\epsilon_a - \epsilon_b)}|a\rangle V_{ab}\langle b| = \sum_\omega e^{is\omega}\sum_{a,b}\delta(\omega - \epsilon_a + \epsilon_b)|a\rangle V_{ab}\langle b|$$

$$\equiv \sum_\omega e^{is\omega}\hat{V}_\omega = \sum_\omega e^{-is\omega}\hat{V}_\omega^\dagger. \quad (268)$$

We use the frequency decomposition to transform the previously obtained operators $\hat{\Lambda}$,

$$\hat{\Lambda} = \int_0^\infty d\tau \langle\!\langle F(\tau)F\rangle\!\rangle\hat{V}_c(-\tau) = \sum_\omega \left( \int_0^\infty \langle\!\langle F(\tau)F\rangle\!\rangle e^{-i\omega\tau}d\tau \right)\hat{V}_\omega \equiv \sum_\omega \gamma_\omega\hat{V}_\omega, \quad (269)$$

as well as the super-qumulant,

$$\int_0^t \mathcal{L}_R(s)ds = \int_0^t ds\,\mathcal{U}_S'(-s)[\hat{V}, \hat{\Lambda}\bullet - \bullet\hat{\Lambda}^\dagger]\mathcal{U}_S'(s)$$

$$= \int_0^t ds\left(\hat{U}_S'(-s)\hat{V}\hat{U}_S'(s)\hat{U}_S'(-s)\hat{\Lambda}\hat{U}_S'(s)\bullet - \hat{U}_S'(-s)\hat{\Lambda}\hat{U}_S'(s)\bullet\hat{U}_S'(-s)\hat{V}\hat{U}_S'(s)\right.$$

$$\left. + \bullet\hat{U}_S'(-s)\hat{\Lambda}^\dagger\hat{U}_S'(s)\hat{U}_S'(-s)\hat{V}\hat{U}_S'(s) - \hat{U}_S'(-s)\hat{V}\hat{U}_S'(s)\bullet\hat{U}_S'(-s)\hat{\Lambda}^\dagger\hat{U}_S'(s)\right)$$

$$= \sum_{\omega,\omega'}\left(\gamma_\omega\hat{V}_{\omega'}^\dagger\hat{V}_\omega\bullet + \bullet\hat{V}_{\omega'}^\dagger\hat{V}_\omega\gamma_{\omega'}^* - (\gamma_\omega + \gamma_{\omega'}^*)\hat{V}_\omega\bullet\hat{V}_{\omega'}^\dagger\right)\int_0^t e^{is(\omega - \omega')}ds. \quad (270)$$

As a result, we have confined the totality of $t$-dependence to phase factors.

The approximation we are about to implement is based on the observation that when the frequencies match up, $\omega = \omega'$, the phase factor simplifies to $t$,

$$\int_0^t e^{is(\omega-\omega')}ds \bigg|_{\omega=\omega'} = \int_0^t ds = t\,, \tag{271}$$

which is a very different behavior than the mismatched case, $\omega \neq \omega'$,

$$\int_0^t e^{i(\omega-\omega')s}ds = e^{\frac{1}{2}i(\omega-\omega')t}\, t \operatorname{sinc}\left[\frac{(\omega-\omega')t}{2}\right]. \tag{272}$$

Therefore, the phase factors with matching frequencies—*on-resonance* terms—overshadow the *off-resonance* terms on a sufficiently long time scale,

$$\frac{\left|\int_0^t e^{is(\omega-\omega')}ds\right|}{\left|\int_0^t ds\right|} = \left|\operatorname{sinc}\left[\frac{(\omega-\omega')t}{2}\right]\right| \xrightarrow{|\omega-\omega'|t \gg 2\pi} 0\,. \tag{273}$$

To obtain this effect for all off-resonance terms, $t$ has to be long in comparison to the slowest time-scale in $S$, or

$$\min_{\omega\neq\omega'}|\omega-\omega'|t \equiv \frac{2\pi}{T_S}t \gg 2\pi \;\Rightarrow\; t \gg T_S\,. \tag{274}$$

When this condition is in effect, then all "off-diagonal" terms where $\omega \neq \omega'$ can be neglected in comparison with the "diagonal" $\omega = \omega'$ terms,

$$\int_0^t \mathcal{L}_R(s)ds \approx t\sum_\omega \left(\gamma_\omega \hat{V}_\omega^\dagger \hat{V}_\omega \bullet + \bullet \hat{V}_\omega^\dagger \hat{V}_\omega \gamma_\omega^* - \left(\gamma_\omega + \gamma_\omega^*\right)\hat{V}_\omega \bullet \hat{V}_\omega^\dagger\right)$$

$$= it\left[\sum_\omega \operatorname{Im}\{\gamma_\omega\}\hat{V}_\omega^\dagger \hat{V}_\omega, \bullet\right] - t\sum_\omega 2\operatorname{Re}\{\gamma_\omega\}\left(\hat{V}_\omega \bullet \hat{V}_\omega^\dagger - \frac{1}{2}\{\hat{V}_\omega^\dagger \hat{V}_\omega, \bullet\}\right)$$

$$\equiv -t\mathcal{L}_D\,. \tag{275}$$

Now note the following: (i) the operator $\sum_\omega \operatorname{Im}\{\gamma_\omega\}\hat{V}_\omega^\dagger \hat{V}_\omega$ is hermitian; (ii) the coefficients $\operatorname{Re}\{\gamma_\omega\}$ are positive because

$$2\operatorname{Re}\{\gamma_\omega\} = \int_0^\infty \left(\langle\!\langle F(\tau)F\rangle\!\rangle e^{-i\omega\tau} + \langle\!\langle F(\tau)F\rangle\!\rangle^* e^{i\omega\tau}\right)d\tau$$

$$= \int_0^\infty \langle\!\langle F(\tau)F\rangle\!\rangle e^{-i\omega\tau}d\tau + \int_{-\infty}^0 \langle\!\langle F(-\tau)F\rangle\!\rangle^* e^{-i\omega\tau}d\tau$$

$$= \int_0^\infty \langle\!\langle F(\tau)F\rangle\!\rangle e^{-i\omega\tau}d\tau + \int_{-\infty}^0 \langle\!\langle F(\tau)F\rangle\!\rangle e^{-i\omega\tau}d\tau$$

$$= \int_{-\infty}^\infty \langle\!\langle F(\tau)F\rangle\!\rangle e^{-i\omega\tau}d\tau = \int_{-\infty}^\infty [C_F(\tau,0) - iK_F(\tau,0)]e^{-i\omega\tau}d\tau$$

$$= S(\omega) - i\kappa(\omega) = S(\omega)\left(1 + \frac{e^{-\beta\omega}-1}{e^{-\beta\omega}+1}\right) = \frac{2S(\omega)}{e^{\beta\omega}+1}\,, \tag{276}$$

and the spectral density $S(\omega)$ cannot be negative or, otherwise, it would violate the second law of thermodynamics (we used the fluctuation-dissipation theorem to eliminate susceptibility $\kappa(\omega)$, see assignment 36 of Sec. 4.7). Note that it is possible to prove the positivity of $\operatorname{Re}\{\gamma_\omega\}$ without resorting to physical arguments [8].

Thus, the approximated map we get at the end,

$$\langle \mathcal{U} \rangle_c(t,0) \approx e^{\lambda^2 t \mathcal{L}_D}, \tag{277}$$

has the GKLS form; indeed, we see that $\lambda^2 \mathcal{L}_D = \mathcal{L}_{\text{GKLS}}$ when we identify

$$\hat{H} \longleftrightarrow -\lambda^2 \sum_\omega \text{Im}\{\gamma_\omega\} \hat{V}_\omega^\dagger \hat{V}_\omega,$$

$$\{\hat{L}_\alpha\}_\alpha \longleftrightarrow \{\hat{V}_\omega\}_\omega,$$

$$\gamma_{\alpha,\beta} \longleftrightarrow \gamma_{\omega,\omega'} = 2\lambda^2 \text{Re}\{\gamma_\omega\} \delta_{\omega,\omega'}.$$

The matrix $\gamma_{\omega,\omega'}$ is, of course, positive because its eigenvalues, $2\lambda^2 \text{Re}\{\gamma_\omega\}$, are non-negative numbers.

The corresponding master equation for the open system density matrix,

$$\begin{aligned}
\frac{d}{dt}\hat{\rho}_S(t) &= \frac{d}{dt}\mathcal{U}_S'(t) e^{\lambda^2 t \mathcal{L}_D}\hat{\rho}_S = -i\mathcal{H}_S'\hat{\rho}_S(t) + \lambda^2 \mathcal{U}_S'(t)\mathcal{L}_D\mathcal{U}_S'(-t)\hat{\rho}_S(t) \\
&= \left(-i\mathcal{H}_S' + \lambda^2 \mathcal{L}_D\right)\hat{\rho}_S(t) \\
&= -i\left[\hat{H}_S + \lambda\langle F\rangle\hat{V} - \lambda^2\sum_\omega \text{Im}\{\gamma_\omega\}\hat{V}_\omega^\dagger\hat{V}_\omega,\ \hat{\rho}_S(t)\right] \\
&\quad + \lambda^2\sum_\omega \text{Re}\{\gamma_\omega\}\left(2\hat{V}_\omega\hat{\rho}_S(t)\hat{V}_\omega^\dagger - \{\hat{V}_\omega^\dagger\hat{V}_\omega, \hat{\rho}_S(t)\}\right),
\end{aligned} \tag{278}$$

is called the *Davies master equation*.

**Assignment 37 (Solution)** *Show that*

$$\mathcal{U}_S'(t)\mathcal{L}_D\mathcal{U}_S'(-t) = \mathcal{L}_D.$$

**Assignment 38 (Solution)** *Assuming that the Hamiltonian $\hat{H}_S' = \sum_a \epsilon_a |a\rangle\langle a|$ is non-degenerate,*

$$\forall_{a\neq b}\epsilon_a \neq \epsilon_b,$$

*derive the Pauli master equation: the rate equation obtained from Davies equation by restricting it to the diagonal elements of the density matrix,*

$$\begin{aligned}
\frac{d}{dt}\langle a|\hat{\rho}_S(t)|a\rangle &= \sum_b M_{a,b}\langle b|\hat{\rho}_S(t)|b\rangle \\
&= \sum_b \lambda^2\left(\gamma_{a,b} - \delta_{a,b}\sum_c \gamma_{c,a}\right)\langle b|\hat{\rho}_S(t)|b\rangle,
\end{aligned}$$

*with $\gamma_{a,b} = 2|V_{ab}|^2 S(\omega_{ab})/(e^{\beta\omega_{ab}} + 1)$.*

Note that the thermal equilibrium state with respect to $\hat{H}_S'$,

$$\hat{p} = \frac{e^{-\beta\hat{H}_S'}}{\text{tr}_S\left(e^{-\beta\hat{H}_S'}\right)} = \frac{1}{Z}\sum_a e^{-\beta\epsilon_a}|a\rangle\langle a|,$$

is a steady state solution to Pauli equation. This can be easily showed by using the *detailed balance* property of the transition matrix,

$$\gamma_{b,a} = |V_{ba}|^2\frac{2S(\omega_{ba})}{e^{\beta\omega_{ba}} + 1} = e^{-\beta\omega_{ba}}|V_{ab}|^2\frac{2S(-\omega_{ab})}{1 + e^{-\beta\omega_{ba}}} = e^{\beta\omega_{ab}}|V_{ab}|^2\frac{2S(\omega_{ab})}{e^{\beta\omega_{ab}} + 1} = e^{\beta\omega_{ab}}\gamma_{a,b},$$

to transform the Pauli equation to

$$\frac{d}{dt}\langle a|\hat{\rho}_S(t)|a\rangle = \lambda^2 \langle a|\mathcal{L}_D \hat{\rho}_S(t)|a\rangle$$

$$= \lambda^2 \sum_b \left(\gamma_{a,b}\langle b|\hat{\rho}_S(t)|b\rangle - \gamma_{b,a}\langle a|\hat{\rho}_S(t)|a\rangle\right)$$

$$= \lambda^2 \sum_b \gamma_{a,b}\left(\langle b|\hat{\rho}_S(t)|b\rangle - e^{\beta\omega_{ab}}\langle a|\hat{\rho}_S(t)|a\rangle\right).$$

Then, when we substitute $\hat{p}$ for $\hat{\rho}_S(t)$ on the RHS we get

$$\lambda^2 \langle a|\mathcal{L}_D\hat{p}|a\rangle = \frac{\lambda^2}{Z}\sum_b \gamma_{a,b}\left(e^{-\beta\epsilon_b} - e^{\beta(\epsilon_a-\epsilon_b)}e^{-\beta\epsilon_a}\right) = 0.$$

In addition, it can be showed that any solution of Pauli master equation satisfies

$$\lim_{t\to\infty}\langle a|\hat{\rho}_S(t)|a\rangle = \langle a|\hat{p}|a\rangle.$$

Moreover, with some additional assumptions about operator $\hat{V}$, one can prove that *all* solutions of the Davies equation tend to the equilibrium state as the time goes to infinity, which gives some insight into the process of thermalization. We will talk about this topic in more detail later in Chapter 6.

## 5.4 Nakajima–Zwanzig equation

In previous section we have derived the Davies master equation starting from the super-qumulant expansion; this is not a standard textbook route, however. The typical textbook derivation (e.g., see [2]) starts with the von Neumann equation for the density matrix of the total system,

$$\frac{d}{dt}\hat{R}_{SE}(t) = -i\lambda[\hat{V}_c(t)\otimes\hat{X}_I(t), \hat{R}_{SE}(t)], \tag{279}$$

where $\hat{R}_{SE}(t) = \mathcal{U}'_S(-t)\otimes e^{it[\hat{H}_E,\bullet]}\hat{\rho}_{SE}(t)$ is the total density matrix in the central picture. The equation can be formally integrated with the assumed initial condition $\hat{R}_{SE}(0) = \hat{\rho}_S\otimes\hat{\rho}_E$,

$$\hat{R}_{SE}(t) = \hat{\rho}_S\otimes\hat{\rho}_E - i\lambda\int_0^t [\hat{V}_c(s)\otimes\hat{X}_I(s), \hat{R}_{SE}(s)]ds. \tag{280}$$

This formal solution is then substituted back into RHS of the equation and the partial trace over $E$ is taken,

$$\frac{d}{dt}\hat{\rho}_c(t) = -\lambda^2\int_0^t \text{tr}_E\left([\hat{V}_c(t)\otimes\hat{X}_I(t), [\hat{V}_c(s)\otimes\hat{X}_I(s), \hat{R}_{SE}(s)]]\right)ds, \tag{281}$$

where, of course, $\text{tr}_E(\hat{R}_{SE}(t)) = \hat{\rho}_c(t)$. The issue is that the above formula is *not* a dynamical equation because there is no $\hat{\rho}_c$ on the RHS; before we can proceed, something has to be done about it.

Here comes the pivotal—and the most suspect—moment of the derivation: to "close" the equation it is assumed that the total density matrix preserves the product form at all times, plus, that the state on the $E$-side remains static,

$$\hat{R}_{SE}(s) \approx \hat{\rho}_c(s)\otimes\hat{\rho}_E. \tag{282}$$

This step is typically referred to as the *Born-Markov approximation*—we will discuss later how one might try to justify this ansatz and if "approximation" is a correct descriptor for it. The ansatz itself, of course, closes the equation,

$$\frac{d}{dt}\hat{\rho}_c(t) \approx -\lambda^2 \int_0^t \mathrm{tr}_E\left([\hat{V}_c(t)\otimes\hat{X}_I(t),[\hat{V}_c(s)\otimes\hat{X}_I(s),\hat{\rho}_c(s)\otimes\hat{\rho}_E]]\right)ds$$

$$= -\lambda^2 \int_0^t \langle\!\langle \mathcal{W}_c(X(t),\bar{X}(t),t)\mathcal{W}_c(X(s),\bar{X}(s),s)\rangle\!\rangle\hat{\rho}_c(s)ds. \tag{283}$$

It is not over yet, as the equation is non-local in time, i.e. $\hat{\rho}_c(s)$ on RHS depends on the integrated variable. To fix this problem, the density matrix has to be brought outside the integral,

$$\frac{d}{dt}\hat{\rho}_c(t) \approx -\lambda^2 \int_0^t \langle\!\langle \mathcal{W}_c(X(t),\bar{X}(t),t)\mathcal{W}_c(X(s),\bar{X}(s),s)\rangle\!\rangle ds\,\hat{\rho}_c(t) = -\lambda^2 \mathcal{L}_c^{(2)}(t)\hat{\rho}_c(t). \tag{284}$$

A rough estimation shows that the resultant error is $\sim \lambda\tau_c$ times smaller than the leading term, hence, it can be neglected when the coupling is weak, $\lambda\tau_c \ll 1$. The equation—now local in time—can be solved without difficulty,

$$\hat{\rho}_c(t) = T_{\mathcal{L}_c^{(k)}}e^{-\lambda^2\int_0^t \mathcal{L}_c^{(2)}(s)ds}\hat{\rho}_S = \langle\mathcal{U}\rangle^{(2)}(t,0)\hat{\rho}_S. \tag{285}$$

At last, we have arrived at the first step of the SQ-based derivation. From this point on, we can get the Davies equation by simply replicating all the steps described in the previous section.

Now let us get back to the Born-Markov approximation. Typically, textbooks try to argue that the form (282) is a genuine approximation, and some of the arguments do sound semi-convincingly. The issue is that it has been demonstrated in numerical experiments that the assumption about product form is simply incorrect [11]—the Born-Markov ansatz does not describe the real form of the total density matrix. So, if it is not an adequate approximation, then why it is still invoked as such an integral part of the textbook derivation? The honest answer is that it is the simplest way to force the issue of closing the Eq. (281). Remember, the SQ expansion—which does not even have this kind of problems because it produces dynamical equation by design—is too recent to make it to traditional textbooks.

A less problematic, but orders of magnitude more complicated, derivation still makes use of the Born-Markov ansatz but it abandons completely any notions that it is some form of approximation. Instead, the ansatz is implemented as a formal super-operator acting in the total *SE* operator space,

$$\mathcal{P}\hat{A}_{SE} = (\mathrm{tr}_E(\bullet)\otimes\hat{\rho}_E)\hat{A}_{SE} = \mathrm{tr}_E(\hat{A}_{SE})\otimes\hat{\rho}_E. \tag{286}$$

For example, when $\mathcal{P}$ acts on the total density matrix $\hat{R}_{SE}(s)$, it effectively enforces the ansatz. The projection super-operator is then used on the von Neumann equation, and after some algebra and the most rudimentary calculus [12], one is lead to the *Nakajima-Zwanzig equation* (NZE),

$$\frac{d}{dt}\mathcal{P}\hat{R}_{SE}(t) = -\lambda^2 \int_0^t \mathcal{P}[\hat{V}_c(t)\otimes\hat{X}_I(t),\bullet]T_{\mathcal{G}}e^{-i\lambda\int_s^t \mathcal{G}(u)du}\mathcal{G}(s)\mathcal{P}\hat{R}_{SE}(s)ds, \tag{287}$$

where $\mathcal{G}(s) = (\bullet - \mathcal{P})[\hat{V}_c(s)\otimes\hat{X}_I(s),\bullet]$.

Taking the partial trace over *E* transforms NZE into a time-non-local equation for the system's density matrix,

$$\frac{d}{dt}\hat{\rho}_c(t) = -\lambda^2 \int_0^t \mathrm{tr}_E\left([\hat{V}_c(t)\otimes\hat{X}_I(t),T_{\mathcal{G}}e^{-i\lambda\int_s^t \mathcal{G}(u)du}\mathcal{G}(s)\bullet\otimes\hat{\rho}_E]\right)\hat{\rho}_c(s)ds. \tag{288}$$

Now, when we take the above equation as an entry point instead of Eq. (281), then we can get to the closed equation (283) by expanding the exp into the power series,

$$T_{\mathcal{G}} e^{-i\lambda \int_s^t \mathcal{G}(u)du} = \bullet - i\lambda \int_s^t \mathcal{G}(u)du + \dots, \tag{289}$$

and then cutting off everything above the lowest $\lambda^2$-order,

$$\begin{aligned}
\frac{d}{dt}\hat{\rho}_c(t) &\approx -\lambda^2 \int_0^t \mathrm{tr}_E \left( [\hat{V}_c(t)\otimes\hat{X}_I(t), (\bullet-\mathcal{P})[\hat{V}_c(s)\otimes\hat{X}_I(s), \bullet\otimes\hat{\rho}_E]] \right)\hat{\rho}_c(s)ds \\
&= -\lambda^2 \int_0^t \langle\langle \mathcal{W}_c(X(t),\bar{X}(t),t)\mathcal{W}_c(X(s),\bar{X}(s),s)\rangle\rangle \hat{\rho}_c(s)ds \\
&\quad + \lambda^2 \int_0^t \langle X(t)\rangle\langle X(s)\rangle [\hat{V}_c(t),[\hat{V}_c(s),\hat{\rho}_c(s)]ds \\
&= -\lambda^2 \int_0^t \langle\langle \mathcal{W}_c(X(t),\bar{X}(t),t)\mathcal{W}_c(X(s),\bar{X}(s),s)\rangle\rangle \hat{\rho}_c(s)ds.
\end{aligned} \tag{290}$$

And thus, we get the same result as before but without invoking the Born-Markov "approximation". Overall, it can be shown that the solution to the Davies master equation (278)—the end point of all those deliberations—tends to the solution of Nakajima–Zwanzig equation asymptotically in the so-called *weak coupling limit*, where $\lambda \to 0$ while $t \to \infty$ at the rate $\sim \lambda^{-2}$ [12].

# 6 Dynamics of thermalisation

## 6.1 Thermalisation

Thermalisation is a process where the state of system $S$ is brought over time to the thermal equilibrium due to interactions with the other system $E$ that already is in the state of equilibrium. An intuitive expectation is that the thermalisation occurs for most systems open to thermal baths (i.e. "large" $E$ initialized in thermal equilibrium), especially when the coupling between $S$ and $E$ is weak. It should be kept in mind, however, that thermalization is not guaranteed—it is quite possible that some peculiarities of the structure of energy levels of $S$ or the symmetries of the coupling will close off the path to the equilibrium. Here we will discuss the necessary conditions for the thermalisation and we will describe the dynamics of the process itself using the formalism of the open system theory. We will restrict our considerations to the weak coupling regime, and thus, the results and conclusions presented below do not exhaust the topic by any means, shape, or form. Thermalisation is still an open problem and a very much active field of research.

Formally, we say that the system thermalizes when

$$\forall_{\hat{\rho}_S} \langle\mathcal{U}\rangle(t,0)\hat{\rho}_S \xrightarrow{t\to\infty} \hat{p} = \frac{1}{\sum_a e^{-\beta\epsilon_a}} \sum_a e^{-\beta\epsilon_a}|a\rangle\langle a| = \frac{1}{Z}\sum_a e^{-\beta\epsilon_a}|a\rangle\langle a|, \tag{291}$$

where $\{\epsilon_a\}_a$ and $\{|a\rangle\}_a$ are the eigenvalues and eigenkets of the operator representing the energy in system $S$ (this operator does not have be equal to the free Hamiltonian $\hat{H}_S$ of $S$) and $\beta = (k_B T)^{-1}$ is the inverse temperature determined by the initial thermal state of $E$,

$$\hat{\rho}_E = \frac{e^{-\beta\hat{H}_E}}{\mathrm{tr}(e^{-\beta\hat{H}_E})}. \tag{292}$$

In the case considered here, the dynamical map $\langle \mathcal{U} \rangle$ is induced by the total Hamiltonian

$$\hat{H}_{SE} = \hat{H}_S \otimes \hat{1} + \hat{1} \otimes \hat{H}_E + \lambda \hat{V} \otimes \hat{F}, \tag{293}$$

and we assume that the coupling is weak, so that the Born approximation (see Sec. 5.3) is sufficiently accurate,

$$\langle \mathcal{U} \rangle(t, 0) \approx e^{-it[\hat{H}_S + \lambda\langle F \rangle \hat{V}, \bullet]} T_{\mathcal{L}_c^{(k)}} e^{-\lambda^2 \int_0^t \mathcal{L}_c^{(2)}(s) ds}. \tag{294}$$

Since the process of thermalisation occurs over long period of time we also naturally satisfy the conditions for the Markov and secular approximations from Sec. 5.3 that lead us to the Davies generator,

$$-\frac{1}{t}\int_0^t \mathcal{L}_c^{(2)}(s)ds \to \mathcal{L}_D = -i[\hat{H}_D, \bullet] + \sum_{\omega \in \Omega} \frac{2S(\omega)}{e^{\beta\omega}+1}\left(\hat{V}_\omega \bullet \hat{V}_\omega^\dagger - \frac{1}{2}\{\hat{V}_\omega^\dagger \hat{V}_\omega, \bullet\}\right) \equiv i\mathcal{J} + \mathcal{D}, \tag{295}$$

Here, $\Omega$ is the set of unique frequencies of form $\omega_{ab} = \epsilon_a - \epsilon_b$ where $\epsilon_a$'s are the eigenvalues of the shifted Hamiltonian $\hat{H}_S' = \hat{H}_S + \lambda\langle F \rangle \hat{V}$, $\hat{H}_S'|a\rangle = \epsilon_a|a\rangle$. The hermitian part of the generator reads

$$\hat{H}_D = -\sum_{\omega \in \Omega} \text{Im}\left\{\int_0^\infty \langle\langle F(\tau)F\rangle\rangle e^{-i\omega\tau} d\tau\right\} \hat{V}_\omega^\dagger \hat{V}_\omega = -\sum_{\omega \in \Omega} h(\omega)\hat{V}_\omega^\dagger \hat{V}_\omega, \tag{296}$$

and the frequency decomposition of the coupling operator is defined as

$$\hat{V}_\omega = \sum_{a,b} \delta(\omega - \omega_{ab})|a\rangle V_{ab}\langle b| = \sum_{(a,b)\in I(\omega)} |a\rangle\langle b|V_{ab} = \hat{V}_{-\omega}^\dagger. \tag{297}$$

Here, the index pair $(a, b)$ belonging to the set of pairs $I(\omega)$ (an energy shell), is such that $\omega_{ab} = \omega$; note, e.g., that if $(a, b) \in I(\omega)$, then $(b, a) \notin I(\omega)$, unless $\omega = 0$.

## 6.2 Necessary conditions for thermalisation

There is a theorem cited in a number of textbooks [2,3] asserting that the Davies map generated by $\mathcal{L}_D$ thermalizes the system with respect to the energy operator $\hat{H}_S' = \sum_a \epsilon_a|a\rangle\langle a|$,

$$\forall_{\hat{\rho}_S} e^{\lambda^2 t \mathcal{L}_D}\hat{\rho}_S \xrightarrow{t\to\infty} \hat{p} = \frac{e^{-\beta\hat{H}_S'}}{\text{tr}(e^{-\beta\hat{H}_S'})} = \frac{1}{Z}\sum_a e^{-\beta\epsilon_a}|a\rangle\langle a|, \tag{298}$$

when the following conditions are met:

1. the system $S$ is *ergodic*:

$$\forall_{\omega\in\Omega} [\hat{A}, \hat{V}_\omega] = 0 \implies \hat{A} \propto \hat{1}, \tag{299}$$

2. the dissipative part of the Davies generator, $\mathcal{D}$, satisfies the *quantum detailed balance*,

$$\forall_{\hat{A},\hat{B}} \text{ tr}\left(\hat{p}\hat{A}^\dagger \mathcal{D}^\dagger \hat{B}\right) = \text{tr}\left(\hat{p}(\mathcal{D}^\dagger \hat{A})^\dagger \hat{B}\right). \tag{300}$$

These somewhat cryptic conditions can be clarified with a more down-to-earth parameterization. We will show here that the thermalisation occurs when

i. Hamiltonian $\hat{H}_S'$ is not degenerate,

$$\forall_{a\neq b} \epsilon_a \neq \epsilon_b, \tag{301}$$

ii. the coupling operator $\hat{V}$ does not have zero off-diagonal matrix elements,

$$\forall_{a \neq b} \; V_{ab} = \langle a|\hat{V}|b\rangle \neq 0. \tag{302}$$

As we have shown in assignment 38 of Sec 5.3, when $\hat{H}'_S$ is not degenerate, the matrix of $\mathcal{L}_D$ in basis $\mathcal{B}'_S = \{|a\rangle\langle b|\}_{a,b}$ has the block-diagonal form with respect to the split between subspaces spanned by the projectors $\mathcal{B}_P = \{|a\rangle\langle a|\}_a$ and coherences $\mathcal{B}_C = \{|a\rangle\langle b|\}_{a \neq b}$,

$$\mathcal{L}_D = \begin{bmatrix} [\mathcal{M}]_{\mathcal{B}_P} & 0 \\ 0 & [i\mathcal{J} + \mathcal{G}]_{\mathcal{B}_C} \end{bmatrix}_{\mathcal{B}'_S}, \tag{303}$$

where the generator of the Pauli master equation (see assignment 38) is given by,

$$M_{a,b} = \text{tr}\left(|a\rangle\langle a|\mathcal{L}_D|b\rangle\langle b|\right) = \text{tr}\left(|a\rangle\langle a|\mathcal{M}|b\rangle\langle b|\right) = \gamma_{a,b} - \delta_{a,b} \sum_c \gamma_{c,a}, \tag{304}$$

and the two-part generator in the coherence subspace,

$$iJ_{ab,cd} = \text{tr}\left(|b\rangle\langle a|i\mathcal{J}|c\rangle\langle d|\right) = i\delta_{a,c}\delta_{b,d} \sum_e \left(h(\omega_{ea})|V_{ea}|^2 - h(\omega_{eb})|V_{eb}|^2\right); \tag{305}$$

$$\begin{aligned} G_{ab,cd} &= \text{tr}\left(|b\rangle\langle a|\mathcal{D}|c\rangle\langle d|\right) = \text{tr}\left(|b\rangle\langle a|\mathcal{G}|c\rangle\langle d|\right) \\ &= 2V_{ac}^* V_{bd} \delta(\omega_{ab} - \omega_{cd}) \frac{S(\omega_{ac})}{e^{\beta\omega_{ac}} + 1} - \delta_{a,b}\delta_{c,d} \sum_e \frac{\gamma_{e,a} + \gamma_{e,b}}{2}, \end{aligned} \tag{306}$$

with the transition matrix given by

$$\gamma_{a,b} = |V_{ab}|^2 \frac{2S(\omega_{ab})}{e^{\beta\omega_{ab}} + 1}. \tag{307}$$

Our set of assumptions is consistent with the assumptions of the general theorem. First, we can show that i and ii lead to ergodic system:

$$\begin{aligned} [\hat{A}, \hat{V}_\omega] &= \sum_{(a,b) \in I(\omega)} \sum_{c,d} A_{cd} V_{ab}[|c\rangle\langle d|, |a\rangle\langle b|] \\ &= \sum_{(a,b) \in I(\omega)} V_{ab}\left(|a\rangle\langle b|(A_{aa} - A_{bb}) + \sum_{c \neq a} A_{ca}|c\rangle\langle b| - \sum_{d \neq b} A_{bd}|a\rangle\langle d|\right) = 0, \end{aligned} \tag{308}$$

which then implies

$$\forall_{(a,b) \in I(\omega)} : \left\{A_{aa} = A_{bb}, \quad \text{and} \quad \forall_{c \neq a}\forall_{d \neq b} : A_{ca} = A_{bd} = 0\right\}, \tag{309}$$

because $V_{ab} \neq 0$ (assumption ii) and, if $(a,b) \in I(\omega)$, then $(c,b), (a,d) \notin I(\omega)$ for $c \neq a, d \neq b$ (assumption i). When we check the condition for commutation for every $\omega \in \Omega$, then we get

$$\begin{aligned} &\forall_{\omega \in \Omega}\forall_{(a,b) \in I(\omega)} \left(A_{aa} = A_{bb} \text{ and } \forall_{c \neq a}\forall_{d \neq b} A_{ca} = A_{bd} = 0\right) \\ &\Leftrightarrow \forall_{a,b} \left(A_{aa} = A_{bb} \text{ and } \forall_{c \neq a}\forall_{d \neq b} A_{ca} = A_{bd} = 0\right) \\ &\Rightarrow \hat{A} \propto \hat{1}. \end{aligned} \tag{310}$$

That is, the condition of ergodicity is satisfied.

To show that the the quantum detailed balance is also satisfied, first we need to find the symmetries of generators,

$$M_{b,a} = \gamma_{b,a} - \delta_{b,a} \sum_e \gamma_{e,b} = \frac{e^{-\beta\epsilon_b}}{e^{-\beta\epsilon_a}} \left( \gamma_{a,b} - \delta_{a,b} \sum_e \gamma_{e,a} \right) = \frac{p_b}{p_a} M_{a,b}\,; \tag{311a}$$

$$\begin{aligned}
G_{cd,ab} &= 2V_{ca}^* V_{db} \delta(\omega_{cd} - \omega_{ab}) \frac{S(\omega_{ca})}{e^{\beta\omega_{ca}} + 1} - \delta_{c,a}\delta_{d,b} \sum_e \frac{\gamma_{e,c} + \gamma_{e,d}}{2} \\
&= 2V_{ac} V_{bd}^* \delta(\omega_{ab} - \omega_{cd}) \frac{S(-\omega_{ac})}{e^{-\beta\omega_{ca}} + 1} \frac{1}{e^{\beta\omega_{ca}}} - \delta_{a,c}\delta_{b,d} \sum_e \frac{\gamma_{e,a} + \gamma_{e,b}}{2} \\
&= \left( 2V_{ac} V_{bd}^* \delta(\omega_{ab} - \omega_{cd}) \frac{S(\omega_{ac})}{e^{\beta\omega_{ac}} + 1} - \delta_{a,c}\delta_{b,d} \sum_e \frac{\gamma_{e,a} + \gamma_{e,b}}{2} \right) \frac{e^{-\beta\epsilon_c}}{e^{-\beta\epsilon_a}} \\
&= \frac{p_c}{p_a} G_{ab,cd}^*\,; \tag{311b}
\end{aligned}$$

$$\begin{aligned}
iJ_{cd,ab} &= i\delta_{a,c}\delta_{b,d} \sum_e \left( h(\omega_{ec})|V_{ec}|^2 - h(\omega_{ed})|V_{ed}|^2 \right) \\
&= i\frac{e^{-\beta\epsilon_c}}{e^{-\beta\epsilon_a}} \delta_{a,c}\delta_{b,d} \sum_e \left( h(\omega_{ea})|V_{ea}|^2 - h(\omega_{ea})|V_{eb}|^2 \right) = -\frac{p_c}{p_a}(iJ_{ab,cd})^*\,. \tag{311c}
\end{aligned}$$

Now, the proof is straightforward,

$$\begin{aligned}
\text{tr}\left( \hat{p}\hat{A}^\dagger \mathcal{D}^\dagger \hat{B} \right) &= \sum_a p_a A_{aa}^* \left( \sum_b M_{b,a} B_{bb} \right) + \sum_{a\neq b} p_b A_{ab}^* \left( \sum_{c\neq d} G_{cd,ab}^* B_{cd} \right) \\
&= \sum_{a,b} p_b \frac{p_a}{p_b} M_{b,a} A_{aa}^* B_{bb} + \sum_{a\neq b}\sum_{c\neq d} p_d \frac{p_b p_c}{p_d p_a} \frac{p_a}{p_c} G_{cd,ab}^* A_{ab}^* B_{cd} \\
&= \sum_b p_b \left( \sum_a M_{a,b} A_{aa} \right)^* + \sum_{c\neq d} p_d \left( \sum_{a\neq b} e^{\beta(\omega_{ab} - \omega_{cd})} G_{ab,cd}^* A_{ab} \right)^* B_{cd} \\
&= \sum_b p_b \left( \sum_a M_{a,b} A_{aa} \right)^* + \sum_{c\neq d} p_d \left( \sum_{a\neq b} G_{ab,cd}^* A_{ab} \right)^* B_{cd} \\
&= \text{tr}\left( \hat{p}(\mathcal{D}^\dagger \hat{A})^\dagger \hat{B} \right)\,, \tag{312}
\end{aligned}$$

where we have used the fact that $G_{ab,cd} \propto \delta(\omega_{ab} - \omega_{cd})$.

## 6.3 H-theorem

The diagonal elements of the system's density matrix,

$$\rho_a(t) = \langle a|\hat{\rho}_S(t)|a\rangle = \langle a|e^{\lambda^2 t \mathcal{L}_D}\hat{\rho}_S|a\rangle = \text{tr}\left( |a\rangle\langle a|e^{\lambda^2 t \mathcal{M}} \sum_b \rho_b(0)|b\rangle\langle b| \right), \tag{313}$$

are positive and normalized, $\sum_a \rho_a(t) = \text{tr}(\exp(\lambda^2 t \mathcal{L}_D)\hat{\rho}_S) = \text{tr}(\hat{\rho}_S) = 1$—therefore, $\rho_a(t)$ can be treated as a time-dependent probability distribution.

One can define the non-negative quantity called the *relative entropy* between the equilibrium distribution $p_a = \langle a|\hat{p}|a\rangle$ and $\rho_a(t)$,

$$H(t) = \sum_a \rho_a(t) \ln\left( \frac{\rho_a(t)}{p_a} \right) \geqslant 0\,. \tag{314}$$

It can be shown that $H(t) = 0$, if and only if, $\rho_a = p_a$ for all $a$—i.e. $H$ vanishes only when $\rho_a(t)$ is identical with the equilibrium distribution.

The so-called *H-theorem* states that $H(t)$ is a monotonically decreasing function of time,

$$
\begin{aligned}
\frac{1}{\lambda^2} \frac{dH(t)}{dt} &= \frac{1}{\lambda^2} \sum_a \frac{d\rho_a(t)}{dt} \ln\left(\frac{\rho_a(t)}{p_a}\right) + \frac{1}{\lambda^2} \sum_a \rho_a(t) \frac{p_a}{\rho_a(t)} \frac{1}{p_a} \frac{d\rho_a(t)}{dt} \\
&= \sum_{a,b} M_{a,b} \rho_b(t) \ln\left(\frac{\rho_a(t)}{p_a}\right) + \frac{1}{\lambda^2} \frac{d}{dt} \sum_a \rho_a(t) \\
&= \frac{1}{2} \sum_{a,b} M_{a,b} \rho_b(t) \ln\left(\frac{\rho_a(t)}{p_a}\right) + \frac{1}{2} \sum_{a,b} M_{b,a} \rho_a(t) \ln\left(\frac{\rho_b(t)}{p_b}\right) \\
&= \frac{1}{2} \sum_{a,b} M_{a,b} p_b \left[ \frac{\rho_b(t)}{p_b} \ln\left(\frac{\rho_a(t)}{p_a}\right) + \frac{\rho_a(t)}{p_a} \ln\left(\frac{\rho_b(t)}{p_b}\right) \right] \\
&= \frac{1}{2} \sum_{a,b} \left( \gamma_{a,b} - \delta_{a,b} \sum_c \gamma_{c,a} \right) p_b \left[ \frac{\rho_b(t)}{p_b} \ln\left(\frac{\rho_a(t)}{p_a}\right) + \frac{\rho_a(t)}{p_a} \ln\left(\frac{\rho_b(t)}{p_b}\right) \right] \\
&= \frac{1}{2} \sum_{a,b} \gamma_{a,b} p_b \left[ \frac{\rho_b(t)}{p_b} \ln\left(\frac{\rho_a(t)}{p_a}\right) + \frac{\rho_a(t)}{p_a} \ln\left(\frac{\rho_b(t)}{p_b}\right) \right] \\
&\quad - \frac{1}{2} \sum_{a,b} \gamma_{b,a} p_a \frac{\rho_a(t)}{p_a} \ln\left(\frac{\rho_a(t)}{p_a}\right) - \frac{1}{2} \sum_{a,b} \gamma_{a,b} p_b \frac{\rho_b(t)}{p_b} \ln\left(\frac{\rho_b(t)}{p_b}\right) \\
&= \frac{1}{2} \sum_{a,b} \gamma_{a,b} p_b \left[ \frac{\rho_b(t)}{p_b} \ln\left(\frac{\rho_a(t)}{p_a}\right) + \frac{\rho_a(t)}{p_a} \ln\left(\frac{\rho_b(t)}{p_b}\right) \right. \\
&\quad \left. - \frac{\rho_a(t)}{p_a} \ln\left(\frac{\rho_a(t)}{p_a}\right) - \frac{\rho_b(t)}{p_b} \ln\left(\frac{\rho_b(t)}{p_b}\right) \right] \\
&= \frac{1}{2} \sum_{a,b} \gamma_{a,b} p_b \left( \frac{\rho_a(t)}{p_a} - \frac{\rho_b(t)}{p_b} \right) \left( \ln\left(\frac{\rho_b(t)}{p_b}\right) - \ln\left(\frac{\rho_a(t)}{p_a}\right) \right) \leqslant 0.
\end{aligned}
\tag{315}
$$

This is because $\gamma_{a,b} > 0$ and

$$
(y-x)(\ln x - \ln y) = (y-x) \ln\left(\frac{x}{y}\right) \leqslant (y-x)\left(\frac{x}{y} - 1\right) = \frac{(y-x)(x-y)}{y} \leqslant 0,
\tag{316}
$$

for any $x \neq y$—the consequence of the concavity of $\ln$,

$$
\ln a - \ln b \leqslant (a-b) \ln' b = \frac{a-b}{b} \xrightarrow{b=1, a=x/y} \ln\left(\frac{x}{y}\right) \leqslant \frac{x}{y} - 1.
\tag{317}
$$

The immediate conclusion from H-theorem is that $H(t)$ keeps decreasing until it reaches its $t \to \infty$ limit value $H(\infty) \geqslant 0$. We can determine when the limit is reached by checking at which point the derivative vanishes. There is only one circumstance when $dH/dt = 0$: all ratios $\rho_a(t)/p_a$ are mutually equal. This can be achieved only by setting $\rho_a(t) = p_a$ for each $a$, i.e. when $\rho_a(t)$ becomes the equilibrium distribution. In other words, the distribution $\rho_a(t)$ evolves under the action of Davies map until it thermalises, then the evolution stops because $p_a$ is the stationary solution of Pauli master equation, $\sum_b M_{a,b} p_b = 0$ (see assignment 38 of Sec. 5.3).

## 6.4 Eigensystem analysis

The H-theorem had shown us that the diagonal of the density matrix always ends up at the thermal equilibrium distribution under the action of Davies map. However, we also want to describe the dynamics of this process. The most transparent analysis is based on the eigensystem of generator $\mathcal{M}$.

First, observe that the matrix $M_{a,b}$ is hermitian, but in a specific sense. Consider a scalar product that "embraces" the special symmetry of the Davies generator (311)

$$\langle\!\langle \hat{A}|\hat{B}\rangle\!\rangle \equiv \sum_{a,b} e^{\beta \epsilon_a} A_{ab}^* B_{ab} \, . \tag{318}$$

For this scalar product we have

$$
\begin{aligned}
\langle\!\langle \hat{A}|\mathcal{M}\hat{B}\rangle\!\rangle &= \sum_a e^{\beta \epsilon_a} A_{aa}^* \sum_b M_{a,b} B_{bb} = \sum_b e^{\beta \epsilon_b} \left( \sum_a \frac{e^{\beta \epsilon_a}}{e^{\beta \epsilon_b}} M_{a,b} A_{aa} \right)^* B_{bb} \\
&= \sum_b e^{\beta \epsilon_b} \left( \sum_a \frac{p_b}{p_a} M_{a,b} A_{aa} \right)^* B_{bb} = \sum_b e^{\beta \epsilon_b} \left( \sum_a M_{b,a} A_{aa} \right)^* B_{bb} \\
&= \langle\!\langle \mathcal{M}\hat{A}|\hat{B}\rangle\!\rangle \, ,
\end{aligned} \tag{319}
$$

which is the definition of hermicity. From basic algebra we know that hermitian matrices have real eigenvalues and the corresponding eigenvectors (or eigenoperators in the case of matrices of super-operators) form an orthogonal basis in the subspace the matrix operates in, i.e.

$$\mathcal{M}\hat{q}(\mu) = \mu\hat{q}(\mu) \, , \tag{320}$$

where $\mu \in \mathbb{R}$, the eigenoperators live in the subspace $\mathcal{B}_P$, $\hat{q}(\mu) = \sum_a |a\rangle q_a(\mu)\langle a|$, and they are mutually orthogonal $\langle\!\langle \hat{q}(\mu)|\hat{q}(\mu')\rangle\!\rangle \propto \delta_{\mu,\mu'}$, and they form basis,

$$\bullet_{\mathcal{B}_P} = \sum_\mu \frac{\hat{q}(\mu)\langle\!\langle \hat{q}(\mu)|\bullet\rangle\!\rangle}{\langle\!\langle \hat{q}(\mu)|\hat{q}(\mu)\rangle\!\rangle} \, , \tag{321}$$

where $\bullet_{\mathcal{B}_P}$ is the identity in subspace $\mathcal{B}_P$.

We already know that one of the eigenvalues equals zero and that it corresponds to the equilibrium state, $\mathcal{M}\hat{p} = 0$ (see assignment 38 of Sec. 5.3). When we consider the projection onto $\hat{p} = \hat{q}(0)$ we get an interesting result,

$$\hat{p} \frac{\langle\!\langle \hat{p}|\bullet\rangle\!\rangle}{\langle\!\langle \hat{p}|\hat{p}\rangle\!\rangle} = \hat{p} \frac{Z^{-1} \sum_a e^{\beta\epsilon_a} e^{-\beta\epsilon_a} \langle a|\bullet|a\rangle}{Z^{-2} \sum_a e^{\beta\epsilon_a} e^{-2\beta\epsilon_a}} = \hat{p} \sum_a \langle a|\bullet|a\rangle = \hat{p} \operatorname{tr}(\bullet) \, , \tag{322}$$

which means that all other eigenoperators $\hat{q}(\mu)$ are traceless because of orthogonality,

$$0 = \langle\!\langle \hat{p}|\hat{q}(\mu)\rangle\!\rangle = \frac{1}{Z} \operatorname{tr}(\hat{q}(\mu)) \, .$$

All non-zero eigenvalues are negative because $\mathcal{M}$ is a non-positive matrix,

$$
\begin{aligned}
\langle\!\langle \hat{A}|\mathcal{M}\hat{A}\rangle\!\rangle &= \sum_a e^{\beta\epsilon_a} A_{aa}^* \left( \sum_b M_{a,b} A_{bb} \right) = \sum_{a,b} e^{\beta\epsilon_a} \gamma_{a,b} A_{aa}^* A_{bb} - \sum_a e^{\beta\epsilon_a} |A_{aa}|^2 \sum_e \gamma_{e,a} \\
&= \sum_{a>b} \left( e^{\beta\epsilon_a} \gamma_{a,b} A_{aa}^* A_{bb} + e^{\beta\epsilon_b} \gamma_{b,a} A_{bb}^* A_{aa} \right) - \sum_a e^{\beta\epsilon_a} |A_{aa}|^2 \sum_{e\neq a} \gamma_{e,a} \\
&= \sum_{a>b} \left[ e^{\beta\epsilon_a} \gamma_{a,b} \left( A_{aa} A_{bb}^* + A_{aa} A_{bb}^* \right) - \sum_{c\neq b} e^{\beta\epsilon_c} |A_{cc}|^2 \sum_{e\neq a} \gamma_{e,c} - \sum_{c\neq a} e^{\beta\epsilon_c} |A_{cc}|^2 \sum_{e\neq b} \gamma_{e,c} \right] \\
&= \sum_{a>b} \left[ e^{\beta\epsilon_a} \gamma_{a,b} \left( A_{aa} A_{bb}^* + A_{aa} A_{bb}^* - e^{-\beta\epsilon_b} |A_{aa}|^2 - e^{\beta\epsilon_b} |A_{bb}|^2 \right) \right. \\
&\qquad \left. - \sum_{c\neq a,b} e^{\beta\epsilon_c} |A_{cc}|^2 \sum_{e\neq a,b} \gamma_{e,c} \right] \\
&= -\sum_{a>b} \left( e^{\beta\epsilon_a} \gamma_{a,b} |A_{aa} e^{-\beta\epsilon_b} - A_{bb} e^{\beta\epsilon_b}|^2 + \sum_{c\neq a,b} e^{\beta\epsilon_c} |A_{cc}|^2 \sum_{e\neq a,b} \gamma_{e,c} \right) \leqslant 0 \, . \tag{323}
\end{aligned}
$$

With all these facts in hand, we can write the explicit form of the Davies map in the projector subspace,

$$e^{\lambda^2 t \mathcal{M}} = \exp\left[\lambda^2 t \sum_\mu \mu \frac{\hat{q}(\mu)\langle\langle\hat{q}(\mu)|\bullet\rangle\rangle}{\langle\langle\hat{q}(\mu)|\hat{q}(\mu)\rangle\rangle}\right] = \sum_\mu e^{\lambda^2 \mu t} \frac{\hat{q}(\mu)\langle\langle\hat{q}(\mu)|\bullet\rangle\rangle}{\langle\langle\hat{q}(\mu)|\hat{q}(\mu)\rangle\rangle}$$

$$= \hat{p}\,\mathrm{tr}(\bullet) + \sum_{\mu<0} e^{-\lambda^2|\mu|t} \frac{\hat{q}(\mu)\langle\langle\hat{q}(\mu)|\bullet\rangle\rangle}{\langle\langle\hat{q}(\mu)|\hat{q}(\mu)\rangle\rangle}\,, \tag{324}$$

and thus

$$e^{\lambda^2 t \mathcal{M}} \sum_a |a\rangle \rho_a(0)\langle a| = \hat{p}\,\mathrm{tr}(\hat{\rho}_S) + \sum_{\mu<0} e^{-\lambda^2|\mu|t}\hat{q}(\mu)\frac{\langle\langle\hat{q}(\mu)|\hat{\rho}_S\rangle\rangle}{\langle\langle\hat{q}(\mu)|\hat{q}(\mu)\rangle\rangle} \xrightarrow{t\to\infty} \hat{p}\,. \tag{325}$$

We knew from the H-theorem that there is only one zero eigenvalue: if there was another $\mu = 0$, then the corresponding eigenoperator would also be a stationary solution, and thus, not all initial distributions would tend to $p_a$, which would contradict the theorem.

## 6.5 Dephasing

For the thermalisation to be complete, bringing the diagonal of the density matrix to the equilibrium distribution is not enough—we also need to have the total dephasing of the off-diagonal elements,

$$e^{\lambda^2 t(i\mathcal{J}+\mathcal{G})} \sum_{a\neq b} |a\rangle \rho_{ab}(0)\langle b| \xrightarrow{t\to\infty} 0\,. \tag{326}$$

Since the off-diagonal elements,

$$\rho_{ab}(t) = \mathrm{tr}(|b\rangle\langle a|e^{\lambda^2 t(i\mathcal{J}+\mathcal{G})}\hat{\rho}_S)\,, \tag{327}$$

are complex numbers, they cannot be interpreted as probability distribution, and thus, there is no H-theorem for the coherence subspace. Still, the best way to demonstrate that the map in $\mathcal{B}_C$ eliminates everything over time is to investigate the eigenvalues of the generator. However, there are some complications. Even though $\mathcal{G}$ is hermitian,

$$\langle\langle\hat{A}|\mathcal{G}\hat{B}\rangle\rangle = \sum_{a\neq b} e^{\beta\epsilon_a} A_{ab}^* \left(\sum_{c\neq d} G_{ab,cd} B_{cd}\right) = \sum_{c\neq d} e^{\beta\epsilon_c}\left(\sum_{a\neq b}\frac{e^{\beta\epsilon_a}}{e^{\beta\epsilon_c}}G_{ab,cd}A_{ab}^*\right)B_{cd}$$

$$= \sum_{c\neq d} e^{\beta\epsilon_c}\left(\sum_{a\neq b}\frac{p_c}{p_a}G_{ab,cd}^*A_{ab}\right)^* B_{cd} = \sum_{c\neq d} e^{\beta\epsilon_c}\left(\sum_{a\neq b}G_{cd,ab}A_{ab}\right)^* B_{cd}$$

$$= \langle\langle\mathcal{G}\hat{A}|\hat{B}\rangle\rangle\,, \tag{328}$$

and $i\mathcal{J}$ is anti-hermitian

$$\langle\langle\hat{A}|i\mathcal{J}\hat{B}\rangle\rangle = -\langle\langle i\mathcal{J}\hat{A}|\hat{B}\rangle\rangle\,, \tag{329}$$

so that both have full set of orthogonal eigenoperators that can be used as a basis, the two generators do not commute,

$$[i\mathcal{J},\mathcal{G}] \neq 0\,. \tag{330}$$

Therefore, we need to solve the eigenproblem for the total generator, $i\mathcal{J} + \mathcal{G}$, and, of course, the sum of hermitian and anti-hermitian matrix is neither of those. This means that there is

no guarantee that the matrix of the total generator has enough eigenoperators to form a basis, and so, we cannot rely on getting a map with form similar to (324). All is not lost, however; even if matrix does not have enough eigenoperators to form a basis, one can still express the map in terms of eigenvalues $\mu$ and exponential factors $\exp(\lambda^2 \mu t)$.

Mathematically, eigenvalues $\mu$ are the roots of the *characteristic polynomial*,

$$\det\left(i\mathcal{J} + \mathcal{G} - \mu \bullet_{\mathcal{B}_C}\right) = 0. \tag{331}$$

According to the fundamental theorem of algebra, every polynomial of degree $n$ has as many roots; however, it is always possible that some roots (or eigenvalues) have the same value,

$$\det\left(i\mathcal{J} + \mathcal{G} - \mu \bullet_{\mathcal{B}_C}\right) = \prod_k (\mu_k - \mu) = (\mu_1 - \mu)^{n_1}(\mu_2 - \mu)^{n_2}\cdots. \tag{332}$$

When $n_k > 1$ we say that the eigenvalue $\mu_k$ is degenerate and $n_k$ is its degree of degeneracy.

The issues come up when the number of linearly independent eigenvectors corresponding to degenerate eigenvalue is smaller than its degree of degeneracy (the two numbers are always equal for hermitian, anti-hermitian, and, in general, normal matrices). If this is the case, then there is not enough (linearly independent) eigenvectors to form a basis and we call the matrix *defective*.

The simplest example of defective matrix is

$$A = \begin{bmatrix} a & 1 \\ 0 & a \end{bmatrix}.$$

Its degenerate eigenvalue is easy to find,

$$0 = \det(A - \mu I) = (a - \mu)^2 \;\Rightarrow\; \mu_1 = a,\; n_1 = 2\,;$$

and the corresponding eigenvector,

$$\begin{bmatrix} 0 \\ 0 \end{bmatrix} = (A - aI)\, r(a) = \begin{bmatrix} 0 & 1 \\ 0 & 0 \end{bmatrix}\begin{bmatrix} r_1 \\ r_2 \end{bmatrix} = \begin{bmatrix} r_2 \\ 0 \end{bmatrix} \;\Rightarrow\; r(a) = \begin{bmatrix} 1 \\ 0 \end{bmatrix}.$$

And so, there is only one eigenvector—not enough to form a basis.

When the matrix $B$ is not defective, one can switch to the basis of its eigenvectors to diagonalize it,

$$S^{-1}BS = \begin{bmatrix} \lambda_1 & & & & \\ & \lambda_2 & & & \\ & & \lambda_2 & & \\ & & & \lambda_2 & \\ & & & & \ddots \end{bmatrix},$$

where $S$ is a transformation to the basis of eigenvectors, and $\lambda_i$ are the eigenvalues (visible here, $\lambda_2$ is degenerate and $\lambda_1$ is not degenerate). Once diagonalized, it is a trivial matter to calculate any function of the matrix, e.g., the exp,

$$\exp(tB) = \sum_{k=0}^{\infty} \frac{1}{k!} t^k B^k = \sum_{k=0}^{\infty} \frac{1}{k!} t^k S(S^{-1}BS)^k S^{-1}$$

$$= S \begin{bmatrix} e^{\lambda_1 t} & & & & \\ & e^{\lambda_2 t} & & & \\ & & e^{\lambda_2 t} & & \\ & & & e^{\lambda_2 t} & \\ & & & & \ddots \end{bmatrix} S^{-1}.$$

Of course, it is impossible to diagonalize a defective matrix. Nevertheless, there is an algorithm for constructing the missing basis elements—the so-called *generalized eigenvectors*—in such a way that the matrix in this new basis is closest to the diagonal form as possible. Let $C$ be defective matrix with a defective eigenvalue $\mu$ of some degeneration degree $n > 1$, then it is possible to find a transformation $N$ such that

$$N^{-1}CN = \begin{bmatrix} \ddots \\ & \mu & 1 \\ & & \mu & 1 \\ & & & \ddots & \ddots \\ & & & & \mu & 1 \\ & & & & & \mu \\ & & & & & & \ddots \end{bmatrix}.$$

The $n \times n$ block with $\mu$ on diagonal, 1's on super-diagonal and 0's everywhere else, is called the *Jordan block*. Using the basis completed with generalized eigenvectors one can now calculate functions of $C$, in particular, the exp,

$$\exp(tC) = N \exp(tN^{-1}CN)N^{-1}$$

$$= N \begin{bmatrix} \ddots \\ & e^{\mu t} & te^{\mu t} & \frac{t^2}{2!}e^{\mu t} & \cdots & \frac{t^n}{n!}e^{\mu t} \\ & & e^{\mu t} & te^{\mu t} & \cdots & \frac{t^{n-1}}{(n-1)!}e^{\mu t} \\ & & & \ddots & \ddots & \vdots \\ & & & & e^{\mu t} & te^{\mu t} \\ & & & & & e^{\mu t} \\ & & & & & & \ddots \end{bmatrix} N^{-1}.$$

As we can see, in the case of exp, each non-zero matrix element within the Jordan block scales with $t$ as a polynomial times $\exp(\mu t)$—this is what we were looking for.

Now let us come back to the generator $i\mathcal{J} + \mathcal{G}$. To make sure that the map it generates leads to the total dephasing, we need to show that the real part of its eigenvalues, $\mathrm{Re}\,\mu$ ($\mu$ are complex, in general), are *strictly* negative; $\mathrm{Re}\,\mu = 0$ would not do, because then there would be some blocks that do not decay to zero over time.

As it turns out, we just need to prove that

$$\forall_{\hat{A}} \; \langle\langle \hat{A} | \mathcal{G} \hat{A} \rangle\rangle < 0 \,, \tag{333}$$

i.e. that the dissipative part of the generator is a negative matrix. Let us proceed then,

$$\langle\langle \hat{A} | \mathcal{G} \hat{A} \rangle\rangle = \sum_{a \neq b} \sum_{c \neq d} \delta(\omega_{ab} - \omega_{cd}) 2 e^{\beta \epsilon_a} A_{ab}^* A_{cd} V_{ac}^* V_{bd} \frac{S(\omega_{ac})}{e^{\beta \omega_{ac}} + 1}$$

$$- \sum_{a \neq b} e^{\beta \epsilon_a} |A_{ab}|^2 \sum_e \left( \frac{|V_{ea}|^2 S(\omega_{ea})}{e^{\beta \omega_{ea}} + 1} + \frac{|V_{eb}|^2 S(\omega_{eb})}{e^{\beta \omega_{eb}} + 1} \right). \tag{334}$$

Take two index pairs $(a, b)$ and $(c, d)$ belonging to one energy shell $I(\omega_{ab})$ (i.e. $\omega_{ab} = \omega_{cd}$) and focus on those terms only,

$$2 A_{ab}^* A_{cd} V_{ac}^* V_{bd} e^{\beta \epsilon_a} \frac{S(\omega_{ac})}{e^{\beta \omega_{ac}} + 1} + 2 A_{cd}^* A_{ab} V_{ca}^* V_{db} e^{\beta \epsilon_c} \frac{S(\omega_{ca})}{e^{\beta \omega_{ca}} + 1}$$

$$= \frac{e^{\beta \epsilon_a} S(\omega_{ac})}{e^{\beta \omega_{ac}} + 1} \left( 2 A_{ab}^* A_{cd} V_{ac}^* V_{bd} + 2 A_{ab} A_{cd}^* V_{ac} V_{bd}^* \right), \tag{335}$$

where we have used the symmetries $S(\omega_{ac}) = S(-\omega_{ca}) = S(\omega_{ca})$, $V_{ca}^* = V_{ac}$. The same index pairs in the second sum gives us

$$-|A_{ab}|^2 e^{\beta\epsilon_a} \sum_e \left( \frac{|V_{ea}|^2 S(\omega_{ea})}{e^{\beta\omega_{ea}}+1} + \frac{|V_{eb}|^2 S(\omega_{eb})}{e^{\beta\omega_{eb}}+1} \right)$$

$$-|A_{cd}|^2 e^{\beta\epsilon_c} \sum_e \left( \frac{|V_{ec}|^2 S(\omega_{ec})}{e^{\beta\omega_{ec}}+1} + \frac{|V_{ed}|^2 S(\omega_{ed})}{e^{\beta\omega_{ed}}+1} \right)$$

$$\equiv -|A_{ab}|^2 e^{\beta\epsilon_a} R_{ab} - |A_{cd}|^2 e^{\beta\epsilon_c} R_{cd}. \tag{336}$$

Note that $R_{ab} > 0$ because we are assuming that $V_{ab} \neq 0$. The "mixed" terms (335) are problematic because their sign is unspecified. Fortunately, we can take care of them by absorbing them into full moduli squares; to do this we need to draw a few terms from $R_{ab}$ and $R_{cd}$ in the second term, namely

$$-|A_{ab}|^2 e^{\beta\epsilon_a} \left( \frac{|V_{ca}|^2 S(\omega_{ca})}{e^{\beta\omega_{ca}}+1} + \frac{|V_{db}|^2 S(\omega_{db})}{e^{\beta\omega_{db}}+1} \right) - |A_{cd}|^2 e^{\beta\epsilon_c} \left( \frac{|V_{ac}|^2 S(\omega_{ac})}{e^{\beta\omega_{ac}}+1} + \frac{|V_{bd}|^2 S(\omega_{bd})}{e^{\beta\omega_{bd}}+1} \right)$$

$$= -\frac{e^{\beta\epsilon_a} S(\omega_{ac})}{e^{\beta\omega_{ac}}+1} \left( e^{\beta\omega_{ac}} |A_{ab}|^2 + e^{-\beta\omega_{ac}} |A_{cd}|^2 \right) \left( |V_{ac}|^2 + |V_{bd}|^2 \right), \tag{337}$$

Now we can combine it with the mixed terms to get

$$(335) + (337)$$

$$= -\frac{e^{\beta\epsilon_a} S(\omega_{ac})}{e^{\beta\omega_{ac}}+1} \left( |e^{\frac{\beta\omega_{ac}}{2}} A_{ab} V_{ac} - e^{-\frac{\beta\omega_{ac}}{2}} A_{cd} V_{bd}|^2 + |e^{\frac{\beta\omega_{ac}}{2}} A_{ab} V_{bd}^* - e^{-\frac{\beta\omega_{ac}}{2}} A_{cd} V_{ac}^*|^2 \right) \leqslant 0. \tag{338}$$

Given that $\hat{H}_S'$ is non-degenerate, there will always be at least two strictly positive terms left in $R_{ab}$ after we draw from it all the terms needed to balance the mixed terms within each energy shell $I(\omega_{ab})$—this means that $\langle\!\langle \hat{A} | \mathcal{G}\hat{A} \rangle\!\rangle < 0$ for any non-zero $\hat{A}$.

Of course, the inequality (333) is also satisfied for all eigenoperators $\hat{r}(\mu)$ of the total generator, so that we can write,

$$(\mathrm{Re}\,\mu + i\,\mathrm{Im}\,\mu)\langle\!\langle \hat{r}(\mu) | \hat{r}(\mu) \rangle\!\rangle = \langle\!\langle \hat{r}(\mu) | (i\mathcal{J} + \mathcal{G})\hat{r}(\mu) \rangle\!\rangle = i\langle\!\langle \hat{r}(\mu) | \mathcal{J}\hat{r}(\mu) \rangle\!\rangle + \langle\!\langle \hat{r}(\mu) | \mathcal{G}\hat{r}(\mu) \rangle\!\rangle, \tag{339}$$

then, since $\langle\!\langle \hat{A} | \hat{A} \rangle\!\rangle > 0$ and $\langle\!\langle \hat{A} | \mathcal{J}\hat{A} \rangle\!\rangle, \langle\!\langle \hat{A} | \mathcal{G}\hat{A} \rangle\!\rangle \in \mathbb{R}$ ($\mathcal{J}$ is hermitian because $i\mathcal{J}$ is anti-hermitian) for any non-zero $\hat{A}$, we get

$$\mathrm{Re}\,\mu = \frac{\langle\!\langle \hat{r}(\mu) | \mathcal{G}\hat{r}(\mu) \rangle\!\rangle}{\langle\!\langle \hat{r}(\mu) | \hat{r}(\mu) \rangle\!\rangle} < 0. \tag{340}$$

Therefore, when $\hat{H}_S'$ is non-degenerate and all $V_{ab} \neq 0$, we have the total dephasing,

$$\sum_{c\neq d} [e^{\lambda^2 t(i\mathcal{J}+\mathcal{G})}]_{ab,cd} \rho_{cd}(0)$$

$$= \sum_{c\neq d} [N \begin{bmatrix} e^{-\lambda^2 t |\mathrm{Re}\,\mu_1|} e^{i\lambda^2 t\,\mathrm{Im}\,\mu_1} B_1(t) & & \\ & e^{-\lambda^2 t |\mathrm{Re}\,\mu_2|} e^{i\lambda^2 t\,\mathrm{Im}\,\mu_2} B_2(t) & \\ & & \ddots \end{bmatrix} N^{-1}]_{ab,cd} \rho_{cd}(0)$$

$$\xrightarrow{t\to\infty} 0, \tag{341}$$

where

$$B_i(t) = \begin{bmatrix} 1 & t & \cdots & \frac{1}{n_i!} t^{n_i} \\ & 1 & t & \cdots \\ & & \ddots & \\ & & & 1 \end{bmatrix}, \tag{342}$$

if the eigenvalue $\mu_i$ is defective, and $B_i(t) = I$ when it is not.

## 6.6 Physics of thermalisation

Here is a bold but surprisingly apt analogy: the process of thermalisation can be compared to the spread of infectious disease. When a "diseasesed" system $E$, that manifests symptoms in a form of $\hat{\rho}_E \propto \exp(-\beta\hat{H}_E)$, comes into contact with other system $S$, the "pathogen" is passed on through the coupling $\lambda\hat{V} \otimes \hat{F}$. Over time, the infection takes hold and the state of $S$ starts to display symptoms, $\hat{\rho}_S(t) \to \hat{p} = \exp(-\beta\hat{H}_S')/Z$. Then, $S$ becomes a potential vector for a further spread of the infection.

If we are willing to fully embrace this analogy, then we could ask what is this "pathogen" that causes the thermalisation to spread from system to system? For example, the thermal state itself, $\hat{\rho}_E \propto \exp(-\beta\hat{H}_E)$, is not it; like we said, the final form of the density matrix is more of a symptom rather than the cause. Indeed, the dynamical map we have been investigating in this chapter does not contain any explicit reference to the initial state of $E$. Therefore, the process of thermalisation is not directly influenced by particular form of the thermal state.

Similarly, the process of thermalisation is largely independent of a detailed form of the coupling operator $\hat{V}$. However, this does not mean that $\hat{V}$ has no impact at all. For example, we had to constraint the $S$-side of the coupling with the condition $V_{ab} \neq 0$ to ensure the total dephasing, and thus, to complete the thermalisation. If any $V_{ab} = 0$, then it would open a possibility that $\text{Re}\,\mu = 0$ for some of the eigenvalues $\mu$ of generator $i\mathcal{J}+\mathcal{G}$, and that would lead to pockets of perpetually oscillating off-diagonal elements of the density matrix. So, within our analogy, the shape of $\hat{V}$ would be one of the factors affecting how resistant the system is to the transmission of the pathogen. If all energy levels are directly coupled (i.e. $V_{ab} \neq 0$), the pathogen spreads freely and the diseases of thermalisation overcomes the system totally. If some energy levels are unreachable (a corresponding $V_{ab} = 0$), then parts of the system will remain "healthy" and it will display less sever symptoms (i.e. some non-zero off-diagonal elements of density matrix). This is, most likely, an artifact of Born approximation: since we have restricted the map to the lowest order in $\lambda\tau_c$, only the direct transmission $V_{ab}$ is available. Although, at the moment, it is only a speculation on our part, it seems highly probable that going beyond Born would enable indirect transmission paths, e.g., $V_{ac}V_{cb}$ to connect $a$ with $b$ through $c$ when the direct path is blocked, $V_{ab} = 0$. If this intuition is correct, then the disease could defeat system's resistance in the end.

Finally, we have the qumulant $\langle\!\langle F(t)F \rangle\!\rangle$, or rather, the real part of its Fourier transform [e.g., see Eq. (276)],

$$\int_{-\infty}^{\infty} \text{Re}\{\langle\!\langle F(\tau)F \rangle\!\rangle\}e^{-i\omega\tau}d\tau = S(\omega) - i\kappa(\omega).$$

As we have already noted, there are no direct references to the thermal state in the Davies map. If so, then how does $S$ know what is the temperature it is supposed to thermalise to? This is where qumulant comes in: in Davies map, it is the only "channel" able to transmit the information about $\beta$ from $E$ to $S$. To be more precise, this information is not passed on by the spectral density $S(\omega)$ or the susceptibility $\kappa(\omega)$ individually (see Sec. 4.7 and assignment 36), but rather it is encoded in the very specific relation between them—the fluctuation-dissipation theorem (FDT)—which gives us the omnipresent temperature-dependent factor

$$S(\omega) - i\kappa(\omega) = \frac{2S(\omega)}{e^{\beta\omega}+1}.$$

The spectral density $S(\omega)$, which, essentially, contains in itself $\hat{F}$ and $\hat{H}_E$, does not really influence the process of thermalisation in any qualitative way; it mostly (together with $\hat{V}$) determines the magnitude of the eigenvalues of generator $\mathcal{L}_D$, which in turn determine the speed of the process. The thermal factor $(e^{\beta\omega}+1)^{-1}$ is the key element here: it is the reason

for the symmetries of $\gamma_{a,b}$ and $\mathcal{L}_D$ [see, Eq. (311)]. Precisely because of these symmetries, the thermal state $\hat{p} \propto \exp(-\beta \hat{H}'_S)$ is the steady-state solution of Davies equation (or, $\mu = 0$ eigenoperator of $\mathcal{L}_D$) and we have the H-theorem. Therefore, if we go back to our analogy, the thermal factor can be pointed out as a cause of the disease. Still, we think that the factor is not the pathogen analogue, rather it is only an expression of the actual pathogen—the fluctuation-dissipation theorem. But how a *concept* like FDT can be analogous to something like virus? Consider this: a typical virus consists of a protein and lipid structure and the genetic material enclosed inside. The purpose of the protein structure is to transport the genetic material and help with depositing it into its target—the host's cell. So, in a sense, the structure is just a part of the transmission mechanism, along with aerosol droplets, bodily fluids, or any other intermediary between the genetic material and the cell's nucleus. The part of the virus that is indispensable for it to be a pathogen is the genetic material (in fact, a primitive kind of viruses called viroids are made of genetic material only). Even the genetic material—a strand of DNA or RNA—can be though of as merely a vessel for the *information* it carries. It is this information that is the essence of virus as a pathogen: it contains the instructions how to hijack the cell, how to use it to replicate itself, and how to facilitate its further spread. The FDT is like this abstract viral information carried in the genetic material; it is behind the transmission mechanism in the form of the thermal factor, and it takes care of self-replication by leading to thermal state, which is the necessary and sufficient condition for FDT (see assignment 36).

We are not done with our analogy yet. We now see that the thermal equilibrium spreads like a disease from infected system to healthy systems (i.e. system not in thermal state). But how this epidemic started? How the "patient zero" become infected? In other words, how the thermal equilibrium gets established when the system is closed and insulated? It is a very interesting and yet unsolved problem. Fortunately, this subject received a lot of attention over the years; if you wish to read more about it, we recommend looking up Eigenstate Thermalization Hypothesis as a starting point.

# 7 The origin of external fields in quantum mechanics

## 7.1 Where do external fields in quantum mechanics come from?

In classical mechanics, a natural way to simplify a many-body problem is to "replace" some of the elements of the composite system with a surrogate *force fields*. Take for example a motion of a planet orbiting around its star. Formally, it is a two-body problem and to find the exact motion of the planet one has to solve the coupled equations for the trajectories of both involved parties. However, the formalism of the classical theory allows, through straightforward change of coordinate system, to effectively eliminate the massive star from the picture and, instead, depict the planet as if it moved through space filled with an external (i.e. independent of the planet) field of force. In fact, it is always possible to transform the coupled equations of motion into a coordinate system where the trajectories are decoupled and constrained by external (sometimes time-dependent) fields.

In general, tricks like that do not work in quantum mechanics: due to the product structure of the composite quantum systems' Hilbert spaces and the state-focused description, it is virtually impossible to decouple dynamical equations with a simple change of the coordinate system (although, there are specific contexts where it can be done). Nevertheless, the external fields are routinely used to accurately model the dynamics of quantum systems, presumably in contact with their quantum environments, as systems that are closed but not insulated.

Examples range from simple Pauli model of a spin-1/2 precssing in an external magnetic field,

$$\hat{H} = \sum_{\alpha=x,y,z} \frac{\mu}{2} \hat{\sigma}_\alpha B_\alpha \,, \tag{343}$$

to stochastic models we have discussed in detail in Chapter 3.

Here is the conundrum. Given the general incompatibility of the formalism of quantum mechanics with replacing-subsystems-with-force-fields, how is it possible that the surrogate field models work at all? What is the mechanism behind it? What kind of conditions the composite system must satisfy for such models to be valid? Since the point of surrogate field models is to "eliminate" a coupled subsystem from the description (and replace it with an external field), it seems obvious that the open system theory should be able to give us answers; after all, "eliminating" a subsystem (the environment) and accounting for its influence on what is left is *the thing* the theory is all about.

## 7.2 Surrogate field representation

The surrogate field model, where the subsystem $E$ of a composite total system $SE$ is superseded with an external surrogate field $F(t)$, is formally implemented by replacing the full $SE$ Hamiltonian with a $S$-only (stochastic) generator,

$$\hat{H}_{SE} = \hat{H}_S \otimes \hat{1} + \hat{1} \otimes \hat{H}_E + \lambda \hat{V} \otimes \hat{F} \quad \rightarrow \quad \hat{H}_F[F](t) = \hat{H}_S + \lambda F(t)\hat{V} \,. \tag{344}$$

The field $F(t)$ is, in general, a stochastic process—remember that deterministic field [like the magnetic field in the Pauli model (343)] is also a stochastic process but with a trivial probability distribution, $P_F[f] = \delta[f - f_0]$.

The model, or the *surrogate field representation* [13] as we will call it, is valid when the stochastic dynamics generated by $\hat{H}_F$ is a good approximation (or an accurate simulation) of the dynamical map originating from the actual $SE$ Hamiltonian, i.e.

$$\left\langle T_{\mathcal{W}} e^{-i\lambda \int_0^t \mathcal{W}(F(s),\bar{F}(s),s)ds} \right\rangle \approx \overline{T_{\mathcal{W}} e^{-i\lambda \int_0^t \mathcal{W}(F(s),F(s),s)ds}} \,. \tag{345}$$

In other words, the surrogate field representation is valid when the quasi-average over joint quasi-probability distributions of the coupling operator $\hat{F}$, $\{Q_F^{(k)}\}_{k=1}^\infty$, can be well approximated with a stochastic average over surrogate field $F(t)$.

Given what we have learned about the structural parallelism between quasi- and stochastic averages, it should be clear that the validity of the surrogate representation depends on whether the quasi-probability distributions,

$$Q_F^{(k)}(\boldsymbol{f}, \bar{\boldsymbol{f}}, \boldsymbol{s}) \equiv Q_F^{(k)}(f_1, \bar{f}_1, s_1; \ldots; f_k, \bar{f}_k, s_k) \,, \tag{346}$$

can be treated as a stochastic-process-defining joint probability distributions we have discussed in chapter 3. Indeed, as we have already hinted at in Sec. 4.3, the dynamical map becomes structurally identical to stochastic map when $Q_F^{(k)}(\boldsymbol{f}, \bar{\boldsymbol{f}}, \boldsymbol{s}) \to \delta_{\boldsymbol{f}, \bar{\boldsymbol{f}}} P_F^{(k)}(\boldsymbol{f}, \boldsymbol{s})$; let us investigate in more detail what is the mechanism behind such transformation.

First, let us define the function representing the "diagonal" part of joint quasi-probabilities — by diagonal we mean here that all $\bar{f}_i$'s are set to be equal to the corresponding $f_i$'s —

$$P_F^{(k)}(\boldsymbol{f}, \boldsymbol{s}) \equiv Q_F^{(k)}(\boldsymbol{f}, \boldsymbol{f}, \boldsymbol{s}) \geqslant 0 \,. \tag{347}$$

Functions $P_F^{(k)}$ are not only non-negative but also normalized (i.e. $\sum_{\boldsymbol{f}} P_F^{(k)}(\boldsymbol{f}, \boldsymbol{s}) = 1$), therefore, the diagonal part of $Q_F^{(k)}$ can serve as a probability distribution for a multi-component

stochastic variable $F = (F_1, F_2, \ldots, F_k)$. Also, recall our previous discussion on possible physical interpretations of joint quasi-probability distributions (see Sec. 4.5); there we have already shown, albeit in a completely different context, that diagonal part of quasi-probability alone can play the role of proper probability distribution. Despite that, at this point, $P_F^{(k)}$'s are not necessarily *joint* probability distributions, and so the components of stochastic vector $F_1, F_2, \ldots$ cannot be interpreted as a consecutive values of a stochastic process—to upgrade stochastic vector $F$ to stochastic process $F(t)$, $P_F^{(k)}$'s would have to satisfy the consistency relation (84), but, in general, they do not.

**Assignment 39 (Solution)** *Show that the diagonal part of joint quasi-probability is non-negative.*

**Assignment 40 (Solution)** *Show that the diagonal part of joint quasi-probability is normalized.*

Of course, the total joint quasi-distribution also has the "off-diagonal" part, where at least one pair of arguments is not matched, $\bar{f}_i \neq f_i$; we will refer to this part as the *interference term* $\Phi_F^{(k)}$ (again, recall the discussion in Sec. 4.5). Formally, the interference term is defined through the following decomposition,

$$Q_F^{(k)}(\boldsymbol{f}, \bar{\boldsymbol{f}}, \boldsymbol{s}) = \delta_{\boldsymbol{f}, \bar{\boldsymbol{f}}} P_F^{(k)}(\boldsymbol{f}, \boldsymbol{s}) + \Phi_F^{(k)}(\boldsymbol{f}, \bar{\boldsymbol{f}}, \boldsymbol{s}). \tag{348}$$

The non-zero interference term is the reason why the family $\{P_F^{(k)}\}_{k=1}^\infty$ is not consistent by itself. Indeed, the consistency of $\{Q_F^{(k)}\}_{k=1}^\infty$ is one of the defining properties of joint quasi-probability distributions [see Sec. 4.3, Eq. (197)],

$$\sum_{f_i, \bar{f}_i} Q_F^{(k)}(\boldsymbol{f}, \bar{\boldsymbol{f}}, \boldsymbol{s}) = Q_F^{(k-1)}(\ldots; f_{i-1}, \bar{f}_{i-1}, s_{i-1}; f_{i+1}, \bar{f}_{i+1}, s_{i+1}; \ldots), \tag{349}$$

but given the decomposition (348), the consistency of $Q_F^{(k)}$'s implies that

$$\sum_{f_i} P_F^{(k)}(\boldsymbol{f}, \boldsymbol{s}) = P_F^{(k-1)}(\ldots; f_{i-1}, s_{i-1}; f_{i+1}, s_{i+1}; \ldots) - \left(\prod_{j \neq i} \delta_{f_j, \bar{f}_j}\right) \sum_{f_i, \bar{f}_i} \Phi_F^{(k)}(\boldsymbol{f}, \bar{\boldsymbol{f}}, \boldsymbol{s}). \tag{350}$$

This brings us to the following conclusion: if the interference terms vanish, then

1. the joint quasi-probability distributions become probability distributions,

$$\forall_k \, \Phi_F^{(k)} \approx 0 \quad \Rightarrow \quad \forall_k \, Q_F^{(k)}(\boldsymbol{f}, \bar{\boldsymbol{f}}, \boldsymbol{s}) \approx \delta_{\boldsymbol{f}, \bar{\boldsymbol{f}}} P_F^{(k)}(\boldsymbol{f}, \boldsymbol{s}),$$

2. the family $\{P_F^{(k)}\}_{k=1}^\infty$ becomes consistent,

$$\forall_k \, \Phi_F^{(k)} \approx 0 \quad \Rightarrow \quad \sum_{f_i} P_F^{(k)}(\boldsymbol{f}, \boldsymbol{s}) = P_F^{(k-1)}(\ldots; f_{i-1}, s_{i-1}; f_{i+1}, s_{i+1}; \ldots).$$

In that case, the family of now joint probability distributions $\{P_F^{(k)}\}$ define a stochastic process

$F(t)$, and this process is, in fact, the surrogate field,

$$
\begin{aligned}
\Big\langle & T_{\mathcal{W}} e^{-i\lambda \int_0^t \mathcal{W}(F(s),\bar{F}(s),s)ds} \Big\rangle \\
&= \sum_{k=0}^{\infty}(-i\lambda)^k \int_0^t ds_1 \cdots \int_0^{s_{k-1}} ds_k \sum_{f,\bar{f}} Q_F^{(k)}(f,\bar{f},s) \mathcal{W}(f_1,\bar{f}_1,s_1) \cdots \mathcal{W}(f_k,\bar{f}_k,s_k) \\
&\approx \sum_{k=0}^{\infty}(-i\lambda)^k \int_0^t ds_1 \cdots \int_0^{s_{k-1}} ds_k \sum_{f,\bar{f}} \delta_{f,\bar{f}} P_F^{(k)}(f,s) \mathcal{W}(f_1,\bar{f}_1,s_1) \cdots \mathcal{W}(f_k,\bar{f}_k,s_k) \\
&= \sum_{k=0}^{\infty}(-i\lambda)^k \int_0^t ds_1 \cdots \int_0^{s_{k-1}} ds_k \, \overline{\mathcal{W}(F(s_1),F(s_1),s_1) \cdots \mathcal{W}(F(s_k),F(s_k),s_k)} \\
&= \overline{T_{\mathcal{W}} e^{-i\lambda \int_0^t \mathcal{W}(F(s),F(s),s)ds}} \,.
\end{aligned}
\tag{351}
$$

### 7.3  Examples of valid surrogate field representations

#### 7.3.1  Static coupling

The simplest class of dynamics with a valid surrogate representation is characterized by the relation

$$
[\hat{F},\hat{H}_E] = 0 \,.
\tag{352}
$$

When two operators commute as above, then both can be diagonalized in the same basis; therefore,

$$
\hat{F} = \sum_f f \hat{P}(f) = \sum_f \sum_n \delta_{f,f(n)} f(n) |n\rangle\langle n| = \sum_n f(n)|n\rangle\langle n| \;\Rightarrow\; \hat{H}_E = \sum_n \epsilon(n)|n\rangle\langle n| \,,
\tag{353}
$$

which, in turn, implies that the projection operators are unchanged by the interaction picture,

$$
\begin{aligned}
\hat{P}_I(f,s) &= e^{is\hat{H}_E}\hat{P}(f)e^{-is\hat{H}_E} = \sum_n \delta_{f,f(n)} e^{is\hat{H}_E}|n\rangle\langle n| e^{-is\hat{H}_E} = \sum_n \delta_{f,f(n)} e^{is\epsilon(n)}|n\rangle\langle n|e^{-is\epsilon(n)} \\
&= \sum_n \delta_{f,f(n)}|n\rangle\langle n| = \hat{P}(f) \,.
\end{aligned}
\tag{354}
$$

The interference terms are automatically eliminated when the projection operators are static,

$$
\begin{aligned}
Q_F^{(k)}(f,\bar{f},s) &= \operatorname{tr}_E\big(\hat{P}_I(f_1,s_1)\cdots\hat{P}_I(f_k,s_k)\hat{\rho}_E\hat{P}_I(\bar{f}_k,s_k)\cdots\hat{P}_I(\bar{f}_1,s_1)\big) \\
&= \operatorname{tr}_E\big(\hat{P}(f_1)\cdots\hat{P}(f_k)\hat{\rho}_E\hat{P}(\bar{f}_k)\cdots\hat{P}(\bar{f}_1)\big) = \prod_{i=2}^k \delta_{f_1,f_i} \prod_{j=2}^k \delta_{\bar{f}_1,\bar{f}_j} \operatorname{tr}_E\big(\hat{P}(f_1)\hat{\rho}_E\hat{P}(\bar{f}_1)\big) \\
&= \delta_{f_1,\bar{f}_1} \prod_{i=2}^k \delta_{f_1,f_i} \prod_{j=2}^k \delta_{\bar{f}_1,\bar{f}_j} \operatorname{tr}\big(\hat{P}(f_1)\hat{\rho}_E\big) = \delta_{f,\bar{f}} \prod_{i=2}^k \delta_{f_1,f_i} \operatorname{tr}\big(\hat{P}(f_1)\hat{\rho}_E\big) \,.
\end{aligned}
\tag{355}
$$

The delta $\delta_{f,\bar{f}}$ indicates that only the diagonal part of quasi-probability can be non-zero. Moreover, the resultant joint probability distributions are rather trivial,

$$
P_F^{(k)}(f,s) = P_F^{(1)}(f_1,0) \prod_{i=2}^k \delta_{f_1,f_i} = \operatorname{tr}_E\big(\hat{P}(f_1)\hat{\rho}_E\big) \prod_{i=2}^k \delta_{f_1,f_i} \,,
\tag{356}
$$

which means that the obtained surrogate field is downgraded from stochastic process to time-independent stochastic *variable F* (see Sec. 3.1.1) with probability distribution defined by the initial state of $E$,

$$P_F(f) = \text{tr}_E \left( \hat{P}(f) \hat{\rho}_E \right) . \tag{357}$$

Therefore, the open system $S$ coupled with such a surrogate field undergoes a relatively simple evolution,

$$\langle \mathcal{U} \rangle_I(t,0) = \overline{\mathcal{U}}_I(t,0) = \overline{T_{\mathcal{V}_I} e^{-i\lambda F \int_0^t \mathcal{V}_I(s)ds}} = \sum_f P_F(f) T_{\mathcal{V}_I} e^{-i\lambda f \int_0^t \mathcal{V}_I(s)ds}, \tag{358}$$

where $\mathcal{V}_I(s) = [\hat{V}_I(s), \bullet]$. If the initial state is chosen as an eigenstate of $\hat{F}$ (or, equivalently, an eigenstate of $\hat{H}_E$), so that $P_F(f) = \delta_{f,B}$, then the dynamical map has the form identical to that of a closed system coupled to the constant external field $B$,

$$\langle \mathcal{U} \rangle_I(t,0) = T_{\mathcal{V}_I} e^{-i\lambda B \int_0^t \mathcal{V}_I(s)ds} \Rightarrow \langle \mathcal{U} \rangle(t,0) = e^{-it[\hat{H}_S, \bullet]} \langle \mathcal{U} \rangle_I(t,0) = e^{-it[\hat{H}_S + \lambda B\hat{V}, \bullet]}. \tag{359}$$

Hence, this class of surrogate representations can be though of as one part of the explanation for the appearance of external fields in quantum mechanics, like, e.g., in a one-axis Pauli model, $\hat{H} = \mu B \hat{\sigma}_z/2$.

However, can we use the same kind of surrogate representation to explain a general three-axis Pauli model (343)? To answer this question we must consider a multi-component surrogate field representation for the general form of $SE$ coupling

$$\lambda \sum_\alpha \hat{V}_\alpha \otimes \hat{F}_\alpha . \tag{360}$$

In assignment 34 of Sec. 4.4 we have derived the marginal joint quasi-probability distributions associated with the set of coupling operators $\{\hat{F}_\alpha\}_\alpha$,

$$\begin{aligned} Q^{(k)}_{\alpha_1,\dots,\alpha_k} & (f_1^{\alpha_1}, \bar{f}_1^{\alpha_1}, s_1; \dots; f_k^{\alpha_k}, \bar{f}_k^{\alpha_k}, s_k) \\ & = \text{tr}_E \left( \hat{P}_{\alpha_1}(f_1^{\alpha_1}, s_1) \cdots \hat{P}_{\alpha_k}(f_k^{\alpha_k}, s_k) \hat{\rho}_E \hat{P}_{\alpha_k}(\bar{f}_k^{\alpha_k}, s_k) \cdots \hat{P}_{\alpha_1}(\bar{f}_1^{\alpha_1}, s_1) \right) , \end{aligned} \tag{361}$$

where the spectral decomposition for each coupling operator reads as follows,

$$\hat{F}_\alpha(s) = \sum_{f^\alpha} f^\alpha \hat{P}_\alpha(f^\alpha, s) = \sum_{f^\alpha} f^\alpha e^{is\hat{H}_E} \hat{P}_\alpha(f^\alpha) e^{-is\hat{H}_E} . \tag{362}$$

The condition for valid multi-component representation, where each operator $\hat{F}_\alpha$ is replaced by a component $F_\alpha(t)$ of a vector stochastic process $\boldsymbol{F}(t)$, is a straightforward generalization of the single-component condition,

$$Q^{(k)}_{\alpha_1,\dots,\alpha_k}(f_1^{\alpha_1}, \bar{f}_1^{\alpha_1}, s_1; \dots; f_k^{\alpha_k}, \bar{f}_k^{\alpha_k}, s_k) \approx \prod_{i=1}^k \delta_{f_i^{\alpha_i}, \bar{f}_i^{\alpha_i}} Q^{(k)}_{\alpha_1,\dots,\alpha_k}(f_1^{\alpha_1}, f_1^{\alpha_1}, s_1; \dots; f_k^{\alpha_k}, f_k^{\alpha_k}, s_k), \tag{363}$$

i.e. only the diagonal part is allowed to survive. The static coupling, $\forall_\alpha [\hat{F}_\alpha, \hat{H}_E] = 0$, is not enough to satisfy the above condition, however. This is because, in general,

$$\hat{P}_\alpha(f^\alpha) \hat{P}_{\alpha'}(f^{\alpha'}) \neq \hat{P}_{\alpha'}(f^{\alpha'}) \hat{P}_\alpha(f^\alpha) \neq \delta_{f^\alpha, f^{\alpha'}} \hat{P}_\alpha(f^\alpha), \tag{364}$$

for $\alpha \neq \alpha'$, which opens a crack for the interference terms to slip through. The simplest way to close this loophole is to demand that, in addition to commutation with $\hat{H}_E$, operators $\hat{F}_\alpha$ couple to independent sub-spaces in $E$, e.g.,

$$\hat{F}_\alpha = \hat{1}^{\otimes(\alpha-1)} \otimes \hat{f}_\alpha \otimes \hat{1}^{\otimes(N-\alpha)} = \sum_{f^\alpha} f^\alpha \, \hat{1}^{\otimes(\alpha-1)} \otimes \hat{P}_\alpha(f^\alpha) \otimes \hat{1}^{\otimes(N-\alpha)}; \qquad (365a)$$

$$\hat{\rho}_E = \bigotimes_{\alpha=1}^{N} \hat{\rho}_\alpha. \qquad (365b)$$

For such a form of couplings and the initial state, the marginal quasi-probabilities simply factorize and the condition for valid surrogate representation is trivially satisfied, e.g.,

$$\begin{aligned}
Q_{\alpha_1,\alpha_2,\alpha_2}^{(3)} &(f_1^{\alpha_1}, \bar{f}_1^{\alpha_1}, s_1; f_2^{\alpha_2}, \bar{f}_2^{\alpha_2}, s_2; f_3^{\alpha_2}, \bar{f}_3^{\alpha_2}, s_3) \\
&= Q_{F_{\alpha_1}}^{(1)}(f_1^{\alpha_1}, \bar{f}_1^{\alpha_1}, s_1) Q_{F_{\alpha_2}}^{(2)}(f_2^{\alpha_2}, \bar{f}_2^{\alpha_2}, s_2; f_3^{\alpha_2}, \bar{f}_3^{\alpha_2}, s_3) \\
&= \left( \delta_{f_1^{\alpha_1}, \bar{f}_1^{\alpha_1}} \prod_{i=2}^{3} \delta_{f_i^{\alpha_2}, \bar{f}_i^{\alpha_2}} \right) P_{F_{\alpha_1}}^{(1)}(f_1^{\alpha_1}, 0) \delta_{f_2^{\alpha_2}, f_3^{\alpha_2}} P_{F_{\alpha_2}}^{(1)}(f_2^{\alpha_2}, 0), \qquad (366)
\end{aligned}$$

for $\alpha_1 \neq \alpha_2$. If we also assume that the initial state is a product of eigenstates, so that

$$\text{tr}_E \left( \hat{1}^{\otimes(\alpha-1)} \hat{P}_\alpha(f^\alpha) \otimes \hat{1}^{\otimes(N-\alpha)} \bigotimes_\alpha \hat{\rho}_\alpha \right) = \text{tr}_\alpha(\hat{P}_\alpha(f^\alpha)\hat{\rho}_\alpha) = \delta_{f^\alpha, B_\alpha}, \qquad (367)$$

then we get a multi-component constant external field,

$$\langle \mathcal{U} \rangle(t,0) = e^{-it\left[\hat{H}_S + \lambda \sum_\alpha B_\alpha \hat{V}_\alpha, \bullet\right]}. \qquad (368)$$

### 7.3.2 Environment of least action

In our second example we are considering an environment with a basis parametrized by a continues variable, so that we can write

$$\hat{P}(f) = \int_{\Gamma(f)} dx |x\rangle\langle x|, \qquad (369)$$

where $\Gamma(f)$ is an interval characterizing the degenerate subspace corresponding to the discrete eigenvalue $f$. Before we proceed further, we rewrite the joint quasi-probabilities in terms of probability amplitudes,

$$\phi_{s_e - s_b}(x_e | x_b) = \langle x_e | e^{-i(s_e - s_b)\hat{H}_E} | x_b \rangle, \qquad (370)$$

that we have discussed previously in Sec. 4.5,

$$
\begin{aligned}
Q_F^{(k)}(\boldsymbol{f},\bar{\boldsymbol{f}},\boldsymbol{s}) &= \mathrm{tr}_E\left(\hat{P}_I(f_1,s_1)\cdots\hat{P}_I(f_k,s_k)\hat{\rho}_E\hat{P}_I(\bar{f}_k,s_k)\cdots\hat{P}_I(\bar{f}_1,s_1)\right) \\
&= \left(\prod_{i=1}^{k}\int_{\Gamma(f_i)}dx_i\int_{\Gamma(\bar{f}_i)}d\bar{x}_i\right)\delta(x_1-\bar{x}_1)\left(\prod_{i=1}^{k-1}\phi_{s_i-s_{i+1}}(x_i|x_{i+1})\right)\left(\prod_{i=1}^{k-1}\phi_{s_i-s_{i+1}}(\bar{x}_i|\bar{x}_{i+1})\right)^* \\
&\quad \times\langle x_k|e^{-is_k\hat{H}_E}\hat{\rho}_E e^{is_k\hat{H}_E}|\bar{x}_k\rangle \\
&= \left(\prod_{i=1}^{k}\int_{\Gamma(f_i)}dx_i\int_{\Gamma(\bar{f}_i)}d\bar{x}_i\right)\delta(x_1-\bar{x}_1)\int_{-\infty}^{\infty}dx_0 d\bar{x}_0\,\langle x_0|\hat{\rho}_E|\bar{x}_0\rangle \\
&\quad \times\left(\phi_{s_k}(x_k|x_0)\prod_{i=1}^{k-1}\phi_{s_i-s_{i+1}}(x_i|x_{i+1})\right)\left(\phi_{s_k}(\bar{x}_k|\bar{x}_0)\prod_{i=1}^{k-1}\phi_{s_i-s_{i+1}}(\bar{x}_i|\bar{x}_{i+1})\right)^*.
\end{aligned}
\tag{371}
$$

In his classic works, R.P. Feynman showed that the probability amplitude can be expressed as a *path integral* [14],

$$
\phi_{s_e-s_b}(x_e|x_b) = \langle x_e|e^{-i(s_e-s_b)\hat{H}_E}|x_b\rangle = \int_{x(s_b)=x_b}^{x(s_e)=x_e}e^{\frac{i}{\hbar}S[x]}[Dx],
\tag{372}
$$

where the functional $S[x]$, that takes paths $x(s)$ as its argument, is called the *action*, the limits of the integral indicate that all paths we integrate over begin in point $x_b$ and end in $x_e$, and $\hbar$ is the Planck constant. The particular form of $S[x]$ and the nature of paths $x(s)$ is, of course, determined by the Hamiltonian $\hat{H}_E$ and the structure of eigenkets $\{|x\rangle\}_{x=-\infty}^{\infty}$. The concept of action originates in classical mechanics, where it is defined as

$$
S_{\mathrm{cl}}[x] = \int_{s_b}^{s_e}[(\text{kinetic energy at } x(s)) - (\text{potential energy at } x(s))]\,ds,
\tag{373}
$$

and, in this context, the path $x(s)$ is a continuous function of time representing the position of the mechanical system. The classical action is the namesake for the famous *principle of least action* which states that only the path $x_{\min}(s)$ that minimizes the action, $\forall_{x(s)}S_{\mathrm{cl}}[x] \geqslant S_{\mathrm{cl}}[x_{\min}]$, describes the actual motion of the system. In other words, the path $x_{\min}(s)$ is also the unique solution to the equations of motion of the mechanical system. Remarkably, for some types of quantum systems the functional $S[x]$ appearing in the path integral (372) has exactly the same form as the classical action (373) of a corresponding mechanical system; one example of such a system is the quantum harmonic oscillator and its classical counterpart. This correspondence, when coupled with the principle of least action, has some intriguing implications for the problem of the emergence of classical world from fundamentally quantum physical laws. Following Feynman's own reasoning, we observe that when the action $S[x]$ of our quantum system is large (in comparison to $\hbar$), then the phase factors under the integral (372) will vary widely from path to path. Such a rapid variation causes the phase factors to cancel each other out due to destructive interference. However, the interference effect is turned from destructive to constructive when the integral approaches the vicinity of the stationary point (or rather, stationary path) of $S[x]$; there, the action nears its extremum where its variation slows downs considerably and the nearby phase factors tend to be aligned in phase. Consequently, for large action, the only contribution to the probability amplitude $\phi_{s_e-s_b}(x_e|x_b)$ comes from the immediate vicinity of stationary points. When we have the correspondence between our

quantum system and its classical counterpart (i.e. $S[x] = S_{cl}[x]$), we can easily identify such a point as the path involved in the least action principle, $x_{min}(s)$, because, obviously, it is the minimum of $S[x]$. Then, we can harness the interference effects to calculate the path integral using the steepest descent method,

$$\phi_{s_e-s_b}(x_e|x_b) \propto e^{\frac{i}{\hbar}S_{cl}(x_e,s_e|x_b,s_b)}, \tag{374}$$

where $S_{cl}(x_e,s_e|x_b,s_b) = S[x_{min}]$ with $x_{min}(s_{e/b}) = x_{e/b}$. This result shows us that when the quantum system approaches its classical limit (e.g., the mass of harmonic oscillator tends to macroscopic scale), the quantum and classical modes of the description of motion start to overlap.

The effect we have highlighted above can also have a significant impact on the issue of the surrogate representation's validity. Let us come back to our joint quasi-probability and assume that the approximation (374) is applicable to our environment,

$$Q_F^{(k)}(f,\bar{f},s) \propto \left(\prod_{i=1}^{k} \int_{\Gamma(f_i)} dx_i \int_{\Gamma(\bar{f}_i)} d\bar{x}_i\right) \delta(x_1 - \bar{x}_1) \int_{-\infty}^{\infty} dx_0 d\bar{x}_0 \, \langle x_0|\hat{\rho}_E|\bar{x}_0\rangle \tag{375}$$

$$\times \left(e^{\frac{i}{\hbar}S_{cl}(x_k,s_k|x_0,0)} \prod_{i=1}^{k-1} e^{\frac{i}{\hbar}S_{cl}(x_i,s_i|x_{i+1},s_{i+1})}\right) \left(e^{\frac{i}{\hbar}S_{cl}(\bar{x}_k,s_k|\bar{x}_0,0)} \prod_{i=1}^{k-1} e^{\frac{i}{\hbar}S_{cl}(\bar{x}_i,s_i|\bar{x}_{i+1},s_{i+1})}\right)^*.$$

However, this change in form does not exhaust all the ramifications brought about by the interference effects (both destructive and constructive) that were so crucial for the least action principle approximation. Indeed, the interference between phase factors is still taking place here because of the integrals that are "stitching" the adjacent probability amplitudes, e.g.,

$$\int_{\Gamma(f_i)} e^{\frac{i}{\hbar}S_{cl}(x_{i-1},s_{i-1}|x_i,s_i)+\frac{i}{\hbar}S_{cl}(x_i,s_i|x_{i+1},s_{i+1})} dx_i. \tag{376}$$

The general principle from before applies here as well: the destructive interference will suppress the integral unless the interval $\Gamma(f_i)$ includes the stationary point of the phase. Mathematically, the point is considered to be a stationary point when the derivative vanishes at its location, i.e.

$$\frac{\partial}{\partial x_i}\left[S_{cl}(x_{i-1},s_{i-1}|x_i,s_i) + S_{cl}(x_i,s_i|x_{i+1},s_{i+1})\right]\Big|_{x_i=x_i^{st}} = 0 \iff (x_i^{st} \text{ is a stationary point}). \tag{377}$$

In the current context, this formal condition has a clear physical interpretation. The classical theory tells us that the derivative of the action over the endpoints equals the momentum at that point,

$$\frac{\partial S_{cl}(x_{i-1},s_{i-1}|x_i,s_i)}{\partial x_i} = -p_{x_i \to x_{i-1}}^{beg}; \quad \frac{\partial S_{cl}(x_i,s_i|x_{i+1},s_{i+1})}{\partial x_i} = p_{x_{i+1} \to x_i}^{end}. \tag{378}$$

Here, the subscript $x_i \to x_{i-1}$ ($x_{i+1} \to x_i$) and the superscript 'beg' ('end') indicate that it is the initial (terminal) momentum for the path that goes from $x_i$ to $x_{i-1}$ (from $x_{i+1}$ to $x_i$). In general, for an arbitrary value of $x_i$, there is no reason why the terminal momentum of path $x_{i+1} \to x_i$ should be equal the initial momentum of the following path $x_i \to x_{i-1}$; even though the two path segments begin or end at the same position, they are otherwise independent of each other. There is one unique situation, however, when the two momenta align perfectly. The momentum is always continuous (i.e. there are no "jumps") along any one classical path, like, e.g., $x_{i+1} \to x_i$ or $x_i \to x_{i-1}$; therefore, $p_{x_i \to x_{i-1}}^{beg} = p_{x_{i+1} \to x_i}^{end}$ only when the position $x_i$ happens

to lie on the path that goes *directly* from $x_{i+1}$ to $x_{i-1}$. That is, the interference effects will not kill the integral of form (376), only when the eigenvalue $f_i$ and the corresponding interval $\Gamma(f_i)$ intersects with the classical path $x_{i-1} \to x_{i+1}$. We can, of course, apply the exact same reasoning to all of the integrals over the in-between points $x_i$ and $\bar{x}_i$. As a result, we arrive at the conclusion that the quasi-probability does not vanish due to destructive interference only for one sequence of arguments $f_1, f_2, \ldots, f_k$ and the corresponding sequence of intervals $\Gamma(f_1), \ldots, \Gamma(f_k)$ that happen to intersect with the single classical path that goes from $x_0$ directly to $x_1$; the same is true for $\bar{f}_1, \ldots, \bar{f}_k$ and the path $\bar{x}_0 \to \bar{x}_1$. The terminal point of $x_0 \to x_1$ and $\bar{x}_0 \to \bar{x}_1$ is the same by default, $x_1 = \bar{x}_1$ [see Eq. (371)], but the initial positions, $x_0$ and $\bar{x}_0$, are determined by the initial state. Therefore, if, in addition to the least action approximation, we demand that the environment is *not* allowed to be initialized in a Schrödinger's cat type of state, i.e. $\langle x_0|\hat{\rho}_E|\bar{x}_0\rangle = \delta(x_0 - \bar{x}_0)\rho(x_0)$, then the paths will overlap, $(x_0 \to x_1) = (\bar{x}_0 \to \bar{x}_1)$, and thus, the corresponding arguments will overlap as well, $\forall_i f_i = \bar{f}_i$. This, of course, means that the interference terms $\Phi_F^{(k)}$ vanish and the surrogate field representation is valid.

### 7.3.3 Open environment

In our last example, we have an environment that can be further split into two interacting subsystems, $A$ and $B$,

$$\hat{H}_E = \hat{H}_{AB} = \hat{H}_A \otimes \hat{1} + \hat{1} \otimes \hat{H}_B + \mu \hat{V}_A \otimes \hat{V}_B \,; \quad \hat{\rho}_E = \hat{\rho}_A \otimes \hat{\rho}_B \,. \tag{379}$$

However, only $A$ is in contact with the outside world, i.e. the coupling $\hat{F}$ is an $A$-only observable,

$$\hat{F}_I(s) = \sum_f f \hat{P}_I(f,s) \otimes \hat{1} = \sum_f f e^{is\hat{H}_A}\hat{P}(f)e^{-is\hat{H}_A} \otimes \hat{1} \,. \tag{380}$$

Essentially, in this model, $A$ is an environment to $S$, and $B$ is an environment to $A$ but not to $S$.

We rewrite the quasi-probability associated with $\hat{F}$ in a form that will prove to be more convenient in the current context,

$$Q_F^{(k)}(\boldsymbol{f}, \bar{\boldsymbol{f}}, \boldsymbol{s}) = \text{tr}_{A+B}\left(\hat{P}(f_1) \otimes \hat{1}e^{-i(s_1-s_2)\hat{H}_{AB}} \cdots \hat{\rho}_A \otimes \hat{\rho}_B \cdots e^{i(s_1-s_2)\hat{H}_{AB}}\hat{P}(\bar{f}_1) \otimes \hat{1}\right)$$
$$= \text{tr}_{A+B}\left(\mathcal{P}(f_1, \bar{f}_1) \otimes \bullet \, e^{-i(s_1-s_2)[\hat{H}_{AB}, \bullet]} \cdots \mathcal{P}(f_k, \bar{f}_k) \otimes \bullet \, e^{-is_k[\hat{H}_{AB}, \bullet]}\hat{\rho}_A \otimes \hat{\rho}_B\right), \tag{381}$$

where we have introduced the $A$-only projection super-operator

$$\mathcal{P}(f, \bar{f}) = \hat{P}(f) \bullet \hat{P}(\bar{f}) \,, \tag{382}$$

and, for future reference, we also define its interaction picture,

$$\mathcal{P}_I(f, \bar{f}, s) = \hat{P}_I(f, s) \bullet \hat{P}_I(\bar{f}, s) = e^{is[\hat{H}_A, \bullet]}\mathcal{P}(f, \bar{f})e^{-is[\hat{H}_A, \bullet]} \,. \tag{383}$$

Next, we switch the $AB$ unitary maps in between the projectors to their interaction pictures (see assignment 10 of Sec. 2.3)

$$e^{-i(s_i-s_{i+1})[\hat{H}_{AB}, \bullet]} = e^{-is_i[\hat{H}_A, \bullet]} \otimes e^{-is_i[\hat{H}_B, \bullet]}T_{\mathcal{V}_{AB}}e^{-i\mu\int_{s_{i+1}}^{s_i}\mathcal{V}_{AB}(u)du}e^{is_{i+1}[\hat{H}_A, \bullet]} \otimes e^{is_{i+1}[\hat{H}_B, \bullet]} \,; \tag{384a}$$

$$\mathcal{V}_{AB}(u) = [(e^{iu\hat{H}_A}\hat{V}_A e^{-iu\hat{H}_A}) \otimes (e^{iu\hat{H}_B}\hat{V}_B e^{-iu\hat{H}_B}), \bullet] = [\hat{V}_A(u) \otimes \hat{V}_B(u), \bullet] \,. \tag{384b}$$

Given the spectral decomposition of $B$-side coupling operator $\hat{V}_B$,

$$\hat{V}_B(u) = \sum_b b\hat{P}_I(b, u) = \sum_b b e^{iu\hat{H}_B}\hat{P}(b)e^{-iu\hat{H}_B} \,; \tag{385a}$$

$$\mathcal{P}(b, \bar{b}) = \hat{P}(b) \bullet \hat{P}(\bar{b}) \,; \tag{385b}$$

$$\mathcal{P}_I(b, \bar{b}, u) = \hat{P}_I(b, u) \bullet \hat{P}_I(\bar{b}, u) = e^{iu[\hat{H}_B, \bullet]}\mathcal{P}(b, \bar{b})e^{-iu[\hat{H}_B, \bullet]} \,, \tag{385c}$$

we can rewrite the interaction picture maps in a format that closely resembles some of the intermediate steps we have went through when deriving the quasi-probability representation for dynamical maps (see assignment 28 of Sec. 4.3),

$$
T_{\mathcal{V}_{AB}} e^{-i\mu \int_{s_{i+1}}^{s_i} \mathcal{V}_{AB}(u)du} = \sum_{n=0}^{\infty} (-i\mu)^n \int_{s_{i+1}}^{s_i} du_1 \cdots \int_{s_{i+1}}^{u_{n-1}} du_n
$$
$$
\times \sum_{b,\bar{b}} \mathcal{W}_A(b_1,\bar{b}_1,u_1)\cdots\mathcal{W}_A(b_n,\bar{b}_n,u_n) \otimes \mathcal{P}_I(b_1,\bar{b}_1,u_1)\cdots\mathcal{P}_I(b_n,\bar{b}_n,u_n),
$$
(386)

where $\mathcal{W}_A$ is an $A$-only super-operator defined as

$$
\mathcal{W}_A(b,\bar{b},u) = b\hat{V}_A(u)\bullet - \bullet \hat{V}_A(u)\bar{b}.
$$
(387)

We substitute this form for every instance of unitary map in $Q_F^{(k)}$ and we get

$$
Q_F^{(k)}(\boldsymbol{f},\bar{\boldsymbol{f}},\boldsymbol{s}) = \sum_{n_1=0}^{\infty}\cdots\sum_{n_k=0}^{\infty}(-i\mu)^{\sum_{i=1}^k n_i}\left(\int_{s_2}^{s_1}du_1^1\cdots\int_{s_2}^{u_{n_1-1}^1}du_{n_1}^1\right)\cdots\left(\int_0^{s_k}du_1^k\cdots\int_0^{u_{n_k-1}^k}du_{n_k}^k\right)
$$
$$
\times \sum_{\boldsymbol{b}^1,\bar{\boldsymbol{b}}^1}\cdots\sum_{\boldsymbol{b}^k,\bar{\boldsymbol{b}}^k}\mathrm{tr}_B\left(\mathcal{P}_I(b_1^1,\bar{b}_1^1,u_1^1)\cdots\mathcal{P}_I(b_{n_k}^k,\bar{b}_{n_k}^k,u_{n_k}^k)\hat{\rho}_B\right)
$$
$$
\times \mathrm{tr}_A\left(\mathcal{P}_I(f_1,\bar{f}_1,s_1)\mathcal{W}_A(b_1^1,\bar{b}_1^1,u_1^1)\cdots\mathcal{W}_A(b_{n_k}^k,\bar{b}_{n_k}^k,u_{n_k}^k)\hat{\rho}_A\right).
$$
(388)

Although the expression looks quite intimidating, we have achieved our goal: the totality of dependence on $B$ in now confined into the joint quasi-probability distribution associated with the $B$-side of $AB$ coupling $\hat{V}_B$,

$$
Q_{V_B}^{(n_1+\cdots+n_k)}(b_1^1,\bar{b}_1^1,u_1^1;\ldots;u_{n_k}^k,\bar{b}_{n_k}^k,u_{n_k}^k)
$$
$$
= \mathrm{tr}_B\left(\hat{P}_I(b_1^1,u_1^1)\cdots\hat{P}_I(b_{n_k}^k,u_{n_k}^k)\hat{\rho}_B\hat{P}_I(\bar{b}_{n_k}^k,u_{n_k}^k)\cdots\hat{P}_I(\bar{b}_1^1,u_1^1)\right)
$$
$$
= \mathrm{tr}_B\left(\mathcal{P}_I(b_1^1,\bar{b}_1^1,u_1^1)\cdots\mathcal{P}_I(b_{n_k}^k,\bar{b}_{n_k}^k,u_{n_k}^k)\hat{\rho}_B\right).
$$
(389)

Given how the newly discovered quasi-probability distributions enter the expression, we can adopt here the quasi-average notation,

$$
Q_F^{(k)}(\boldsymbol{f},\bar{\boldsymbol{f}},\boldsymbol{s}) = \mathrm{tr}_A\left(\left\langle\mathcal{P}_I(f_1,\bar{f}_1,s_1)T_{\mathcal{W}_A}e^{-i\mu\int_{s_2}^{s_1}\mathcal{W}_A(B(u),\bar{B}(u),u)du}\cdots\right.\right.
$$
$$
\left.\left.\cdots\mathcal{P}_I(f_k,\bar{f}_k,s_k)T_{\mathcal{W}_A}e^{-i\mu\int_0^{s_k}\mathcal{W}_A(B(u),\bar{B}(u),u)du}\hat{\rho}_A\right\rangle\right).
$$
(390)

This is a general result—it is always possible cast $Q_F^{(k)}$ in this form when the environment itself is an open system.

Environment being open does not yet guarantee a valid surrogate representation, however. Instead, it opens new options for possible mechanisms for suppressing the interference terms in quasi-probabilities $Q_F^{(k)}$; here, we will explore one such option. The first step is to assume the weak coupling,

$$
\mu\tau_B \ll 1,
$$
(391)

where $\tau_B$ is the correlation time in $B$, defined in a standard way using $\{Q_{V_B}^{(k)}\}_{k=1}^{\infty}$ and related qumulant distributions $\{\tilde{Q}_{V_B}^{(k)}\}_{k=0}^{\infty}$ (see Sec. 4.7). This assumption has an immediate effect: a segment of evolution in between projections,

$$\cdots \mathcal{P}_I(f_i, \bar{f}_i, s_i) T_{\mathcal{W}_A} e^{-i\mu \int_{s_{i+1}}^{s_i} \mathcal{W}_A(B(u), \bar{B}(u), u) du} \mathcal{P}_I(f_{i+1}, \bar{f}_{i+1}, s_{i+1}) \cdots , \tag{392}$$

should last for a period much longer than the correlation time, $s_i - s_{i+1} \gg \tau_B$, or otherwise $\mu(s_i - s_{i+1}) \ll 1$ and any effects of the segment would be negligible.

Now, observe that each non-negligible segment involves the quasi-process $(B(u), \bar{B}(u))$ evaluated within the corresponding interval $(s_i, s_{i+1})$. Then, it follows that the values of $(B(u), \bar{B}(u))$ in any one segment *decorrelate* from the values in any other segment, because each interval is much longer than the correlation time. Formally, we can parametrize this effect by writing

$$Q_{V_B}^{(n_1 + \cdots + n_k)}(b_1^1, \bar{b}_1^1, u_1^1; \ldots; b_{n_k}^k, \bar{b}_{n_k}^k, u_{n_k}^k) \approx \prod_{i=1}^{k} Q_{V_B}^{(n_i)}(b_1^i, \bar{b}_1^i, u_1^i; \ldots; b_{n_i}^i, \bar{b}_{n_i}^i, u_{n_i}^i), \tag{393}$$

when $u_1^i, \ldots, u_{n_i}^i \in (s_i, s_{i+1})$. To understand where this equation came from, consider the following example. For simplicity, assume that $B$ is a Gaussian environment; then, the quasi-probability $Q_{V_B}^{(k)}$ can be written as a combination of products of the second-order qumulant distributions, e.g.,

$$\begin{aligned} Q_{V_B}^{(4)}(\boldsymbol{b}, \bar{\boldsymbol{b}}, \boldsymbol{u}) = {} & \tilde{Q}_{V_B}^{(2)}(b_1, \bar{b}_1, u_1; b_2, \bar{b}_2, u_2) \tilde{Q}_{V_B}^{(2)}(b_3, \bar{b}_3, u_3; b_4, \bar{b}_4, u_4) \\ & + \tilde{Q}_{V_B}^{(2)}(b_1, \bar{b}_1, u_1; b_3, \bar{b}_3, u_3) \tilde{Q}_{V_B}^{(2)}(b_2, \bar{b}_2, u_2; b_4, \bar{b}_4, u_4) \\ & + \tilde{Q}_{V_B}^{(2)}(b_1, \bar{b}_1, u_1; b_4, \bar{b}_4, u_4) \tilde{Q}_{V_B}^{(2)}(b_2, \bar{b}_2, u_2; b_3, \bar{b}_3, u_3). \end{aligned} \tag{394}$$

(We are ignoring here $\tilde{Q}_{V_B}^{(1)}$ because we can either assume $\langle B(u) \rangle = 0$ or we can simply switch to the central picture.) Now imagine that half of the arguments belong to interval $(s_1, s_2)$ and the other half to $(s_2, s_3)$; since $\tau_B$ is short, the qumulants that connect times from different intervals will vanish and we are left with,

$$\begin{aligned} & Q_{V_B}^{(4)}(b_1^1, \bar{b}_1^1, u_1^1; b_2^1, \bar{b}_2^1, u_2^1; b_1^2, \bar{b}_1^2, u_1^2; b_2^2, \bar{b}_2^2, u_2^2) \\ & \approx \tilde{Q}_{V_B}^{(2)}(b_1^1, \bar{b}_1^1, u_1^1; b_2^1, \bar{b}_2^1, u_2^1) \tilde{Q}_{V_B}^{(2)}(b_1^2, \bar{b}_1^2, u_1^2; b_2^2, \bar{b}_2^2, u_2^2) \\ & = Q_{V_B}^{(2)}(b_1^1, \bar{b}_1^1, u_1^1; b_2^1, \bar{b}_2^1, u_2^1) Q_{V_B}^{(2)}(b_1^2, \bar{b}_1^2, u_1^2; b_2^2, \bar{b}_2^2, u_2^2). \end{aligned} \tag{395}$$

The edge cases when $u_{n_i}^i$ approaches its lower limit $s_{i+1}$ and it meets with $u_1^{i+1}$ that approaches the upper limit, can be neglected because their contribution to the overall time integral is insignificant in comparison to the bulk of integrations sweeping the interior of the intervals $(s_i, s_{i+1})$. The same kind of argument as above (but with more terms) can also be made for non-Gaussian $B$.

Ultimately, the factorization of distributions (393) means that the quasi-average of the whole sequence of projections and evolution segments (390) can be approximated with an independent quasi-average of each individual segment,

$$\begin{aligned} Q_F^{(k)}(\boldsymbol{f}, \bar{\boldsymbol{f}}, \boldsymbol{s}) \approx \mathrm{tr}_A \Big( & \mathcal{P}_I(f_1, \bar{f}_1, s_1) \Big\langle T_{\mathcal{W}_A} e^{-i\mu \int_{s_2}^{s_1} \mathcal{W}_A(B(u), \bar{B}(u), u) du} \Big\rangle \cdots \\ & \cdots \mathcal{P}_I(f_k, \bar{f}_k, s_k) \Big\langle T_{\mathcal{W}_A} e^{-i\mu \int_0^{s_k} \mathcal{W}_A(B(u), \bar{B}(u), u) du} \Big\rangle \hat{\rho}_A \Big). \end{aligned} \tag{396}$$

Now that the segments are quasi-averaged, their form is the same as a dynamical map defined for the respective time interval $(s_i, s_{i+1})$. Moreover, since we are already assuming $\mu \tau_B \ll 1$

and $s_i - s_{i+1} \gg \tau_B$, we are obliged to apply to those pseudo-maps the full suite of Davies approximations from Sec. 5.3,

$$\left\langle T_{\mathcal{W}_A} e^{-i\mu \int_{s_{i+1}}^{s_i} \mathcal{W}_A(B(u),\bar{B}(u),u)du} \right\rangle \approx e^{\mu^2(s_i-s_{i+1})\mathcal{L}_D}, \tag{397}$$

which, as a side effect, allows us to treat the segments as a proper dynamical maps in the GKLS form. In summary, we have shown that, for open environment, when $\mu\tau_B \ll 1$, the quasi-probability associated with coupling $\hat{F}$ has a general form

$$Q_F^{(k)}(\boldsymbol{f},\bar{\boldsymbol{f}},\boldsymbol{s}) \approx \mathrm{tr}_A\left(\mathcal{P}(f_1,\bar{f}_1)e^{(s_1-s_2)\mathcal{L}_{\mathrm{GKLS}}} \cdots \mathcal{P}(f_k,\bar{f}_k)e^{s_k\mathcal{L}_{\mathrm{GKLS}}}\hat{\rho}_A\right). \tag{398}$$

Again, this model, when the form of generator $\mathcal{L}_{\mathrm{GKLS}}$ is arbitrary, does not guarantee surrogate representation by default. The most straightforward way to obtain a valid surrogate field from this point is to demand that the generator is block-diagonal with respect to the split between sub-spaces spanned by projectors and cohereneces, i.e.

$$\forall_{f,f'\neq\bar{f}'}\ \mathcal{P}(f,f)\mathcal{L}_{\mathrm{GKLS}}\mathcal{P}(f',\bar{f}') = 0. \tag{399}$$

It is easy to verify by direct calculation that when this condition is satisfied, the interference terms in (398) are automatically eliminated. A simple example of such a generator is found for a two-level system $A$,

$$\hat{F} = \frac{1}{2}\hat{\sigma}_z = \sum_{f=\pm 1/2} f\,|\mathrm{sign}(f)\rangle\langle\mathrm{sign}(f)|;\quad \mathcal{L}_{\mathrm{GKLS}} = -\frac{w}{2}[\hat{\sigma}_x,[\hat{\sigma}_x,\bullet]]. \tag{400}$$

The resultant joint probabilities have a familiar form,

$$Q_F^{(k)}(\boldsymbol{f},\bar{\boldsymbol{f}},\boldsymbol{s}) = \delta_{\boldsymbol{f},\bar{\boldsymbol{f}}}\left(\prod_{i=1}^{k-1}\frac{1+\mathrm{sign}(f_i)\,\mathrm{sign}(f_{i+1})e^{-2w(s_i-s_{i+1})}}{2}\right)\langle\mathrm{sign}(f_k)|e^{s_k\mathcal{L}_{\mathrm{GKLS}}}\hat{\rho}_A|\mathrm{sign}(f_k)\rangle, \tag{401}$$

that is, we have arrived here at the distribution of random telegraph noise (e.g., Sec. 3.1.3).

## 7.4 Objectivity of surrogate fields

As we have pointed out previously (e.g., see Sec. 4.5), joint quasi-probability distributions are completely independent of the open system $S$. Aside from any other arguments for this statement, the most straightforward proof comes from their very definition,

$$Q_F^{(k)}(\boldsymbol{f},\bar{\boldsymbol{f}},\boldsymbol{s}) = \mathrm{tr}_E\left(\hat{P}_I(f_1,s_1)\cdots\hat{P}_I(f_k,s_k)\hat{\rho}_E\hat{P}_I(\bar{f}_k,s_k)\cdots\hat{P}_I(\bar{f}_1,s_1)\right),$$

where $\hat{P}_I(f,s) = e^{is\hat{H}_E}\hat{P}(f)e^{-is\hat{H}_E}$ and $\hat{F} = \sum_f f\hat{P}(f)$. Clearly, the quasi-probability depends exclusively on the dynamical properties of the environment ($\hat{H}_E$ and $\hat{\rho}_E$) and the observable in $E$, $\hat{F}$, the distribution is associated with. Therefore, whether the surrogate field representation is valid, is determined solely by the relationship between the dynamics of $E$ and the eigenket structure of operator $\hat{F}$; no property of $S$, even the $S$-side coupling $\hat{V}$, has any influence over the surrogate's validity nor any of its features. Consequently, when the observable $\hat{F}_I(t)$ has a valid surrogate representation $F(t)$, the dynamics of any arbitrary open system $S$ (defined by the choice of $\hat{H}_S$ and $\hat{V}$), that couples to $E$ via the interaction $\lambda\hat{V}\otimes\hat{F}$, can be simulated with the stochastic map driven by the same stochastic process $F(t)$. If we were to consider various systems placed in the role of $S$ as observers of the surrogate field, then they all would report experiencing the same stochastic process. Hence, we can say that the surrogate representation is *inter-subjective*—all observers of the surrogate field are in the agreement about their subjective experiences.

However, to consider the surrogate field as a truly *objective* entity (in a conventionally classical sense) it should also be possible to observe/measure the field—its trajectories—directly, without the involvement of any intermediaries like the open system $S$. Of course, it is not, by any means, obvious that the surrogate field (or its trajectories) could even be considered outside the specific context of open system dynamics. On the face of it, it is quite possible that the field's trajectories are nothing more than the artifact of the surrogate representation. After all, within the context of open systems, the state of $S$ is always averaged over trajectories, and so, there is no measurement on $\hat{\rho}_S(t)$ that could give access to a single realization of the field. However, we have already seen that joint quasi-probabilities can be found in contexts other than open systems, see Sec. 4.5. So, if we do not want to probe the surrogate with the help of intermediary system $S$, we could try measuring the observable $\hat{F}$ directly. Let us then consider a setup for an observation of $\hat{F}$ over period of time. We will carry out a sequence of projective measurements of observable $\hat{F}$ preformed directly on system $E$ [15]. We assume that the physical process of single measurement takes so little time to complete that it can be considered instantaneous. We also assume that the measurement is not destructive, so that the state of $E$ continues to evolve freely after the procedure is concluded, albeit the act of measurement changes the density matrix in line with the collapse rule.

We begin with the first measurement performed after duration $t_1 > 0$. According to the Born rule, the probability of measuring the result $f_1$ is then given by,

$$\text{Prob}(f_1) = \text{tr}\left(\hat{P}(f_1) e^{-it_1 \hat{H}_E} \hat{\rho}_E \, e^{it_1 \hat{H}_E}\right) = \text{tr}\left(\hat{P}_I(f_1, t_1) \hat{\rho}_E\right) = Q_F^{(1)}(f_1, f_1, t_1) = P_F^{(1)}(f_1, t_1), \quad (402)$$

and the *posterior* density matrix, after the act of measuring eigenvalue $f_1$ collapses the state, is

$$\hat{\rho}(t|f_1) = \frac{e^{-i(t-t_1)\hat{H}_E}\hat{P}(f_1)e^{-it_1\hat{H}_E}\hat{\rho}_E \, e^{it_1\hat{H}_E}\hat{P}(f_1)e^{i(t-t_1)\hat{H}_E}}{P_F^{(1)}(f_1, t_1)} = \frac{e^{-it\hat{H}_E}\hat{q}(f_1, t_1)e^{it\hat{H}_E}}{P_F^{(1)}(f_1, t_1)}. \quad (403)$$

(The operator $\hat{q}$ was first used in the solution to the assignment 39 of Sec. 7.2.) After a period of free evolution, we follow up with the second measurement performed at time $t_2$; the probability of measuring the result $f_2$ at time $t_2$, given that $f_1$ was measured at $t_1 < t_2$, is then given by,

$$\text{Prob}(f_2|f_1) = \text{tr}\left(\hat{P}(f_2)\hat{\rho}(t_2|f_1)\right) = \frac{Q_F^{(2)}(f_2, f_2, t_2; f_1, f_1, t_1)}{P_F^{(1)}(f_1, t_1)} = \frac{P_F^{(2)}(f_2, t_2; f_1, t_1)}{P_F^{(1)}(f_1, t_1)}. \quad (404)$$

Therefore, the probability of measuring a sequence of results $(f_2, f_1)$, such that $f_1$ was obtained at $t_1$ and $f_2$ at $t_2$, is calculated according to the Bayes law

$$\text{Prob}[(f_2, f_1)] = \text{Prob}(f_2|f_1)\text{Prob}(f_1) = \frac{P_F^{(2)}(f_2, t_2; f_1, t_1)}{P_F^{(1)}(f_1, t_1)}P_F^{(1)}(f_1, t_1) = P_F^{(2)}(f_2, t_2; f_1, t_1). \quad (405)$$

Predictably, the *posterior* state is given by

$$\hat{\rho}(t|f_1, f_2) = \frac{e^{-i(t-t_2)\hat{H}_E}\hat{P}(f_2)\hat{\rho}(t_2|f_1)\hat{P}(f_2)e^{i(t-t_2)\hat{H}_E}}{\text{Prob}(f_2|f_1)} = \frac{e^{-it\hat{H}_E}\hat{q}(f_2, t_2; f_1, t_1)e^{it\hat{H}_E}}{P_F^{(2)}(f_2, t_2; f_1, t_1)}. \quad (406)$$

The emerging patter should now be clear. When this procedure is continued over consecutive steps, we obtain the sequence of results $(f_k, f_{k-1}, \ldots, f_1)$, each taken at the corresponding point in time, $t_1 < t_2 < \cdots < t_k$; the probability of measuring any such a sequence reads

$$\text{Prob}[(f_k, \ldots, f_1)] = P_F^{(k)}(f_k, t_k; \ldots; f_1, t_1) = Q_F^{(k)}(f_k, f_k, t_k; \ldots; f_1, f_1, t_1). \quad (407)$$

That is, the probability of measuring the sequence of length $k$ equals the diagonal part of the joint quasi-probability distribution $Q_F^{(k)}$.

So far, we have not specified if the surrogate representation of $\hat{F}$ is valid. Hence, the above result is general—we have just identified another context where the joint quasi-probability distributions appear naturally. When we do assume that the representation is valid, then $\text{Prob}[(f_k, \ldots, f_1)]$ becomes identical with the joint probability distribution for the trajectory $f(t)$ of the surrogate field $F(t)$ to pass through all the measured values at the corresponding times,

$$f_1 = f(t_1),\ f_2 = f(t_2),\ \ldots\ f_k = f(t_k).$$

In other words, when the surrogate representation is valid, the sequential projective measurements of $\hat{F}$ are equivalent—in the sense of equal probability distributions—to sampling a single trajectory of the surrogate field on a discrete time grid.

Does all of this mean that it is possible to observe a single trajectory of a surrogate field? Was this a strong enough argument for the objectivity of the surrogate fields? At the moment, we are not sure that we can easily answer those, ultimately, philosophical questions. However, there is something that we can say here for certain. The fact that trajectories and measured sequences are identically distributed has important practical implications for problems involving stochastic maps, and stochastic processes, in general. As we have shown previously (see Sec. 3.6.4), having access to the ensemble of sample trajectories trivializes any computation of stochastic averages by enabling the sample average approximation. From the technical point of view, it does not matter what is the exact nature of the used samples; the only thing that does matter, is that the samples are distributed identically to the probability distribution the sample average is supposed to approximate. Of course, this is exactly what we have shown here: the measured sequences have the same distribution as the joint probability distributions of the surrogate field. Therefore, for the purpose of the sample average, the measured sequences can be used in place of the sample trajectories.

# A Solutions to assignments

**Assignment 1 (Sec 2.1)** *Use the definition* (2),

$$\hat{U}(t) = e^{-it\hat{H}} = \hat{1} + \sum_{k=1}^{\infty} \frac{(-it)^k}{k!} \hat{H}^k = \sum_{k=0}^{\infty} \frac{(-it)^k}{k!} \hat{H}^k,$$

*to calculate the evolution operator for the Hamiltonian*

$$\hat{H} = \frac{\omega}{2} \hat{\sigma}_z,$$

*where $\hat{\sigma}_z$ is one of the Pauli operators,*

$$\hat{\sigma}_z = \begin{bmatrix} 1 & 0 \\ 0 & -1 \end{bmatrix}; \quad \hat{\sigma}_x = \begin{bmatrix} 0 & 1 \\ 1 & 0 \end{bmatrix}; \quad \hat{\sigma}_y = \begin{bmatrix} 0 & -i \\ i & 0 \end{bmatrix}.$$

*Solution*: Pauli operators are special because the set $\{\hat{\sigma}_x, \hat{\sigma}_y, \hat{\sigma}_z, \hat{\sigma}_0 = \hat{1}\}$ is closed with respect to matrix multiplication, i.e. products of Pauli matrices can always be expressed as a linear combination of themselves (plus the identity); in particular,

$$\hat{\sigma}_a \hat{\sigma}_b = \delta_{a,b} \hat{1} + i \sum_{c=x,y,z} \epsilon_{abc} \hat{\sigma}_c.$$

Here, the Levi-Civita symbol $\epsilon_{abc}$ is totally antisymmetric in its indexes (when any pair of indexes is exchanged the symbol changes sign), $\epsilon_{xyz} = 1$, and $\epsilon_{abc} = 0$ when any indexes repeat (the consequence of total antisymmetry).

According to the definition (2), the evolution operator is given by a power series,

$$e^{-i\frac{\omega t}{2}\hat{\sigma}_z} = \sum_{k=0}^{\infty} \frac{(-i)^k}{k!}\left(\frac{\omega t}{2}\right)^k \hat{\sigma}_z^k.$$

This can be simplified because we can express an arbitrary power of Pauli operator, $\hat{\sigma}_z^k$, through combinations of the first powers of Pauli operators; in our case we note

$$\hat{\sigma}_z^{2n} = (\hat{\sigma}_z^2)^n = \hat{1}^n = \hat{1}.$$

Now, we split the series into the even and the odd branch,

$$\sum_{k=0}^{\infty} \frac{(-i)^k}{k!}\left(\frac{\omega t}{2}\right)^k \hat{\sigma}_z^k = \sum_{n=0}^{\infty} \frac{(-1)^n}{(2n)!}\left(\frac{\omega t}{2}\right)^{2n} \hat{\sigma}_z^{2n} - i\hat{\sigma}_z \sum_{n=0}^{\infty} \frac{(-1)^n}{(2n+1)!}\left(\frac{\omega}{2}\right)^{2n+1} \hat{\sigma}_z^{2n}$$

$$= \cos\left(\frac{\omega t}{2}\right)\hat{1} - i\sin\left(\frac{\omega t}{2}\right)\hat{\sigma}_z,$$

and thus,

$$\hat{U}(t) = \cos\left(\frac{\omega t}{2}\right)\hat{1} - i\sin\left(\frac{\omega t}{2}\right)\hat{\sigma}_z = \begin{bmatrix} e^{-i\frac{\omega t}{2}} & 0 \\ 0 & e^{i\frac{\omega t}{2}} \end{bmatrix}.$$

As we mentioned before, Pauli matrices are special; do not expect to be able to compute the series (2) using similar tricks for general Hamiltonians.

**Assignment 2 (Sec. 2.2.1)** *Show that neither*

$$e^{-it\hat{H}(t)}|\Psi(0)\rangle,$$

*nor*

$$e^{-i\int_0^t \hat{H}(s)ds}|\Psi(0)\rangle,$$

*are the solution to time-dependent Schrödinger equation.*

*Solution*: Compute the derivative of $\exp[-it\hat{H}(t)]|\Psi(0)\rangle$,

$$\frac{d}{dt}e^{-it\hat{H}(t)}|\Psi(0)\rangle = \sum_{k=1}^{\infty} \frac{d}{dt}\left[\frac{(-it)^k}{k!}\hat{H}(t)^k\right]|\Psi(0)\rangle$$

$$= \sum_{k=1}^{\infty}\left[-i\frac{(-it)^{k-1}}{(k-1)!}\hat{H}^k(t) + \frac{(-it)^k}{k!}\sum_{l=1}^{k}\hat{H}(t)^{l-1}\frac{d\hat{H}(t)}{dt}\hat{H}(t)^{k-l}\right]|\Psi(0)\rangle$$

$$= -i\hat{H}(t)e^{-it\hat{H}(t)}|\Psi(0)\rangle + \sum_{k=1}^{\infty}\sum_{l=1}^{k}\frac{(-it)^k}{k!}\hat{H}(t)^{l-1}\frac{d\hat{H}(t)}{dt}\hat{H}(t)^{k-l}|\Psi(0)\rangle$$

$$\neq -i\hat{H}(t)e^{-it\hat{H}(t)}|\Psi(0)\rangle.$$

As indicated, the RHS is not of the form of Eq. (10). Similarly,

$$\frac{d}{dt}e^{-i\int_0^t \hat{H}(s)ds}|\Psi(0)\rangle = \sum_{k=1}^{\infty}\frac{(-i)^k}{k!}\sum_{l=1}^{\infty}\left(\int_0^t \hat{H}(s)ds\right)^{l-1}\hat{H}(t)\left(\int_0^t \hat{H}(s)ds\right)^{k-l}|\Psi(0)\rangle,$$

also does not satisfy the Schrödinger equation, unless $[\hat{H}(t),\hat{H}(s)] = 0$ for all $t,s$.

**Assignment 3 (Sec. 2.2.1)** *Verify by direct calculation that Eq. (11),*

$$|\Psi(t)\rangle = \hat{U}(t,s)|\Psi(s)\rangle,$$

*where*

$$\hat{U}(t,s) = \hat{1} + \sum_{k=1}^{\infty} (-i)^k \int_s^t du_1 \cdots \int_s^{u_{k-1}} du_k \, \hat{H}(u_1) \cdots \hat{H}(u_k),$$

*is the solution to the time-dependent Schrödinger equation and show that $\hat{U}(t,s)$ satisfies the time-dependent dynamical equation*

$$\frac{d}{dt}\hat{U}(t,s) = -i\hat{H}(t)\hat{U}(t,s),$$

*with the initial condition $\hat{U}(s,s) = \hat{1}$.*

*Solution*: The derivative over $t$ affects only the upper limit of the first integral in each term of the series,

$$
\begin{aligned}
\frac{d}{dt}|\Psi(t)\rangle &= \frac{d}{dt}\hat{U}(t,s)|\Psi(s)\rangle = \sum_{k=1}^{\infty}(-i)^k \frac{d}{dt}\int_s^t du_1 \cdots \int_s^{u_{k-1}} du_k \, \hat{H}(u_1)\cdots\hat{H}(u_k)|\Psi(s)\rangle \\
&= \sum_{k=1}^{\infty}\left(-i\hat{H}(t)\right)(-i)^{k-1}\int_s^t du_2 \cdots \int_s^{u_{k-1}} du_k \, \hat{H}(u_2)\cdots\hat{H}(u_k)|\Psi(s)\rangle \\
&= -i\hat{H}(t)\left(\hat{1} + \sum_{l=1}^{\infty}(-i)^l \int_s^t du_2 \cdots \int_s^{u_l} du_{l+1}\hat{H}(u_2)\cdots\hat{H}(u_{l+1})\right)|\Psi(s)\rangle \\
&= -i\hat{H}(t)\hat{U}(t,s)|\Psi(s)\rangle.
\end{aligned}
$$

Since the above equation is true for any initial state-ket $|\Psi(s)\rangle$, it is also true for the evolution operator itself, hence, the dynamical equation (13).

**Assignment 4 (Sec. 2.2.1)** *Find all $3! = 6$ orderings for $\int_s^t du_1 du_2 du_3$.*

*Solution*:

$$
\begin{aligned}
\int_s^t du_1 du_2 du_3 &= \left( \int_s^t du_1 \int_s^{u_1} du_2 + \int_s^t du_2 \int_s^{u_2} du_1 \right) \int_s^t du_3 \\
&= \left( \int_s^t du_1 \int_s^{u_1} du_3 + \int_s^t du_3 \int_s^{u_3} du_1 \right) \int_s^{u_1} du_2 \\
&\quad + \left( \int_s^t du_2 \int_s^{u2} du_3 + \int_s^t du_3 \int_s^{u_3} du_2 \right) \int_s^{u_2} du_1 \\
&= \int_s^t du_1 \int_s^{u_1} du_2 du_3 + \int_s^t du_3 \int_s^{u_3} du_1 \int_s^{u_1} du_2 + \int_s^t du_2 \int_s^{u_2} du_1 du_3 \\
&\quad + \int_s^t du_3 \int_s^{u_3} du_2 \int_s^{u_2} du_1 \\
&= \int_s^t du_1 \int_s^{u_1} du_2 \int_s^{u_2} du_3 + \int_s^t du_1 \int_s^{u_1} du_3 \int_s^{u_3} du_2 + \int_s^t du_3 \int_s^{u_3} du_1 \int_s^{u_1} du_2 \\
&\quad + \int_s^t du_2 \int_s^{u_2} du_1 \int_s^{u_1} du_3 + \int_s^t du_2 \int_s^{u_2} du_3 \int_s^{u_3} du_1 + \int_s^t du_3 \int_s^{u_3} du_2 \int_s^{u_2} du_1 .
\end{aligned}
$$

**Assignment 5 (Sec. 2.2.1)** *Compute the evolution operator for $\hat{H}(t) = \hat{H}_0$ (a constant Hamiltonian) and $\hat{H}(t) = f(t)\hat{H}_0$ and show that in these cases the time-order exp reduces to unordered exp.*

*Solution*: First, the constant Hamiltonian, $\hat{H}(t) = \hat{H}_0$,

$$
\begin{aligned}
\hat{U}(t,s) &= \hat{1} + \sum_{k=1}^{\infty} (-i)^k \int_s^t du_1 \cdots \int_s^{u_{k-1}} du_k \hat{H}_0^k = \hat{1} + \sum_{k=1}^{\infty} \frac{(-i)^k}{k!} \hat{H}_0^k \int_s^t du_1 \cdots du_k \\
&= \hat{1} + \sum_{k=1}^{\infty} \frac{(-i)^k}{k!} (t-s)^k \hat{H}_0^k = e^{-i(t-s)\hat{H}_0} = \hat{U}(t-s).
\end{aligned}
$$

Next, $\hat{H}(t) = f(t)\hat{H}_0$,

$$
\begin{aligned}
\hat{U}(t,s) &= \hat{1} + \sum_{k=1}^{\infty} (-i)^k \int_s^t du_1 \cdots \int_s^{u_{k-1}} du_k f(u_1)\hat{H}_0 \cdots f(u_k)\hat{H}_0 \\
&= \hat{1} + \sum_{k=1}^{\infty} \frac{(-i)^k}{k!} \hat{H}_0^k \left( \int_s^t f(u)du \right)^k = e^{-i\hat{H}_0 \int_s^t f(u)du}.
\end{aligned}
$$

**Assignment 6 (Sec. 2.2.1)** *Show that $\hat{U}^{\dagger}(t,s) = \hat{U}(s,t)$.*

*Solution*: Note that the dagger operation (the hermitian conjugate) reverses the order of operator products,

$$
\hat{U}^{\dagger}(t,s) = \left( \sum_{k=0}^{\infty} (-i)^k \int_s^t du_1 \cdots \int_s^{u_{k-1}} du_k\, \hat{H}(u_1) \cdots \hat{H}(u_k) \right)^{\dagger}
$$

$$
= \sum_{k=0}^{\infty} (-i)^k (-1)^k \int_s^t du_1 \cdots \int_s^{u_{k-1}} du_k\, \hat{H}(u_k) \cdots \hat{H}(u_1).
$$

In what follows, we will make an extensive use of the identity,

$$
\int_s^t du_i \int_s^{u_i} du_j = \int_s^t du_j \int_{u_j}^t du_i, \tag{A.1}
$$

which can be proven by comparing the two equivalent decompositions of unordered integrals: $\int_s^t du_i du_j = \int_s^t du_i \int_s^{u_i} du_j + \int_s^t du_j \int_s^{u_j} du_i$ and $\int_s^t du_i du_j = \int_s^t du_i \left( \int_{u_i}^t du_j + \int_s^{u_i} du_j \right)$. When one iterates this identity over a number of ordered integrals one obtains

$$
\int_s^t du_1 \cdots \int_s^{u_{k-1}} du_k = \int_s^t du_1 \cdots \int_s^{u_{k-3}} du_{k-2} \int_s^{u_{k-2}} du_k \int_{u_k}^{u_{k-2}} du_{k-1}
$$

$$
= \int_s^t du_1 \cdots \int_s^{u_{k-4}} du_{k-3} \int_s^{u_{k-3}} du_k \int_{u_k}^{u_{k-3}} du_{k-2} \int_{u_k}^{u_{k-2}} du_{k-1} = \ldots
$$

$$
= \int_s^t du_k \left( \int_{u_k}^t du_1 \cdots \int_{u_k}^{u_{k-2}} du_{k-1} \right)
$$

$$
= \int_s^t du_k \int_{u_k}^t du_{k-1} \left( \int_{u_{k-1}}^t du_3 \cdots \int_{u_{k-1}}^{u_{k-3}} du_{k-2} \right) = \ldots
$$

$$
= \int_s^t du_k \int_{u_k}^t du_{k-1} \cdots \int_{u_2}^t du_1,
$$

an effective reversal of the ordering. Going back to the series, we reverse the direction of each integration (thus absorbing the additional $(-1)^k$ factor), and we rename the integration variables,

$$
\hat{U}^{\dagger}(t,s) = \sum_{k=0}^{\infty} (-i)^k (-1)^k \int_s^t du_k \int_{u_k}^t du_{k-1} \cdots \int_{u_2}^t du_1\, \hat{H}(u_k) \cdots \hat{H}(u_1)
$$

$$
= \sum_{k=0}^{\infty} (-i)^k \int_t^s du_k \int_t^{u_k} du_{k-1} \cdots \int_t^{u_2} du_1\, \hat{H}(u_k) \cdots \hat{H}(u_1)
$$

$$
= \sum_{k=0}^{\infty} (-i)^k \int_t^s dv_1 \int_t^{v_1} dv_2 \cdots \int_t^{v_{k-1}} dv_k\, \hat{H}(v_1) \cdots \hat{H}(v_k)
$$

$$
= \hat{U}(s,t).
$$

**Assignment 7 (Sec. 2.2.2)** *Using only algebraic methods prove the composition rule,*

$$
\hat{U}(t,w)\hat{U}(w,s) = \hat{U}(t,s).
$$

*Solution*: Start with the series representation (21) of the composition of evolution operators,

$$
\hat{U}(t,w)\hat{U}(w,s) = \left( \hat{1} + \sum_{n=1}^{\infty} (-i)^n \int_w^t du_1 \cdots \int_w^{u_{n-1}} du_n \, \hat{H}(u_1)\cdots\hat{H}(u_n) \right)
$$
$$
\times \left( \hat{1} + \sum_{m=1}^{\infty} (-i)^m \int_s^w dv_1 \cdots \int_s^{v_{m-1}} dv_m \, \hat{H}(v_1)\cdots\hat{H}(v_m) \right)
$$
$$
= \hat{1} + \sum_{k=1}^{\infty} \left[ \int_w^t d\tau_1 \cdots \int_w^{\tau_{k-1}} d\tau_k + \sum_{j=1}^{k} \int_w^t d\tau_1 \cdots \int_w^{\tau_{j-2}} d\tau_{j-1} \int_s^w d\tau_j \cdots \int_s^{\tau_{k-1}} d\tau_k \right]
$$
$$
\times (-i)^k \hat{H}(\tau_1)\cdots\hat{H}(\tau_k).
$$

On the other hand, we have the series representation of $\hat{U}(t,s)$,

$$
\hat{U}(t,s) = \hat{1} + \sum_{k=1}^{\infty} \int_s^t d\tau_1 \cdots \int_s^{\tau_{k-1}} d\tau_k (-i)^k \hat{H}(\tau_1)\cdots\hat{H}(\tau_k).
$$

Now take the $k$-fold ordered integral and start bisecting the integration intervals,

$$
\int_s^t d\tau_1 \cdots \int_s^{\tau_{k-1}} d\tau_k = \left( \int_w^t + \int_s^w \right) d\tau_1 \int_s^{\tau_1} d\tau_2 \cdots \int_s^{\tau_{k-1}} d\tau_k
$$
$$
= \int_s^w d\tau_1 \cdots \int_s^{\tau_{k-1}} d\tau_k + \int_w^t d\tau_1 \left( \int_w^{\tau_1} + \int_s^w \right) d\tau_2 \cdots \int_s^{\tau_{k-1}} d\tau_k
$$
$$
= \int_s^w d\tau_1 \cdots \int_s^{\tau_{k-1}} d\tau_k + \int_w^t d\tau_1 \int_w^{\tau_1} d\tau_2 \cdots \int_s^{\tau_{k-1}} d\tau_k
$$
$$
+ \int_w^t d\tau_1 \int_w^{\tau_1} d\tau_2 \left( \int_w^{\tau_2} + \int_s^w \right) d\tau_3 \cdots \int_s^{\tau_{k-1}} d\tau_k
$$
$$
= \cdots
$$
$$
= \int_w^t d\tau_1 \cdots \int_w^{\tau_{k-1}} d\tau_k + \sum_{j=1}^{k} \int_w^t d\tau_1 \cdots \int_w^{\tau_{j-2}} d\tau_{j-1} \int_s^w d\tau_j \cdots \int_s^{\tau_{k-1}} d\tau_k.
$$

Hence, the $k$-fold integral splits into the combination identical to the one found in $k$th order term of the composition; this proves that the composition rule is correct.

**Assignment 8 (Sec. 2.3)** *Prove the disentanglement theorem by transforming the dynamical equation* (4),

$$
\frac{d}{dt}\hat{U}(t) = -i(\hat{H}_0 + \hat{V})\hat{U}(t),
$$

*to the interaction picture.*

*Solution*: Write down the LHS of the dynamical equation and substitute the form (33),

$$\hat{U}(t) = e^{-it\hat{H}_0}\hat{U}_I(t,0),$$

for $\hat{U}(t) = \exp[-it(\hat{H}_0 + \hat{V})]$,

$$\text{LHS} = \frac{d}{dt}e^{-it(\hat{H}_0+\hat{V})} = \frac{d}{dt}e^{-it\hat{H}_0}\hat{U}_I(t,0) = -i\hat{H}_0 e^{-it\hat{H}_0}\hat{U}_I(t,0) + e^{-it\hat{H}_0}\frac{d}{dt}\hat{U}_I(t,0),$$

then do the same for the RHS,

$$\text{RHS} = -i(\hat{H}_0 + \hat{V})e^{-it(\hat{H}_0+\hat{V})} = -i\hat{H}_0 e^{-it\hat{H}_0}\hat{U}_I(t,0) - i\hat{V}e^{-it\hat{H}_0}\hat{U}_I(t,0).$$

Multiply both sides by $\hat{U}_0^\dagger(t) = \hat{U}_0(-t) = \exp(it\hat{H}_0)$ and equate them afterwards,

$$e^{it\hat{H}_0}\,\text{LHS} = e^{it\hat{H}_0}\,\text{RHS} \;\Rightarrow$$

$$-ie^{it\hat{H}_0}\hat{H}_0 e^{-it\hat{H}_0}\hat{U}_I(t,0) + \frac{d}{dt}\hat{U}_I(t,0) = -ie^{it\hat{H}_0}\hat{H}_0 e^{-it\hat{H}_0}\hat{U}_I(t,0) - ie^{it\hat{H}_0}\hat{V}e^{-it\hat{H}_0}\hat{U}_I(t,0) \Rightarrow$$

$$\frac{d}{dt}\hat{U}_I(t,0) = -i\hat{V}_I(t)\hat{U}_I(t,0).$$

The last equation has the form of dynamical equation with time-dependent generator,

$$\hat{V}_I(t) = \hat{U}_0^\dagger(t)\hat{V}\hat{U}_0(t),$$

hence, the solution is a time-ordered exponential (35),

$$\hat{U}_I(t,0) = Te^{-i\int_0^t \hat{V}_I(s)ds} = \hat{1} + \sum_{k=1}^{\infty}(-i)^k \int_0^t ds_1 \cdots \int_0^{s_{k-1}} ds_k \hat{V}_I(s_1)\cdots\hat{V}_I(s_k).$$

**Assignment 9 (Sec. 2.3)** *Prove the disentanglement theorem for time-dependent Hamiltonian using only algebraic methods.*

*Solution*: We start by rearranging the terms in the series representation of evolution operator,

$$
\hat{U}(t,0) = \hat{1} + \sum_{k=0}^{\infty} (-i)^k \int_0^t ds_1 \cdots \int_0^{s_{k-1}} ds_k (\hat{H}_0(s_1) + \hat{V}(s_1)) \ldots (\hat{H}_0(s_k) + \hat{V}(s_k))
$$

$$
= \hat{1} + \sum_{k=1}^{\infty} (-i)^k \int_0^t ds_1 \cdots \int_0^{s_{k-1}} ds_k \, \hat{H}_0(s_1) \cdots \hat{H}_0(s_k)
$$

$$
+ \sum_{k=1}^{\infty} \sum_{j=1}^{k} (-i)^k \int_0^t ds_1 \cdots \int_0^{s_{k-1}} ds_k \, \hat{H}_0(s_1) \cdots \hat{V}(s_j) \cdots \hat{H}_0(s_k)
$$

$$
+ \sum_{k=2}^{\infty} \sum_{j=2}^{k} \sum_{i=1}^{j-1} (-i)^k \int_0^t ds_1 \cdots \int_0^{s_{k-1}} ds_k \, \hat{H}_0(s_1) \cdots \hat{V}(s_i) \cdots \hat{V}(s_j) \cdots \hat{H}_0(s_k)
$$

$$
+ \ldots
$$

$$
= \hat{U}_0(t,0)
$$

$$
+ \sum_{j=1}^{\infty} (-i)^j \int_0^t ds_1 \cdots \int_0^{s_{j-1}} ds_j \, \hat{H}_0(s_1) \cdots \hat{V}(s_j) \hat{U}_0(s_j,0)
$$

$$
+ \sum_{j=2}^{\infty} \sum_{i=1}^{j-1} (-i)^j \int_0^t ds_1 \cdots \int_0^{s_{j-1}} ds_j \, \hat{H}_0(s_1) \cdots \hat{V}(s_i) \cdots \hat{V}(s_j) \hat{U}_0(s_j,0)
$$

$$
+ \ldots
$$

Now, we make use of the identity (A.1) from Assignment 6,

$$
\int_0^{s_{j-2}} ds_{j-1} \int_0^{s_{j-1}} ds_j = \int_0^{s_{j-2}} ds_j \int_{s_j}^{s_{j-2}} ds_{j-1} \,,
$$

to move the variables of *V*'s to the front of the ordered integrals in accordance with

$$
\int_0^t ds_1 \cdots \int_0^{s_{j-1}} ds_j = \int_0^t ds_1 \cdots \int_0^{s_{j-3}} ds_{j-2} \int_0^{s_{j-2}} ds_j \int_{s_j}^{s_{j-2}} ds_{j-1} = \ldots = \int_0^t ds_j \int_{s_j}^t ds_1 \int_{s_j}^{s_1} ds_2 \cdots \int_{s_j}^{s_{j-2}} ds_{j-1} \,.
$$

When we apply this transformation to the series we get

$$\hat{U}(t,0) = \hat{U}_0(t,0)$$

$$+ \sum_{j=1}^{\infty}(-i)^j \int_0^t ds_j \int_{s_j}^t ds_1 \cdots \int_{s_j}^{s_{j-2}} ds_{j-1} \hat{H}_0(s_1)\cdots\hat{H}_0(s_{j-1})\hat{V}(s_j)\hat{U}_0(s_j,0) + \ldots$$

$$= \hat{U}_0(t,0) - i\int_0^t \hat{U}_0(t,s)\hat{V}(s)\hat{U}_0(s,0)ds$$

$$+ \sum_{j=1}^{\infty}(-i)^{j+1}\int_0^t ds\, \hat{U}_0(t,s)\hat{V}(s)\int_0^s ds_1 \cdots \int_0^{s_{j-1}} ds_j\, \hat{H}_0(s_1)\cdots\hat{H}_0(s_{j-1})\hat{V}(s_j)\hat{U}_0(s_j,0)$$

$$+ \ldots$$

$$= \hat{U}_0(t,0)\left(\hat{1} - i\int_0^t \hat{U}_0(0,s)\hat{V}(s)\hat{U}_0(s,0)ds\right)$$

$$+ \sum_{j=1}^{\infty}(-i)^{j+1}\int_0^t ds \int_0^s ds_j\, \hat{U}_0(t,s)\hat{V}(s)\int_{s_j}^s ds_1 \cdots \int_{s_j}^{s_{j-2}} ds_{j-1}\, \hat{H}_0(s_1)\cdots\hat{H}_0(s_{j-1})\hat{V}(s_j)\hat{U}_0(s_j,0)$$

$$+ \ldots$$

$$= \hat{U}_0(t,0)\left(\hat{1} - i\int_0^t \hat{V}_I(s)ds\right)$$

$$+ (-i)^2\int_0^t ds_1 \int_0^{s_1} ds_2\, \hat{U}_0(t,s_1)\hat{V}(s_1)\hat{U}_0(s_1,s_2)\hat{V}(s_2)\hat{U}_0(s_2,0) + \ldots$$

$$= \hat{U}_0(t,0)\left(\hat{1} - i\int_0^t \hat{V}_I(s)ds + (-i)^2\int_0^t ds_1 \int_0^{s_1} ds_2\, \hat{V}_I(s_1)\hat{V}_I(s_2) + \ldots\right)$$

$$= \hat{U}_0(t,0)T_{V_I}e^{-i\int_0^t \hat{V}_I(s)ds}.$$

**Assignment 10 (Sec. 2.3)** *Show that*

$$\hat{U}(t,s) = \hat{U}_0(t,s)T_{V_I}e^{-i\int_s^t \hat{V}_I(u,s)du} = \hat{U}_0(t,0)\left(T_{V_I}e^{-i\int_s^t \hat{V}_I(u)du}\right)\hat{U}_0^{\dagger}(s,0),$$

*where*

$$\hat{V}_I(u,s) \equiv \hat{U}_0^{\dagger}(u,s)\hat{V}(u)\hat{U}_0(u,s) \neq \hat{V}_I(u), \quad when \quad s \neq 0.$$

*Solution*: The solution to the dynamical equation (13) for $\hat{H}(t) = \hat{H}_0(t) + \hat{V}(t)$ when we set the initial condition to $\hat{U}(s,s) = \hat{1}$ is given by

$$\hat{U}(t,s) = T_H e^{-i\int_s^t \hat{H}(u)du}.$$

When we apply to this operator the disentanglement theorem to factor out

$$\hat{U}_0(t,s) = T_{H_0}e^{-i\int_s^t \hat{H}_0(u)du},$$

we get

$$
\begin{aligned}
\hat{U}(t,s) &= \hat{U}_0(t,s) T_{V_I} e^{-i\int_s^t \hat{V}_I(u,s)du} \\
&= \hat{U}_0(t,s) + \sum_{k=1}^{\infty}(-i)^k \int_s^t du_1 \cdots \int_s^{u_{k-1}} du_k\, \hat{U}_0(t,s)\hat{V}_I(u_1,s)\cdots\hat{V}_I(u_k,s) \\
&= \hat{U}_0(t,s) + \sum_{k=1}^{\infty}(-i)^k \int_s^t du_1 \cdots \int_s^{u_{k-1}} du_k\, \hat{U}_0(t,s) \\
&\quad \times \hat{U}_0(s,u_1)\hat{V}(u_1)\hat{U}_0(u_1,s)\hat{U}_0(s,u_2)\hat{V}(u_2)\hat{U}_0(u_2,s)\cdots\hat{U}_0(s,u_k)\hat{V}(u_k)\hat{U}_0(u_k,s) \\
&= \hat{U}_0(t,s) + \sum_{k=1}^{\infty}(-i)^k \int_s^t du_1 \cdots \int_s^{u_{k-1}} du_k \\
&\quad \times \hat{U}_0(t,u_1)\hat{V}(u_1)\hat{U}_0(u_1,u_2)\hat{V}(u_2)\cdots\hat{U}_0(u_{k-1},u_k)\hat{V}(u_k)\hat{U}_0(u_k,s),
\end{aligned}
\tag{A.2}
$$

where we have use the inverse relation $\hat{U}_0^{\dagger}(t,s) = \hat{U}_0(s,t)$ and the composition rule,

$$
\hat{U}_0(u_i,s)\hat{U}_0(s,u_{i+1}) = \hat{U}_0(u_i,u_{i+1}).
$$

Note that we can split the evolution operators in between $\hat{V}$'s at an arbitrary point; in particular, we can chose $\hat{U}_0(u_i,u_{i+1}) = \hat{U}_0(u_i,0)\hat{U}_0(0,u_{i+1})$. When we do that, we can rewrite Eq. (A.2) as

$$
\begin{aligned}
\hat{U}(t,s) &= \hat{U}_0(t,0)\hat{U}_0(0,s) + \sum_{k=1}^{\infty}(-i)^k \int_s^t du_1 \cdots \int_s^{u_{k-1}} du_k\, \hat{U}_0(t,0) \\
&\quad \times \hat{U}_0(0,u_1)\hat{V}(u_1)\hat{U}_0(u_1,0)\hat{U}_0(0,u_2)\hat{V}(u_2)\cdots\hat{U}_0(0,u_k)\hat{V}(u_k)\hat{U}_0(u_k,0)\hat{U}_0(0,s) \\
&= \hat{U}_0(t,0)\left(\hat{1} + \sum_{k=1}^{\infty}(-i)^k \int_s^t du_1 \cdots \int_s^{u_{k-1}} du_k\, \hat{V}_I(u_1)\cdots\hat{V}_I(u_k)\right)\hat{U}_0^{\dagger}(s,0) \\
&= \hat{U}_0(t,0)\left(T_{V_I} e^{-i\int_s^t \hat{V}_I(u)du}\right)\hat{U}_0^{\dagger}(s,0).
\end{aligned}
$$

**Assignment 11 (Sec. 2.4)** *Write a code that integrates numerically the dynamical equation*

$$
\frac{d}{dt}\hat{U}(t,0) = -i\frac{\mu}{2}\left[\cos(\omega t)\hat{\sigma}_x - \sin(\omega t)\hat{\sigma}_y\right]\hat{U}(t,0),
\tag{A.3}
$$

*with the initial condition $\hat{U}(0,0) = \hat{1}$ and constant $\mu$, $\omega$.*

*Compare the results obtained with three integration methods: (i) the Euler method, (ii) the Runge-Kutta method, and (iii) the Crank-Nicolson method.*

*Solution*: In short, the numerical integration is an iterative method where the value of the solution in the next time step, $\hat{U}(t+h,0)$ ($h$ is the chosen step size), is computed using the current, already computed, value $\hat{U}(t,0)$ and the RHS of the dynamical equation,

$$
\text{RHS}(t) = -i\hat{H}(t)\hat{U}(t,0).
$$

1. The Euler method is the simplest scheme of numerical integration where the next value $\hat{U}(jh+h,0) = \hat{U}_{j+1}$ is computed with the previous value according to formula

$$
\frac{\hat{U}_{j+1} - \hat{U}_j}{h} = -i\hat{H}(jh)\hat{U}_j \;\Rightarrow\; \hat{U}_{j+1} = \hat{U}_j - ih\hat{H}(jh)\hat{U}_j.
$$

The method originates form a simple approximation of the derivative with a finite difference,

$$\frac{d}{dt}\hat{U}(t,0) \approx \frac{\hat{U}(t+h,0)-\hat{U}(t,0)}{h},$$

with very small $h$.

2. The Runge-Kutta method calculates $\hat{U}_{j+1}$ using a significantly more complex formula,

$$\hat{U}_{j+1} = \hat{U}_j + \frac{h}{6}\left(\hat{R}_1 + \hat{R}_2 + \hat{R}_3 + \hat{R}_4\right),$$

where

$$\begin{aligned}
\hat{R}_1 &= -i\hat{H}(jh)\hat{U}_j\,; \\
\hat{R}_2 &= -i\hat{H}\left(jh+\tfrac{h}{2}\right)\left(\hat{U}_j + \frac{h}{2}\hat{R}_1\right); \\
\hat{R}_3 &= -i\hat{H}\left(jh+\tfrac{h}{2}\right)\left(\hat{U}_j + \frac{h}{2}\hat{R}_2\right); \\
\hat{R}_4 &= -i\hat{H}(jh+h)\left(\hat{U}_j + h\hat{R}_3\right).
\end{aligned}$$

3. Crank-Nicolson is an implicit method, which means that the next step is given in the form of linear equation that has to be solved for $\hat{U}_{j+1}$; in the simplest incarnation of the method, the equation is defined as

$$\frac{\hat{U}_{j+1}-\hat{U}_j}{h} + i\frac{\hat{H}(jh+h)\hat{U}_{j+1}+\hat{H}(jh)\hat{U}_j}{2} = 0.$$

In this case, its solution can be formally written as

$$\hat{U}_{j+1} = \left(\hat{1}+i\tfrac{h}{2}\hat{H}(jh+h)\right)^{-1}\left(\hat{1}-i\tfrac{h}{2}\hat{H}(jh)\right)\hat{U}_j.$$

As it turns out, Eq. (A.3) can be solved analytically. Note that the Hamiltonian can be written as

$$\hat{H}(t) = \frac{\mu}{2}\left[\cos(\omega t)\hat{\sigma}_x - \sin(\omega t)\hat{\sigma}_y\right] = \frac{\mu}{2}e^{i\frac{\omega t}{2}\hat{\sigma}_z}\hat{\sigma}_x\,e^{-i\frac{\omega t}{2}\hat{\sigma}_z},$$

so that, $\hat{H}(t)$ has the form of operator $\hat{\sigma}_x$ in the interaction picture with respect to operator $(\omega/2)\hat{\sigma}_z$, i.e.

$$e^{-i\frac{t}{2}(\omega\hat{\sigma}_z+\mu\hat{\sigma}_x)} = e^{-i\frac{\omega t}{2}\hat{\sigma}_z}T_H\exp\left[-i\int_0^t e^{i\frac{\omega s}{2}\hat{\sigma}_z}\hat{\sigma}_x\,e^{-i\frac{\omega s}{2}\hat{\sigma}_z}ds\right] = e^{-i\frac{\omega t}{2}\hat{\sigma}_z}\hat{U}(t,0).$$

Therefore, we can re-entangle the exps,

$$\hat{U}(t,0) = T_H e^{-i\int_0^t \hat{H}(s)ds} = e^{i\frac{\omega t}{2}\hat{\sigma}_z}e^{-\frac{1}{2}it(\omega\hat{\sigma}_z+\mu\hat{\sigma}_x)}.$$

Now that the generators of exps are time-independent, we can diagonalize them and compute the evolution operator,

$$\langle\uparrow|\hat{U}(t,0)|\uparrow\rangle = e^{+i\frac{\omega t}{2}}\left[\cos\left(\frac{t}{2}\sqrt{\omega^2+\mu^2}\right) - \frac{i\omega}{\sqrt{\omega^2+\mu^2}}\sin\left(\frac{t}{2}\sqrt{\omega^2+\mu^2}\right)\right];$$

$$\langle\downarrow|\hat{U}(t,0)|\downarrow\rangle = e^{-i\frac{\omega t}{2}}\left[\cos\left(\frac{t}{2}\sqrt{\omega^2+\mu^2}\right) + \frac{i\omega}{\sqrt{\omega^2+\mu^2}}\sin\left(\frac{t}{2}\sqrt{\omega^2+\mu^2}\right)\right];$$

$$\langle\uparrow|\hat{U}(t,0)|\downarrow\rangle = -\frac{i\mu e^{+i\frac{\omega t}{2}}}{\sqrt{\omega^2+\mu^2}}\sin\left(\frac{t}{2}\sqrt{\omega^2+\mu^2}\right);$$

$$\langle\downarrow|\hat{U}(t,0)|\uparrow\rangle = -\frac{i\mu e^{-i\frac{\omega t}{2}}}{\sqrt{\omega^2+\mu^2}}\sin\left(\frac{t}{2}\sqrt{\omega^2+\mu^2}\right).$$

You can use this exact solution to benchmark the results of the numerical integration.

**Assignment 12 (Sec. 2.6.1)** *Find the interaction picture with respect to $\hat{H}_0(t)$ of the unitary map for the time-dependent Hamiltonian $\hat{H}(t) = \hat{H}_0(t) + \hat{V}(t)$.*

*Solution*: Given the Hamiltonian $\hat{H}(t)$, the time-dependent generator of the unitary map is given by

$$\mathcal{H}(t) = [\hat{H}_0(t) + \hat{V}(t), \bullet] = [\hat{H}_0(t), \bullet] + [\hat{V}(t), \bullet] = \mathcal{H}_0(t) + \mathcal{V}(t).$$

The map in a form of time-ordered exponent can be factorized according to the rules of the disentanglement theorem introduced in Sec. 2.3,

$$\mathcal{U}(t,0) = T_{\mathcal{H}}e^{-i\int_0^t \mathcal{H}(s)ds} = \mathcal{U}_0(t,0)\mathcal{U}_I(t,0),$$

where the disentangled and the interaction picture maps are

$$\mathcal{U}_0(t,0) = T_{\mathcal{H}_0}e^{-i\int_0^t \mathcal{H}_0(s)ds},$$
$$\mathcal{U}_I(t,0) = T_{\mathcal{V}_I}e^{-i\int_0^t \mathcal{V}_I(s)ds},$$

with the interaction picture of $\mathcal{V}(t)$ that reads

$$\mathcal{V}_I(t) = \mathcal{U}_0^\dagger(t,0)\mathcal{V}(t)\mathcal{U}_0(t,0) = \mathcal{U}_0(0,t)\mathcal{V}(t)\mathcal{U}_0(t,0).$$

Using the relation (63) for $\mathcal{U}_0$, the super-operator $\mathcal{V}_I(t)$ can further simplified,

$$\begin{aligned}\mathcal{V}_I(t) &= \hat{U}_0(0,t)[\hat{V}(t),\hat{U}_0(t,0)\bullet\hat{U}_0^\dagger(t,0)]\hat{U}_0^\dagger(0,t)\\ &= \hat{U}_0(0,t)\hat{V}(t)\hat{U}_0(t,0)\bullet\hat{U}_0(0,t)\hat{U}_0(t,0) - \hat{U}_0(0,t)\hat{U}_0(t,0)\bullet\hat{U}_0(0,t)\hat{V}(t)\hat{U}_0(t,0)\\ &= \hat{U}_0(0,t)\hat{V}(t)\hat{U}_0(t,0)\bullet - \bullet\hat{U}_0(0,t)\hat{V}(t)\hat{U}_0(t,0)\\ &= [\hat{V}_I(t),\bullet].\end{aligned}$$

Here, we have made use of the identity $\hat{U}_0^\dagger(t,0) = \hat{U}_0(0,t)$ and the definition of the operator version of interaction picture (38).

**Assignment 13 (Sec. 2.6.2)** *Show that $(\mathcal{C}^\dagger)_{nm} = \mathcal{C}^*_{mn}$.*

*Solution*: The dagger (Hermitian conjugate) is defined with respect to the inner product,

$$(\hat{A}|\mathcal{C}\hat{B}) = (\mathcal{C}^\dagger\hat{A}|\hat{B}).$$

Now, knowing that $(\mathcal{C}^\dagger)^\dagger = \mathcal{C}$ we calculate the matrix element,

$$\begin{aligned}
(\mathcal{C}^\dagger)_{nm} = (\hat{E}_n|\mathcal{C}^\dagger\hat{E}_m) = (\mathcal{C}\hat{E}_n|\hat{E}_m) &= \mathrm{tr}\left(\left(\mathcal{C}\hat{E}_n\right)^\dagger\hat{E}_m\right) \\
&= \mathrm{tr}\left[\left(\left(\mathcal{C}\hat{E}_n\right)^\dagger\hat{E}_m\right)^\dagger\right]^* = \mathrm{tr}\left[\hat{E}_m^\dagger(\mathcal{C}\hat{E}_n)\right]^* = (\hat{E}_m|\mathcal{C}\hat{E}_n)^* \\
&= \mathcal{C}_{mn}^*.
\end{aligned}$$

**Assignment 14 (Sec. 2.6.2)** *Find the super-operator matrix representation of quantum expectation value,*

$$\langle A(t)\rangle = \mathrm{tr}\left[\hat{A}\hat{\rho}(t)\right] = \mathrm{tr}\left[\hat{A}\mathcal{U}(t,0)\hat{\rho}\right],$$

*where $\hat{A}$ is a hermitian operator ($\hat{A}^\dagger = \hat{A}$).*

*Solution*: We chose an orthonormal basis in the space of operators, $\mathcal{B} = \{\hat{E}_i\}_{i=1}^{d^2}$, such that

$$(\hat{E}_i|\hat{E}_j) = \mathrm{tr}(\hat{E}_i^\dagger\hat{E}_j) = \delta_{i,j}.$$

We assumed that the ket-space is $d$-dimensional, so that the vector space of operators is $d \times d = d^2$ dimensional—hence, there are $d^2$ basis elements.

Using the decomposition of identity (71),

$$\bullet = \sum_n \hat{E}_n(\hat{E}_n|\bullet) = \sum_n \hat{E}_n\,\mathrm{tr}\left(\hat{E}_n^\dagger\bullet\right),$$

we decompose the observable $\hat{A}$ and the initial state $\hat{\rho}$ in the basis $\mathcal{B}$,

$$\hat{A} = \sum_i \hat{E}_i\,\mathrm{tr}(\hat{E}_i^\dagger\bullet)\hat{A} = \sum_i \mathrm{tr}(\hat{E}_i^\dagger\hat{A})\hat{E}_i = \sum_i A_i\hat{E}_i = \begin{bmatrix} A_1 \\ A_2 \\ \vdots \\ A_{d^2} \end{bmatrix}_{\mathcal{B}};$$

$$\hat{\rho} = \sum_i \hat{E}_i\,\mathrm{tr}(\hat{E}_i^\dagger\bullet)\hat{\rho} = \sum_i \mathrm{tr}(\hat{E}_i^\dagger\hat{\rho})\hat{E}_i = \sum_i r_i\hat{E}_i = \begin{bmatrix} r_1 \\ r_2 \\ \vdots \\ r_{d^2} \end{bmatrix}_{\mathcal{B}}.$$

The column notation reminds us that operators count as vectors in the space of operators. The unitary map $\mathcal{U}(t,0)$ is a super-operator, hence, it is a square matrix acting on column vectors in the space of operators,

$$\begin{aligned}
\mathcal{U}(t,0) &= \sum_i \hat{E}_i\,\mathrm{tr}(\hat{E}_i^\dagger\bullet)\mathcal{U}(t,0)\sum_j \hat{E}_j\,\mathrm{tr}(\hat{E}_j^\dagger\bullet) \\
&= \sum_{i,j} \mathrm{tr}[\hat{E}_i^\dagger\mathcal{U}(t,0)\hat{E}_j]\hat{E}_i\,\mathrm{tr}(\hat{E}_j^\dagger\bullet) = \sum_{i,j} U_{i,j}(t,0)\hat{E}_i\,\mathrm{tr}(\hat{E}_j^\dagger\bullet) \\
&= \begin{bmatrix} U_{1,1}(t,0) & U_{1,2}(t,0) & \cdots & U_{1,d^2}(t,0) \\ U_{2,1}(t,0) & U_{2,2}(t,0) & \cdots & U_{2,d^2}(t,0) \\ \vdots & & \ddots & \\ U_{d^2,1}(t,0) & U_{d^2,2}(t,0) & \cdots & U_{d^2,d^2}(t,0) \end{bmatrix}_{\mathcal{B}}.
\end{aligned}$$

Finally, we rewrite the expectation value using the decomposition of identity and the above matrix representations,

$$\langle A(t)\rangle = \mathrm{tr}\big[\hat{A}\,\mathcal{U}(t,0)\hat{\rho}\big]$$

$$= \mathrm{tr}\left[\left(\sum_i \hat{E}_i\,\mathrm{tr}(\hat{E}_i^\dagger\bullet)\hat{A}^\dagger\right)^\dagger \sum_j \hat{E}_j\,\mathrm{tr}(\hat{E}_j^\dagger\bullet)\mathcal{U}(t,0)\sum_k \hat{E}_k\,\mathrm{tr}(\hat{E}_k^\dagger\bullet)\hat{\rho}\right]$$

$$= \sum_{i,j,k} \mathrm{tr}(\hat{E}_i^\dagger\hat{E}_j)\,\mathrm{tr}(\hat{E}_i^\dagger\hat{A})^*\,\mathrm{tr}[\hat{E}_j^\dagger\mathcal{U}(t,0)\hat{E}_k]\,\mathrm{tr}[\hat{E}_k^\dagger\hat{\rho}]$$

$$= \sum_{i,j,k} \delta_{i,j}A_i^* U_{j,k}(t,0)r_k = \sum_{i,j} A_i^* U_{i,j}(t,0)r_j$$

$$= \begin{bmatrix} A_1^* & A_2^* & \cdots & A_{d^2}^* \end{bmatrix}_{\mathcal{B}} \begin{bmatrix} U_{1,1}(t,0) & U_{1,2}(t,0) & \cdots & U_{1,d^2}(t,0) \\ U_{2,1}(t,0) & U_{2,2}(t,0) & \cdots & U_{2,d^2}(t,0) \\ \vdots & & \ddots & \\ U_{d^2,1}(t,0) & U_{d^2,2}(t,0) & \cdots & U_{d^2,d^2}(t,0) \end{bmatrix}_{\mathcal{B}} \begin{bmatrix} r_1 \\ r_2 \\ \vdots \\ r_{d^2} \end{bmatrix}_{\mathcal{B}}.$$

The matrix of dynamical map is multiplied from left by the row vector obtained from the column vector by transposing it and taking the complex conjugate of its elements,

$$\begin{bmatrix} A_1^* & A_2^* & \cdots & A_{d^2}^* \end{bmatrix}_{\mathcal{B}} = \left(\begin{bmatrix} A_1 \\ A_2 \\ \vdots \\ A_{d^2} \end{bmatrix}_{\mathcal{B}}^T\right)^* = \begin{bmatrix} A_1 \\ A_2 \\ \vdots \\ A_{d^2} \end{bmatrix}_{\mathcal{B}}^\dagger.$$

Note that that the expectation value can be written in terms of operator space inner product,

$$\langle A(t)\rangle = (\hat{A}|\mathcal{U}(t,0)\hat{\rho}).$$

**Assignment 15 (Sec. 3.1.3)** *Calculate the average value and the auto-correlation function for random telegraph noise.*

*Solution*: The average value,

$$\overline{R(t)} = \sum_{r=\pm 1} r P_R^{(1)}(r,t) = \sum_{r=\pm 1} r\left(\frac{1+rpe^{-2wt}}{2}\right) = pe^{-2wt}.$$

Then we calculate the second moment (we assume here $t_1 > t_2 > 0$ so that we can use joint distribution),

$$\overline{R(t_1)R(t_2)} = \sum_{r_1,r_2=\pm 1} r_1 r_2 P_R^{(2)}(r_1,t_1;r_2,t_2)$$

$$= \sum_{r_1,r_2} r_1\left(\frac{1+r_1 r_2 e^{-2w(t_1-t_2)}}{2}\right) r_2 P_R^{(1)}(r_2,t_2)$$

$$= e^{-2w(t_1-t_2)} \sum_{r_2=\pm 1} r_2^2 P_R^{(1)}(r_2,t_2) = e^{-2w(t_1-t_2)} \sum_{r_2} P_R^{(1)}(r_2,t_2)$$

$$= e^{-2w(t_1-t_2)}.$$

The argument of the auto-correlation function does not have to be ordered,

$$C_R(t_1,t_2) = \theta(t_1-t_2)\overline{R(t_1)R(t_2)} + \theta(t_2-t_1)\overline{R(t_2)R(t_1)}$$

$$= \theta(t_1-t_2)e^{-2w(t_1-t_2)} + \theta(t_2-t_1)e^{-2w(t_2-t_1)} - p^2 e^{-2w(t_1+t_2)}$$

$$= e^{-2w|t_1-t_2|} - p^2 e^{-2w(t_1+t_2)},$$

where $\theta(x) = 1$ when $x > 0$ and 0 otherwise.

**Assignment 16 (Sec. 3.1.3)** *Note the following property of joint distributions of RTN,*

$$\sum_{r_1,r_2=\pm 1} r_1 r_2 P_R^{(k+2)}(r_1,t_1;\ldots;r_k,t_k) = e^{-2w(t_1-t_2)} P_R^{(k)}(r_3,t_3;\ldots;r_k,t_k).$$

*Prove it and use it to calculate an arbitrary moment* $\overline{R(t_1)\cdots R(t_k)}$.

*Solution*: First, the proof:

$$\sum_{r_1,r_2=\pm 1} r_1 r_2 P_R^{(k+2)}(r_1,t_1;\ldots;r_k,t_k)$$

$$= \sum_{r_1,r_2=\pm 1} r_1 \left( \frac{1 + r_1 r_2 e^{-2w(t_1-t_2)}}{2} \right) r_2 P_R^{(k+1)}(r_2,t_2;\ldots;r_k,t_k)$$

$$= e^{-2w(t_1-t_2)} \sum_{r_2=\pm 1} r_2^2 P_R^{(k+1)}(r_2,t_2;\ldots;r_k,t_k) = e^{-2w(t_1-t_2)} \sum_{r_2} P_R^{(k+1)}(r_2,t_2;r_3,t_3;\ldots)$$

$$= e^{-2w(t_1-t_2)} P_R^{(k)}(r_3,t_3;\ldots;r_k,t_k),$$

where, in the last line, we have used the consistency relation (84).

Now, we can iterate the just proven relation to calculate the moments,

$$\sum_{r_i} \left( \prod_{i=1}^{2k} r_i \right) P_R^{(2k)}(r_1,t_1;\ldots;r_{2k},t_{2k}) = \prod_{j=1}^{k} e^{-2w(t_{2j-1}-t_{2j})},$$

$$\sum_{r_i} \left( \prod_{i=1}^{2k+1} r_i \right) P_R^{(2k+1)}(r_1,t_1;\ldots;r_{2k+1},t_{2k+1}) = p e^{-2wt_{2k+1}} \prod_{j=1}^{k} e^{-2w(t_{2j-1}-t_{2j})}.$$

**Assignment 17 (Sec. 3.1.3)** *Check that RTN is a Markovian process, i.e. using the Bayes law, show that the probability for the trajectory to reach value $r_1$ at time $t_1$ depends only on the value $r_2$ attained in the latest moment in past $t_2 < t_1$, but not on all the earlier values $r_3$ at $t_3$, ..., $r_k$ at $t_k$ ($t_1 > t_2 > t_3 > \cdots > t_k$).*

*Solution*: According to the Bayes law, the probability that the trajectory of a stochastic process $F(t)$ will reach value $f_1$ at time $t_1$, under the condition that it has passed through values $f_2$ at $t_2$, ..., and $f_k$ at $t_k$ (assuming $t_1 > t_2 > \cdots t_k$), is given by

$$P_F^{(1|k-1)}(f_1,t_1|f_2,t_2;\ldots;f_k,t_k) = \frac{P_F^{(k)}(f_1,t_1;f_2,t_2;\ldots;f_k,t_k)}{P_F^{(k-1)}(f_2,t_2;\ldots;f_k,t_k)}.$$

In the case of RTN $R(t)$, the conditional probability reads,

$$P_R^{(1|k-1)}(r_1,t_1|r_2,t_2;\ldots;r_k,t_k) = \frac{P_R^{(1)}(r_k,t_k) \prod_{j=1}^{k-1} \left( \frac{1 + r_j r_{j+1} e^{-2w(t_j-t_{j+1})}}{2} \right)}{P_R^{(1)}(r_k,t_k) \prod_{j=2}^{k-1} \left( \frac{1 + r_j r_{j+1} e^{-2w(t_j-t_{j+1})}}{2} \right)}$$

$$= \frac{1 + r_1 r_2 e^{-2w(t_1-t_2)}}{2} = P_R^{(1|1)}(r_1,t_1|r_2,t_2),$$

i.e. only the value $r_2$ in the immediate past $t_2 < t_1$ affects the probability. In general, the processes for which

$$P_F^{(1|k-1)}(f_1,t_1|f_2,t_2;\ldots;f_k,t_k) = P_F^{(1|1)}(f_1,t_1|f_2,t_2),$$

is said to be *Markovian*; of course, as we have just demonstrated, RTN is Markovian.

A Markovian process is fully characterized by the first conditional probability $P_F^{(1|1)}$ and the first joint distribution $P_F^{(1)}$. Indeed, inverting the Bayes rule we can write

$$P_F^{(k)}(f_1, t_1; \ldots; f_k, t_k) = P_F^{(1|k-1)}(f_1, t_1|f_2, t_2; \ldots; f_k, t_k)P_F^{(k-1)}(f_2, t_2; \ldots; f_k, t_k)$$
$$= P_F^{(1|1)}(f_1, t_1|f_2, t_2)P_F^{(k-1)}(f_2, t_2; \ldots; f_k, t_k).$$

Now, we can iterate this relation until we reduce the joint distribution on RHS to $P_F^{(1)}$,

$$P_F^{(k)}(f_1, t_1; \ldots; f_k, t_k) = P_F^{(1|1)}(f_1, t_1|f_2, t_2)P_F^{(1|1)}(f_2, t_2|f_3, t_3) \cdots$$
$$\cdots P_F^{(1|1)}(f_{k-1}, t_{k-1}|f_k, t_k)P_F^{(1)}(f_k, t_k)$$
$$= P_F^{(1)}(f_k, t_k)\prod_{j=1}^{k-1} P_F^{(1|1)}(f_j, t_j|f_{j+1}, t_{j+1}). \tag{A.4}$$

In case of RTN, when the conditional probability is given by

$$P_R^{(1|1)}(r, t; r', t') = \frac{1 + rr'e^{-2w(t-t')}}{2} \equiv u_{r,r'}(t-t'),$$

we recognize in Eq. (A.4) the formula for joint distribution we have provided in (90).

**Assignment 18 (Sec. 3.1.3)** *Calculate the joint probability distributions for the dichotomic process $T(t)$ that switches between 1 and $-1$ but the rates of switching are unequal.*

*Process $T(t)$ is Markovian and its first conditional probability, $P_T^{(1|1)}(r, t|r', t') = u_{r,r'}(t-t')$, is defined by the rate equation,*

$$\dot{u}_{r_3,r_1}(t) = \sum_{r_2=\pm 1} \left[ w_{r_3,r_2}u_{r_2,r_1}(t) - w_{r_2,r_3}u_{r_3,r_1}(t) \right],$$

*where $w_{r,r'}$ is the rate of switching from $r'$ to $r$.*

*Solution*: Let us write the rate equations in an explicit form,

$$\dot{u}_{1,1}(t) = w_+ u_{-1,1}(t) - w_- u_{1,1}(t);$$
$$\dot{u}_{-1,-1}(t) = w_- u_{1,-1}(t) - w_+ u_{-1,-1}(t);$$
$$\dot{u}_{1,-1}(t) = w_+ u_{-1,-1}(t) - w_- u_{1,-1}(t);$$
$$\dot{u}_{-1,1}(t) = w_- u_{1,1}(t) - w_+ u_{-1,1}(t);$$

$$u_{r,r'}(0) = \delta_{r,r'} \quad \text{(the initial condition)};$$

where $w_+ = w_{1,-1}, w_- = w_{-1,1}, w_{\pm 1, \pm 1}$ have canceled out. This system of differential equations can be solved exactly,

$$U(t) = \begin{bmatrix} u_{1,1}(t) & u_{1,-1}(t) \\ u_{-1,1}(t) & u_{-1,-1}(t) \end{bmatrix} = \frac{1}{2\overline{w}} \begin{bmatrix} w_+ + w_- e^{-2\overline{w}t} & w_+ \left(1 - e^{-2\overline{w}t}\right) \\ w_- \left(1 - e^{-2\overline{w}t}\right) & w_- + w_+ e^{-2\overline{w}t} \end{bmatrix},$$

where $\overline{w} = (w_+ + w_-)/2$. The first joint distribution $p(t) = [P_T^{(1)}(1, t), P_T^{(1)}(-1, t)]^T$ can be calculated using Markovianity and the consistency relation (84),

$$p(t) = \begin{bmatrix} \sum_r P_T^{(2)}(1, t; r, 0) \\ \sum_r P_T^{(2)}(-1, t; r, 0) \end{bmatrix} = \begin{bmatrix} \sum_r u_{1,r}(t)p_r(0) \\ \sum_r u_{-1,r}(t)p_r(0) \end{bmatrix} = U(t) \cdot p(0),$$

and we can set an arbitrary initial distribution $p_{\pm 1}(0) = (1 \pm p)/2$ with $p \in [0,1]$.

The joint distribution can now be written using the matrix elements of $U$ and $p$,

$$P_T^{(k)}(r_1, t_1; \ldots; r_k, t_k) = p_{r_k}(t_k) \prod_{j=1}^{k-1} u_{r_j, r_{j+1}}(t_j - t_{j+1}).$$

**Assignment 19 (Sec. 3.1.3)** *Write a code to generate trajectories of RTN.*

*Solution*: Naturally, the trajectories generated by the program will be spanned on a discrete grid, $\{r(0), r(\Delta t), r(2\Delta t), \ldots, r(n\Delta t)\}$. The program should make an extensive use of the Markov property of RTN to generate the values of trajectories iteratively. The next value $r_{i+1}$ can be represented by the stochastic variable $R_{i+1}$ with the probability distribution derived from the conditional probability $P_R^{(1|1)}$ when we substitute for the previous value the already generated $r_i$,

$$P_{R_{i+1}}(r) = P_R^{(1|1)}(r, (i+1)\Delta t | r_i, i\Delta t).$$

To obtain an actual number to be assigned for $r_{i+1}$, one draws it at random form the distribution $P_{R_{i+1}}$.

In this way, the Markovianity of RTN is a real boon; when a processes does not have Markov property, then the only way to sample its trajectories is to draw them at random from full joint distributions, which is technically challenging.

Below, we give an example of a pseudocode implementing the algorithm described above,

```
inputs:
   time step dt,
   number of steps n,
   probability the initial value is one p,
   switching rate w
outputs:
   the array of values r[i] = r(iΔt)

def u( r1 , r2 , dt ):  {
   return 0.5 ( 1 + r1 r2 exp( -2 w dt ) ) }
def rand( p ):  {
   val := (chose at random:
   1 with probability p, or -1 with probability 1-p);
   return val;}
r[0] := rand( p );
for( i = 1; i <= n ; i++ ){
   r[i] := rand( u( 1, r[i-1], dt ) );}
return r;
```

**Assignment 20 (Sec. 3.2)** *Verify that $\overline{\mathcal{U}}$ is a proper map from density matrices to density matrices, i.e. check if the operator $\hat{\rho}(t) = \overline{\mathcal{U}}(t,0)\hat{\rho}$ is positive and have unit trace.*

*Solution*: First, let us show that $\overline{\mathcal{U}}$ preserves the trace of the initial density matrix,

$$\mathrm{tr}[\hat{\rho}(t)] = \mathrm{tr}[\overline{\mathcal{U}}(t,0)\hat{\rho}] = \int P_F[f] \, \mathrm{tr}[\mathcal{U}[f](t,0)\hat{\rho}] \, [Df]$$

$$= \left( \int P_F[f][Df] \right) \mathrm{tr}(\hat{\rho}(0)) = 1.$$

To obtain the above result we have used two facts: (i) the unitary maps—such as the trajectory-wise $\mathcal{U}[f](t,0)$—preserve the trace, see Eq. (66); (ii) $P_F[f]$ is normalized to unity, see Eq. (82).

Second, we check if $\hat{\rho}(t)$ is a positive operator,

$$
\overline{\mathcal{U}}(t,0)\hat{\rho} = \int P_F[f]\,\mathcal{U}[f](t,0)\hat{O}_\rho^\dagger \hat{O}_\rho[Df] = \int \left(\sqrt{P_F[f]}\hat{O}[f]\right)^\dagger \left(\sqrt{P_F[f]}\hat{O}[f]\right)[Df] \geqslant 0,
$$

where we have used the fact that unitary maps preserve positivity, see Eq. (67) and $P_F \geqslant 0$.

**Assignment 21 (Sec. 3.4.3)** *Calculate the coherence function $W(t)$ for random telegraph noise.*

*Solution*: Recall that we have already calculated the explicit form of RTN moments, see Eqs. (93) and (94). A one way to proceed from here would be to compute explicitly the time-ordered integrals in Eq. (115), get an analytical formula for each term, and then try to identify what function has the same series representation as the one we just calculated. There is another, more clever way but it only works for RTN. If you inspect closely Eq. (115) after we substitute the explicit form of moments, but before we perform the integrations, you might notice that each term has a fractal-like structure. When you see something like this, you should try a derivative over $t$ and check if the expression reconstitutes itself on the RHS; if it does, then you will obtain a differential equation that might happen to be solvable! Let us check then,

$$
\dot{W}(t) = i\lambda\overline{R(t)} - \lambda^2 \int_0^t e^{-2w(t-s)}W(s)ds = ip\lambda e^{-2wt} - \lambda^2 \int_0^t e^{-2w(t-s)}W(s). \tag{A.5}
$$

The intuition was correct, $W$ has reappeared on the RHS. A good news is that this is a linear equation (with inhomogeneity); a bad news is that it is not time-local. However, since the memory kernel (the function that multiplies $W(s)$ under the integral) is an exp of $t$, there is a good chance we can recover time-locality when we do the second derivative:

$$
\ddot{W}(t) = -i2wp\lambda e^{-2wt} - \lambda^2 W(t) + 2w\lambda^2 \int_0^t e^{-2w(t-s)}W(s)ds = -\lambda^2 W(t) - 2w\dot{W}(t),
$$

As anticipated, we were able to use Eq. (A.5) to eliminate the time-non-local term. Thus, we now have the second-order linear inhomogeneous local differential equation to solve with the initial conditions $W(0) = 1$ and $\dot{W}(0) = ip\lambda$. The solution is found with elementary methods and it is given by Eq. (116).

**Assignment 22 (Sec. 3.6.2)** *Verify equation* (150),

$$
\overline{\mathcal{U}}(t,0) = T_{\mathcal{V}_I} e^{\sum_{k=1}^\infty (-i\lambda)^k \int_0^t ds_1 \cdots \int_0^{s_{k-1}} ds_k\, C_F^{(k)}(s_1,\ldots,s_k)\mathcal{V}_I(s_1)\cdots\mathcal{V}_I(s_k)}.
$$

*Solution*: First, let us establish how the $k$th moment $\overline{F(s_1)\cdots F(s_k)}$ expresses through cumulants. To do this, we compare the power series expansion of the coherence function,

$$
\overline{e^{i\lambda \int_0^t F(s)ds}} = 1 + \sum_{k=1}^\infty (i\lambda)^k \int_0^t ds_1 \cdots \int_0^{s_{k-1}} ds_k \overline{F(s_1)\cdots F(s_k)},
$$

and the cumulant series,

$$
e^{\chi_F(t)} = e^{\sum_{n=1}^{\infty} \frac{(i\lambda)^n}{n!} \int_0^t C_F^{(n)}(s_1,\ldots,s_n) ds_1 \cdots ds_n}
$$

$$
= 1 + \sum_{m=1}^{\infty} \frac{1}{m!} \left( \sum_{n=1}^{\infty} \frac{(i\lambda)^n}{n!} \int_0^t ds_1 \cdots ds_n \, C_F^{(n)}(s_1,\ldots,s_n) \right)^m
$$

$$
= 1 + \sum_{k=1}^{\infty} (i\lambda)^k \int_0^t ds_1 \cdots ds_k
$$

$$
\times \Bigg[ \frac{1}{1!} \frac{C_F^{(k)}(s_1,\ldots,s_k)}{k!}
$$

$$
+ \frac{1}{2!} \left( \frac{C_F^{(k-1)}(s_1,\ldots,s_{k-1})}{(k-1)!} \frac{C_F^{(1)}(s_k)}{1!} + \frac{C_F^{(1)}(s_1)}{1!} \frac{C_F^{(k-1)}(s_2,\ldots,s_k)}{(k-1)!} + \cdots \right)
$$

$$
+ \frac{1}{3!} \left( \frac{C^{(k-2)}(s_1,\ldots,s_{k-2})}{(k-2)!} \frac{C^{(1)}(s_{k-1})}{1!} \frac{C^{(1)}(s_k)}{1!} \right.
$$

$$
\left. + \frac{1}{3!} \frac{C^{(1)}(s_1)}{1!} \frac{C^{(k-2)}(s_2,\ldots,s_{k-1})}{(k-2)!} \frac{C^{(1)}(s_k)}{1!} + \cdots \right) + \cdots \Bigg]
$$

$$
= 1 + \sum_{k=1}^{\infty} (i\lambda)^k \int_0^t ds_1 \cdots ds_k
$$

$$
\times \sum_{r=1}^{k} \frac{1}{r!} \sum_{k_1=1}^{k} \cdots \sum_{k_r=1}^{k} \delta\left( k - \sum_{i=1}^{r} k_i \right) \frac{C_F^{(k_1)}(s_1,\ldots,s_{k_1})}{k_1!} \cdots \frac{C_F^{(k_r)}(s_{k_{r-1}+1},\ldots,s_k)}{k_r!}
$$

$$
= 1 + \sum_{k=1}^{\infty} (i\lambda)^k \int_0^t ds_1 \cdots \int_0^{s_{k-1}} ds_k
$$

$$
\times \sum_{r=1}^{k} \frac{1}{r!} \sum_{k_1=1}^{k} \cdots \sum_{k_r=1}^{k} \delta\left( k - \sum_{i=1}^{r} k_i \right) \sum_{\sigma \in \Sigma(1,\ldots,k)} \frac{C_F^{(k_1)}(s_{\sigma(1)},\ldots)}{k_1!} \cdots \frac{C_F^{(k_r)}(\ldots,s_{\sigma(k)})}{k_r!}.
$$

Comparing the terms of the same order in $\lambda$ we get

$$
\overline{F(s_1)\cdots F(s_k)} = \sum_{r=1}^{k} \frac{1}{r!} \sum_{k_1=1}^{k} \cdots \sum_{k_r=1}^{k} \delta\left( k - \sum_{i=1}^{r} k_i \right) \sum_{\sigma \in \Sigma(1,\ldots,k)} \frac{C_F^{(k_1)}(s_{\sigma(1)},\ldots)}{k_1!} \cdots \frac{C_F^{(k_r)}(\ldots,s_{\sigma(k)})}{k_r!}.
$$

The sum over $k_i$'s and the delta assures that the products of cumulants have a total order $k$; the sum over permutations comes from switching from unordered integration to ordered integration.

Now, we expand into power series the LHS of Eq. (150),

$$
\text{LHS} = \bullet + \sum_{k=1}^{\infty} (-i\lambda)^k \int_0^t ds_1 \cdots \int_0^{s_{k-1}} ds_k \, \overline{F(s_1)\cdots F(s_k)} \, \mathcal{V}_I(s_1) \cdots \mathcal{V}_I(s_k),
$$

and we want to compare it with the RHS,

$$\text{RHS} = T_{\mathcal{V}_I} e^{\sum_{n=1}^{\infty}(-i\lambda)^n \int_0^t ds_1 \cdots \int_0^{s_{n-1}} ds_n C_F^{(n)}(s_1,\ldots,s_n)\mathcal{V}_I(s_1)\cdots\mathcal{V}_I(s_n)}$$

$$= \sum_{m=0}^{\infty} \frac{1}{m!} T_{\mathcal{V}_I} \left\{ \left( \sum_{n=1}^{\infty}(-i\lambda)^n \int_0^t ds_1 \cdots \int_0^{s_{n-1}} ds_n \, C_F^{(n)}(s_1,\ldots,s_n)\mathcal{V}_I(s_1)\cdots\mathcal{V}_I(s_n) \right)^m \right\}$$

$$= \bullet + \sum_{k=1}^{\infty}(-i\lambda)^k \sum_{r=1}^{k} \frac{1}{r!} \sum_{k_1=1}^{k} \cdots \sum_{k_r=1}^{k} \delta\left(k - \sum_{i=1}^{r} k_i\right)$$

$$\times \left( \prod_{j=1}^{r} \int_0^t ds_1^{(j)} \cdots \int_0^{s_{k_j-1}^{(j)}} ds_{k_j}^{(j)} C_F^{(k_j)}(s_1^{(j)},\ldots,s_{k_j}^{(j)}) \right) T_{\mathcal{V}_I} \left\{ \prod_{i=1}^{r} \prod_{j=1}^{k_i} \mathcal{V}_I(s_j^{(i)}) \right\}$$

$$= \bullet + \sum_{k=1}^{\infty}(-i\lambda)^k \sum_{r=1}^{k} \frac{1}{r!} \sum_{k_1=1}^{k} \cdots \sum_{k_r=1}^{k} \delta\left(k - \sum_{i=1}^{r} k_i\right)$$

$$\times \left( \prod_{j=1}^{r} \int_0^t ds_1^{(j)} \cdots ds_{k_j}^{(j)} \frac{C_F^{(k_j)}(s_1^{(j)},\ldots,s_{k_j}^{(j)})}{k_j!} \right) T_{\mathcal{V}_I} \left\{ \prod_{i=1}^{r} \prod_{j=1}^{k_i} \mathcal{V}_I(s_j^{(i)}) \right\}$$

$$= \bullet + \sum_{k=1}^{\infty}(-i\lambda)^k \int_0^t ds_1 \cdots \int_0^{s_{k-1}} ds_k \, T_{\mathcal{V}_I}\{\mathcal{V}_I(s_1)\cdots\mathcal{V}_I(s_k)\}$$

$$\times \sum_{r=1}^{k} \frac{1}{r!} \sum_{k_1=1}^{k} \cdots \sum_{k_r=1}^{k} \delta\left(k - \sum_{i=1}^{r} k_i\right) \sum_{\sigma \in \Sigma(1,\ldots,k)} \frac{C_F^{(k_1)}(s_{\sigma(1)},\ldots)}{k_1!} \cdots \frac{C_F^{(k_r)}(\ldots,s_{\sigma(k)})}{k_r!}$$

$$= \bullet + \sum_{k=1}^{\infty}(-i\lambda)^k \int_0^t ds_1 \cdots \int_0^{s_{k-1}} ds_k \, \mathcal{V}_I(s_1)\cdots\mathcal{V}_I(s_k)\overline{F(s_1)\cdots F(s_k)} = \text{LHS}.$$

We have used here the fact that time-ordered product of super-operators and cumulants are symmetric under reordering of their time arguments so that we could switch to ordered integration.

**Assignment 23 (Sec. 3.6.3)** *Find the 4th super-cumulant $\mathcal{C}^{(4)}$ for non-Gaussian $F(t)$; assume that $\overline{F(t)} = 0$.*

*Solution*: Let us start by expanding the LHS and RHS of Eq. (153) into power series in $\omega$,

$$\text{LHS} = T_{\mathcal{V}_I} e^{\sum_{k=1}^{\infty}(-i\lambda)^k \int_0^t ds_1 \cdots \int_0^{s_{k-1}} ds_k C_F^{(k)}(s_1,\ldots,s_k)\mathcal{V}_I(s_1)\cdots\mathcal{V}_I(s_k)}$$

$$= \bullet + \sum_{k=1}^{\infty}(-i\lambda)^k \sum_{r=1}^{k} \frac{1}{r!} \sum_{k_1=1}^{k} \cdots \sum_{k_r=1}^{k} \delta\left(k - \sum_{i=1}^{r} k_i\right)$$

$$\times \left( \prod_{j=1}^{r} \int_0^t ds_1^{(j)} \cdots \int_0^{s_{k_j-1}^{(j)}} ds_{k_j}^{(j)} C_F^{(k_j)}(s_1^{(j)},\ldots,s_{k_j}^{(j)}) \right) T_{\mathcal{V}_I} \left\{ \prod_{i=1}^{r} \prod_{j=1}^{k_i} \mathcal{V}_I(s_j^{(i)}) \right\} ;$$

$$\text{RHS} = T_{\mathcal{C}^{(k)}} e^{\sum_{n=1}^{\infty}(-i\lambda)^n \int_0^t \mathcal{C}^{(n)}(s)ds}$$

$$= \bullet + \sum_{k=1}^{\infty}(-i\lambda)^k \sum_{r=1}^{k} \frac{1}{r!} \sum_{k_1=1}^{k} \cdots \sum_{k_r=1}^{k} \delta\left(k - \sum_{i=1}^{r} k_i\right) \int_0^t ds_1 \cdots ds_r \, T_{\mathcal{C}^{(k)}} \left\{ \prod_{i=1}^{r} \mathcal{C}^{(k_i)}(s_i) \right\} .$$

The $\lambda^1$-order terms are very simple and we get the 1st order super-cumulant without any difficulty,

$$\mathcal{C}^{(1)}(s) = C_F^{(1)}(s)\mathcal{V}_I(s). \tag{A.6}$$

The $\lambda^2$-order term on LHS reads,

$$\int_0^t ds_1 \int_0^{s_1} ds_2 \, C_{12} \mathcal{V}_1 \mathcal{V}_2 + \frac{1}{2!} \int_0^t ds_1 ds_2 \, C_1 C_2 \, T_{\mathcal{V}_I} \{\mathcal{V}_1 \mathcal{V}_2\} = \int_0^t ds_1 \int_0^{s_1} ds_2 \, [C_{12} + C_1 C_2] \mathcal{V}_1 \mathcal{V}_2 \,.$$

and the corresponding order on RHS is

$$\int_0^t ds_1 \mathcal{C}_1^{(2)} + \frac{1}{2!} \int_0^t ds_1 ds_2 \, T_{\mathcal{C}^{(k)}} \left\{ \mathcal{C}_1^{(1)} \mathcal{C}_2^{(1)} \right\} = \int_0^t ds_1 \mathcal{C}_1^{(2)} + \int_0^t ds_1 \int_0^{s_1} ds_2 \, C_1 C_2 \mathcal{V}_1 \mathcal{V}_2 \,,$$

where we used a shorthand notation: $C_{i_1 \cdots i_k} = C_F^{(k)}(s_{i_1}, \ldots, s_{i_k})$, $\mathcal{C}_i^{(k)} = \mathcal{C}^{(k)}(s_i)$, and $\mathcal{V}_i = \mathcal{V}_I(s_i)$. Comparing the two terms we obtain the 2nd order super-cumulant,

$$\mathcal{C}^{(2)}(s) = \mathcal{V}_I(s) \int_0^s C_F^{(2)}(s, s_1) \mathcal{V}_I(s_1) ds_1 \,. \tag{A.7}$$

The $\lambda^3$-order on LHS,

$$\int_0^t ds_1 \int_0^{s_1} ds_2 \int_0^{s_2} ds_3 \, C_{123} \mathcal{V}_1 \mathcal{V}_2 \mathcal{V}_3 + \frac{1}{3!} \int_0^t ds_1 ds_2 ds_3 \left( \prod_{i=1}^3 C_i \right) T_{\mathcal{V}_I} \left\{ \prod_{i=1}^3 \mathcal{V}_i \right\}$$

$$+ \int_0^t ds_1 \int_0^{s_1} ds_2 \, C_{12} \int_0^t ds_3 \, C_3 \, T_{\mathcal{V}_I} \left\{ \prod_{i=1}^3 \mathcal{V}_i \right\}$$

$$= \int_0^t ds_1 \int_0^{s_1} ds_2 \int_0^{s_2} ds_3 \, [C_{123} + C_1 C_2 C_3 + C_1 C_{23} + C_2 C_{13} + C_3 C_{12}] \mathcal{V}_1 \mathcal{V}_2 \mathcal{V}_3 \,,$$

while one the RHS,

$$\int_0^t ds_1 \mathcal{C}_1^{(3)} + \int_0^t ds_1 \int_0^{s_1} ds_2 \int_0^{s_2} ds_3 \, C_1 C_2 C_3 \mathcal{V}_1 \mathcal{V}_2 \mathcal{V}_3 + \int_0^t ds_1 ds_2 \, T_{\mathcal{C}} \left\{ \mathcal{C}_1^{(1)} \mathcal{C}_2^{(2)} \right\}$$

$$= \int_0^t ds_1 \mathcal{C}_1^{(3)} + \int_0^t ds_1 \int_0^{s_1} ds_2 \int_0^{s_2} ds_3 \, C_1 C_2 C_3 \mathcal{V}_1 \mathcal{V}_2 \mathcal{V}_3 + \int_0^t ds_1 \int_0^{s_1} ds_2 \, \left[ \mathcal{C}_1^{(1)} \mathcal{C}_2^{(2)} + \mathcal{C}_1^{(2)} \mathcal{C}_2^{(1)} \right]$$

$$= \int_0^t ds_1 \mathcal{C}_1^{(3)} + \int_0^t ds_1 \int_0^{s_1} ds_2 \int_0^{s_2} ds_3 \, C_1 C_2 C_3 \mathcal{V}_1 \mathcal{V}_2 \mathcal{V}_3$$

$$+ \int_0^t ds_1 \int_0^{s_1} ds_2 \int_0^{s_2} ds_3 \, C_1 C_{23} \mathcal{V}_1 \mathcal{V}_2 \mathcal{V}_3 + \int_0^t ds_1 \int_0^{s_1} ds_2 \int_0^{s_1} ds_3 \, C_{12} C_3 \mathcal{V}_1 \mathcal{V}_2 \mathcal{V}_3$$

$$= \int_0^t ds_1 \mathcal{C}_1^{(3)} + \int_0^t ds_1 \int_0^{s_1} ds_2 \int_0^{s_2} ds_3 \, [C_1 C_2 C_3 + C_1 C_{23} + C_2 C_{13} + C_3 C_{12}] \mathcal{V}_1 \mathcal{V}_2 \mathcal{V}_3$$

$$- \int_0^t ds_1 \int_0^{s_1} ds_2 \int_0^{s_1} ds_3 \, C_2 C_{13} \mathcal{V}_1 (\mathcal{V}_2 \mathcal{V}_3 - \mathcal{V}_3 \mathcal{V}_2) \,.$$

Comparing the two we get the 3rd order super-cumulant,

$$\mathcal{C}^{(3)}(s) = \mathcal{V}_I(s) \int_0^s ds_1 \int_0^{s_1} ds_2 \Big[ C_F^{(3)}(s, s_1, s_2) \mathcal{V}_I(s_1) \mathcal{V}_I(s_2)$$

$$+ C_F^{(1)}(s_1) C_F^{(2)}(s, s_2) \big( \mathcal{V}_I(s_1) \mathcal{V}_I(s_2) - \mathcal{V}_I(s_2) \mathcal{V}_I(s_1) \big) \Big]$$

$$= \int_0^s ds_1 \int_0^{s_1} ds_2 \mathcal{V}_I(s) \Big( C_F^{(3)}(s, s_1, s_2) \mathcal{V}_I(s_1) \mathcal{V}_I(s_2) + C_F^{(1)}(s_1) C_F^{(2)}(s, s_2) [\mathcal{V}_I(s_1), \mathcal{V}_I(s_2)] \Big) \,.$$

$$\tag{A.8}$$

The $\lambda^4$-order; we start with the LHS,

$$\int_0^t ds_1 \cdots \int_0^{s_3} ds_4 \mathcal{V}_1 \cdots \mathcal{V}_4 \big[ C_{1234} + C_{12}C_{34} + C_{13}C_{24} + C_{14}C_{23}$$
$$+ C_1 C_2 C_{34} + C_1 C_{23} C_4 + C_1 C_{24} C_3 + C_{12} C_3 C_4 + C_{13} C_2 C_4 + C_{14} C_2 C_3$$
$$+ C_1 C_{234} + C_2 C_{134} + C_3 C_{124} + C_4 C_{123} + C_1 C_2 C_3 C_4 \big]$$
$$= \int_0^t ds_1 \cdots \int_0^{s_3} ds_4 \, \mathcal{V}_1 \cdots \mathcal{V}_4 \big[ C_{1234} + C_{12}C_{34} + C_{13}C_{24} + C_{14}C_{23} \big] \,,$$

where we have finally used the assumption $\overline{F(t)} = C_F^{(1)}(t) = 0$ to simplify the expressions. The corresponding order term on the RHS reads,

$$\int_0^t ds\, \mathcal{C}^{(4)}(s) + \int_0^t ds_1 \int_0^{s_1} ds_2\, \mathcal{C}^{(2)}(s_1) \mathcal{C}^{(2)}(s_2)$$

$$= \int_0^t ds\, \mathcal{C}^{(4)}(s) + \int_0^t ds_1 \int_0^{s_1} ds_2 \int_0^{s_1} ds_3 \int_0^{s_3} ds_4\, C_{12} \mathcal{V}_1 \mathcal{V}_2 C_{34} \mathcal{V}_3 \mathcal{V}_4$$

$$= \int_0^t ds\, \mathcal{C}^{(4)}(s) + \int_0^t ds_1 \int_0^{s_1} ds_2 \int_0^{s_2} ds_3 \int_0^{s_3} ds_4\, C_{12} C_{34} \mathcal{V}_1 \cdots \mathcal{V}_4$$
$$+ \int_0^t ds_1 \int_0^{s_1} ds_2 \int_0^{s_2} ds_3 \int_0^{s_2} ds_4\, C_{13} C_{24} \mathcal{V}_1 \mathcal{V}_3 \mathcal{V}_2 \mathcal{V}_4$$

$$= \int_0^t ds\, \mathcal{C}^{(4)}(s) + \int_0^t ds_1 \int_0^{s_1} ds_2 \int_0^{s_2} ds_3 \int_0^{s_3} ds_4\, C_{12} C_{34} \mathcal{V}_1 \cdots \mathcal{V}_4$$
$$+ \int_0^t ds_1 \int_0^{s_1} ds_2 \int_0^{s_2} ds_3 \int_0^{s_3} ds_4\, C_{13} C_{24} \mathcal{V}_1 \mathcal{V}_3 \mathcal{V}_2 \mathcal{V}_4$$
$$+ \int_0^t ds_1 \int_0^{s_1} ds_2 \int_0^{s_2} ds_4 \int_0^{s_4} ds_3\, C_{14} C_{23} \mathcal{V}_1 \mathcal{V}_4 \mathcal{V}_2 \mathcal{V}_3$$

$$= \int_0^t ds\, \mathcal{C}^{(4)}(s) + \int_0^t ds_1 \int_0^{s_1} ds_2 \int_0^{s_2} ds_3 \int_0^{s_3} ds_4\, \mathcal{V}_1 \cdots \mathcal{V}_4 \big[ C_{12} C_{34} + C_{13} C_{24} + C_{14} C_{23} \big]$$
$$+ \int_0^t ds_1 \int_0^{s_1} ds_2 \int_0^{s_2} ds_3 \int_0^{s_3} ds_4\, C_{13} C_{24} \mathcal{V}_1 \left( \mathcal{V}_3 \mathcal{V}_2 - \mathcal{V}_2 \mathcal{V}_3 \right) \mathcal{V}_4$$
$$+ \int_0^t ds_1 \int_0^{s_1} ds_2 \int_0^{s_2} ds_4 \int_0^{s_4} ds_3\, C_{14} C_{23} \mathcal{V}_1 \left( \mathcal{V}_4 \mathcal{V}_2 \mathcal{V}_3 - \mathcal{V}_2 \mathcal{V}_3 \mathcal{V}_4 \right)$$

$$= \int_0^t ds\, \mathcal{C}^{(4)}(s) + \int_0^t ds_1 \int_0^{s_1} ds_2 \int_0^{s_2} ds_3 \int_0^{s_3} ds_4\, \mathcal{V}_1 \cdots \mathcal{V}_4 \big[ C_{12} C_{34} + C_{13} C_{24} + C_{14} C_{23} \big]$$
$$- \int_0^t ds_1 \int_0^{s_1} ds_2 \int_0^{s_2} ds_3 \int_0^{s_3} ds_4\, \mathcal{V}_1 \left( C_{13} C_{24} [\mathcal{V}_2, \mathcal{V}_3] \mathcal{V}_4 + C_{14} C_{23} [\mathcal{V}_2 \mathcal{V}_3, \mathcal{V}_4] \right) \,.$$

Comparing the two sides we arrive at

$$\mathcal{C}^{(4)}(s) = \int_0^s ds_1 \int_0^{s_1} ds_2 \int_0^{s_2} ds_3\, \mathcal{V}_I(s) \Big( C_F^{(4)}(s, s_1, s_2, s_3) \mathcal{V}_I(s_1) \mathcal{V}_I(s_2) \mathcal{V}_I(s_3)$$

$$+ C_F^{(2)}(s, s_2) C_F^{(2)}(s_1, s_3) [\mathcal{V}_I(s_1), \mathcal{V}_I(s_2)] \mathcal{V}_I(s_3)$$

$$+ C_F^{(2)}(s, s_3) C_F^{(2)}(s_1, s_2) [\mathcal{V}_I(s_1) \mathcal{V}_I(s_2), \mathcal{V}_I(s_3)] \Big) \,. \qquad (A.9)$$

**Assignment 24 (Sec. 3.6.4)** *Consider the stochastic Hamiltonian,*

$$\hat{H}[F](t) = \frac{\Omega}{2}\hat{\sigma}_z + \frac{\lambda}{2}F(t)\hat{\sigma}_x.$$

*Compute the stochastic map generated by this Hamiltonian using the sample average and the 2nd-order super-cumulant expansion methods for $F(t) = R_0(t)$—the random telegraph noise with zero average ($p = 0$ in $P_R^{(k)}$'s). In addition, do the same for*

$$F(t) = G(t) = \sum_{i=1}^{N} \frac{1}{\sqrt{N}} R_i(t),$$

*where $R_i(t)$ are independent, zero-average RTNs and $N \gg 1$, so that $G(t)$ approaches Gaussian process.*

*Solution*: To compute the map using the sample average method we need two ingredients: the set of sample trajectories of $R_0$, and a code to compute the unitary map for each sample. Numerical computation of unitary map was covered in Assignment 11. The sample trajectories of RTN can be generated using the code from Assignment 19; we just set the input "the probability the initial value is one" to 0.5 (this will result in zero average RTN), and we make sure that the "time step $dt$" is set to be compatible with the chosen method for computing unitary maps, e.g., if we choose Crank-Nicolson method, then $dt = h$, but for Runge-Kutta method we need $dt = h/2$ because, there, the algorithm evaluates the Hamiltonian in the midpoint between steps. To achieve a decent accuracy of the sample average, the number of samples, $N_0$, should be $\sim 10^3$.

The sample trajectories of $G(t) = (1/\sqrt{N})\sum_{i=1}^{N} R_i(t)$ can be generated using the same code as before. Since we assume here that all $R_i(t)$'s are identical (the same switch rate $w$ etc.), we simply generate $N \times N_0$ trajectories, we partition them into $N_0$ subsets, add the trajectories from each subset together and divide the sum by $\sqrt{N}$—now we have $N_0$ samples of $G(t)$.

To compute the map with the super-cumulant methods, first, we need to compute the matrix of the 2nd-order super-cumulant (here $\mathcal{C}^{(1)} = 0$),

$$\mathcal{C}^{(2)}(s) = \int_0^s du\, C_{R_0}^{(2)}(s,u)\left[\tfrac{1}{2}e^{is\frac{\Omega}{2}\hat{\sigma}_z}\hat{\sigma}_x e^{-is\frac{\Omega}{2}\hat{\sigma}_z}, \bullet\right]\left[\tfrac{1}{2}e^{iu\frac{\Omega}{2}\hat{\sigma}_z}\hat{\sigma}_x e^{-iu\frac{\Omega}{2}\hat{\sigma}_z}, \bullet\right]$$

$$= \frac{1}{4}\int_0^s du\, e^{-2w(s-u)}\left[\cos(\Omega s)\hat{\sigma}_x - \sin(\Omega s)\hat{\sigma}_y, \bullet\right]\left[\cos(\Omega u)\hat{\sigma}_x - \sin(\Omega u)\hat{\sigma}_y, \bullet\right].$$

In the basis $\mathcal{B} = \{|\uparrow\rangle\langle\uparrow|, |\downarrow\rangle\langle\downarrow|, |\uparrow\rangle\langle\downarrow|, |\downarrow\rangle\langle\uparrow|\}$, the matrix representation of the commutator super-operators read,

$$[\cos(\Omega u)\hat{\sigma}_x - \sin(\Omega u)\hat{\sigma}_y, \bullet] = \begin{bmatrix} 0 & 0 & -e^{-i\Omega u} & e^{i\Omega u} \\ 0 & 0 & e^{-i\Omega u} & -e^{i\Omega u} \\ -e^{-i\Omega u} & e^{i\Omega u} & 0 & 0 \\ e^{-i\Omega u} & -e^{i\Omega u} & 0 & 0 \end{bmatrix}_{\mathcal{B}},$$

and thus

$$\int_0^s du\, e^{-2w(s-u)}[\cos(\Omega u)\hat{\sigma}_x - \sin(\Omega u)\hat{\sigma}_y, \bullet] =$$

$$= \begin{bmatrix} 0 & 0 & \frac{e^{-2ws}-e^{-i\Omega s}}{2w-i\Omega} & -\frac{e^{-2ws}-e^{i\Omega s}}{2w+i\Omega} \\ 0 & 0 & -\frac{e^{-2ws}-e^{-i\Omega s}}{2w-i\Omega} & \frac{e^{-2ws}-e^{i\Omega s}}{2w+i\Omega} \\ \frac{e^{-2ws}-e^{i\Omega s}}{2w+i\Omega} & -\frac{e^{-2ws}-e^{i\Omega s}}{2w+i\Omega} & 0 & 0 \\ -\frac{e^{-2ws}-e^{-i\Omega s}}{2w-i\Omega} & \frac{e^{-2ws}-e^{-i\Omega s}}{2w-i\Omega} & 0 & 0 \end{bmatrix}_{\mathcal{B}}.$$

Obtaining from here the explicit matrix of $\mathcal{C}^{(2)}(s)$ is trivial. Given the matrix representation of the 2nd-order super-cumulant, we can now employ one of the numerical methods to integrate the dynamical equation,

$$\frac{d}{dt}\overline{\mathcal{U}}^{(2)}(t,0) = -\lambda^2 \mathcal{C}^{(2)}(t)\overline{\mathcal{U}}^{(2)}(t,0); \quad \overline{\mathcal{U}}^{(2)}(0,0) = \bullet,$$

and thus, compute the approximated stochastic map.

Since,

$$C_G^{(2)} = \frac{1}{N} C_{R_1+...+R_N}^{(2)} = \frac{1}{N}\sum_{i=1}^{N} C_{R_i}^{(2)} = C_{R_0}^{(2)},$$

the 2nd-order super-cumulant computation for $G(t)$ is identical to the previously obtained result.

**Assignment 25 (Sec. 4.2)** *Show that*

$$e^{-it[\hat{H}_S\otimes\hat{1}+\hat{1}\otimes\hat{H}_E,\bullet]} = e^{-it[\hat{H}_S,\bullet]} \otimes e^{-it[\hat{H}_E,\bullet]}.$$

*Solution*: First, recall that unitary maps can be expressed with unitary operators,

$$e^{-it[\hat{H}_S\otimes\hat{1}+\hat{1}\otimes\hat{H}_E,\bullet]} = e^{-it(\hat{H}_S\otimes\hat{1}+\hat{1}\otimes\hat{H}_E)} \bullet e^{it(\hat{H}_S\otimes\hat{1}+\hat{1}\otimes\hat{H}_E)}.$$

Next, since the free Hamiltonians $\hat{H}_S \otimes \hat{1}$ and $\hat{1} \otimes \hat{H}_E$ commute,

$$[\hat{H}_S \otimes \hat{1}, \hat{1} \otimes \hat{H}_E] = (\hat{H}_S \otimes \hat{1})(\hat{1} \otimes \hat{H}_E) - (\hat{1} \otimes \hat{H}_E)(\hat{H}_S \otimes \hat{1}) = \hat{H}_S \otimes \hat{H}_E - \hat{H}_S \otimes \hat{H}_E = 0,$$

the operator exp in unitary operator can be factorized like number exp,

$$e^{-it(\hat{H}_S\otimes\hat{1}+\hat{1}\otimes\hat{H}_E)} = e^{-it\hat{H}_S\otimes\hat{1}}e^{-it\,\hat{1}\otimes\hat{H}_E},$$

and, in each exp, the identity operators can be further factored out,

$$e^{-it\hat{H}_S\otimes\hat{1}} = \sum_{k=0}^{\infty}\frac{(-it)^k}{k!}(\hat{H}_S\otimes\hat{1})^k = \sum_{k=0}^{\infty}\frac{(-it)^k}{k!}\hat{H}_S\otimes\hat{1}\cdots\hat{H}_S\otimes\hat{1}$$

$$= \sum_{k=0}^{\infty}\frac{(-it)^k}{k!}\hat{H}_S^k\otimes\hat{1}^k = \left(\sum_{k=0}^{\infty}\frac{(-it)^k}{k!}\hat{H}_S^k\right)\otimes\hat{1} = e^{-it\hat{H}_S}\otimes\hat{1};$$

$$e^{-it\hat{1}\otimes\hat{H}_E} = \hat{1}\otimes e^{-it\hat{H}_E}.$$

All this leads to

$$e^{-it[\hat{H}_S\otimes\hat{1}+\hat{1}\otimes\hat{H}_E,\bullet]} = e^{-it(\hat{H}_S\otimes\hat{1}+\hat{1}\otimes\hat{H}_E)} \bullet e^{it(\hat{H}_S\otimes\hat{1}+\hat{1}\otimes\hat{H}_E)}$$

$$= \left(e^{-it\hat{H}_S}\otimes\hat{1}\right)\left(\hat{1}\otimes e^{-it\hat{H}_E}\right)\bullet\left(e^{it\hat{H}_S}\otimes\hat{1}\right)\left(\hat{1}\otimes e^{it\hat{H}_E}\right)$$

$$= \left(e^{-it\hat{H}_S}\otimes e^{-it\hat{H}_E}\right)\bullet\left(e^{it\hat{H}_S}\otimes e^{it\hat{H}_E}\right).$$

Let us now take an arbitrary operator in $SE$ space [see Eq. (166)],

$$\hat{A}_{SE} = \sum_{n,j}\sum_{n',j'}A_{nj,n'j'}|n\rangle\langle n'|\otimes|j\rangle\langle j'|,$$

and act on it with the map,

$$
\begin{aligned}
e^{-it[\hat{H}_S\otimes\hat{1}+\hat{1}\otimes\hat{H}_E,\bullet]}\hat{A}_{SE} &= \sum_{n,j}\sum_{n'j'}A_{nj,n'j'}\left(e^{-it\hat{H}_S}\otimes e^{-it\hat{H}_E}\right)\left(|n\rangle\langle n'|\otimes|j\rangle\langle j'|\right)\left(e^{it\hat{H}_S}\otimes e^{it\hat{H}_E}\right) \\
&= \sum_{n,j}\sum_{n'j'}A_{nj,n'j'}\left(e^{-it\hat{H}_S}|n\rangle\langle n'|e^{it\hat{H}_S}\right)\otimes\left(e^{-it\hat{H}_E}|j\rangle\langle j'|e^{it\hat{H}_E}\right) \\
&= \sum_{n,j}\sum_{n'j'}A_{nj,n'j'}\left(e^{-it[\hat{H}_S,\bullet]}|n\rangle\langle n'|\right)\otimes\left(e^{-it[\hat{H}_E,\bullet]}|j\rangle\langle j'|\right) \\
&= \sum_{n,j}\sum_{n'j'}A_{nj,n'j'}\left(e^{-it[\hat{H}_S,\bullet]}\otimes e^{-it[\hat{H}_E,\bullet]}\right)\left(|n\rangle\langle n'|\otimes|j\rangle\langle j'|\right) \\
&= \left(e^{-it[\hat{H}_S,\bullet]}\otimes e^{-it[\hat{H}_E,\bullet]}\right)\sum_{n,j}\sum_{n'j'}A_{nj,n'j'}|n\rangle\langle n'|\otimes|j\rangle\langle j'| \\
&= e^{-it[\hat{H}_S,\bullet]}\otimes e^{-it[\hat{H}_E,\bullet]}\hat{A}_{SE}\,.
\end{aligned}
$$

Since $\hat{A}_{SE}$ was arbitrary, we conclude that (178) is true.

**Assignment 26 (Sec. 4.2)** *Find the state $\hat{\rho}_S(t)$ of a qubit S open to the environment E composed of a single qubit. The total Hamiltonian is given by*

$$
\hat{H}_{SE} = \frac{\lambda}{2}\hat{\sigma}_z\otimes\hat{\sigma}_z\,,
$$

*and the initial state is*

$$
\hat{\rho}_{SE} = |X\rangle\langle X|\otimes|X\rangle\langle X|\,,
$$

*where $|X\rangle = (|1\rangle+|-1\rangle)/\sqrt{2}$ and $\hat{\sigma}_z|\pm1\rangle = \pm|\pm1\rangle$.*

*Solution*: Since $\hat{H}_{SE}$ is constant, the Hamiltonian can be diagonalized. It is quite obvious that $\hat{\sigma}_z\otimes\hat{\sigma}_z$ is diagonal in a composite basis made out of eigenkets of $\hat{\sigma}_z$, $B_{SE} = \{|s\rangle\otimes|e\rangle\}_{s,e=\pm1}$,

$$
\hat{H}_{SE}|s\rangle\otimes|e\rangle = se\frac{\lambda}{2}|s\rangle\otimes|e\rangle \;\Rightarrow\; e^{-it\hat{H}_{SE}}|s\rangle\otimes|e\rangle = e^{-ise\frac{\lambda}{2}}|s\rangle\otimes|e\rangle\,.
$$

Given the decomposition of $|X\rangle$ in the basis of $\hat{\sigma}_z$, we can easily compute the evolution of the state of the total system,

$$
\begin{aligned}
\hat{\rho}_{SE}(t) &= e^{-it\hat{H}_{SE}}|X\rangle\langle X|\otimes|X\rangle\langle X|e^{it\hat{H}_{SE}} = \frac{1}{4}\sum_{s,s',e,e'=\pm1}e^{-it\hat{H}_{SE}}|s\rangle\langle s'|\otimes|e\rangle\langle e'|e^{it\hat{H}_{SE}} \\
&= \frac{1}{4}\sum_{s,s',e,e'=\pm1}\left(e^{-it\hat{H}_{SE}}|s\rangle\otimes|e\rangle\right)\left(\langle s'|\otimes\langle e'|e^{it\hat{H}_{SE}}\right) \\
&= \frac{1}{4}\sum_{s,s',e,e'=\pm1}e^{-\frac{1}{2}i(se-s'e')\lambda t}\left(|s\rangle\otimes|e\rangle\right)\left(\langle s'|\otimes\langle e'|\right) \\
&= \frac{1}{4}\begin{bmatrix} 1 & e^{-i\lambda t} & e^{-i\lambda t} & 1 \\ e^{i\lambda t} & 1 & & e^{i\lambda t} \\ e^{i\lambda t} & & 1 & e^{i\lambda t} \\ 1 & e^{-i\lambda t} & e^{-i\lambda t} & 1 \end{bmatrix}_{B_{SE}}.
\end{aligned}
$$

Now, we can calculate the partial trace over $E$ to obtain the reduced state of $S$,

$$\hat{\rho}_S(t) = \mathrm{tr}_E(\hat{\rho}_{SE}(t)) = \frac{1}{4} \sum_{e''=\pm1} \sum_{s,s',e,e'=\pm1} e^{-\frac{1}{2}i(se-s'e')\lambda t} \left(\langle e''|e\rangle|s\rangle\right)\left(\langle s'|\langle e'|e''\rangle\right)$$

$$= \frac{1}{4} \sum_{e''=\pm1} \sum_{s,s'=\pm1} e^{-\frac{1}{2}i(s-s')e''\lambda t}|s\rangle\langle s'| = \frac{1}{2} \sum_{s,s'=\pm1} \cos\left[\frac{(s-s')\lambda t}{2}\right]|s\rangle\langle s'|$$

$$= \frac{1}{2} \begin{bmatrix} 1 & \cos(\lambda t) \\ \cos(\lambda t) & 1 \end{bmatrix}_{B_S},$$

where $B_S = \{|1\rangle, |-1\rangle\}$ is a basis in $S$ space.

*Bonus*: Consider a variation on the above setup: the environment $E$ is now composed of $n \gg 1$ independent qubits that couple to $S$ in the same $\hat{\sigma}_z \otimes \hat{\sigma}_z$ way,

$$\hat{H}_{SE} = \frac{1}{2} \frac{\lambda}{\sqrt{n}} \hat{\sigma}_z \otimes \left(\sum_{k=1}^{n} \hat{1}^{\otimes(k-1)} \otimes \hat{\sigma}_z \otimes \hat{1}^{\otimes(n-k+1)}\right).$$

Note that the coupling constant scales with the size of $E$. The initial state of the total system is $\hat{\rho}_{SE} = |X\rangle\langle X|^{\otimes(n+1)}$.

Repeating the steps we have followed previously leads us to the reduced state of $S$,

$$\hat{\rho}_S(t) = \frac{1}{2} \begin{bmatrix} 1 & \cos^n\left(\frac{\lambda}{\sqrt{n}}t\right) \\ \cos^n\left(\frac{\lambda}{\sqrt{n}}t\right) & 1 \end{bmatrix}_{B_S} \xrightarrow{n\to\infty} \frac{1}{2} \begin{bmatrix} 1 & e^{-\frac{\lambda^2 t^2}{2}} \\ e^{-\frac{\lambda^2 t^2}{2}} & 1 \end{bmatrix}_{B_S}.$$

**Assignment 27 (Sec. 4.2)** *Calculate the entropy of state of the open qubit system from Assignment 26.*

*Solution*: Using elementary methods we find the eigenvalues of $\hat{\rho}_S(t)$:

$$(\text{eigenvalues}) = \left\{\cos^2\left(\frac{\lambda t}{2}\right), \sin^2\left(\frac{\lambda t}{2}\right)\right\}.$$

Therefore, the entropy of the state after the evolution period $t$ is given by

$$S(\hat{\rho}_S(t)) = -\cos^2\left(\frac{\lambda t}{2}\right)\ln\left[\cos^2\left(\frac{\lambda t}{2}\right)\right] - \sin^2\left(\frac{\lambda t}{2}\right)\ln\left[\sin^2\left(\frac{\lambda t}{2}\right)\right].$$

Initially, the open system is in pure state, $\hat{\rho}_S(0) = \mathrm{tr}_E(|X\rangle\langle X| \otimes |X\rangle\langle X|) = |X\rangle\langle X|$, so its entropy is minimal,

$$S(\hat{\rho}_S(0)) = S(|X\rangle\langle X|) = 0.$$

Depending on the duration of the evolution, the reduced density matrix oscillates between pure and completely mixed state, in particular

$$S\left(\hat{\rho}_S\left(n\frac{\pi}{\lambda}\right)\right) = 0;$$

$$S\left(\hat{\rho}_S\left(\left(n+\tfrac{1}{2}\right)\frac{\pi}{\lambda}\right)\right) = S\left(\tfrac{1}{2}\hat{1}\right) = \ln(2),$$

for $n \in$ Integers.

This example shows that dynamical maps *do not* preserve the purity of the initial state. Although, in this particular case the entropy is capped from below by $S(\hat{\rho}_S(0))$ [even for mixed $\hat{\rho}_S(0)$], in general, the open system dynamics can change the entropy in either direction in comparison to the initial value.

**Assignment 28 (Sec. 4.3)** *Prove the joint quasi-probability representation for dynamical maps* (193),

$$\langle \mathcal{U} \rangle_I(t,0) = \bullet + \sum_{k=1}^{\infty} (-i\lambda)^k \int_0^t ds_1 \cdots \int_0^{s_{k-1}} ds_k$$
$$\times \sum_{f_1, \bar{f}_1} \cdots \sum_{f_k, \bar{f}_k} Q_F^{(k)}(f_1, \bar{f}_1, s_1; \ldots; f_k, \bar{f}_k, s_k) \mathcal{W}(f_1, \bar{f}_1, s_1) \cdots \mathcal{W}(f_k, \bar{f}_k, s_k) \,.$$

*Solution*: Recall the series representation of dynamical map,

$$\langle \mathcal{U} \rangle_I(t,0) = \sum_{k=0}^{\infty} (-i\lambda)^k \mathrm{tr}_E \left( \int_0^t ds_1 [\hat{V}_I(s_1) \otimes \hat{F}_I(s_1), \bullet] \cdots \int_0^{s_{k-1}} ds_k [\hat{V}_I(s_k) \otimes \hat{F}_I(s_k), \bullet \otimes \hat{\rho}_E] \right)$$

$$= \bullet + \sum_{k=1}^{\infty} (-i\lambda)^k \int_0^t ds_1 \cdots \int_0^{s_{k-1}} ds_k \sum_{f_1} \cdots \sum_{f_k} f_1 \cdots f_k$$
$$\times \mathrm{tr}_E \left( [\hat{V}_I(s_1) \otimes \hat{P}(f_1, s_1), \bullet] \cdots [\hat{V}_I(s_k) \otimes \hat{P}_I(f_k, s_k), \bullet \otimes \hat{\rho}_E] \right),$$

where we have substituted for $\hat{F}_I(s)$ its spectral decomposition (191). We focus now on the action of one of the commutator super-operators; first, we act with $f[\hat{V}_I(s) \otimes \hat{P}_I(f, s), \bullet]$ on an arbitrary operator in a product form $\hat{S} \otimes \hat{E}$,

$$\sum_f f[\hat{V}_I(s) \otimes \hat{P}_I(f, s), \hat{S} \otimes \hat{E}] = \sum_f f \left( \hat{V}_I(s)\hat{S} \otimes \hat{P}_I(f, s)\hat{E} - \hat{S}\hat{V}_I(s) \otimes \hat{E}\hat{P}_I(f, s) \right)$$

$$= \sum_f f \left( \hat{V}_I(s)\hat{S} \otimes \hat{P}_I(f, s)\hat{E}e^{is\hat{H}_E} \sum_{\bar{f}} \hat{P}(\bar{f})e^{-is\hat{H}_E} - \hat{S}\hat{V}_I(s) \otimes e^{is\hat{H}_E} \sum_{\bar{f}} \hat{P}(\bar{f})e^{-is\hat{H}_E}\hat{E}\hat{P}_I(f, s) \right)$$

$$= \sum_{f, \bar{f}} \left( f\hat{V}_I(s)\hat{S} \otimes \hat{P}_I(f, s)\hat{E}\hat{P}_I(\bar{f}, s) - \bar{f}\hat{S}\hat{V}_I(s) \otimes \hat{P}_I(f, s)\hat{E}\hat{P}_I(\bar{f}, s) \right)$$

$$= \sum_{f, \bar{f}} \left( f\hat{V}_I(s)\hat{S} - \bar{f}\hat{S}\hat{V}_I(s) \right) \otimes \hat{P}_I(f, s)\hat{E}\hat{P}_I(\bar{f}, s) = \sum_{f, \bar{f}} \mathcal{W}(f, \bar{f}, s)\hat{S} \otimes \hat{P}_I(f, s)\hat{E}\hat{P}_I(\bar{f}, s),$$

where we have used the decomposition of identity $\hat{1} = \sum_{\bar{f}} \hat{P}(\bar{f})$, $\hat{1} = e^{is\hat{H}_E} e^{-is\hat{H}_E}$, the definition of $\mathcal{W}$ (194), and we have switched variables names $f \leftrightarrow \bar{f}$ in one of the terms. Now take an arbitrary basis in $S$- and $E$-operator spaces, $\mathcal{B}_S = \{\hat{S}_i\}_i$ and $\mathcal{B}_E = \{\hat{E}_n\}_n$; create a product basis in $SE$-operator space, $\mathcal{B}_{SE} = \{\hat{S}_i \otimes \hat{E}_n\}_{i,n}$, and decompose in it an arbitrary $SE$ operator $A_{SE}$,

$$\hat{A}_{SE} = \sum_{i,n} \mathrm{tr}\left( \hat{S}_i^{\dagger} \otimes \hat{E}_n^{\dagger} \hat{A}_{SE} \right) \hat{S}_i \otimes \hat{E}_n = \sum_{i,n} A_{in} \hat{S}_i \otimes \hat{E}_n \,.$$

Act with $f[\hat{V}_I(s) \otimes \hat{P}_I(f, s), \bullet]$ on $\hat{A}_{SE}$,

$$\sum_f f[\hat{V}_I(s) \otimes \hat{P}_I(f, s), \hat{A}_{SE}] = \sum_{i,n} A_{in} \sum_f f[\hat{V}_I(s) \otimes \hat{P}_I(f, s), \hat{S}_i \otimes \hat{E}_n]$$

$$= \sum_{i,n} A_{in} \sum_{f, \bar{f}} \mathcal{W}(f, \bar{f}, s)\hat{S}_i \otimes \hat{P}_I(f, s)\hat{E}_n \hat{P}_I(\bar{f}, s)$$

$$= \sum_{f, \bar{f}} \left( \mathcal{W}(f, \bar{f}, s)\bullet \right) \otimes \left( \hat{P}_I(f, s)\bullet \hat{P}_I(\bar{f}, s) \right) \sum_{i,n} A_{in} \hat{S}_i \otimes \hat{E}_n$$

$$= \sum_{f, \bar{f}} \mathcal{W}(f, \bar{f}, s) \otimes \left( \hat{P}_I(f, s)\bullet \hat{P}_I(\bar{f}, s) \right) \hat{A}_{SE} \,.$$

But $\hat{A}_{SE}$ was arbitrary, therefore the above equality is true for the super-operators,

$$\sum_f f[\hat{V}_I(s) \otimes \hat{P}_I(f,s), \bullet] = \sum_{f,\bar{f}} \mathcal{W}(f,\bar{f},s) \otimes \hat{P}_I(f,s) \bullet \hat{P}_I(\bar{f},s).$$

Now, we can apply the commutator super-operators and calculate the partial trace,

$$\langle \mathcal{U} \rangle_I(t,0) = \bullet + \sum_{k=1}^{\infty} (-i\lambda)^k \int_0^t ds_1 \cdots \int_0^{s_{k-1}} ds_k \sum_{f_1,\bar{f}_1} \cdots \sum_{f_k,\bar{f}_k}$$

$$\times \mathrm{tr}_E \left( \mathcal{W}(f_1,\bar{f}_1,s_1) \cdots \mathcal{W}(f_k,\bar{f}_k,s_k) \otimes \hat{P}_I(f_1,s_1) \cdots \hat{P}_I(f_k,s_k) \hat{\rho}_E \hat{P}_I(\bar{f}_k,s_k) \cdots \hat{P}_I(\bar{f}_1,s_1) \right)$$

$$= \bullet + \sum_{k=1}^{\infty} (-i\lambda)^k \int_0^t ds_1 \cdots \int_0^{s_{k-1}} ds_k \sum_{f_1,\bar{f}_1} \cdots \sum_{f_k,\bar{f}_k}$$

$$\times Q_F^{(k)}(f_1,\bar{f}_1,s_1;\ldots;f_k,\bar{f}_k,s_k) \mathcal{W}(f_1,\bar{f}_1,s_1) \cdots \mathcal{W}(f_k,\bar{f}_k,s_k),$$

where $Q_F^{(k)}$ is given by Eq. (195).

**Assignment 29 (Sec. 4.3)** *Prove consistency relation* (197),

$$\sum_{f_i,\bar{f}_i} Q_F^{(k)}(f_1,\bar{f}_1,s_1;\ldots;f_k,\bar{f}_k,s_k) = Q_F^{(k-1)}(\ldots;f_{i-1},\bar{f}_{i-1},s_{i-1};f_{i+1},\bar{f}_{i+1},s_{i+1};\ldots).$$

*Solution*: Compute the LHS of Eq. (197),

$$\sum_{f_i,\bar{f}_i} Q_F^{(k)}(f_1,\bar{f}_1,s_1;\ldots;f_k,\bar{f}_k,s_k)$$

$$= \mathrm{tr}_E \left( \hat{P}_I(f_1,s_1) \cdots \sum_{f_i} \hat{P}_I(f_i,s_i) \cdots \hat{P}_I(f_k,s_k) \hat{\rho}_E \hat{P}_I(\bar{f}_k,s_k) \cdots \sum_{\bar{f}_i} \hat{P}_I(\bar{f}_i,s_i) \cdots \hat{P}_I(\bar{f}_1,s_1) \right)$$

$$= Q_F^{(k-1)}(\ldots;f_{i-1},\bar{f}_{i-1},s_{i-1};f_{i+1},\bar{f}_{i+1},s_{i+1};\ldots),$$

where we used the decomposition of identity (192),

$$\sum_{f_i} \hat{P}_I(f_i,s_i) = e^{is_i\hat{H}_E} \left( \sum_{f_i} \hat{P}(f_i) \right) e^{-is_i\hat{H}_E} = e^{is_i\hat{H}_E} \hat{1} e^{-is_i\hat{H}_E} = \hat{1}.$$

**Assignment 30 (Sec. 4.3)** *Find the explicit form of quasi-moments (assume $t > s$):*

$$\langle F(t) \rangle = \sum_{f_1,\bar{f}_1} Q_F^{(1)}(f_1,\bar{f}_1,t) f_1;$$

$$\langle \bar{F}(t) \rangle = \sum_{f_1,\bar{f}_1} Q_F^{(1)}(f_1,\bar{f}_1,t) \bar{f}_1;$$

$$\langle F(t)F(s) \rangle = \sum_{f_1,\bar{f}_1} \sum_{f_2,\bar{f}_2} Q_F^{(2)}(f_1,\bar{f}_1,t;f_2,\bar{f}_2,s) f_1 f_2;$$

$$\langle F(t)\bar{F}(s) \rangle = \sum_{f_1,\bar{f}_1} \sum_{f_2,\bar{f}_2} Q_F^{(2)}(f_1,\bar{f}_1,t;f_2,\bar{f}_2,s) f_1 \bar{f}_2;$$

$$\langle \bar{F}(t)F(s) \rangle = \sum_{f_1,\bar{f}_1} \sum_{f_2,\bar{f}_2} Q_F^{(2)}(f_1,\bar{f}_1,t;f_2,\bar{f}_2,s) \bar{f}_1 f_2;$$

$$\langle \bar{F}(t)\bar{F}(s) \rangle = \sum_{f_1,\bar{f}_1} \sum_{f_2,\bar{f}_2} Q_F^{(2)}(f_1,\bar{f}_1,t;f_2,\bar{f}_2,s) \bar{f}_1 \bar{f}_2.$$

SciPost Phys. Lect. Notes 68 (2023)

*Solution*: The 1st quasi-moment,

$$\langle F(t)\rangle = \sum_{f_1,\bar{f}_1} Q_F^{(1)}(f_1,\bar{f}_1,t)f_1$$

$$= \mathrm{tr}_E\left(e^{it\hat{H}_E}\left(\sum_{f_1} f_1\hat{P}(f_1)\right)e^{-it\hat{H}_E}\hat{\rho}_E e^{it\hat{H}_E}\left(\sum_{\bar{f}_1}\hat{P}(\bar{f}_1)\right)e^{-it\hat{H}_E}\right)$$

$$= \mathrm{tr}_E\left(\hat{F}_I(t)\hat{\rho}_E\right),$$

where we used the spectral decomposition $\sum_f f\,\hat{P}(f) = \hat{F}$ [see Eq. (191)] and the decomposition of identity $\sum_f \hat{P}(f) = \hat{1}$ [see Eq. (192)]. As a result, we get that the first quasi-moment equals the quantum expectation value of operator $\hat{F}_I(t)$ on the state $\hat{\rho}_E$. For the first quasi-moment, the backwards process $\bar{F}(t)$ gives the same result as the forwards $F(t)$,

$$\langle \bar{F}(t)\rangle = \mathrm{tr}_E\left(\hat{\rho}_E\hat{F}_I(t)\right) = \mathrm{tr}_E\left(\hat{F}_I(t)\hat{\rho}_E\right) = \langle F(t)\rangle.$$

The second quasi-moments:

$$\langle F(t)F(s)\rangle = \sum_{f_1,\bar{f}_1}\sum_{f_2,\bar{f}_2} Q_F^{(2)}(f_1,\bar{f}_1,t;f_2,\bar{f}_2,s)f_1 f_2$$

$$= \mathrm{tr}_E\left(e^{it\hat{H}_E}\left(\sum_{f_1} f_1\hat{P}(f_1)\right)e^{-it\hat{H}_E}e^{is\hat{H}_E}\left(\sum_{f_2} f_2\hat{P}(f_2)\right)e^{-is\hat{H}_E}\hat{\rho}_E\right.$$

$$\left.\times\, e^{is\hat{H}_E}\left(\sum_{\bar{f}_2}\hat{P}(\bar{f}_2)\right)e^{-is\hat{H}_E}e^{it\hat{H}_E}\left(\sum_{\bar{f}_1}\hat{P}(\bar{f}_1)\right)e^{-it\hat{H}_E}\right)$$

$$= \mathrm{tr}_E\left(\hat{F}_I(t)\hat{F}_I(s)\hat{\rho}_E\right);$$

$$\langle \bar{F}(t)\bar{F}(s)\rangle = \mathrm{tr}_E\left(\hat{\rho}_E\hat{F}_I(s)\hat{F}_I(t)\right) = \mathrm{tr}_E\left(\hat{F}_I(s)\hat{F}_I(t)\hat{\rho}_E\right);$$

$$\langle F(t)\bar{F}(s)\rangle = \mathrm{tr}_E\left(\hat{F}_I(t)\hat{\rho}_E\hat{F}_I(s)\right) = \langle \bar{F}(t)\bar{F}(s)\rangle;$$

$$\langle \bar{F}(t)F(s)\rangle = \mathrm{tr}_E\left(\hat{F}_I(s)\hat{\rho}_E\hat{F}_I(t)\right) = \langle F(t)F(s)\rangle.$$

**Assignment 31 (Sec. 4.3)** *Find the explicit form of super-quasi-moments,*

$$\langle \mathcal{W}(F(t),\bar{F}(t),t)\rangle, \quad \langle \mathcal{W}(F(t),\bar{F}(t),t)\mathcal{W}(F(s),\bar{F}(s),s)\rangle \quad (t>s).$$

*Solution*: The 1st order super-quasi-moment:

$$\langle \mathcal{W}(F(t),\bar{F}(t),t)\rangle = \sum_{f,\bar{f}} Q_F^{(1)}(f,\bar{f},t)\left(f\hat{V}_I(t)\bullet - \bullet\hat{V}_I(t)\bar{f}\right) = \mathrm{tr}_E\left(\hat{F}_I(t)\hat{\rho}_E\right)[\hat{V}_I(t),\bullet]$$

$$= \langle F(t)\rangle[\hat{V}_I(t),\bullet];$$

and the 2nd super-quasi-moment,

$$\langle \mathcal{W}(F(t),\bar{F}(t),t)\mathcal{W}(F(s),\bar{F}(s),s)\rangle$$

$$= \langle F(t)F(s)\rangle\left(\hat{V}_I(t)\bullet\right)\left(\hat{V}_I(s)\bullet\right) - \langle F(t)\bar{F}(s)\rangle\left(\hat{V}_I(t)\bullet\right)\left(\bullet\hat{V}_I(s)\right)$$

$$- \langle \bar{F}(t)F(s)\rangle\left(\bullet\hat{V}_I(t)\right)\left(\hat{V}_I(s)\bullet\right) + \langle \bar{F}(t)\bar{F}(s)\rangle\left(\bullet\hat{V}_I(t)\right)\left(\bullet\hat{V}_I(s)\right)$$

$$= \langle F(t)F(s)\rangle\left(\hat{V}_I(t)\hat{V}_I(s)\bullet - \hat{V}_I(s)\bullet\hat{V}_I(t)\right)$$

$$+ \langle \bar{F}(t)\bar{F}(s)\rangle\left(\bullet\hat{V}_I(s)\hat{V}_I(t) - \hat{V}_I(t)\bullet\hat{V}_I(s)\right).$$

**Assignment 32 (Sec. 4.4)** *Find the first two qumulant distributions $\tilde{Q}_F^{(1)}$ and $\tilde{Q}_F^{(2)}$ without invoking the structural parallelism between stochastic and dynamical maps.*

*Solution*: We start with the quasi-moment series for the dynamical map,

$$\langle \mathcal{U} \rangle_I(t,0) = \bullet - i\lambda \int_0^t ds_1 Q_1 \mathcal{W}_1 - \lambda^2 \int_0^t ds_1 \int_0^{s_1} ds_2 Q_{12} \mathcal{W}_1 \mathcal{W}_2 + \dots,$$

where we have introduced the shorthand notation,

$$Q_{1\dots k} \mathcal{W}_1 \cdots \mathcal{W}_k = \sum_{f_1, \bar{f}_1} \cdots \sum_{f_k, \bar{f}_k} Q_F^{(k)}(f_1, \bar{f}_1, s_1; \dots; f_k, \bar{f}_k, s_k) \mathcal{W}(f_1, \bar{f}_1, s_1) \cdots \mathcal{W}(f_k, \bar{f}_k, s_k).$$

We define implicitly the qumulant expansion,

$$\langle \mathcal{U} \rangle_I(t,0) \equiv T_{\mathcal{W}} e^{\sum_{k=1}^{\infty}(-i\lambda)^k \int_0^t ds_1 \cdots \int_0^{s_{k-1}} ds_k \tilde{Q}_{1\dots k} \mathcal{W}_1 \cdots \mathcal{W}_k}.$$

As usual, we find $\tilde{Q}_F^{(k)}$ by expanding the RHS into power series in $\lambda$ and compare it with the quasi-moment series. The $\lambda^1$-order is very simple,

$$\int_0^t ds_1 Q_1 \mathcal{W}_1 = \int_0^t ds_1 \tilde{Q}_1 \mathcal{W}_1 \Rightarrow \tilde{Q}_F^{(1)}(f_1, \bar{f}_1, s_1) = Q_F^{(1)}(f_1, \bar{f}_1, s_1);$$

the $\lambda^2$-order is more complex,

$$\int_0^t ds_1 \int_0^{s_1} ds_2 Q_{12} \mathcal{W}_1 \mathcal{W}_2 = \frac{1}{2} \int_0^t ds_1 ds_2 \tilde{Q}_1 \tilde{Q}_2 T_{\mathcal{W}} \{\mathcal{W}_1 \mathcal{W}_2\} + \int_0^t ds_1 \int_0^{s_1} ds_2 \tilde{Q}_{12} \mathcal{W}_1 \mathcal{W}_2$$

$$= \int_0^t ds_1 \int_0^{s_1} ds_2 \left[ \tilde{Q}_1 \tilde{Q}_2 + \tilde{Q}_{12} \right] \mathcal{W}_1 \mathcal{W}_2,$$

which implies,

$$\tilde{Q}_F^{(2)}(f_1, \bar{f}_1, s_1; f_2, \bar{f}_2, s_2) = Q_F^{(2)}(f_1, \bar{f}_1, s_1; f_2, \bar{f}_2, s_2) - Q_F^{(1)}(f_1, \bar{f}_1, s_1) Q_F^{(1)}(f_2, \bar{f}_2, s_2).$$

**Assignment 33 (Sec. 4.4)** *Calculate the explicit forms of super-qumulants $\mathcal{L}^{(1)}(s)$ and $\mathcal{L}^{(2)}(s)$.*

*Solution*: We will be employing here the same shorthand notation as in Assignment 32:

$$\mathcal{W}_i = \mathcal{W}(F(s_i), \bar{F}(s_i), s_i),$$

$$\tilde{Q}_{1\dots k} \mathcal{W}_1 \cdots \mathcal{W}_k = \sum_{f_1, \bar{f}_1} \cdots \sum_{f_k, \bar{f}_k} \tilde{Q}_F^{(k)}(f_1, \bar{f}_1, s_1; \dots; f_k, \bar{f}_k, s_k) \mathcal{W}(f_1, \bar{f}_1, s_1) \cdots \mathcal{W}(f_k, \bar{f}_k, s_k).$$

The first super-qumulant,

$$\mathcal{L}^{(1)}(s_1) = \langle\langle \mathcal{W}_1 \rangle\rangle = \tilde{Q}_1 \mathcal{W}_1 = Q_1 \mathcal{W}_1 = \langle F(s_1) \rangle \hat{V}_I(s_1) \bullet - \bullet \hat{V}_I(s_1) \langle \bar{F}(s_1) \rangle$$

$$= \text{tr}_E\left(\hat{F}_I(s_1)\hat{\rho}_E\right)[\hat{V}_I(s_1), \bullet] = \langle F(s_1) \rangle [\hat{V}_I(s_1), \bullet];$$

the second super-qumulant,

$$\mathcal{L}^{(2)}(s_1) = \int_0^{s_1} \langle\!\langle \mathcal{W}_1 \mathcal{W}_2 \rangle\!\rangle ds_2 = \int_0^{s_1} (Q_{12}\mathcal{W}_1\mathcal{W}_2 - Q_1\mathcal{W}_1 Q_2\mathcal{W}_2) ds_2$$

$$= \int_0^{s_1} \langle F(s_1)F(s_2)\rangle \left(\hat{V}_I(s_1)\hat{V}_I(s_2)\bullet - \hat{V}_I(s_2)\bullet \hat{V}_I(s_1)\right) ds_2$$

$$+ \int_0^{s_1} \langle \bar{F}(s_1)\bar{F}(s_2)\rangle \left(\bullet\hat{V}_I(s_2)\hat{V}_I(s_1) - \hat{V}_I(s_1)\bullet \hat{V}_I(s_2)\right) ds_2$$

$$- \int_0^{s_1} \langle F(s_1)\rangle \langle F(s_2)\rangle [\hat{V}_I(s_1),[\hat{V}_I(s_2),\bullet]] ds_2,$$

where, for $t > s$,

$$\langle F(t)F(s)\rangle = \mathrm{tr}_E\left(\hat{F}_I(t)\hat{F}_I(s)\hat{\rho}_E\right),$$
$$\langle \bar{F}(t)\bar{F}(s)\rangle = \mathrm{tr}_E\left(\hat{F}_I(s)\hat{F}_I(t)\hat{\rho}_E\right).$$

**Assignment 34 (Sec. 4.4)** *Find the quasi-probability representation of the dynamical map for general SE coupling*

$$\hat{V}_{SE} = \lambda \sum_\alpha \hat{V}_\alpha \otimes \hat{F}_\alpha,$$

*where $\hat{V}_\alpha$ and $\hat{F}_\alpha$ are hermitian operators.*
   *Note that this also includes couplings of form*

$$\hat{V}_{SE} = \lambda \left(\hat{S} \otimes \hat{E} + \hat{S}^\dagger \otimes \hat{E}^\dagger\right),$$

*where $\hat{S}$ and $\hat{E}$ are non-hermitian. Operator like this can always be transformed into a combination of hermitian products:*

$$\hat{S} \otimes \hat{E} + \hat{S}^\dagger \otimes \hat{E}^\dagger = \frac{\hat{S}+\hat{S}^\dagger}{\sqrt{2}} \otimes \frac{\hat{E}+\hat{E}^\dagger}{\sqrt{2}} + \frac{\hat{S}-\hat{S}^\dagger}{i\sqrt{2}} \otimes \frac{\hat{E}^\dagger-\hat{E}}{i\sqrt{2}}.$$

*Solution*: The power series representation of the dynamical map,

$$\langle \mathcal{U}\rangle_I(t,0) = \sum_{k=0}^\infty (-i\lambda)^k \mathrm{tr}_E\left(\int_0^t ds_1[\hat{V}_{SE}(s_1),\bullet]\cdots\int_0^{s_{k-1}} ds_k[\hat{V}_{SE}(s_k),\bullet\otimes\hat{\rho}_E]\right)$$

$$= \sum_{k=0}^\infty (-i\lambda)^k \sum_{\alpha_1,\dots,\alpha_k}$$

$$\times \mathrm{tr}_E\left(\int_0^t ds_1[\hat{V}_{\alpha_1}(s_1)\otimes\hat{F}_{\alpha_1}(s_1),\bullet]\cdots\int_0^{s_{k-1}} ds_k[\hat{V}_{\alpha_k}(s_k)\otimes\hat{F}_{\alpha_k}(s_k),\bullet\otimes\hat{\rho}_E]\right),$$

where we have dropped $I$ subscript from interaction picture operators,

$$\hat{V}_{SE}(s) = \left(e^{is\hat{H}_S}\otimes e^{is\hat{H}_E}\right)\hat{V}_{SE}\left(e^{-is\hat{H}_S}\otimes e^{-is\hat{H}_E}\right),$$
$$\hat{V}_\alpha(s) = e^{is\hat{H}_S}\hat{V}_\alpha e^{-is\hat{H}_S},$$
$$\hat{F}_\alpha(s) = e^{is\hat{H}_E}\hat{F}_\alpha e^{-is\hat{H}_E}.$$

We will need the spectral decomposition for each $\hat{F}_\alpha$ operator,

$$\hat{F}_\alpha(s) = e^{is\hat{H}_E} \sum_{f^\alpha} f^\alpha \hat{P}_\alpha(f^\alpha) e^{-is\hat{H}_E} \equiv \sum_{f^\alpha} f^\alpha \hat{P}_\alpha(f^\alpha, s).$$

Substituting the decomposed form back into the dynamical map we get

$$\langle \mathcal{U} \rangle_I(t,0) = \bullet + \sum_{k=1}^\infty (-i\lambda)^k \int_0^t ds_1 \cdots \int_0^{s_{k-1}} ds_k \sum_{\alpha_1,\ldots,\alpha_k} \sum_{f_1^{\alpha_1}} \cdots \sum_{f_k^{\alpha_k}} f_1^{\alpha_1} \cdots f_k^{\alpha_k}$$
$$\times \operatorname{tr}_E \left( [\hat{V}_{\alpha_1}(s_1) \otimes \hat{P}_{\alpha_1}(f_1^{\alpha_1}, s_1), \bullet] \cdots [\hat{V}_{\alpha_k}(s_k) \otimes \hat{P}_{\alpha_k}(f_k^{\alpha_k}, s_k), \bullet \otimes \hat{\rho}_E] \right),$$

and we focus on the action of one of the commutator super-operators,

$$\sum_{f^\alpha} f^\alpha [\hat{V}_\alpha(s) \otimes \hat{P}_\alpha(f^\alpha, s), \hat{S} \otimes \hat{E}]$$
$$= \sum_{f^\alpha} \sum_{\bar{f}^\alpha} \left( f^\alpha \hat{V}_\alpha(s) \hat{S} - \bar{f}^\alpha \hat{S} \hat{V}_\alpha(s) \right) \otimes \hat{P}_\alpha(f^\alpha, s) \hat{E} \hat{P}_\alpha(\bar{f}^\alpha, s)$$
$$\equiv \sum_{f^\alpha} \sum_{\bar{f}^\alpha} \mathcal{W}_\alpha(f^\alpha, \bar{f}^\alpha, s) \hat{S} \otimes \hat{P}_\alpha(f^\alpha, s) \hat{E} \hat{P}_\alpha(\bar{f}^\alpha, s).$$

Insert this into the dynamical map and compute the partial trace to get the quasi-probability representation,

$$\langle \mathcal{U} \rangle_I(t,0) = \bullet + \sum_{k=1}^\infty (-i\lambda)^k \int_0^t ds_1 \cdots \int_0^{s_{k-1}} ds_k \sum_{\alpha_1,\ldots,\alpha_k} \sum_{f_1^{\alpha_1}} \sum_{\bar{f}_1^{\alpha_1}} \cdots \sum_{f_k^{\alpha_k}} \sum_{\bar{f}_k^{\alpha_k}}$$
$$\times Q_{\alpha_1,\ldots,\alpha_k}^{(k)}(f_1^{\alpha_1}, \bar{f}_1^{\alpha_1}, s_1; \ldots; f_k^{\alpha_k}, \bar{f}_k^{\alpha_k}, s_k) \mathcal{W}_{\alpha_1}(f_1^{\alpha_1}, \bar{f}_1^{\alpha_1}, s_1) \cdots \mathcal{W}_{\alpha_k}(f_k^{\alpha_k}, \bar{f}_k^{\alpha_k}, s_k),$$

where the *marginal* joint quasi-probability distributions are defined as

$$Q_{\alpha_1,\ldots,\alpha_k}^{(k)}(f_1^{\alpha_1}, \bar{f}_1^{\alpha_1}, s_1; \ldots; f_k^{\alpha_k}, \bar{f}_k^{\alpha_k}, s_k)$$
$$= \operatorname{tr}_E \left( \hat{P}_{\alpha_1}(f_1^{\alpha_1}, s_1) \cdots \hat{P}_{\alpha_k}(f_k^{\alpha_k}, s_k) \hat{\rho}_E \hat{P}_{\alpha_k}(\bar{f}_k^{\alpha_k}, s_k) \cdots \hat{P}_{\alpha_1}(\bar{f}_1^{\alpha_1}, s_1) \right).$$

Naturally, marginal quasi-distributions satisfy the consistency relation,

$$\sum_{f_i^{\alpha_i}} \sum_{\bar{f}_i^{\alpha_i}} Q_{\alpha_1,\ldots,\alpha_k}^{(k)}(f_1^{\alpha_1}, \bar{f}_1^{\alpha_1}, s_1; \ldots; f_k^{\alpha_k}, \bar{f}_k^{\alpha_k}, s_k)$$
$$= Q_{\ldots,\alpha_{i-1},\alpha_{i+1},\ldots}^{(k-1)}(\ldots; f_{i-1}^{\alpha_{i-1}}, \bar{f}_{i-1}^{\alpha_{i-1}}, s_{i-1}; f_{i+1}^{\alpha_{i+1}}, \bar{f}_{i+1}^{\alpha_{i+1}}, s_{i+1}; \ldots).$$

**Assignment 35 (Sec. 4.4)** *Calculate the explicit forms of super-qumulants $\mathcal{L}^{(1)}(s)$ and $\mathcal{L}^{(2)}(s)$ for the general SE coupling,*

$$\hat{V}_{SE} = \lambda \sum_\alpha \hat{V}_\alpha \otimes \hat{F}_\alpha,$$

*where $\hat{V}_\alpha$ and $\hat{F}_\alpha$ are hermitian operators.*

*Solution*:

$$\mathcal{L}^{(1)}(s) = \sum_\alpha \langle F_\alpha(s) \rangle \left[ \hat{V}_\alpha(s), \bullet \right];$$

$$\mathcal{L}^{(2)}(s) = \sum_{\alpha,\beta} \int_0^s \langle F_\alpha(s) F_\beta(u) \rangle \left( \hat{V}_\alpha(s) \hat{V}_\beta(u) \bullet - \hat{V}_\beta(u) \bullet \hat{V}_\alpha(s) \right) du$$

$$+ \sum_{\alpha,\beta} \int_0^s \langle \bar{F}_\beta(s) \bar{F}_\alpha(u) \rangle \left( \bullet \hat{V}_\alpha(u) \hat{V}_\beta(s) - \hat{V}_\beta(s) \bullet \hat{V}_\alpha(u) \right) du$$

$$- \sum_{\alpha,\beta} \int_0^s \langle F_\alpha(s) \rangle \langle F_\beta(u) \rangle \left[ \hat{V}_\alpha(s), [\hat{V}_\beta(u), \bullet] \right] du,$$

where, for $t > s$,

$$\langle F_\alpha(t) F_\beta(s) \rangle = \text{tr}_E \left( \hat{F}_\alpha(t) \hat{F}_\beta(s) \hat{\rho}_E \right), \quad \langle \bar{F}_\beta(t) \bar{F}_\alpha(s) \rangle = \text{tr}_E \left( \hat{F}_\alpha(s) \hat{F}_\beta(t) \hat{\rho}_E \right).$$

**Assignment 36 (Sec. 4.7)** *Prove the Fluctuation-Dissipation theorem,*

$$S(\omega) = i \left( \frac{1 + e^{-\beta\omega}}{1 - e^{-\beta\omega}} \right) \kappa(\omega),$$

*where $\beta = 1/k_\text{B}T$, and*

$$S(\omega) = \int_{-\infty}^\infty C_F(\tau, 0) e^{-i\omega\tau} d\tau,$$

$$\kappa(\omega) = \int_{-\infty}^\infty K_F(\tau, 0) e^{-i\omega\tau} d\tau.$$

*Solution*: Assume that $E$ is initially in the thermal state,

$$\hat{\rho}_E = \frac{e^{-\beta\hat{H}_E}}{\text{tr}_E(e^{-\beta\hat{H}_E})} = \frac{1}{Z} \sum_a e^{-\beta\epsilon_a} |a\rangle\langle a|,$$

where $\hat{H}_E |a\rangle = \epsilon_a |a\rangle$, and define the centered coupling operator,

$$\hat{X}(t) \equiv \hat{F}_I(t) - \langle F(t) \rangle \hat{1} = \hat{F}_I(t) - \text{tr}_E \left( e^{it\hat{H}_E} \hat{F} e^{-it\hat{H}_E} \frac{e^{-\beta\hat{H}_E}}{Z} \right) \hat{1} = \hat{F}_I(t) - \text{tr}_E \left( \hat{F} \hat{\rho}_E \right) \hat{1}$$

$$= \hat{F}_I(t) - \langle F \rangle \hat{1}.$$

Since $\hat{H}_E$ commutes with $\hat{\rho}_E$, the correlation function and the susceptibility depend only on the difference of their arguments, e.g.,

$$C_F(t, s) = \frac{1}{2} \text{tr}_E \left[ \left( e^{it\hat{H}_E} \hat{F} e^{i(t-s)\hat{H}_E} \hat{F} e^{-is\hat{H}_E} + e^{is\hat{H}_E} \hat{F} e^{i(t-s)\hat{H}_E} \hat{F} e^{-it\hat{H}_E} \right) \frac{e^{-\beta\hat{H}_E}}{Z} \right] - \langle F \rangle^2$$

$$= \frac{1}{2} \text{tr}_E \left[ \left( e^{i(t-s)\hat{H}_E} \hat{F} e^{i(t-s)\hat{H}_E} \hat{F} + \hat{F} e^{i(t-s)\hat{H}_E} \hat{F} e^{-i(t-s)\hat{H}_E} \right) \frac{e^{-\beta\hat{H}_E}}{Z} \right] - \langle F \rangle^2$$

$$= \frac{1}{2} \text{tr}_E \left[ \left( \hat{X}(t-s) \hat{X}(0) + \hat{X}(0) \hat{X}(t-s) \right) \hat{\rho}_E \right] = C_F(t-s, 0).$$

Now we proceed to prove the theorem; we start with the decomposition of $\hat{X}$ in the basis of Hamiltonian's eigenkets,

$$\hat{X}(t) = \sum_{a,b} e^{i(\epsilon_a - \epsilon_b)t} |a\rangle X_{ab} \langle b| \equiv \sum_{a,b} e^{i\omega_{ab}t} |a\rangle X_{ab} \langle b|.$$

Then we substitute this form into $C_F$ and $K_F$ to obtain,

$$C_F(\tau, 0) = \sum_{a,b} \frac{\left(e^{i\omega_{ab}\tau} X_{ab} X_{ba} + e^{i\omega_{ba}\tau} X_{ab} X_{ba}\right) e^{-\beta \epsilon_a}}{2} \frac{e^{-\beta \epsilon_a}}{Z} = \sum_{a,b} \frac{e^{-\beta \epsilon_a}}{Z} |X_{ab}|^2 \left(\frac{e^{i\omega_{ab}\tau} + e^{-i\omega_{ab}\tau}}{2}\right);$$

$$K_F(\tau, 0) = \sum_{a,b} \frac{e^{-\beta \epsilon_a}}{Z} |X_{ab}|^2 \left(\frac{e^{-i\omega_{ab}\tau} - e^{i\omega_{ab}\tau}}{2i}\right).$$

Next step is to calculate the Fourier transforms of $C_F$ and $K_F$,

$$S(\omega) = \int_{-\infty}^{\infty} C_F(\tau, 0) e^{-i\omega\tau} d\tau$$

$$= \pi \sum_{a,b} \frac{e^{-\beta \epsilon_a}}{Z} |X_{ab}|^2 \delta(\omega_{ab} - \omega) + \pi \sum_{a,b} \frac{e^{-\beta \epsilon_a}}{Z} |X_{ab}|^2 \delta(\omega_{ab} + \omega)$$

$$= \pi \sum_{a,b} \frac{1}{Z} |X_{ab}|^2 \delta(\omega_{ab} - \omega) \left(e^{-\beta \epsilon_a} + e^{-\beta \epsilon_b}\right)$$

$$= \pi \sum_{a,b} \frac{e^{-\beta \epsilon_b}}{Z} |X_{ab}|^2 \delta(\omega_{ab} - \omega) \left(1 + e^{-\beta \omega_{ab}}\right)$$

$$= \left(1 + e^{-\beta \omega}\right) \pi \sum_{a,b} \frac{e^{-\beta \epsilon_b}}{Z} |X_{ab}|^2 \delta(\omega_{ab} - \omega);$$

$$\kappa(\omega) = \int_{-\infty}^{\infty} K_F(\tau, 0) e^{-i\omega\tau} d\tau = -i\pi \sum_{a,b} \frac{e^{-\beta \epsilon_a}}{Z} |X_{ab}|^2 \left[\delta(\omega_{ab} + \omega) - \delta(\omega_{ab} - \omega)\right]$$

$$= -i\left(1 - e^{-\beta \omega}\right) \pi \sum_{a,b} \frac{e^{-\beta \epsilon_b}}{Z} |X_{ab}|^2 \delta(\omega_{ab} - \omega),$$

which gives us the theorem.

**Assignment 37 (Sec. 5.3)** *Show that*

$$\mathcal{U}_S'(t) \mathcal{L}_D \mathcal{U}_S'(-t) = \mathcal{L}_D.$$

*Solution*: For the definition of frequency decomposition (268) we get the following relations,

$$\hat{U}_S'(t) \hat{V}_\omega \hat{U}_S'(-t) = e^{-i\omega t} \hat{V}_\omega;$$

$$\hat{U}_S'(t) \hat{V}_\omega^\dagger \hat{U}_S'(-t) = e^{i\omega t} \hat{V}_\omega^\dagger.$$

Now, let us investigate how $\mathcal{U}_S'$ affects each part of the super-operator $\mathcal{L}_D$:

$$\mathcal{U}_S'(t)[\hat{V}_\omega^\dagger \hat{V}_\omega, \bullet] \mathcal{U}_S'(-t) = \hat{U}_S'(t)[\hat{V}_\omega^\dagger \hat{V}_\omega, \hat{U}_S'(-t) \bullet \hat{U}_S'(t)] \hat{U}_S'(-t)$$

$$= [\hat{U}_S'(t) \hat{V}_\omega^\dagger \hat{U}_S'(-t) \hat{U}_S'(t) \hat{V}_\omega \hat{U}_S'(-t), \bullet]$$

$$= [\left(e^{+i\omega t} \hat{V}_\omega^\dagger\right)\left(e^{-i\omega t} \hat{V}_\omega\right), \bullet] = [\hat{V}_\omega^\dagger \hat{V}_\omega, \bullet];$$

$$\mathcal{U}_S'(t)\{\hat{V}_\omega^\dagger \hat{V}_\omega, \bullet\} \mathcal{U}_S'(-t) = \{\hat{V}_\omega^\dagger \hat{V}_\omega, \bullet\};$$

$$\mathcal{U}_S'(t)\left(\hat{V}_\omega \bullet \hat{V}_\omega^\dagger\right) \mathcal{U}_S'(-t) = \hat{U}_S'(t) \hat{V}_\omega \hat{U}_S'(-t) \bullet \hat{U}_S'(t) \hat{V}_\omega^\dagger \hat{U}_S'(-t) = \hat{V}_\omega \bullet \hat{V}_\omega^\dagger,$$

i.e. $\mathcal{U}_S'$ does not change any super-operator within $\mathcal{L}_D$, therefore, $\mathcal{U}_S'(t) \mathcal{L}_D \mathcal{U}_S'(-t) = \mathcal{L}_D$.

**Assignment 38 (Sec. 5.3)** *Assuming that the Hamiltonian $\hat{H}'_S = \sum_a \epsilon_a |a\rangle\langle a|$ is non-degenerate,*

$$\forall_{a \neq b} \epsilon_a \neq \epsilon_b,$$

*derive the Pauli master equation: the rate equation obtained from Davies equation by restricting it to the diagonal elements of the density matrix,*

$$\frac{d}{dt}\langle a|\hat{\rho}_S(t)|a\rangle = \sum_b M_{a,b}\langle b|\hat{\rho}_S(t)|b\rangle.$$

*Solution*: First, we calculate the action of the Davies generator $\mathcal{L}_D$ on the projectors $|a\rangle\langle a|$,

$$\left(\sum_\omega 2\,\mathrm{Re}\{\gamma_\omega\}\hat{V}_\omega \bullet \hat{V}_\omega^\dagger\right)|a\rangle\langle a|$$

$$= \sum_\omega 2\,\mathrm{Re}\{\gamma_\omega\}\sum_{a',b'}\sum_{a'',b''}\delta(\omega - \epsilon_{a'} + \epsilon_{b'})\delta(\omega - \epsilon_{a''} + \epsilon_{b''})V_{a'b'}V_{b''a''}|a'\rangle\langle b'|a\rangle\langle a|b''\rangle\langle a''|$$

$$= \sum_\omega 2\,\mathrm{Re}\{\gamma_\omega\}\sum_{a',a''}\delta(\omega - \epsilon_{a'} + \epsilon_a)\delta(\omega - \epsilon_{a''} + \epsilon_a)V_{a'a}V_{aa''}|a'\rangle\langle a''|.$$

Now, since the energy levels are non-degenerate, there is a one-to-one correspondence between the eigenket indexes $a, a', \dots$ and eigenvalues $\epsilon_a, \epsilon_{a'}, \dots$, therefore we can unambiguously resolve the deltas,

$$\left(\sum_\omega 2\,\mathrm{Re}\{\gamma_\omega\}\hat{V}_\omega \bullet \hat{V}_\omega^\dagger\right)|a\rangle\langle a| = \sum_\omega 2\,\mathrm{Re}\{\gamma_\omega\}\sum_{a',a''}\delta(\omega - \epsilon_{a'} + \epsilon_a)\delta(\epsilon_{a'} - \epsilon_{a''})V_{a'a}V_{aa''}|a'\rangle\langle a''|$$

$$= \sum_\omega 2\,\mathrm{Re}\{\gamma_\omega\}\sum_b \delta(\omega - \epsilon_b + \epsilon_a)|V_{ab}|^2|b\rangle\langle b|$$

$$= \sum_b \frac{2S(\omega_{ba})}{e^{\beta\omega_{ba}} + 1}|V_{ab}|^2|b\rangle\langle b| \equiv \sum_b \gamma_{b,a}|b\rangle\langle b|,$$

where $\omega_{ba} = \epsilon_b - \epsilon_a$. We proceed to the second part of the generator,

$$\left(\sum_\omega \mathrm{Re}\{\gamma_\omega\}\{\hat{V}_\omega^\dagger\hat{V}_\omega, \bullet\}\right)|a\rangle\langle a|$$

$$= \sum_\omega \mathrm{Re}\{\gamma_\omega\}\left\{\sum_{b,c,b',c'}\delta(\omega - \epsilon_b + \epsilon_c)\delta(\omega - \epsilon_{b'} + \epsilon_{c'})V_{cb}V_{b'c'}|c\rangle\langle b|b'\rangle\langle c'|, |a\rangle\langle a|\right\}$$

$$= \sum_\omega \mathrm{Re}\{\gamma_\omega\}\left\{\sum_{b,c,c'}\delta(\omega - \epsilon_b + \epsilon_c)\delta(\omega - \epsilon_b + \epsilon_{c'})V_{cb}V_{bc'}|c\rangle\langle c'|, |a\rangle\langle a|\right\}$$

$$= \sum_\omega \mathrm{Re}\{\gamma_\omega\}\left\{\sum_{b,c,c'}\delta(\omega - \epsilon_b + \epsilon_c)\delta(\epsilon_{c'} - \epsilon_c)V_{cb}V_{bc'}|c\rangle\langle c'|, |a\rangle\langle a|\right\}$$

$$= \sum_\omega \mathrm{Re}\{\gamma_\omega\}\left\{\sum_{b,c}\delta(\omega - \epsilon_b + \epsilon_c)|V_{bc}|^2|c\rangle\langle c|, |a\rangle\langle a|\right\} = |a\rangle\langle a|\sum_b \gamma_{b,a}.$$

We have just showed that projectors are mapped by $\mathcal{L}_D$ onto combination of projectors. Similar calculation shows the same is true for coherences $|a\rangle\langle b|$ $(a \neq b)$,

$$\left(\sum_\omega 2\,\mathrm{Re}\{\gamma_\omega\}\hat{V}_\omega \bullet \hat{V}_\omega^\dagger\right)|a\rangle\langle b| = \sum_{c,c'}\frac{2S(\omega_{ca})}{e^{\beta\omega_{ca}} + 1}\delta(\epsilon_c - \epsilon_{c'} - (\epsilon_a - \epsilon_b))V_{ca}V_{bc'}|c\rangle\langle c'|$$

$$= \sum_{c,c'}(1 - \delta_{c,c'})\frac{2S(\omega_{ca})}{e^{\beta\omega_{ca}} + 1}\delta(\omega_{cc'} - \omega_{ab})V_{ca}V_{bc'}|c\rangle\langle c'|,$$

and

$$\left(\sum_\omega \mathrm{Re}\{\gamma_\omega\}\{\hat{V}_\omega^\dagger \hat{V}_\omega, \bullet\}\right)|a\rangle\langle b| = |a\rangle\langle b| \sum_c \frac{\gamma_{c,a}+\gamma_{c,b}}{2}\,.$$

Of course, also the Hamiltonian part of the dynamical map maps projector onto projectors and coeherences onto coherence; in this case, it is because $|a\rangle\langle b|$ are its eigenoperators,

$$[\hat{H}_S', \bullet]|a\rangle\langle b| = \omega_{ab}|a\rangle\langle b|\,.$$

Therefore, we can write a closed equation only for diagonal elements (in the basis of $\hat{H}_S'$ eigenkets) of the density matrix—the *Pauli master equation*,

$$\begin{aligned}
\frac{d}{dt}\langle a|\hat{\rho}_S(t)|a\rangle &= \langle a|\frac{d}{dt}\mathcal{U}_S'(t)e^{\lambda^2 t\mathcal{L}_D}\hat{\rho}_S|a\rangle = \lambda^2\langle a|\mathcal{L}_D\hat{\rho}_S(t)|a\rangle \\
&= \lambda^2\langle a|\left(\mathcal{L}_D\sum_b\langle b|\hat{\rho}_S(t)|b\rangle|b\rangle\langle b|\right)|a\rangle \\
&= \lambda^2\sum_b\langle b|\hat{\rho}_S(t)|b\rangle\sum_c\gamma_{c,b}\langle a|\left(-|b\rangle\langle b|+|c\rangle\langle c|\right)|a\rangle \\
&= \lambda^2\sum_b\left[\gamma_{a,b}\langle b|\hat{\rho}_S(t)|b\rangle - \gamma_{b,a}\langle a|\hat{\rho}_S(t)|a\rangle\right] \\
&= \lambda^2\sum_b\left(\gamma_{a,b}-\delta_{a,b}\sum_c\gamma_{c,a}\right)\langle b|\hat{\rho}_S(t)|b\rangle\,.
\end{aligned}$$

**Assignment 39 (Sec. 7.2)** *Show that the diagonal part of joint quasi-probability is non-negative,*

$$P_F^{(k)}(\boldsymbol{f},\boldsymbol{s}) = Q_F^{(k)}(\boldsymbol{f},\boldsymbol{f},\boldsymbol{s}) \geqslant 0\,.$$

*Solution*: Recall that an operator $\hat{A}$ is positive, if and only if, it can be written as $\hat{A}=\hat{O}_A^\dagger\hat{O}_A$. Then, we express the joint quasi-probability as a trace of an operator,

$$\begin{aligned}
Q_F^{(k)}(\boldsymbol{f},\boldsymbol{f},\boldsymbol{s}) &= \mathrm{tr}_E(\hat{q}(\boldsymbol{f},\boldsymbol{s}))\,; \\
\hat{q}(\boldsymbol{f},\boldsymbol{s}) &= \hat{P}_I(f_1,s_1)\cdots\hat{P}_I(f_k,s_k)\hat{\rho}_E\hat{P}_I(f_k,s_k)\cdots\hat{P}_I(f_1,s_1)\,,
\end{aligned}$$

where $\hat{P}_I(f,s) = e^{is\hat{H}_E}\hat{P}(f)e^{-is\hat{H}_E}$ and $\hat{F}=\sum_f f\hat{P}(f)$.

Since the initial density matrix is positive, it can be decomposed as $\hat{\rho}_E=\hat{O}_\rho^\dagger\hat{O}_\rho$; when we substitute this form into $\hat{q}$ we get

$$\begin{aligned}
\hat{q}(\boldsymbol{f},\boldsymbol{s}) &= \left(\hat{P}_I(f_1,s_1)\cdots\hat{P}_I(f_k,s_k)\hat{O}_\rho^\dagger\right)\left(\hat{O}_\rho\hat{P}_I(f_k,s_k)\cdots\hat{P}_I(f_1,s_1)\right) \\
&= \left(\hat{O}_\rho\hat{P}_I(f_k,s_k)\cdots\hat{P}_I(f_1,s_1)\right)^\dagger\left(\hat{O}_\rho\hat{P}_I(f_k,s_k)\cdots\hat{P}_I(f_1,s_1)\right) = \hat{O}_q^\dagger\hat{O}_q\,.
\end{aligned}$$

Therefore, $\hat{q}$ is positive (and also hermitian), and thus, its eigenvalues, $q_i$, are non-negative. This immediately implies that $Q_F$ is non-negative because

$$Q_F^{(k)}(\boldsymbol{f},\boldsymbol{f},\boldsymbol{s}) = \mathrm{tr}_E(\hat{q}(\boldsymbol{f},\boldsymbol{s})) = \sum_i q_i \geqslant 0\,.$$

**Assignment 40 (Sec. 7.2)** *Show that the diagonal part of joint quasi-probability is normalized,*

$$\sum_{f_1,\dots,f_k} Q_F^{(k)}(\boldsymbol{f},\boldsymbol{f},\boldsymbol{s}) = 1\,.$$

*Solution*: Using the cyclic property of the trace and the operator $\hat{q}$ defined in the previous assignment 39 we write

$$\sum_{f_1} Q_F^{(k)}(\boldsymbol{f},\boldsymbol{f},\boldsymbol{s}) = \sum_{f_1} \mathrm{tr}_E\left(\hat{P}_I(f_1,s_1)\hat{q}(f_2,s_2;\ldots;f_k,s_k)\hat{P}_I(f_1,s_1)\right)$$

$$= \mathrm{tr}_E\left(e^{is_1\hat{H}_E}\left(\sum_{f_1}\hat{P}(f_1)\right)e^{-is_1\hat{H}_E}\hat{q}(f_2,s_2;\ldots;f_k,s_k)\right)$$

$$= \mathrm{tr}_E\left(e^{is_1\hat{H}_E}\hat{1}e^{-is_1\hat{H}_E}\hat{q}(f_2,s_2;\ldots;f_k,s_k)\right)$$

$$= \mathrm{tr}_E\left(\hat{q}(f_2,s_2;\ldots;f_k,s_k)\right) = Q_F^{(k-1)}(f_2,f_2,s_2;\ldots;f_k,f_k,s_k).$$

Summing over consecutive arguments $f_2, f_3, \ldots$, etc., and using the above relation in each step we arrive at

$$\sum_{f_1,\ldots,f_k} Q_F^{(k)}(\boldsymbol{f},\boldsymbol{f},\boldsymbol{s}) = \sum_{f_k} Q_F^{(1)}(f_k,f_k,s_k) = \sum_{f_k} \mathrm{tr}_E\left(\hat{P}(f_k)e^{-is_k\hat{H}_E}\hat{\rho}_E\,e^{is_k\hat{H}_E}\right) = \mathrm{tr}_E\left(\hat{\rho}_E\right) = 1.$$

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
