# Peer review of "Introduction to the theory of open quantum systems"

_SciPost Physics Lecture Notes, doi:SciPost Phys. Lect. Notes 68 (2023)_

## Round 3 · Referee Report · Anonymous · 2022-12-24

Strengths
1- Self-contained and original introduction in the field of open quantum systems
2- Many exercises with solutions
3- Clear style and "easy" to read
Weaknesses
1- Calculations are too verbose at times.
Report
The lecture note are a clear introduction to the field of open quantum systems. The presentation is original and on a topic of ongoing interest. In principle, I am positive towards recommending publication. However, I do believe that the presentation could and should be improved in several ways.
In particular, the abstract should be rewritten. It should be explained what the notes contain and what is novel (so far the novelty is only claimed and it is left to the reader to figure out what the author means).
Given that chapter 1 covers the topic that is typically covered in graduate quantum mechanics, I would propose to streamline/sharpen it a bit. There are too many assignments that break the flow of the text. The author should concentrate on the points (such as "generalized" time-ordering) that is important for the rest of the text.
In assignment 3 and 8, expressions are proved by showing that the "two sides" fulfill the same differential equation with the same initial condition. The main reason for the time-ordering is that taking the time-derivative so that Eq. (15) holds. I do believe that some of the proves (also in late chapters) can be made more crisp by showing that LHS and RHS fulfill the same ODE (rather than manipulating long strings of time-ordered Taylor expansion). I do understand that the author has certain reservations (see word of caution after Eq. (30) ), but I do not think that this speaks agains the alternative way of prove I propose. In principle, I would encourage to get rid of manipulations like in Assignment 7 as much as possible (maybe doing it once or so). This mainly concerns assignments (like assignments 10, 22, 25, ...)
I would consider to exchange chapters 4 and 5. The reason is that chapter 5 in some respects is "simpler" than chapter 4. Students are cannot be expected to already know about master equations. So it would be nice to first show what "nice" equations there are with potential "derivations" and then show how the formalism that has been introduced in 1-3 may serve to derive these results more coherently.
Requested changes
1- Rewrite the abstract to explain explicitely what is done in the lecture notes
2-Shorten chapter 1, in particular reduce the number of assignments
3- Consider showing some results by showing that both sides solve the same ODE with the same initial condition to reduce the algebra involved
4- Consider swapping chapters 4 and 5
5- There are some mathematical inaccuracies in Sec. 2.5: It is not true that non-negativity implies Hermiticity (as claimed in point 2). In fact, decomposing an arbitrary operator $O= H+A$ in Hermitian $H= (O+O^\dagger)/2$ and anti-Hermitian part $A= (O-O^\dagger)/2$, we see that point 1 says nothing about $A$. The discussion around Eq. (55) has to be improved as well. As I understand the general (non-orthogonal) functions $|\phi_i\rangle$ are states (thus normalized). But then it does not follow that $\sum_i q_i=1$. The fact that the decomposition of a non-pure state is not unique can be best explained or example on the Bloch sphere, where a point in the interior can be written as a convex combination of points on the surface in many different ways.
6- The correlation function of Eq. (95) are at times used for arbitrary time arguments later. It would be good to explain in Eq. (95) the symmetry of the correlation functions and the moments when exchanging the time arguments.
7- The presentation is very clear. However, I have notices some typos. Maybe reading the text once more can still improve the presentation (e.g., "Even more fundamental axiom of quantum theory ", "with integration of dynamical equation", "longest time" -> "largest time ", "your self" -> "yourself" ). Also please replace the colloquial '“fancy” font' with 'caligraphic font'. In assignment 26, there $\sqrt 2$ should be in the denominator. The statement "Formally, the density matrix is also a vector" does not make sense. I believe that it should be explained in more details. It would also make sense to call Eq. (63) the "Hilbert-Schmidt" inner-product, to allow for consulting additional literature.
8- It would be good to put the tensor product in bracket whenever ambiguities in the expression are possible. This involves for example Eqs. (186) -(191).
9- It would be nice to state explicitly *before* Eq. (166) that in this expression the super-cumulants $C^{(k)}$ are defined implicitly.
10- I believe that the random unitary maps are not always unitary. If so, I would explain this important fact explicitly, e.g., in the paragraph following Eq. (109).
11- The expression Eq. (96) in Sec. 3.1.3 falls from the sky. I would propose to first introduce the Markovian property and then describe that it only relies on the transition probability. One then could simply state the transition probability for a telegraph process, or even derive it from the rate equation. Then Eq. (96) is much more clear.
12- I would encourage to change the sign of $\lambda$ in the expression of the coherence function $W(t)$ to conform with the standard definition of the characteristic function in statistics.
13- From a graphics point of view, it would be nice to get the two lines for the cumulant average closer together if possible.
14- In Eq. (84), the sum should either go over the Img(X) or it should be explained that Img(X) is not only discrete but a subset of the integers.
15- At times the formulas go into the margin: reformatting the expression should be done to improve the layout.
Author: Piotr Szankowski on 2022-12-29 [id 3193]
(in reply to Report 1 on 2022-12-24)I would like to thank the Reviewer for careful evaluation of this, admittedly voluminous, manuscript. Many of the comments and critiques raised by the Reviewer have inspired me to reconsider some of the choices I have made composing this script. The most impactful was the criticism of the approach to the presentation of the assignments and their solutions: as the Reviewer has correctly pointed out, some of the solutions take too much space and because of that they quite severely disrupt the natural flow the text. To remedy this issue I have decided to move all of the solutions to the dedicated appendix at the end of the manuscript—this is the largest change I have made but the resultant improvements cannot be overstated. The detailed responses to all the comments of the Reviewer all listed below.
The Reviewer comment 1:
Reply I have made an attempt (hopefully it was successful) to summarize in the span of two sentences the quasi-probability/super-qumulant approach presented in chapter 4. I believe this should be enough to indicate to the experienced reader where to expect the advertised ‘novelty’ of the approach.
Comments 2 and 3
Reply To fix the issue with assignments taking too much space and breaking up the flow, I have decided to move the solutions from the main body of the text to the dedicated appendix at the end of the manuscript. To ease the navigation through the document (at least, in pdf format) I have created a system of hyperlinks allowing for jumping back-and-forth between the solution in the appendix and the assignment placed in the main text. I agree with the Reviewer that some of the solution could be made ‘leaner’ and simpler; however, the efficiency was not my goal here. I have decided to go with the more laborsome route of algebraic transformations to showcase some useful tricks for handling typical calculations involving time-ordered integrals. Ultimately, my intention was to, in a way, ‘demystify’ time-ordering and to equip the students with tools and know-how necessary to employ a brute-force approach to these kinds of problems; brute-force methods are never elegant, but they are always reliable.
Comment 4
Reply The ordering of chapters was carefully premeditated and it serves a concrete purpose. The first four chapters were composed as a part of a single ‘plot arc’ or a ‘thread’. Chapter 2 (Dynamics of closed systems) introduces the language of super-operators, it explains the concept of ‘specialized’ time-ordering, and it points to the issue of computability and how it is solved via dynamical equation. Then, chapter 3 (Stochastic dynamics) shows that not all dynamics can be described with unitary maps; stochastic maps are used to introduce trajectories and their distributions, plus the moments and cumulants in the least complicated setting possible. The issue of computablility is further expanded on and it is showed how the technology of specialized time-ordering and cumulants can used to construct an approximated solvable dynamical equations (i.e., the super-cumulants). Finally, in chapter 4 (Dynamics of open systems), aside of defining what an open quantum system is, all the pieces from previous chapters are gathered together to show how to solve in a systematic way the problem of computability of dynamical maps: joint quasi-probabilities are introduced as a generalization of joint probabilities and quasi-cumulants (qumulants) as a generalization of stochastic cumulants. Super-qumulants, the generalization of super-cumulants, are found thanks to the structural parallelism with stochastic maps, and then, are used to derive approximated dynamical equations mirroring the approach from the previous chapter. Overall, the goal was to teach the students how to handle a general problem of finding the dynamics of an open system: how to determine what kind of information about the environment is needed, how to actually compute the dynamical map in practice, and how to control the accuracy of used approximations. Essentially, one could consider the program of the course to be concluded already in chapter 4.
Chapter 5 (Master equation) is a kind of historic review of the traditional textbook program where the theory is constructed axiomatically with the concept of CP maps at its core. (See the discussion at the beginning of Sec. 5.1.) It is true that the master equation (GKLS form, dynamical semi-groups etc.) is much simpler than super-qumulants, but after preparation with chapters 2 and 3, super-qumulants are a more natural progression. If chapters 4 and 5 were swapped, then the sudden ‘gear shift’ from computability, dynamical equations, cumulants etc. to abstract maps, complete positvity, GKLS theorem etc. would be quite jarring, in my opinion. Not to mention that without the super-qumulant method, the traditional way to obtain the master equation from first principles is the Born-Markov ansatz route; if the super-qumulants were introduced after that, it would be supremely awkward to go back to the derivation and explain why it was incorrect and we should have used this new method in the first place.
Also, in the defense of the current ordering, the students I have subjected to this program seemed to take it quite well; in fact, the relatively simple master equation was a good kind of palate cleanser after the main course of super-qumulants.
Comment 5a
Reply According to my understanding, non-negativity (or positive semi-definiteness) does imply Hermicity. This can be shown on the example of the decomposition brought up by the Reviewer: $ O = H + i{A}'$, where $ H=(O+ O^\dagger)/2$ is Hermitian ($ H^\dagger = H$) and $\hat A = ( O- O^\dagger)/2 = i( O- O^\dagger)/2i = i A'$ is anti-Hermitian ($ A^\dagger = - A$), and thus,$ A'$ is Hermitian. Since both $ H$ and $ A'$ are Hermitian, an expectation value on an arbitrary state of those operators is a real number: $h(\phi) = \langle \phi| H|\phi\rangle$ , $a(\phi) = \langle\phi|A'|\phi\rangle$, and $h, a \in \mathrm{Reals}$. Therefore, an expectation value of $O$ is, in general, a complex number: $\langle \phi| O|\phi\rangle = h(\phi) + i a(\phi)$. Now, if we assume that $ O$ is non-negative, i.e., $\langle\phi| O|\phi\rangle \geq 0$ for all $|\phi\rangle$, then, it follows that the imaginary part $a(\phi)$ must be zero for any $|\phi\rangle$ because non-real numbers cannot be non-negative. Hence, the ani-hermitian part of the operator, $ A$, must vanish, and so, $ O$ is Hermitian.
Comment 5b
Reply I have rewritten the whole discussion about mixed density matrices to make the explanation clearer.
Comment 6
Reply I have added to Sec. 3.1.2 the explanation why moments of stochastic process are completely symmetric with respect to their time arguments.
Comment 7
Reply I have fixed the typos. I have also rewritten the passage about the Hilbert space of operators to make it more clear.
Comment 8
Reply: Fixed.
Comment 9
Reply: Fixed.
Comment 10
Reply: Done.
Comment 11
Reply The formula for the joint probability of RTN seems to come out of nowhere because it is introduced as a definition of the process. There is no way around it, though; the random telegraph noise, or any stochastic process for that matter, is a constructed entity—it cannot be derived from some first principles. Of course, one could chose to construct it in a less direct way; like the Reviews suggests, instead of giving the explicit formula, one could demand that joint distributions factorize (i.e., have Markov property), and that the conditional probability is a solution to a given rate equation. In my opinion the explicit route, where the formula is supplied at the beginning, minimizes the chance that the listener will be confused or overburdened by unnecessary additional information and factoids.
It is certainly true that whether the stochastic process in question is Markovian is a key information for the study of the process in itself. However, it is only a curiosity from the point of view of the program of this course. This is the main reason I chose not to mention Markovianity in the main text. Of course, I recognize the importance of Markovian processes for the dedicated theory of stochastic process and the compromise I have settled for was to delegate this topic to assignments. In assignment 17 I ask to check that RTN is a markovian process (with a concise explanation what Markovianity entials). Then, in assignment 19, the task is to utilize markovianity to solve a practical problem of numerical generation of process trajectories. Also, in assignment 18 the problem is to find the explicit formula for joint distribution of generalized RTN (the one where the rates of switching are unequal) following the indirect construction where the markovianity is assumed and there is a rate equation to solve.
Given all that, I still think it was the correct choice to introduce the RTN as I did in the manuscript.
Comment 12
Reply: Done.
Comment 13
Reply: I completely agree with the Reviewer on this point; however, I don’t know how to make it better.
Comment 14
Reply: Fixed.
Comment 15
Reply: I have fixed all case I’ve managed to catch.

---

## Round 4 · Referee Report · Anonymous (Referee 1) · 2023-1-14

Report

Having read the reply of the author, I do believe that the manuscript has been substantially improved compared to the first version. In particular, I do like the decision to collect the solutions in an appendix. All errors have been fixed. The author did disagree with some of my suggestions. In each case, the reasons have been explained in details and I do understand the different point of view.

The subject "open systems" is of current ongoing interest. The presentation of the material is original, correct, and may serve as a reference for students and researchers entering the field. As a result, I do believe that the manuscript should be published as a Scipost lecture note.

---

## Round 4 · Author Response

I would like to thank the Reviewer for careful evaluation of this, admittedly voluminous, manuscript. Many of the comments and critiques raised by the Reviewer have inspired me to reconsider some of the choices I have made composing this script. The one with the most impact was the criticism of the approach to the presentation of the assignments and their solutions: as the Reviewer has correctly pointed out, some of the solutions take too much space and because of that they quite severely disrupt the natural flow the text. To remedy this issue I have decided to move all of the solutions to the dedicated appendix at the end of the manuscript—this is the largest change I have made but the resultant improvements cannot be overstated.

---

## Round 4 · List of Changes

• The solutions to assignments has been moved to dedicated appendix; the system of hyperlinks has been setup to ease the navigation in the pdf format of the manuscript.
  • The abstract has been modified to include more explicit information how the approach of this script is novel.
  • In Sec.2.5, the discussion on the decomposition of density matrix into a mixture of projectors has been augmented to make it more clear.
  • In Sec.2.6.1, the explanation how density matrix is a member of a vector space has be rewritten to make it more clear.
  • In Sec.3.1.2, the explanation has been added why moments of stochastic process are completely symmetric functions of their time arguments.
  • The definition of the coherence function W(t) in Sec.3.4.3 has been changed make it consistent with the conventional definition of the characteristic functional (i.e., i was replaced with -i).
  • In Sec.3.2 a comment has been added to immediately point out that, in general, stochastic maps are no longer unitary.
  • In Sec.3.6.3 an additional comment has been added before Eq.(154) to make it cleared that the equation is an implicit definition of super-cumulants \mathcal{C}^{(k)}.
  • Various typos has been fixed thought the manuscript.

---

## Editorial Decision

published